# An atlas of epithelial cell states and plasticity in lung adenocarcinoma

Guangchun Han[1,13], Ansam Sinjab[2,13], Zahraa Rahal[2,13], Anne M. Lynch[3,4,13], Warapen Treekitkarnmongkol[2], Yuejiang Liu[2,5], Alejandra G. Serrano[2], Jiping Feng[2], Ke Liang[2], Khaja Khan[2], Wei Lu[2], Sharia D. Hernandez[2], Yunhe Liu[1], Xuanye Cao[1], Enyu Dai[1], Guangsheng Pei[1], Jian Hu[6], Camille Abaya[2], Lorena I. Gomez-Bolanos[2], Fuduan Peng[1], Minyue Chen[2,5], Edwin R. Parra[2], Tina Cascone[7], Boris Sepesi[8], Seyed Javad Moghaddam[3], Paul Scheet[9], Marcelo V. Negrao[7], John V. Heymach[7], Mingyao Li[6], Steven M. Dubinett[10], Christopher S. Stevenson[11], Avrum E. Spira[11,12], Junya Fujimoto[2], Luisa M. Solis[2], Ignacio I. Wistuba[2], Jichao Chen[3,5,14 ✉], Linghua Wang[1,5,14 ✉] & Humam Kadara[2,5,14 ✉]

Understanding the cellular processes that underlie early lung adenocarcinoma (LUAD) development is needed to devise intervention strategies[1]. Here we studied 246,102 single epithelial cells from 16 early-stage LUADs and 47 matched normal lung samples. Epithelial cells comprised diverse normal and cancer cell states, and diversity among cancer cells was strongly linked to LUAD-specific oncogenic drivers. *KRAS* mutant cancer cells showed distinct transcriptional features, reduced differentiation and low levels of aneuploidy. Non-malignant areas surrounding human LUAD samples were enriched with alveolar intermediate cells that displayed elevated *KRT8* expression (termed *KRT8*[+] alveolar intermediate cells (KACs) here), reduced differentiation, increased plasticity and driver *KRAS* mutations. Expression profiles of KACs were enriched in lung precancer cells and in LUAD cells and signified poor survival. In mice exposed to tobacco carcinogen, KACs emerged before lung tumours and persisted for months after cessation of carcinogen exposure. Moreover, they acquired *Kras* mutations and conveyed sensitivity to targeted KRAS inhibition in KAC-enriched organoids derived from alveolar type 2 (AT2) cells. Last, lineage-labelling of AT2 cells or KRT8[+] cells following carcinogen exposure showed that KACs are possible intermediates in AT2-to-tumour cell transformation. This study provides new insights into epithelial cell states at the root of LUAD development, and such states could harbour potential targets for prevention or intervention.

LUADs are increasingly being detected at earlier pathological stages owing to enhanced screening[2–4]. Yet, patient prognosis remains moderate to poor, which warrants the need for improved early treatment strategies. Decoding the earliest events that drive LUADs can identify ideal targets for modulation. Previous work has shown that smoking leads to pervasive molecular (for example, *KRAS* mutations) and immune changes that are shared between LUADs and their adjacent normal-appearing ecosystems and are strongly associated with the development of lung premalignant lesions and LUAD[1,5–12]. However, most of these reports were based on bulk approaches and focused on tumour and distant sites of normal tissue in the lung. Therefore, the cellular and transcriptional phenotypes of expanded LUAD landscapes

remain understudied. Furthermore, although many lung single-cell RNA sequencing (scRNA-seq) studies have decoded immune and stromal states[13,14], little is known about epithelial cells. This is probably because of their paucity (around 4%) when performing single-cell analyses without enrichment of the epithelial compartment. Consequently, the identities of specific epithelial subsets or how they promote a field of injury, trigger progression of normal lung (NL) to premalignant lesion and promote LUAD pathogenesis remain unclear. Understanding cell-type-specific changes at the root of LUAD initiation will help identify actionable targets and strategies for the prevention of this morbid disease. Here we perform in-depth single-cell interrogation of malignant and normal epithelial cells from early-stage LUAD and

[1]Department of Genomic Medicine, The University of Texas MD Anderson Cancer Center, Houston, TX, USA. [2]Department of Translational Molecular Pathology, The University of Texas MD Anderson Cancer Center, Houston, TX, USA. [3]Department of Pulmonary Medicine, The University of Texas MD Anderson Cancer Center, Houston, TX, USA. [4]Graduate Program in Developmental Biology, Baylor College of Medicine, Houston, TX, USA. [5]The University of Texas Health Houston Graduate School of Biomedical Sciences, Houston, TX, USA. [6]Department of Biostatistics, Epidemiology and Informatics, Perelman School of Medicine, University of Pennsylvania, Philadelphia, PA, USA. [7]Department of Thoracic, Head and Neck Medical Oncology, The University of Texas MD Anderson Cancer Center, Houston, TX, USA. [8]Department of Cardiovascular and Thoracic Surgery, The University of Texas MD Anderson Cancer Center, Houston, TX, USA. [9]Department of Epidemiology, The University of Texas MD Anderson Cancer Center, Houston, TX, USA. [10]Department of Medicine, The University of California Los Angeles, Los Angeles, CA, USA. [11]Lung Cancer Initiative at Johnson & Johnson, Boston, MA, USA. [12]Section of Computational Biomedicine, School of Medicine, Boston University, Boston, MA, USA. [13]These authors contributed equally: Guangchun Han, Ansam Sinjab, Zahraa Rahal, Anne M. Lynch. [14]These authors jointly supervised this work: Jichao Chen, Linghua Wang, Humam Kadara. ✉e-mail: JChen16@mdanderson.org; lwang22@mdanderson.org; hkadara@mdanderson.org

from carcinogenesis and lineage-tracing mouse models that recapitulate the disease, with a focus on how specific populations evolve to give rise to malignant tumours.

## Epithelial transcriptional landscape

Our study combined in-depth scRNA-seq of early-stage LUAD clinical specimens and cross-species analysis and lineage tracing in a human-relevant model of LUAD development following exposure to tobacco carcinogen (Fig. 1a). We used scRNA-seq to study EPCAM-enriched epithelial cell subsets from early-stage LUAD samples from 16 patients and 47 paired NL samples spanning a topographical continuum from the LUADs, that is, tumour-adjacent, tumour-intermediate and tumour-distant locations[15] (Fig. 1a, Supplementary Fig. 1 and Supplementary Tables 1 and 2). We also collected tumour and normal tissue sets from the same regions for whole-exome sequencing (WES) profiling and high-resolution spatial transcriptomics (ST) and protein analyses (Fig. 1a).

Following quality control, 246,102 epithelial cells were retained for analyses (Supplementary Fig. 2 and Supplementary Table 2). Malignant cells ($n$ = 17,064) were distinguished from otherwise non-malignant normal cells ($n$ = 229,038) by integrating information from inferred copy number variation (inferCNV[16]), clustering distribution, lineage-specific gene expression and the presence of reads carrying $KRAS^{G12D}$ somatic mutations (Fig. 1b,c and Supplementary Fig. 3). Analyses of non-malignant clusters identified two major lineages—alveolar and airway—and a small subset of proliferative cells (Extended Data Fig. 1a and Supplementary Table 2). Airway cells ($n$ = 40,607) included basal ($KRT17^+$), ciliated ($FOXJ1^+$) and club and secretory ($SCGB1A1^+$) populations, as well as rare cell types such as ionocytes ($ASCL3^+$), neuroendocrine cells ($ASCL1^+$) and tuft cells ($GNAT3^+$) (Extended Data Fig. 1a and Supplementary Table 3). Alveolar cells ($n$ = 187,768) consisted of alveolar type 1 (AT1) cells ($AGER1^+ETV5^+$), AT2 cells ($SFTPB^+SFTPC^+$), $SCGB1A1^+SFTPC^+$ dual-positive cells and a cluster of alveolar intermediate cells (AICs) that was closely tucked between AT1 and AT2 clusters and shared gene expression features with both major alveolar cell types (Fig. 1b and Extended Data Fig. 1a,b).

Malignant cells showed low-to-no expression of lineage-specific markers and, overall, reduced lineage identity (Fig. 1b, bottom). Malignant cells formed 14 clusters (Fig. 1c) that were primarily patient-specific (Extended Data Fig. 1c, left), which signified strong inter-patient heterogeneity. Overall, malignant cells showed high levels of aneuploidy (Extended Data Fig. 1c, middle). We did not detect any distinct clustering pattern with respect to smoking status (Extended Data Fig. 1d). Annotation based on genomic profiling (by WES) showed that malignant cells from 3 patients with $KRAS$ mutant LUADs (KM-LUADs; patients P2, P10 and P14) clustered closely together. By contrast, malignant cells from other LUADs showed a more dispersed clustering pattern (Fig. 1d, Extended Data Fig. 1c,e and Supplementary Table 1). scRNA-seq analysis confirmed the presence of copy number variations (CNVs) and $KRAS^{G12D}$ mutations in patient-specific tumour clusters and the absence of $KRAS^{G12D}$ in $KRAS$ wild-type LUADs (KW-LUADs) (Extended Data Fig. 1c).

## LUAD malignant transcriptional programs

Malignant cells from KM-LUADs clustered together and distinctively from those of $EGFR$ mutant LUADs (EM-LUADs) or $MET$ mutant LUADs (MM-LUADs) (Fig. 1e). KM-LUADs showed more transcriptomic similarity (that is, shorter Bhattacharyya distances) at both sample and cell levels (Extended Data Fig. 1f, left and right, respectively) compared with other LUADs ($P < 2.2 \times 10^{-16}$). Distances between KM-LUADs (KM–KM) were significantly smaller compared with those between EM-LUADs (EM–EM; $P$ = 0.02) or other LUADs (other–other; $P$ = 0.03; Extended Data Fig. 1f, left). Clustering of malignant cells, following adjustment

for patient-specific effects, showed that cluster 5 was enriched with cells from KM-LUADs (patients P2, P10 and P14; Extended Data Fig. 1g). Most of the $KRAS$ mutant malignant cells clustered separately from other cells, which indicated the presence of distinct transcriptional programs in $KRAS$ mutant cells (Fig. 1f). In line with previous reports[15,17], malignant cells from KM-LUADs were chromosomally more stable than those from EM-LUADs ($P < 2.2 \times 10^{-16}$; Extended Data Fig. 1h, left). CNV burden was significantly higher in malignant cells from patients who smoke than in patients who never smoked ($P < 2.2 \times 10^{-16}$; Extended Data Fig. 1h, right). Differentiation states of malignant cells exhibited high inter-patient heterogeneity. That is, irrespective of tumour mutation load, KM-LUAD cells were the least differentiated, as indicated by their highest CytoTRACE[18] scores, followed by EM-LUADs ($P < 0.001$; Fig. 1g,h and Supplementary Table 4). There was intra-tumour heterogeneity (ITH) in differentiation states (for example, patients P2, P9, P14 and P15), whereby malignant cells from 7 out of the 14 patients with detectable malignant cells exhibited a broad distribution of CytoTRACE scores, with KM-LUADs showing a trend for higher variability in differentiation (greater Wasserstein distances) than EM-LUADs or other LUADs (Fig. 1h and Extended Data Fig. 1i).

Clustering of malignant cells (Meta C1 to Meta C5) based on levels of 23 recurrent meta-programs (MPs)[19] showed that Meta C1 comprised cells mostly from KM-LUADs (92%). Cells in Meta C1 also displayed the highest expression of gene modules associated with $KRAS^{G12D}$ present in pancreatic ductal adenocarcinoma (MP30)[19], epithelial-to-mesenchymal transition (EMT-III; MP14) and epithelial senescence (MP19), and, conversely, the lowest levels of alveolar MP (MP31) (Extended Data Fig. 2a–c and Supplementary Table 5). Notably, malignant cells from patients P2, P10 and P14 with KM-LUADs showed significantly higher expression of MP30 than those from patients with KW-LUADs ($P < 2.2 \times 10^{-16}$; Extended Data Fig. 2d). Malignant cell states also exhibited ITH in KM-LUADs (for example, patient P14; Extended Data Fig. 2e). A subset of $KRAS^{G12D}$ cells showed activation of MP30, and there were diverse activation patterns for other MPs (for example, cell respiration) across the mutant cells (Extended Data Fig. 2e, middle, 2f). Overall, malignant cells bearing $KRAS^{G12D}$ mutations showed reduced differentiation (Extended Data Fig. 2e, right), which was concordant with the loss of alveolar differentiation (MP31) in KM-LUADs (Extended Data Fig. 2a,b). Malignant cell clusters from patient P14 exhibited different levels of CNVs[15], whereby a cluster enriched in $KRAS^{G12D}$ cells harboured relatively late CNV events (for example, chromosome 1p loss, chromosome 8 and chromosome 12 gains) and reduced alveolar signature scores, a result in line with attenuated differentiation (Extended Data Fig. 2g,h). A KRAS signature was derived based on distinct expression features of $KRAS$ mutant malignant cells from our cohort (that is, specific to cluster 5; Extended Data Fig. 1g), which was strongly and significantly correlated with the MP30 signature ($R$ = 0.92, $P < 2.2 \times 10^{-16}$, Extended Data Fig. 2i and Supplementary Table 6). KM-LUADs from The Cancer Genome Atlas (TCGA) cohort and with relatively high expression of our KRAS signature were enriched with activated $KRAS$ MP30 and with other MPs that were increased in Meta C1 (Extended Data Fig. 2j). KW-LUADs in TCGA with a relatively higher expression of the KRAS signature displayed significantly lower overall survival (OS; $P$ = 0.02; Extended Data Fig. 2k). A similar trend was observed when analysing $KRAS^{G12D}$ mutant LUADs alone despite the small cohort size ($P$ = 0.3; Extended Data Fig. 2k). These data highlight the extensive transcriptomic heterogeneity between LUAD cells and transcriptional programs that are biologically and possibly clinically relevant to KM-LUAD.

## AICs in LUAD

In contrast to AT2 cells, which were overall decreased in LUADs compared with multi-region NL samples ($P$ = 0.002), AICs showed the opposite pattern ($P$ = 0.02; Extended Data Fig. 3a,b). AT2 cell

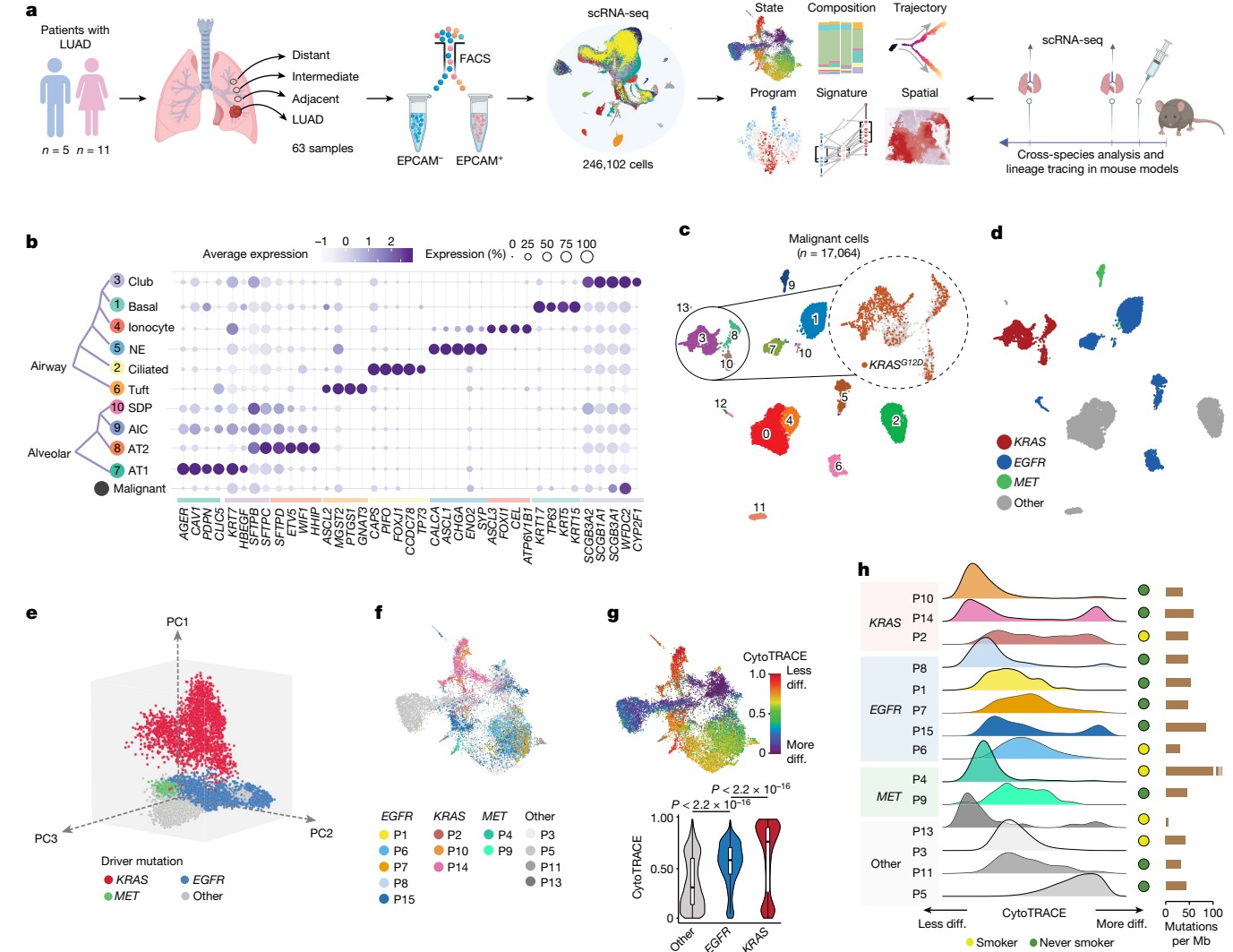

**Fig. 1 | Transcriptional landscape of lung epithelial and malignant cells in early-stage LUAD. a**, Schematic overview of the experimental design and analysis workflow. Composition, composition of cell subsets; Program, transcriptional programs in malignant cells; Spatial, in situ spatial transcriptome and protein analyses; State, cellular transcriptional state. **b**, Proportions and average expression levels (scaled) of selected marker genes for ten normal epithelial and one malignant cell subset. NE, neuroendocrine. **c**, Unsupervised clustering of 17,064 malignant cells coloured by cluster identity. Top right inset shows malignant cells coloured by $KRAS^{G12D}$ mutation status identified by scRNA-seq. **d**, Uniform manifold approximation and projection (UMAP) of malignant cells shown in **c** and coloured by driver mutations identified in each tumour sample using WES. **e**, Principal component analysis (PCA) plot of malignant cells coloured by driver mutations identified in each tumour sample by WES. **f**, UMAP plots of malignant cells coloured by patient identifier and grouped by driver mutation status. **g**, Top, UMAP of malignant cells by differentiation state inferred by CytoTRACE. Bottom, comparison of CytoTRACE scores between malignant cells from samples with different driver mutations. Boxes indicate the median ± interquartile range; whiskers, 1.5× the interquartile range; centre line, median. $n$ cells in each box-and-whisker (left to right): 9,135, 5,457 and 2,472. $P$ values were calculated using two-sided Wilcoxon rank-sum test with Benjamini–Hochberg correction. diff., differentiated. **h**, Per sample distribution of malignant cell CytoTRACE scores. The schematic in **a** was created using BioRender (https://www.biorender.com).

fractions were gradually reduced with increasing tumour proximity across multi-region NL samples from 7 out of the 16 patients with LUAD ($P$ = 0.004; Extended Data Fig. 3c,d). No significant changes in fractions were found for other major lung epithelial cell types (Extended Data Fig. 3e). AICs were intermediary along the AT2-to-AT1 cell developmental and differentiation trajectories (Fig. 2a and Extended Data Fig. 3f,g), a result reminiscent of intermediary alveolar cells in cancer-free mice exposed to acute lung injury[20]. The proportion of least-differentiated AICs in LUAD tissues was higher than that of their more differentiated counterparts (29% compared with 11%, respectively; Extended Data Fig. 3h). Notably, AICs were inferred to transition to malignant cells, including $KRAS$ mutant cells that were more developmentally late relative to $EGFR$ mutant malignant cells ($P < 2.2 \times 10^{-16}$; Fig. 2a and Extended Data Fig. 3f). Further analysis of AICs identified a subpopulation that

had a distinctly high expression of $KRT8$ (Fig. 2b). These KACs had increased expression of $CDKN1A$, $CDKN2A$, $PLAUR$ and the tumour marker $CLDN4$ (Fig. 2b, Extended Data Fig. 3i and Supplementary Table 7). KACs were also significantly less differentiated ($P < 2.2 \times 10^{-16}$; Fig. 2c) and more developmentally late ($P = 1.2 \times 10^{-11}$; Extended Data Fig. 3j) than other AICs. Notably, KACs transitioned to $KRAS$ mutant malignant cells in pseudotime, whereas other AICs were more closely associated with differentiation to AT1 cells (Extended Data Fig. 3j). Proportions of KACs among non-malignant epithelial cells were strongly and significantly increased in LUADs relative to multi-region NL tissues ($P = 2.4 \times 10^{-4}$; Fig. 2d), and were significantly higher in LUADs than in AT1, AT2 or other AIC fractions ($P < 2.2 \times 10^{-16}$; Fig. 2e). Notably, tumour-associated KACs clustered farther away from AICs compared with NL-derived KACs (Extended Data Fig. 3k).

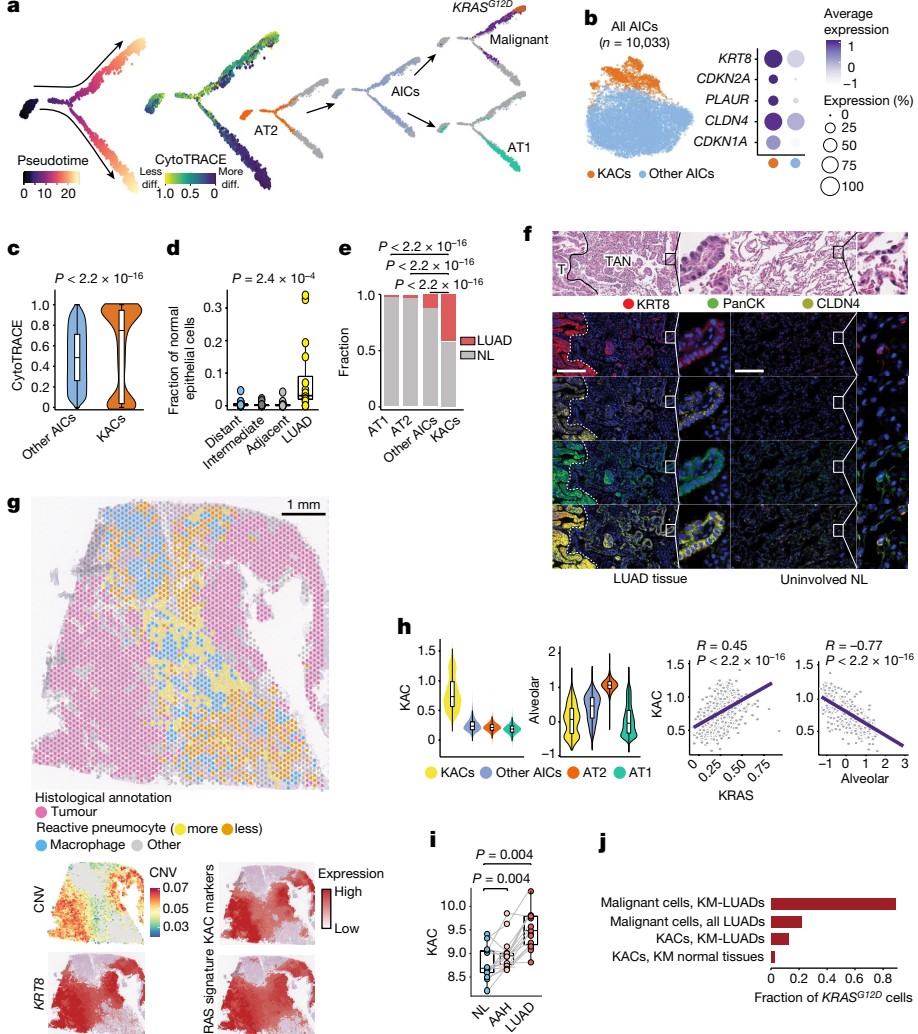

**Fig. 2 | Identification and characterization of KACs in human LUAD.**
**a**, Pseudotime analysis of alveolar and malignant cells. **b**, Left, subclustering analysis of AICs. Right, proportions and average expression levels (scaled) of representative KAC marker genes. **c**, CytoTRACE score in KACs versus other AICs. *n* cells (left to right): 8,591 and 1,440. *P* value was calculated using two-sided Wilcoxon rank-sum test. **d**, Proportion of KACs among non-malignant epithelial cells. *n* samples (left to right): 16, 15, 16 and 16. *P* value was calculated using Kruskal–Wallis test. **e**, Fraction of alveolar cell subsets coloured by sample type. *P* values were calculated using two-sided Fisher's exact tests with Benjamini–Hochberg correction. **f**, Top, haematoxylin and eosin (H&E) staining of LUAD tumour (T), TAN displaying reactive hyperplasia of AT2 cells and uninvolved NL tissue. Bottom, digital spatial profiling showing KRT8, PanCK, CLDN4, Syto13 blue nuclear stain and composite image. Magnification, ×20. Scale bar, 200 μm.

Staining was repeated four times with similar results. Dashed white lines represent the margins separating tumours and TAN regions. **g**, ST analysis of LUAD from patient P14 showing histologically annotated H&E-stained Visium slide (left) and spatial heatmaps (right) depicting CNV score and scaled expression of *KRT8*, KAC markers (**b**) and KRAS signature. **h**, Expression (top) and correlation (bottom) analyses of KAC, KRAS and alveolar signatures. *n* = 1,440 (KACs), 8,593 (other AICs), 146,776 (AT2) and 25,561 (AT1). *R*, Spearman's correlation coefficient. *P* values were calculated using Spearman's correlation test. **i**, KAC signature expression in premalignancy cohort (15 samples each). *P* values were calculated using two-sided Wilcoxon signed-rank test with Benjamini–Hochberg correction. **j**, Fraction of *KRAS*[G12D] cells in different subsets. For **c**,**d**,**h** and **i**, box-and-whisker definitions are the same as Fig. 1g.

High-resolution, multiplex imaging analysis of KRT8, CLDN4 and pan-cytokeratin (PanCK) showed that KACs were enriched in tumour-adjacent normal regions (TANs) and were found immediately next to malignant cells showing high expression of KRT8 and CLDN4 (Fig. 2f and Extended Data Fig. 4a). Although KACs were also found in the uninvolved NL samples, consistent with our scRNA-seq analysis, only in the TANs did they display features of 'reactive' epithelial cells (Fig. 2f and Extended Data Fig. 4a). ST analysis of tumour tissue from patient P14 demonstrated increased expression of *KRT8* in tumour regions (with high CNV scores) and in TAN regions that histologically comprised highly reactive pneumocytes and exhibited moderate-to-low CNV scores (Fig. 2g). Deconvolution showed that KACs were closer to tumour regions relative to alveolar cells (Extended Data Fig. 4b). ST analysis

of a KAC-enriched region showed that KACs were intermediary in the transition of alveolar parenchyma to tumour cells (Extended Data Fig. 4b). Tumour regions had markedly reduced expression of *NKX2-1* and the alveolar signature (Extended Data Fig. 4b), a result in line with reduced alveolar differentiation in KM-LUADs (Extended Data Fig. 2b).

KAC markers (Fig. 2b) were high in tumour regions and in TANs with reactive pneumocytes, and they spatially overlapped with the *KRAS* signature (Fig. 2g). Similar to KRAS, but unlike the AT1 and alveolar signatures, a KAC signature we derived was highest in KACs relative to AT1, AT2 or other AICs (Fig. 2h, Extended Data Fig. 4c,d and Supplementary Table 8). A signature pertinent to other AICs we derived was evidently lower in KACs relative to other AICs (Extended Data Fig. 4e). In KACs from all samples, KAC and KRAS signatures positively correlated

together ($R = 0.45$; $P < 2.2 \times 10^{-16}$) and inversely with their alveolar counterpart ($R = -0.77$; $P < 2.2 \times 10^{-16}$; Fig. 2h). By contrast, there was no correlation between 'other AIC' and KRAS ($R = 0.045$; $P = 3.2 \times 10^{-5}$) or alveolar ($R = -0.11$; $P < 2.2 \times 10^{-16}$) signatures (Extended Data Fig. 4f,g). The KAC signature was significantly higher in KACs and in malignant cells from KM-LUADs than those from EM-LUADs ($P < 2.2 \times 10^{-16}$; Extended Data Fig. 4h). In contrast to 'other AIC' and alveolar signatures, the KAC signature was significantly enriched in TCGA LUADs compared with their matched uninvolved NL samples ($P = 1.9 \times 10^{-15}$; Extended Data Fig. 5a–c). Of note, the KAC signature was significantly and progressively increased along matched NL, premalignant atypical adenomatous hyperplasia (AAH) and invasive LUAD (Fig. 2i), whereas there was no such pattern for the 'other AIC' signature (Extended Data Fig. 5d). The KAC signature was significantly higher in TCGA KM-LUADs than in KW-LUADs ($P = 0.002$; Extended Data Fig. 5e). Also, the KAC signature, but not the 'other AIC' signature, was significantly associated with reduced OS in two independent cohorts (TCGA, $P = 0.005$; PROSPECT, $P = 0.04$; Extended Data Fig. 5f–i). The KAC signature was associated with shortened OS even after accounting for disease stage (false discovery rate (FDR) adjusted $q$ value = 0.034; Extended Data Fig. 5j).

Despite exhibiting lower CNV scores than malignant cells, KACs exhibited moderately increased CNV burdens relative to AT2, AT1 and other AICs (Extended Data Fig. 6a,b). $KRAS^{G12D}$ was present in malignant cells with a variant allele frequency (VAF) of 78% in KM-LUADs (Fig. 2j, Extended Data Fig. 6c and Supplementary Table 9). KACs, but not AT2, AT1 or other AICs, harboured $KRAS^{G12D}$ mutations (Extended Data Fig. 6c,d). $KRAS^{G12D}$ KACs were exclusively found in tissues (primarily tumours) from KM-LUADs and, thus, $KRAS^{G12D}$ VAF (10%) was higher in KACs from KM-LUADs than in KACs from all examined LUADs (5%) or samples (3%) (Fig. 2j and Extended Data Fig. 6c,d). $KRAS^{G12D}$ mutations were detected in KACs of NL samples from patients with KM-LUAD (VAF of 2%). Meanwhile, other $KRAS$ variants ($KRAS^{G12C}$) were detected in NL of one patient with KM-LUAD, which indicated a potential field cancerization effect (Extended Data Fig. 6c,d). Concordantly, the KRAS signature was significantly increased in $KRAS^{G12D}$ KACs relative to $KRAS^{WT}$ counterparts ($P = 3.9 \times 10^{-3}$; Extended Data Fig. 6e). The KRAS signature was also increased in $KRAS^{WT}$ KACs relative to other AICs ($P < 2.2 \times 10^{-16}$) and in other AICs relative to AT2 cells ($P < 2.2 \times 10^{-16}$; Extended Data Fig. 6e). This result points towards increased KRAS signalling along the AT2–AIC–KAC spectrum. KACs from NL or tumours of KM-LUAD but not KW-LUAD cases were consistently and significantly less differentiated than other AICs (all $P < 2.2 \times 10^{-16}$, Extended Data Fig. 6f,g). Together, our findings characterize KACs as an intermediate alveolar cell subset that is highly relevant to the pathogenesis of human LUAD, especially KM-LUAD.

## A KAC state is linked to mouse KM-LUAD

We next performed scRNA-seq analysis of lung epithelial cells from mice in which the lung lineage-specific G protein-coupled receptor a gene, $Gprc5a$, is knocked out ($Gprc5a^{-/-}$)[21,22] and which develop KM-LUADs following tobacco carcinogen exposure. We analysed lungs from $Gprc5a^{-/-}$ mice treated with nicotine-derived nitrosamine ketone (NNK) or saline (as control) at the end of exposure (EOE) and at 7 months after exposure, the time point of KM-LUAD onset ($n = 4$ mice per group and time point; Fig. 3a and Supplementary Fig. 4). Clustering analysis of 9,272 high-quality epithelial cells revealed distinct lineages, including KACs that clustered between AT1 and AT2 cell subsets and close to tumour cells (Extended Data Fig. 7a). Similar to their human counterparts, malignant cells displayed low expression of lineage-specific genes (Extended Data Fig. 7b and Supplementary Table 10). Consistently, cells from the malignant cluster had high CNV scores, expressed $Kras^{G12D}$ mutations and showed increased expression of markers associated with loss of alveolar differentiation ($Kng2$ and $Meg3$) and immunosuppression ($Cd24a$)[23] (Extended Data Fig. 7c,d). Malignant cells were present only at 7 months after NNK treatment and were absent at EOE

to carcinogen and in saline-treated animals (Fig. 3b and Extended Data Fig. 7e,f). KAC fractions were markedly increased at EOE relative to control saline-treated littermates ($P = 0.03$), and they were, for the most part, maintained at 7 months after NNK treatment (Fig. 3b and Extended Data Fig. 7f,g). Immunofluorescence (IF) analysis showed that KRT8+ AT2-derived cells were present in NNK-exposed NL and were nearly absent in the lungs of saline-treated mice (Fig. 3c). LUADs also displayed high expression of KRT8 (Fig. 3c). KACs displayed a markedly increased prevalence of $Kras^{G12D}$ mutations, more so than CNV burden, and increased expression of genes (for example, $Gnk2$) associated with loss of alveolar differentiation[24], albeit to lesser extents compared with malignant cells (Fig. 3d, Extended Data Fig. 7h and Supplementary Table 11). Of note, AT2 cell fractions were reduced with time (Extended Data Fig. 7f,g). ST analysis at 7 months after NNK treatment showed that tumour regions had significantly increased expression of $Krt8$ and $Plaur$ and had spatially overlapping KAC and KRAS signatures (Fig. 3e and Extended Data Fig. 8a,c,e). In line with our human data, $Krt8^{high}$ KACs with increased expression of KAC and KRAS signatures were enriched in 'reactive', non-neoplastic regions surrounding tumours and were themselves intermediary in the transition from normal to tumour cells (Fig. 3e and Extended Data Fig. 8).

Mouse (Extended Data Fig. 9a) and human (Extended Data Fig. 9b) KACs displayed commonly increased activation of pathways, including NF-κB, hypoxia and p53 signalling, among others. A p53 signature we derived was significantly increased in KACs at EOE, and more so at 7 months after exposure to NNK, compared with both AT2 and tumour cells (Extended Data Fig. 9c, left). Similar patterns were noted for the expression of p53 pathway-related genes and senescence markers, including $Cdkn1a$, $Cdkn2b$ and $Bax$, as well as $Trp53$ itself (Extended Data Fig. 9c, right). Of note, activation of p53 has previously been reported in $Krt8^+$ transitional cells[25] during bleomycin-induced alveolar regeneration, and which themselves showed overlapping genes with KACs from our study (32%; Extended Data Fig. 9d). A mouse KAC signature we derived and that was significantly enriched in mouse KACs and malignant cells ($P < 2.2 \times 10^{-16}$, Extended Data Fig. 9e) and in human LUADs ($P = 1.2 \times 10^{-8}$, Extended Data Fig. 9f, left) was also significantly increased in premalignant AAHs ($P = 4.3 \times 10^{-4}$) and further increased in invasive LUADs ($P = 1.5 \times 10^{-3}$) relative to matched NL tissues (Extended Data Fig. 9f, right). Similar to alveolar intermediates in acute lung injury[25,26] and KACs in human LUADs (Fig. 2), mouse KACs were probably AT2 cell-derived, acted as intermediate states in AT2-to-AT1 cell differentiation and were inferred to transition to malignant cells (Fig. 4a, top row, Supplementary Fig. 5 and Supplementary Table 12). KACs assumed an intermediate differentiation state that more closely resembled malignant cells than other alveolar subsets (Fig. 4a, middle). The KAC signature was increased in cancer stem cell and stem cell-like progenitor cells that we had cultured from the MDA-F471 LUAD cell line (derived from a $Gprc5a^{-/-}$ mouse exposed to NNK[27]) relative to parental 2D cells (Extended Data Fig. 10a). KACs at EOE were less differentiated than those at 7 months after exposure (Fig. 4a, bottom right). Notably, the fraction of KACs with $Kras^{G12D}$ mutations was low at EOE (about 0.02) and was increased at 7 months after NNK (about 0.19) (Extended Data Fig. 10b). $Kras^{G12D}$ KACs from the late time point were significantly less differentiated ($P = 7.8 \times 10^{-6}$; Extended Data Fig. 10c) and showed higher expression of KAC signature genes such as $Cldn4$, $Krt8$, $Cavin3$ and $Cdkn2a$ than in $Kras^{WT}$ KACs (Extended Data Fig. 10d). Moreover, $Kras^{WT}$ KACs were more similar to previously reported $Krt8^+$ intermediate cells[25] than $Kras^{G12D}$ KACs (20% overlap compared with 10%, respectively; Extended Data Fig. 10e,f).

We performed integrated scRNA-seq analysis of cells from our mouse cohort with those in mice driven by $Kras^{G12D}$ from two separate studies[28,29]. Cluster C5 comprised cells from all three studies with distinctly high expression of KAC markers and the KAC signature itself (Extended Data Fig. 10g–i). The majority of C5 cells were from our study; however, C5 cells from $Kras^{G12D}$-driven mice still expressed higher levels of the

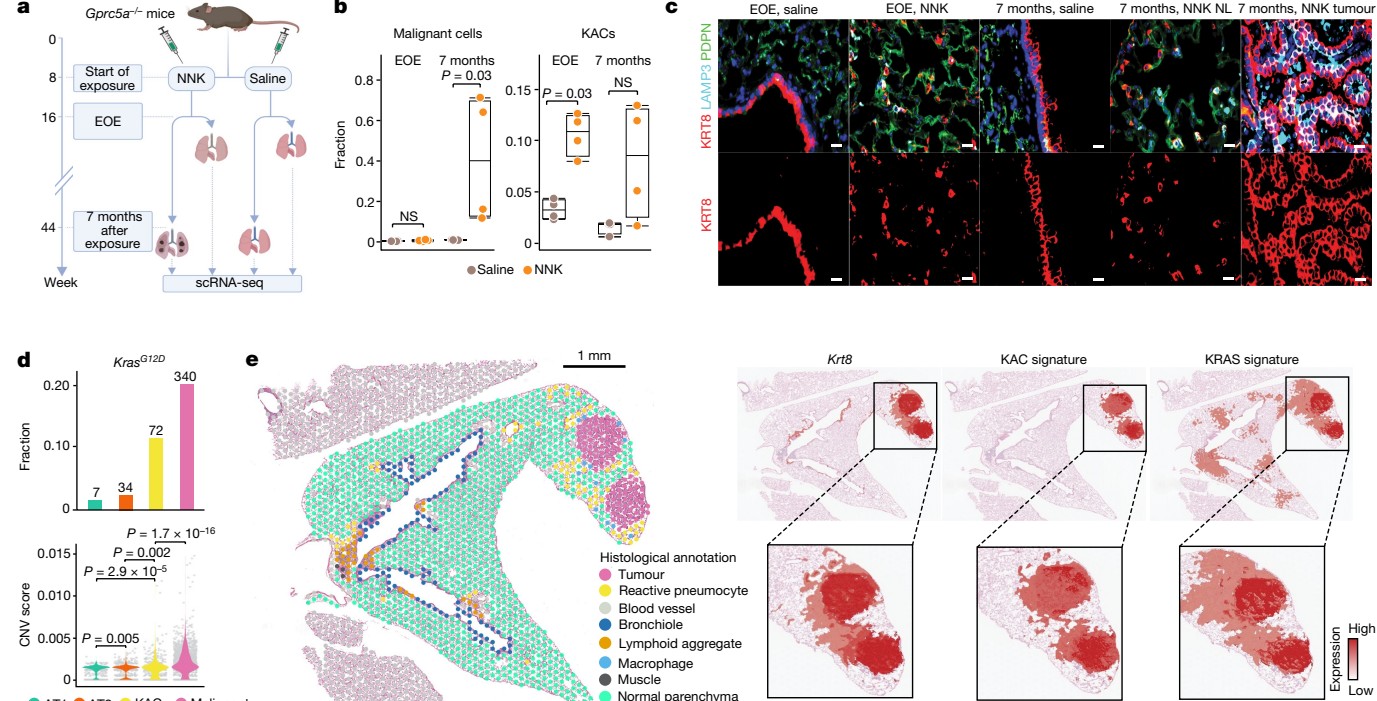

**Fig. 3 | KACs evolve early and before tumour onset during tobacco-associated KM-LUAD pathogenesis. a**, Schematic view of the in vivo experimental design. **b**, Fraction of malignant cells (left) and KACs (right) across treatment groups and time points. Box-and-whisker definitions are same as in Fig. 1g. $n = 4$ biologically independent samples per condition. $P$ values were calculated using two-sided Mann–Whitney $U$-test. NS, not significant. **c**, IF analysis of KRT8, LAMP3 and PDPN in mouse lung tissues. Scale bar, 10 µm. Results are representative of two independent biological replicates per treatment and timepoint. Staining was repeated three times with similar results. **d**, Top, distribution of CNV scores among alveolar and malignant cells. $n$ on top of each

bar denotes the numbers of $Kras^{G12D}$ mutant cells in each cell group. Bottom, fraction of $Kras^{G12D}$ mutant cells in KACs, malignant, AT1 and AT2 subsets. $n = 496$ (AT1), 1,320 (AT2), 512 (KACs) and 1,503 (malignant) cells. $P$ values were calculated using two-sided Mann–Whitney $U$-test with Benjamini–Hochberg correction. **e**, ST analysis of lung tissue at 7 months after exposure to NNK and showing histological annotation of H&E-stained Visium slide (left) and spatial heatmaps showing scaled expression of $KRT8$ as well as KAC and KRAS signatures. ST analysis was done on three different tumour-bearing mouse lung tissues from two mice at 7 months following NNK. The schematic in **a** was created using BioRender (https://www.biorender.com).

mouse KAC signature compared with normal AT2 cells from all studies (Extended Data Fig. 10j). The mouse KAC signature was markedly and significantly increased in human AT2 cells with induced expression of $KRAS^{G12D}$ relative to those with $KRAS^{WT}$ from ref. 29 ($P < 2.2 \times 10^{-16}$; Extended Data Fig. 10k). In agreement with these findings, the mouse KAC signature, like its human counterpart (Extended Data Fig. 4h), was significantly enriched in KACs and in malignant cells from KM-LUADs relative to EM-LUADs ($P = 0.04$ and $P < 2.2 \times 10^{-16}$, respectively; Extended Data Fig. 10l).

We further investigated the biology of KACs using $Gprc5a^{-/-}$ mice with reporter-labelled AT2 cells ($Gprc5a^{-/-};Sftpc^{creER/+};Rosa^{Sun1GFP/+}$; Fig. 4b). GFP+ organoids derived from NNK-exposed but not saline-exposed reporter mice at EOE were enriched in KACs (Extended Data Fig. 11a and Supplementary Fig. 6). GFP+ cells ($n = 3,089$) almost exclusively comprised AT2, early tumour and AT2-like tumour (early–AT2-like tumour) cells, KACs and KAC-like (KAC–KAC-like) cells and a few AT1 cells, all of which were nearly absent in the GFP− fraction (Extended Data Fig. 11b,c and Supplementary Fig. 7). There were markedly increased fractions of GFP+ AT1 cells, KACs and, as expected, tumour cells from NNK-treated mice compared with saline-treated mice (Fig. 4c). GFP expression was almost exclusive to alveolar regions and tumours, the latter of which were almost entirely GFP+ as well as KRT8+ and KAC marker-positive (CLDN4+CAVIN3+) (Supplementary Fig. 8a–c). NL regions included AT2 cell-derived KACs (GFP+KRT8+ and CLDN4+ or CAVIN3+) (Supplementary Fig. 8a–c). GFP+LAMP3+KRT8−/low AT2 cells were also evident, including in normal (non-tumoral) lung regions from NNK-exposed reporter mice (Supplementary Fig. 8d). GFP+ KACs from this time point, which coincides with the formation of preneoplasias[21],

harboured driver $Kras^{G12D}$ mutations at similar fractions when compared with early–AT2-like tumour cells (Extended Data Fig. 11d–f). As seen in $Gprc5a^{-/-}$ mice (Fig. 4a), KACs were closely associated with tumour cells in pseudotime (Extended Data Fig. 11g,h).

GFP+ organoids from reporter mice at 3 months after NNK treatment showed significantly and markedly enhanced growth compared with those from saline-exposed animals, and were almost exclusively composed of cells with KAC markers (KRT8+ and CLDN4+; Extended Data Fig. 12a,e). Given that KACs, like early tumour cells, acquired $Kras$ mutations, we examined the effects of targeted KRAS(G12D) inhibition on these organoids. We first tested effects of the KRAS(G12D) inhibitor MRTX1133 (ref. 30) in vitro and found that it inhibited the growth of mouse MDA-F471 cells and LKR13 cells (derived from $Kras^{LSL-G12D}$ mice[31]) in a dose-dependent manner (Extended Data Fig. 12b). This effect was accompanied by the suppression of phosphorylated levels of ERK1, ERK2 and S6 kinase in both cell lines (Extended Data Fig. 12c and Supplementary Fig. 9). Notably, MRTX1133-treated KAC marker-positive organoids showed significantly reduced sizes and KRT8 and CLDN4 expression intensities relative to DMSO-treated counterparts ($P < 1.5 \times 10^{-10}$; Extended Data Fig. 12d,e).

To further confirm that KACs give rise to tumour cells, we labelled KRT8+ cells in $Gprc5a^{-/-};Krt8\text{-}creER;Rosa^{tdT/+}$ mice. $Krt8\text{-}creER;Rosa^{tdT/+}$ mice were first used to confirm increased tdT+ labelling (that is, higher number of KACs) in the lung parenchyma at EOE to NNK compared with control saline-treated mice (Fig. 4d and Extended Data Fig. 13a). We then analysed lungs of NNK-exposed $Gprc5a^{-/-};Krt8\text{-}creER;Rosa^{tdT/+}$ mice that were injected with tamoxifen immediately after NNK treatment (Fig. 4d). Of note, most tumours showed tdT+KRT8+ cells at varying

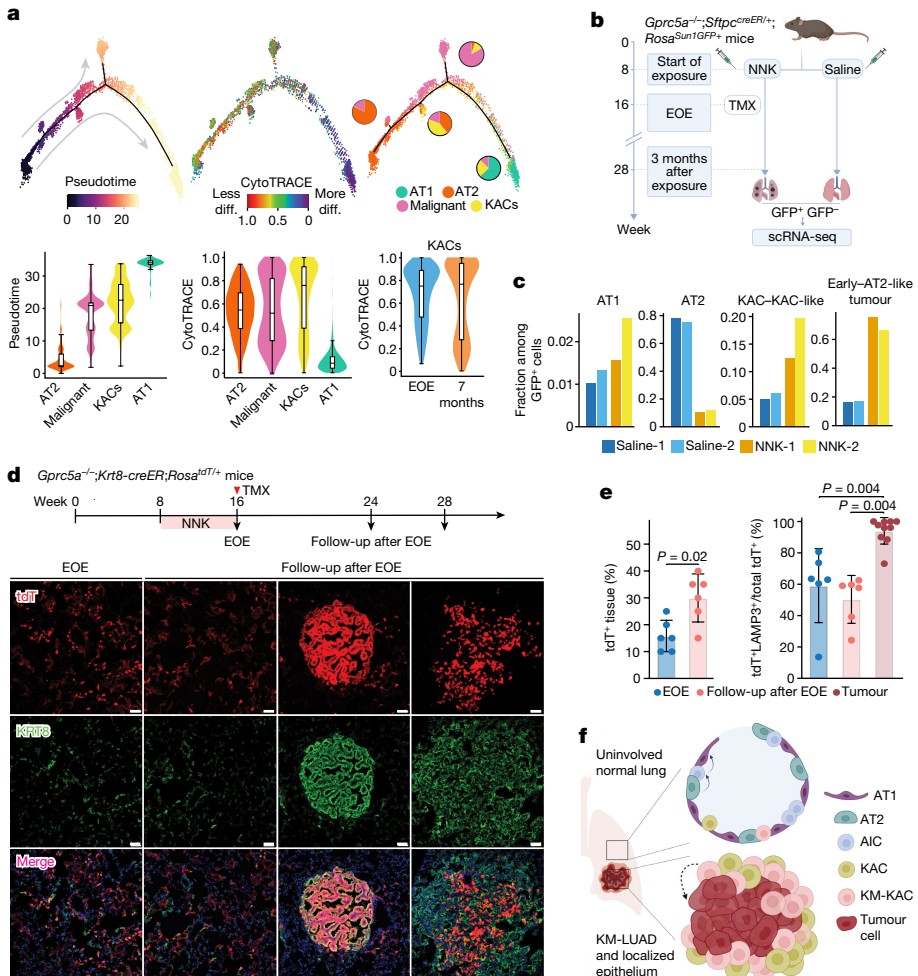

**Fig. 4 | KACs are implicated in the transition of AT2 to *Kras* mutant tumour cells. a**, Trajectories of alveolar and malignant cells coloured by inferred pseudotime, cell differentiation status and cell type (top left to right). Distribution of inferred pseudotime (bottom left) and CytoTRACE (bottom middle) scores across the indicated cell subsets. Bottom right panel shows CytoTRACE score distribution in KACs at the two time points. Box-and-whisker definitions are the same as in Fig. 1g. *n* cells (left to right): 1,791, 1,693, 636, 580, 1,791, 1,693, 636, 580, 301 and 335. **b**, Schematic overview showing analysis of *Gprc5a⁻/⁻* mice with reporter-labelled AT2 cells (*Gprc5a⁻/⁻;Sftpc^creER/+^;Rosa^Sun1GFP/+^*). TMX, tamoxifen. **c**, Fractions of AT1, AT2, KACs and KAC-like cells (KAC–KAC-like) and early tumour and AT2-like tumour cells (early–AT2-like tumour) within GFP⁺ cells from lungs of two NNK-treated and two saline-treated mice analysed at 3 months after exposure. **d**, IF analysis of tdT and KRT8 expression at EOE to NNK (first column; EOE) and at 8–12 weeks following NNK (follow-up after EOE) in normal-appearing regions (second column) and tumours (last two columns) of *Gprc5a⁻/⁻;Krt8-creER;Rosa^tdT/+^* mice. Tamoxifen (1 mg per dose) was delivered immediately after EOE to NNK for six continuous days. Results are representative of three biological replicates per condition. Staining was performed two times with similar results. Magnification, ×20. Scale bar, 10 μm. **e**, Left, percentage of lung tissue areas containing tdT⁺ cells. Right, percentage of tdT⁺LAMP3⁺ cells among total tdT⁺ cells in normal-appearing regions at different time points. Error bars show the mean ± s.d. of *n* biologically independent samples (left to right): 6, 6, 6, 6 and 10. *P* values were calculated using Mann–Whitney *U*-test. **f**, Proposed model for alveolar plasticity, whereby a subset of AICs in the intermediate AT2-to-AT1 differentiation state are KACs and, later, acquire *KRAS^G12D^* mutations and are implicated in KM-LUAD development from a particular region in the lung. The schematics in **b** and **f** were created using BioRender (https://www.biorender.com).

levels, with some tumours showing a strong extent of tdT labelling, which suggested oncogenesis of KRT8⁺ cells (Fig. 4d and Extended Data Fig. 13b,c). Most tdT⁺ tumour cells were AT2 cell-derived (LAMP3⁺) (Fig. 4e and Extended Data Fig. 13b). The fraction of tdT⁺LAMP3⁺ cells out of the total tdT⁺ cells was similar between EOE and follow-up after EOE to NNK (Fig. 4e). Normal-appearing regions also showed tdT⁺ AT1 cells (NKX2-1⁺LAMP3⁻), which indicated the possible turnover of AT2 cells and KACs to AT1 cells (Extended Data Fig. 13a). Taken together, our in vivo analyses identified KACs as an intermediate cell state in the early development of KM-LUAD and following tobacco carcinogen exposure.

## Discussion

Our multi-modal analysis of epithelial cells from early-stage LUADs and the peripheral lung uncovered diverse malignant states, patterns of ITH and cell plasticity programs that are linked to KM-LUAD pathogenesis. Of these, we identified alveolar intermediary cells (KACs) that arise after activation of alveolar differentiation programs and that could act as progenitors for KM-LUAD (Fig. 4f). KACs were evident in normal-appearing areas in the vicinity of lesions in both mouse and patient samples, which suggested that the early appearance of these cells (for example, following tobacco exposure) may represent a 'field of injury'[11]. A pervasive field of injury is relevant to the development of human lung cancer and to the complex spectrum of mutations present in normal-appearing lung tissue[32,33]. We propose that KACs represent injured or mutated cells in the normal-appearing lung that have an increased likelihood of transformation to lung tumour cells (Fig. 4f).

Our analysis uncovered strong links and intimately shared properties between KACs and *KRAS* mutant lung tumour cells, including *KRAS* mutations, reduced differentiation and pathways. Notably, we showed

that growth of KAC-rich and AT2 reporter-labelled organoids derived from lungs with early lesions was highly sensitive to KRAS(G12D) inhibition[34]. Although our in vivo findings are consistent with previous independent reports showing that AT2 cells are the preferential cells of origin in *Kras*-driven LUADs in animals[35–37], they enable a deeper scrutiny of the specific attributes and states of alveolar intermediary cells in the trajectory towards KM-LUADs.

Following acute lung injury, AT2 cells can differentiate into AICs that are characterized by high expression of *Krt8* and are crucial for AT1 regeneration[25,26,38]. We found evidence of KAC-like cells with notable expression of the KAC signature in *Kras*$^{G12D}$-driven mice, albeit at a reduced frequency compared with our tobacco-mediated carcinogenesis model. Thus, it is plausible that KACs can arise owing to an injury stimulus (here tobacco exposure) or mutant *Kras* expression or to both conditions. Our work raises questions that would be important to pursue in future studies. It is not clear whether KACs are a dominant or obligatory path in AT2-to-tumour transformation. Also, we do not know the effects of expressing mutant oncogenes, *Kras* or others, or tumour suppressors on the likelihood of KACs to divert away from mediating AT1 regeneration and, instead, transition to tumour cells. Recent studies suggest that p53 could curtail the oncogenesis of alveolar intermediate cells[39].

Combining in-depth interrogation of early-stage human LUADs and *Kras* mutant lung carcinogenesis models, our study provided an atlas with an expansive number of epithelial cells. This atlas of epithelial and malignant cell states in human and mouse lungs underscores new cell-specific subsets that underlie inception of LUADs. Our discoveries may inspire the derivation of targets (for example, KAC signals such as early KRAS programs) to prevent the initiation and development of LUAD.

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

## Methods

### Multi-regional sampling of human surgically resected LUADs and NL tissues

Study participants were evaluated at the MD Anderson Cancer Center and underwent standard-of-care surgical resection of early-stage LUAD (I–IIIA). Samples from all patients were obtained from banked or residual tissues under informed consent and approved by MD Anderson institutional review board protocols. Residual surgical specimens were then used for derivation of multi-regional samples for single-cell analysis (Supplementary Table 1). Immediately following surgery, resected tissues were processed by an experienced pathologist assistant. One side of the specimen was documented and measured, followed by tumour margin identification. Based on the placement of the tumour within the specimen, incisions were made at defined collection sites in one direction along the length of the specimen and spanning the entire lobe: tumour-adjacent and tumour-distant normal parenchyma at 0.5 cm from the tumour edge and from the periphery of the overall specimen or lobe, respectively. An additional tumour-intermediate normal tissue sample was selected for patients P2–P16 and ranged between 3 and 5 cm from the edge of the tumour. Sample collection was initiated with NL tissues that are farthest from the tumour moving inward towards the tumour to minimize cross-contamination during collection.

### Single-cell isolation from tissue samples

Fresh tissues from human donors and mouse lungs were collected in RPMI medium supplemented with 2% FBS and maintained on ice for immediate processing. Tissues were placed in a cell culture dish containing Hank's balanced salt solution (HBSS) on ice, and extra-pulmonary airways and connective tissue were removed with scissors. Samples were transferred to a new dish on ice and minced into about 1 mm³ pieces followed by enzymatic digestion. For human tissues, the enzymatic solution was composed of collagenase A (10103578001, Sigma Aldrich), collagenase IV (NC9836075, Thermo Fisher Scientific), DNase I (11284932001, Sigma Aldrich), dispase II (4942078001, Sigma Aldrich), elastase (NC9301601, Thermo Fisher Scientific) and pronase (10165921001, Sigma Aldrich) as previously described[40]. For mouse lung digestion, the enzymatic solution was composed of collagenase type I (CLS-1 LS004197, Worthington), elastase (ESL LS002294, Worthington) and DNase I (D LS002007, Worthington). Samples were transferred to 5 ml LoBind Eppendorf tubes and incubated in a 37 °C oven for 20 min with gentle rotation. Samples were then filtered through 70 µm strainers (Miltenyi Biotech, 130-098-462) and washed with ice-cold HBSS. Filtrates were then centrifuged and resuspended in ice-cold ACK lysis buffer (A1049201, Thermo Fisher Scientific) for red blood cell lysis. Following red blood cell lysis, samples were centrifuged and resuspended in ice-cold FBS, filtered (using 40 µm FlowMi tip filters; H13680-0040, Millipore) and an aliquot was taken to count cells and check for viability by Trypan blue (T8154, Sigma Aldrich) exclusion analysis.

### Sorting and enrichment of viable lung epithelial singlets

Single cells from patient P1 were stained with Sytox Blue viability dye (S34857, Life Technologies) and processed on a FACS Aria I instrument. Cells from P2–P16 were stained with anti-EPCAM-PE (347198, BD Biosciences; 1:50 dilution in ice-cold PBS containing 2% FBS) for 30 min with gentle rotation at 4 °C. Mouse lung single cells were similarly stained but with a cocktail of antibodies (1:250 each) against CD45-PE/Cy7 (103114, BioLegend), ICAM2-A647 (A15452, Life Technologies), EPCAM-BV421 (118225, BioLegend) and ECAD-A488 (53-3249-80, eBioscience). Stained cells were then washed, filtered using 40 µm filters, stained with Sytox Blue (human) or Sytox Green (mouse) and processed on a FACS Aria I instrument (gating strategies for epithelial cell sorting are shown in Supplementary Figs. 1 and 4 for human and mouse cells, respectively). Doublets and dead cells were eliminated, and viable (Sytox-negative) epithelial singlets were collected in PBS containing 2% FBS. Cells were washed again to eliminate ambient RNA, and a sample was taken for counting by Trypan Blue exclusion before loading on 10X Genomics Chromium microfluidic chips.

### Preparation of single-cell 5′ gene expression libraries

Up to 10,000 cells per sample were partitioned into nanolitre-scale Gel beads-in-emulsion (GEMs) using a Chromium Next GEM Single Cell 5′ Gel Bead kit v.1.1 (1000169, 10X Genomics) and by loading onto Chromium Next GEM Chips G (1000127, 10X Genomics). GEMs were then recovered to construct single-cell gene expression libraries using a Chromium Next GEM Single Cell 5′ Library kit (1000166, 10X Genomics) according to the manufacturer's protocol. In brief, recovered barcoded GEMs were broken and pooled, followed by magnetic bead clean-up (Dynabeads MyOne Silane, 37002D, Thermo Fisher Scientific). 10X-barcoded full-length cDNA was then amplified by PCR and analysed using a Bioanalyzer High Sensitivity DNA kit (5067-4626, Agilent). Up to 50 ng of cDNA was carried over to construct gene expression libraries and was enzymatically fragmented and size-selected to optimize the cDNA amplicon size before 5′ gene expression library construction. Samples were then subjected to end-repair, A-tailing, adaptor ligation and sample index PCR using Single Index kit T Set A (2000240, 10X Genomics) to generate Illumina-ready barcoded gene expression libraries. Library quality and yield were measured using a Bioanalyzer High Sensitivity DNA kit (5067-4626, Agilent) and a Qubit dsDNA High Sensitivity Assay kit (Q32854, Thermo Fisher Scientific). Indexed libraries were normalized by adjusting for the ratio of the targeted cells per library as well as individual library concentration and then pooled to a final concentration of 10 nM. Library pools were then denatured and diluted as recommended for sequencing on an Illumina NovaSeq 6000 platform.

### scRNA-seq data processing and quality control

Raw scRNA-seq data were pre-processed (demultiplex cellular barcodes, read alignment and generation of gene count matrix) using Cell Ranger Single Cell Software Suite (v.3.0.1) provided by 10X Genomics. For read alignment of human and mouse scRNA-seq data, human reference GRCh38 (hg38) and mouse reference GRCm38 (mm10) genomes were used, respectively. Detailed quality control metrics were generated and evaluated, and cells were carefully and rigorously filtered to obtain high-quality data for downstream analyses[15]. In brief, for basic quality filtering, cells with low-complexity libraries (in which detected transcripts were aligned to <200 genes such as cell debris, empty drops and low-quality cells) were filtered out and excluded from subsequent analyses. Probable dying or apoptotic cells in which >15% of transcripts derived from the mitochondrial genome were also excluded. For scRNA-seq analysis of $Gprc5a^{-/-}$;$Sftpc^{creER/+}$;$Rosa^{Sun1GFP/+}$ mice, cells with ≤500 detected genes or with a mitochondrial gene fraction that is ≥15% were filtered out using Seurat[41].

### Doublet detection and removal, and batch effect evaluation and correction

Probable doublets or multiplets were identified and carefully removed through a multi-step approach as described in previous studies[15,42]. In brief, doublets or multiplets were identified based on library complexity, whereby cells with high-complexity libraries in which detected transcripts are aligned to >6,500 genes were removed and, based on cluster distribution and marker gene expression, whereby doublets or multiplets forming distinct clusters with hybrid expression features and/or exhibiting an aberrantly high gene count were also removed. Expression levels and proportions of canonical lineage-related marker genes in each identified cluster were carefully reviewed. Clusters co-expressing discrepant lineage markers were identified and removed. Doublets or multiplets were also identified using the doublet detection algorithm DoubletFinder[43]. The proportion of expected doublets was estimated based on cell counts obtained before scRNA-seq library

construction. Data normalization was then performed using Seurat[41] on the filtered gene–cell matrix. Statistical assessment of possible batch effects was performed on non-malignant epithelial cells using the R package ROGUE[36], an entropy-based statistic, as described in previous studies[15,42] and Harmony[44] was run with default parameters to remove batch effects present in the PCA space.

## Unsupervised clustering and subclustering analysis

The function FindVariableFeatures of Seurat[41] was applied to identify highly variable genes for unsupervised cell clustering. PCA was performed on the top 2,000 highly variable genes. The elbow plot was generated with the ElbowPlot function of Seurat and, based on which, the number of significant principal components (PCs) was determined. The FindNeighbors function of Seurat was used to construct the shared nearest neighbour (SNN) graph based on unsupervised clustering performed using the Seurat function FindClusters. Multiple rounds of clustering and subclustering analyses were performed to identify major epithelial cell types and distinct cell transcriptional states. Dimensionality reduction and 2D visualization of cell clusters was performed using UMAP[45] and the Seurat function RunUMAP. The number of PCs used to calculate the embedding was the same as that used for clustering. For analysis of human epithelial cells, ROGUE was used to quantify cellular transcriptional heterogeneity of each cluster. Subclustering analysis was then performed for low-purity clusters identified by ROGUE. Hierarchical clustering of major epithelial subsets was performed on the Harmony batch-corrected PCA dimension reduction space. For malignant cells, except for global UMAP visualization, downstream analyses, including identification of large-scale CNVs, inference of cancer cell differentiation states, quantification of meta-program expression, trajectory analysis and mutation analysis, were performed without Harmony batch correction. The hierarchical tree of human epithelial cell lineages was computed based on Euclidean distance using the Ward linkage method, and the dendrogram was generated using the R function plot.hc. For scRNA-seq analysis of *Gprc5a*$^{-/-}$ mice, the top-ranked ten PCs were selected using the elbow-plot function. SNN graph construction was performed with resolution parameter = 0.4, and UMAP visualization was performed with default parameters. For scRNA-seq analysis of *Gprc5a*$^{-/-}$;*Sftpc*$^{creER/+}$;*Rosa*$^{Sun1GFP/+}$ mice, the top-ranked 20 Harmony-corrected PCs were used for SNN graph construction, and unsupervised clustering was performed with resolution parameter = 0.4. UMAP visualization was performed with the RunUMAP function with min.dist = 0.1. Differentially expressed genes (DEGs) of clusters were identified using the FindAllMarkers function with FDR-adjusted *P* value < 0.05 and log$_2$(fold change) > 1.2.

## Identification of malignant cells and mapping *KRAS* codon 12 mutations

Malignant cells were distinguished from non-malignant subsets based on information integrated from multiple sources as described in previous studies[15,42]. The following strategies were used to identify malignant cells. (1) Cluster distribution: owing to the high degree of inter-patient tumour heterogeneity, malignant cells often exhibit distinct cluster distribution compared with normal epithelial cells. Although non-malignant cells derived from different patients are often clustered together by cell type, malignant cells from different patients probably form separate clusters. (2) CNVs: we applied inferCNV[16] (v.1.3.2) to infer large-scale CNVs in each individual cell with T cells as the reference control. To quantify CNVs at the cell level, CNV scores were aggregated using a previously described strategy[16]. In brief, arm-level CNV scores were computed based on the mean of the squares of CNV values across each chromosomal arm. Arm-level CNV scores were further aggregated across all chromosomal arms by calculating the arithmetic mean value of the arm-level scores using the R function mean. (3) Marker gene expression: expression of lung epithelial lineage-specific genes and LUAD-related oncogenes was determined in

epithelial cell clusters. (4) Cell-level expression of *KRAS*$^{G12D}$ mutations: as we previously described[15], BAM files were queried for *KRAS*$^{G12D}$ mutant alleles, which were then mapped to specific cells. *KRAS*$^{G12D}$ mutations, along with cluster distribution, marker gene expression and inferred CNVs as described above, were used to distinguish malignant cells from non-malignant cells. Following clustering of malignant cells from all patients, an absence of malignant cells that were identified from P12 or P16 was noted. This can be possibly attributed to the low number of epithelial cells captured in tumour samples from these patients (Supplementary Table 2).

**Mapping *KRAS* codon 12 mutations.** To map somatic *KRAS* mutations at single-cell resolution, alignment records were extracted from the corresponding BAM files using mutation location information. Unique mapping alignments (MAPQ = 255) labelled as either PCR duplication or secondary mapping were filtered out. The resulting somatic variant carrying reads were evaluated using Integrative Genomics Viewer (IGV)[46] and the CB tags were used to identify cell identities of mutation-carrying reads. To estimate the VAF of *KRAS*$^{G12D}$ mutation and cell fraction of *KRAS*$^{G12D}$-carrying cells within malignant and non-malignant epithelial cell subpopulations (for example, malignant cells from all LUADs, malignant cells from KM-LUADs, KACs from KM-LUADs), reads were first extracted based on their unique cell barcodes and BAM files were generated for each subpopulation using samtools (v.1.15). Mutations were then visualized using IGV, and VAFs were calculated by dividing the number of *KRAS*$^{G12D}$-carrying reads by the total number of uniquely aligned reads for each subpopulation. A similar approach was used to visualize *KRAS*$^{G12C}$-carrying reads and to calculate the VAF of *KRAS*$^{G12C}$ in KACs of normal tissues from KM-LUAD cases. To calculate the mutation-carrying cell fraction, extracted reads were mapped to the *KRAS*$^{G12D}$ locus from BAM files using AlignmentFile and fetch functions in pysam package. Extracted reads were further filtered using the 'Duplicate' and 'Quality' tags to remove PCR duplicates and low-quality mappings. The number of reads with or without *KRAS*$^{G12D}$ mutation in each cell was summarized using the CB tag in read barcodes. Mutation-carrying cell fractions were then calculated as the ratio of the number of cells with at least one *KRAS*$^{G12D}$ read over the number of cells with at least one high-quality read mapped to the locus.

## PCA analysis of malignant cells and quantification of transcriptome similarity

Raw unique molecular identifier counts of identified malignant cells were log-normalized and used for PCA analysis using Seurat (RunPCA function). PCA dimension reduction data were extracted using the Embeddings function. The top three most highly ranked PCs were exported for visualization using JMP (v.15). 3D scatterplots of PCA data were generated using the scatterplot 3D tool in JMP (v.15). Bhattacharyya distances were calculated using the bhattacharyya.dist function from the R package fpc (v.2.2-9). The top 25 highly ranked PCs were used for both patient-level and cell-level distance calculations. For Bhattacharyya distance quantification at the cell level, 100 cells were randomly sampled for each patient group defined by driver mutations (for example, KM-LUADs). The random sampling process was repeated 100 times, and pairwise Bhattacharyya distances were then calculated between patient groups. Differences in Bhattacharyya distances between patient groups were tested using Wilcoxon rank-sum tests, and boxplots were generated using the geom_boxplot function from the R package ggplot2 (v.3.2.0).

## Determination of non-malignant cell types and states

Non-malignant cell types and states were determined based on unsupervised clustering analysis following batch effect correction using Harmony[44]. Two rounds of clustering analysis were performed on non-malignant cells to identify major cell types and cell transcriptional states within major cell types. Clustering and UMAP visualization of

human normal epithelial cells (Extended Data Fig. 1a) were performed using Seurat with default parameters. Specifically, the parameters k.param = 20 and resolution = 0.4 were used for SNN graph construction and cluster identification, respectively. UMAP visualization was performed with default parameters (min.dist = 0.3). For clustering analysis of airway and alveolar epithelial cells, the RunPCA function was used to determine the most contributing top PCs for each subpopulation and similar clustering parameters (k.param = 20 and resolution = 0.4) were used for SNN graph construction and cluster identification. UMAP plots were generated with min.dist = 0.3 using the RunUMAP function in Seurat. Density plots of alveolar intermediate cells were generated using the stat_densit_2d function in the R package ggplot2 (v.3.3.5) with the first two UMAP dimension reduction data as the input. DEGs for each cluster were identified using the FindMarkers function in Seurat with a FDR-adjusted $P < 0.05$ and a fold change cut-off > 1.2. Canonical epithelial marker genes from previously published studies by our group and others[15,47,48] were used to annotate normal epithelial cell types and states. Bubble plots were generated for select DEGs and canonical markers to define AT1 cells ($AGER1^+ETVS^+PDPN^+$), AT2 cells ($SFTPB^+SFTPC^+ETVS^+$), $SCGB1A1^+SFTPC^+$ dual-positive cells, AICs ($AGER1^+ETVS^+PDPN^+$ and $SFTPB^+SFTPC^+$), club and secretory cells ($SCGB1A1^+SCGB3A1^+CYP2F1^+$), basal cells ($KRT5^+TP63^+$), ciliated cells ($CAPS^+PIFO^+FOXJ1^+$), ionocytes ($ASCL3^+FOXI^+$), neuroendocrine cells ($CALCA^+ASCL1^+$) and tuft cells ($ASCL2^+MGST2^+PTGS1^+$). KACs were identified by unsupervised clustering of AICs and defined based on previously reported marker genes[25,26,49], including significant upregulation of the following genes relative to other alveolar cells: $KRT8$, $CLDN4$, $PLAUR$, $CDKN1A$ and $CDKN2A$.

## Scoring of curated gene signatures

Genes in previously defined ITH MPs[19] were downloaded from the original study. Among a total of 41 consensus ITH MPs identified, MPs with unassigned functional annotations (unassigned MPs 38–41; $n = 4$), neural and haematopoietic lineage-specific MPs (MPs 25–29, MPs 33–37; $n = 10$) and cell-type-specific MPs irrelevant to LUAD (MPs 22–24 secreted/cilia, MP 32 skin-pigmentation; $n = 4$) were filtered out, resulting in 23 MPs that closely correlated with hallmarks of cancer and that were used for further analysis. Signature scores were computed using the AddModuleScore function in Seurat as previously described[15,42]. The KRAS signature used in this study was derived by calculating DEGs between the $KRAS$ mutant malignant-cell-enriched cluster and other malignant cells (FDR-adjusted $P$ value < 0.05, log(fold change) > 1.2; Extended Data Fig. 2i). Human and mouse KAC signatures and the human 'other AIC' signature were derived by calculating DEGs using FindAllMarkers among alveolar cells (FDR-adjusted $P$ value < 0.05, log(fold change) > 1.2). Mouse genes in the p53 pathway were downloaded from the Molecular Signature Database (MSigDB; https://www.gsea-msigdb.org/gsea/msigdb/mouse/geneset/HALLMARK_P53_PATHWAY; MM3896). Signature scores for KACs, other AICs, KRAS and p53 were calculated using the AddModuleScore function in Seurat.

## Analysis of alveolar cell differentiation states and trajectories

Analysis of differentiation trajectories of lung alveolar and malignant cells was performed using Monocle 2 (ref. 50) by inferring the pseudotemporal ordering of cells according to their transcriptome similarity. Monocle 2 analysis of malignant cells from P14 was performed using default parameters with the detectGenes function. Detected genes were further required to be expressed by at least 50 cells. For construction of the differentiation trajectory of lineage-labelled epithelial cells (GFP+), the top 150 DEGs (FDR-adjusted $P$ value < 0.05, log(fold change) > 1.5, expressed in ≥50 cells) ranked by fold-change of each cell population from NNK-treated samples were used for ordering cells with the setOrderingFilter function. Trajectories were generated using the reduceDimension function with the method set to 'DDRTree'. Trajectory roots were selected based on the following

criteria: (1) inferred pseudotemporal gradient; (2) CytoTRACE score prediction; and (3) careful manual review of the DEGs along the trajectory. To depict expression dynamics of ITH MPs[19], ITH MP scores were plotted along the pseudotime axis and smoothed lines were generated using the smoother tool in JMP Pro (v.15). Using the raw counts without normalization as input, CytoTRACE[18] was applied with default parameters to infer cellular differentiation states to complement trajectory analysis and further understand cellular differentiation hierarchies. The normalmixEM function from the R package mixtools was used to determine the CytoTRACE score threshold in AICs with $k = 2$. A final threshold of 0.58 was used to dichotomize AICs into high-differentiation and low-differentiation groups. The Wasserstein distance metric was applied using R package transport (v.0.13) to quantify the variability of distribution of CytoTRACE scores. The function wasserstein1d was used to calculate the distance between the distribution of actual CytoTRACE scores of one patient and the distribution of simulated data with identical mean and standard deviation. The robustness of Monocle 2-based pseudotemporal ordering prediction was validated by independent pseudotime prediction tools including Palantir[51], Slingshot[52] and Cellrank[53]. Slingshot (v.2.6.0) pseudotime prediction was performed using slingshot function with reduceDim parameter set to 'PCA' and other parameters set to defaults. Cellrank prediction was performed using the CytoTRACEKernel function with default parameters from Cellrank python package (v.1.5.1). Palantir prediction was performed using Palantir python package (v.1.0.1). A diffusion map was generated using run_diffusion_maps function with n_components = 5. Palantir prediction was generated using run_palantir function with num_waypoints = 500 and other parameters set to defaults. Inferred pseudotime by the three independent methods was then integrated with that generated by Monocle 2 for each single cell, followed by pairwise mapping and correlation analysis. Cell density plots were generated using Contour tool in JMP (v.15) with $n = 10$ gradient levels and contour type parameter set to 'Nonpar Density'. To assess the pseudotime prediction consistency between Monocle 2 and the three independent methods, Spearman's correlation coefficients were calculated and statistically tested using cor.test function in R.

## ST data generation and analysis

ST profiling of formalin fixation and paraffin-embedding (FFPE) tissues from P14 with LUAD and of three lung tissues from two $Gprc5a^{-/-}$ mice was performed using the Visium platform from 10X Genomics according to the manufacturer's recommendations and as previously reported[54]. P14 FFPE tissues were collected from areas adjacent to the tissues analysed by scRNA-seq. Regions of interest per tissue or sample, each comprising a 6.5 × 6.5 mm capture area, were selected based on careful annotation of H&E-stained slides that were digitally acquired using an Aperio ScanScope Turbo slide scanner (Leica Microsystems). HALO software (Indica Labs) was used for pathological annotation (tumour areas, blood vessels, bronchioles, lymphoid cell aggregates, macrophages, muscle tissue, normal parenchyma and reactive pneumocytes) of H&E histology images. Spot-level histopathological annotation and visualization was generated using loupe browser (v.6.3.0). In brief, cloupe files generated from Space Ranger were loaded into the loupe browser. Visualization of annotation was then generated in svg formats using the export plot tool. ST RNA-seq libraries were generated according to the manufacturer's instructions, each with up to about 3,600 uniquely barcoded spots. Libraries were sequenced on an Illumina NovaSeq 6000 platform to achieve a depth of at least 50,000 mean read pairs per spot and at least 2,000 median genes per spot.

Demultiplexed raw sequencing data were aligned, and gene level expression quantification was generated with analysis pipelines as previously described[54]. In brief, demultiplexed clean reads were aligned against the UCSC human GRCh38 (hg38) or the GRCm38 (mm10) mouse reference genomes by Spaceranger (v.1.3.0 for human ST data and v.2.0.0 for mouse ST data) and using default settings. Generated

ST gene expression count matrices were then analysed using Seurat (v.4.1.0) to perform unsupervised clustering analysis. Using default parameters, the top-ranked 30 PCA components were used for SNN graph construction and clustering and for UMAP low-dimension space embedding with default parameters. UMAP analysis was performed using the RunUMAP function. The SpatialDimPlot function was used to visualize unsupervised clustering. The R package inferCNV[16] was used for copy number analysis. Reference spots used in CNV analysis were selected on the basis of careful review of cluster marker genes using the DotPlot function from Seurat and inspection of pathological annotation. CNV scores were calculated by computing the standard deviations of CNVs inferred across 22 autosomes. Loupe browser (v.6.3.0) was used for visualization of pathological annotation results. Expression levels of genes of interest (for example, *KRT8*) as well as signatures of interest (for example, KAC and KRAS) were measured and directly annotated on histology images with pixel-level resolution using the TESLA (v.1.2.2) machine learning framework[55] (https://github.com/jianhuupenn/TESLA). TESLA can compute superpixel-level gene expression and detect unique structures within and surrounding tumours by integrating information from high-resolution histology images. The annotation and visualize_annotation functions were used to annotate regions with high signature signals. *KRT8*, *PLAUR*, *CLDN4*, *CDKN1A* and *CDKN2A* were used for 'KAC markers' signature annotation in the human ST analysis. For mouse ST data, *Krt8*, *Plaur*, *Cldn4*, *Cdkn1a* and *Cdkn2a* were used for 'KAC signature' annotation. Gene level expression visualization of *Krt8* and *Plaur* was generated using the scatter function from scanpy (v.1.9.1). Deconvolution analysis was conducted using CytoSPACE[56] (https://github.com/digitalcytometry/cytospace). Annotated scRNA-seq data were first transformed into a compatible format using function generate_cytospace_from_scRNA_seurat_object. Visium spatial data were prepared using the function generate_cytospace_from_ST_seurat_object. Deconvolution was performed using CytoSpace function (v.1.0.4) with default parameters. To determine neighbouring cell composition for a specific cell population in Visium data, CytoSPACE was first applied to annotate every spot with the most probable cell type. Neighbouring spots were defined as the six spots surrounding each spot and, accordingly, the neighbouring cell composition for specific cell types were computed. Trajectory construction of ST data was performed using Monocle 2 (ref. 18) with the DDRTree method using DEGs with FDR-adjusted *P* value < 0.05.

## Bulk DNA extraction and WES

Total DNA was isolated from homogenized cryosections of human lung tissues and, when available, from frozen peripheral blood mononuclear cells (PBMCs) using a Qiagen AllPrep mini kit (80204) or a DNeasy Blood and Tissue kit (69504), respectively (both from Qiagen) according to the manufacturer's recommendations. Qubit 4 Fluorometer (Thermo Fisher Scientific) was used for measurement of DNA yield. TWIST-WES was performed on a NovaSeq 6000 platform at a depth of 200× for tumour samples and 100× for NL and PBMCs to analyse recurrent driver mutations and using either PBMCs or distant NL tissues when blood draw was not consented, as germline control. WES data were processed and mapped to the human reference genome, and somatic mutations were identified and annotated as previously described[57,58] with further filtration steps. In brief, only MuTect[59] calls marked as 'KEEP' were selected and taken into the next step. Mutations with a low VAF (<0.02) or low alt allele read coverage (<4) were removed. Then, common variants reported by ExAc (the Exome Aggregation Consortium, http://exac.broadinstitute.org), Phase-3 1000 Genome Project (http://phase3browser.1000genomes.org/Homo_sapiens/Info/Index) or the NHLBI GO Exome Sequencing Project (ESP6500) (http://evs.gs.washington.edu/EVS/) with minor allele frequencies greater than 0.5% were further removed. Intronic mutations, mutations at 3′ or 5′ UTR or UTR-flanking regions, and silent mutations were also removed. The mutation load in each tumour was calculated as the number of nonsynonymous somatic mutations (nonsense, missense, splicing, stop gain, stop loss substitutions as well as frameshift insertions and deletions).

## Survival analysis

Analysis of OS in the TCGA LUAD and PROSPECT[60] cohorts was performed as previously described[15]. *KRAS* mutation status in TCGA LUAD samples was downloaded from cBioPortal (https://www.cbioportal.org, study ID: luad_tcga_pan_can_atlas_2018). For TCGA dataset, clinical data were downloaded from the PanCanAtlas study[18]. The logrank test and Kaplan–Meier methods were used to calculate *P* values between groups and to generate survival curves, respectively. Statistical significance testing for all survival analyses was two-sided. To control for multiple hypothesis testing, Benjamini–Hochberg method was applied to correct *P* values, and FDR *q* values were calculated where applicable. Results were considered significant at *P* value or FDR *q* value of <0.05. Multivariate survival analysis was performed using a Cox proportional hazards regression model that calculated the hazard ratio, the 95% confidence interval and *P* values when using pathologic stage, age, KAC and 'other AIC' signatures as covariables.

## Analysis of public datasets

Publicly available datasets were obtained from the Gene Expression Omnibus (GEO) database (https://www.ncbi.nlm.nih.gov/geo/) under accession numbers GSE149813, GSE154989, GSE150263, GSE102511 and GSE219124. Details of the studies[28,29] analysed are as follows: GSE149813 investigated single lung cells from *Kras*[LSL-G12D;LSL-YFP] mice with Ad5CMV-Cre infection[29]; GSE154989 studied AT2 lineage-labelled cells from lungs of *Kras*[LSL-G12D/+];*Rosa26*[LSL-tdTomato/+] mice[28]. Gene expression count matrices of dataset interrogating *Kras*[G12D]-driven mouse model from GSE149813 were pre-processed using Seurat following the same filtering steps in that original report[29]. For the GSE154989 dataset[28], cells used for analysis were the ones labelled as "PASSED_QC" in supplementary table S7 in that study. For the GSE149813 dataset[29], cells with >500 median number of genes detected and <10% fraction of mitochondrial genome derived reads, and according to the pre-processing methods described in their original report[29], were retained for analysis. Cells with >7,500 number of genes detected were further filtered to remove potential doublets or multiplets, resulting in 8,304 cells in total for downstream analysis. Both datasets were integrated with mouse cell data generated in this study using Harmony[18] with default parameters settings. The top ranked 20 Harmony-corrected PCs were used for clustering with the FindClusters function using resolution = 0.4. UMAP dimension reduction embedding was performed using the RunUMAP function with the same set of Harmony-corrected PCs. Gene expression levels and frequencies of representative cluster marker genes were visualized using DotPlot function from Seurat. The KAC signature score was calculated using the AddModuleScore function from Seurat. The mouse KAC signature was also studied in human AT2 cells with and without inducible *KRAS*[G12D] (dataset GSE150263) also from ref. 29. Cell filtration criteria described in the original report[29] were followed to filter out potential dead cells and doublets (number of detected genes > 800 and the percent of mitochondrial gene reads fraction < 25%). The 20 top-ranked PCs were used for clustering using the FindClusters function with resolution = 0.1. UMAP dimension reduction embeddings were computed using the same SNN graph. The KAC signature score was calculated using AddModuleScore function from Seurat package.

The bulk RNA-seq dataset GSE102511 was a previously published dataset by our group and comprised normal lung tissues, precursor AAHs and matched LUADs (*n* = 15, each)[61]. The previously published[62] bulk RNA-seq data GSE219124 were generated on cancer stem cell and stem cell-like progenitor cells, in the form of spheres, and their parental MDA-F471 counterparts (a cell line we had developed and cultured from a KM-LUAD of an NNK-exposed *Gprc5a*[−/−] mouse)[62]. To interrogate the association of KACs with tumour formation, gene expression matrices

of bulk RNA-seq data GSE102511 (TPM count matrix) and GSE219124 (FPKM count matrix) were extracted and used for quantification of KAC signature expression using MCPcounter (v.1.2.0) R package. Heatmaps were generated using pheatmap (v.1.0.12) R package.

Mouse KACs from this study were compared to mouse *Krt8*⁺ transitional cells involved in alveolar regeneration post-acute lung injury from a previous study[25]. Overlapping marker genes between mouse KACs and the previously reported *Krt8*⁺ transitional cells were statistically evaluated using the ggvenn (v.0.1.9) R package using the top-ranked 50 marker genes based on fold change from each study.

## Digital spatial profiling of human tissues

The following antibodies were used for digital spatial profiling (DSP): claudin 4 (clone 3E2C1, AF594, LSBio, LS-C354893, concentration 0.5 µg ml⁻¹) and keratin 8 (clone EP1628Y, AF647, Abcam, ab192468, concentration 0.25 µg ml⁻¹). Optimization of antibodies was performed with different dilutions using colorectal carcinoma and LUAD tissues. IF staining was performed on three cases of matched LUAD and NL using the standard GeoMx DSP protocol for morphology markers only (PanCk: clone AE1/AE3, AF532, concentration 0.25 µg ml⁻¹, from GeoMx Solid Tumour Morp kit HsP, 121300301, Novus Biologicals). Slides were scanned at ×20 using the GeoMx DSP platform (NanoString Technologies). Following scanning, multiplex IF image slides were visualized, adjusting channel thresholds for each fluorophore. Expression of KRT8, PanCK and CLDN4 was assessed in adenocarcinoma cells, adjacent reactive lung tissue and distant non-reactive lung tissue.

## Animal housing and tobacco carcinogen exposure experiments

Animal experiments were conducted according to Institutional Animal Care and Use Committee (IACUC)-approved protocols at the University of Texas MD Anderson Cancer Center. Mice were maintained in a pathogen-free animal facility. No statistical methods were used to predetermine sample sizes. In all animal experiments, sex-matched and age-matched mice were randomized to treatment groups. For all experiments and until end points were reached (up to 7 months after exposure to saline or NNK), mice were monitored for signs of ill health and their body weight was measured to ensure weight loss did not exceed 20% of body weight over 72 h. None of the mice developed these symptoms; therefore, they were all euthanized after reaching IACUC-approved end points. End points permitted by our IACUC protocols were not exceeded in any of the experiments. Analysis of data from animal experiments was performed in a blinded fashion. To study KACs in the context of KM-LUAD pathogenesis in vivo, *Gprc5a*⁻/⁻ mice were interrogated because they form LUADs that are accelerated by tobacco carcinogen exposure and acquire somatic *Kras*^G12D mutations—features that are highly pertinent to KM-LUAD development[21,63,64] and therefore to exploring KACs in this setting. *Gprc5a*⁻/⁻ mice were generated as previously described[21,65]. Sex-matched and age-matched *Gprc5a*⁻/⁻ mice were divided into starting groups of 4 mice per exposure (NNK or saline control) and time point (EOE or 7 months after exposure, *n* = 16 mice in total). Eight-week-old mice were intraperitoneally injected with 75 mg kg⁻¹ of body weight NNK or vehicle 0.9% saline (control), 3 times per week for 8 weeks. At EOE or at 7 months after exposure, lungs were collected for derivation of live single cells for scRNA-seq. Whole lungs from additional mice treated as described above were processed by FFPE and for analysis by IF (*n* = 2 mice per treatment group at EOE and 7 months after exposure, 8 mice in total) and ST (3 lung tissues from *n* = 2 mice at 7 months after NNK exposure).

*Sftpc*^creER/+;*Rosa*^Sun1GFP/+ mice were provided by H. Chapman (University of California, San Francisco) and were crossed to *Gprc5a*⁻/⁻ mice to generate *Gprc5a*⁻/⁻;*Sftpc*^creER/+;*Rosa*^Sun1GFP/+ mice for analysis of lineage-labelled AT2 cells. *Gprc5a*⁻/⁻;*Sftpc*^creER/+;*Rosa*^Sun1GFP/+ mice were treated with 75 mg kg⁻¹ NNK or control saline (intraperitoneally), 3 times per week for 8 weeks. At week 6 of treatment (2 weeks before EOE), mice from both groups received 250 µg (intraperitoneally)

tamoxifen dissolved in corn oil for four consecutive days. At EOE or 3 months after exposure to saline or NNK, lungs were digested to derive live (Sytox Blue-negative) GFP⁺ single cells by flow cytometry using a FACS Aria I instrument as previously described[66] (the gating strategy for GFP cell sorting is shown in Supplementary Fig. 6). Sorted single cells were analysed by scRNA-seq (GFP⁺ and GFP⁻ fractions from *n* = 2 mice per treatment at 3 months after exposure to saline and NNK) or used to derive organoids (GFP⁺ cells from *n* = 4 or 5 mice at EOE to saline or NNK, respectively, and from *n* = 10 or 13 mice at 3 months after saline or NNK, respectively). Whole lungs from additional mice treated with saline or NNK and tamoxifen as described above (*n* = 2 per treatment group) were collected (FFPE) at 3 months after NNK and analysed by IF.

*Krt8-creER*;*Rosa*^tdT/+ animals were used to generate *Gprc5a*⁻/⁻; *Krt8-creER*;*Rosa*^tdT/+ mice for analysis of lineage-labelled KRT8⁺ cells. *Krt8-creER* (stock number 017947) and *Rosa*^tdT/+ (Ai14; stock number 007914) mice were obtained from the Jackson Laboratory. Mice harbouring *Krt8-creER*;*Rosa*^tdT/+ were first used for pilot studies to examine labelling of KRT8⁺ cells. Mice were exposed to control saline (*n* = 2 mice) or to 8 weeks of NNK (*n* = 3 mice) as described above followed by 1 mg tamoxifen for 6 continuous days, after which lungs were analysed at the end of tamoxifen exposure. To examine the relevance of labelled KRT8⁺ cells to tumour development, *Gprc5a*⁻/⁻;*Krt8-creER*;*Rosa*^tdT/+ mice were similarly exposed to NNK for 8 weeks followed by tamoxifen, and lungs were then analysed at 8–12 weeks after NNK exposure (*n* = 3 mice). All lungs were collected and processed for formalin fixation, OCT embedding and IF analysis.

## Histopathological and IF analysis of mouse lung tissues

Lungs of *Gprc5a*⁻/⁻ mice (*n* = 2 per treatment and time point) were inflated with formalin by gravity drip inflation, excised, examined for lung surface lesions by macroscopic observation and processed for FFPE, sectioning and H&E staining. Stained slides were digitally scanned using an Aperio ScanScope Turbo slide scanner (Leica Microsystems) at ×200 magnification, and visualized using ImageScope software (Leica Microsystems). Unstained lung tissue sections were obtained for IF analysis of LAMP3 (clone 391005, Synaptic Systems), KRT8 (TROMA-I clone from the University of Iowa DSHB) and PDPN (clone 8.1.1, from the University of Iowa DSHB). Lung FFPE tissue samples were obtained in the same manner from *Gprc5a*⁻/⁻;*Sftpc*^creER/+;*Rosa*^Sun1GFP/+ mice at 3 months after exposure to saline or NNK (*n* = 2 mice per condition) and following injection with tamoxifen. Tissue sections were obtained for H&E staining and assessment of tumour development, and unstained sections were used for IF analysis using antibodies against GFP (AB13970, Abcam, 1:5000), LAMP3 (391005, Synaptic Systems, 1:10,000), KRT8 (TROMA-I, University of Iowa Developmental Studies Hybridoma Bank, 1:100), PDPN (clone 8.1.1, University of Iowa Developmental Studies Hybridoma Bank, 1:100), claudin 4 (ZMD.306, Invitrogen, 1:250), and PRKCDBP (cavin 3, Proteintech, 1:250). Slides were then stained with fluorophore-conjugated secondary antibodies and 4′,6′-diamidino-2-phenylindole (DAPI). Sections were mounted with Aquapolymount (18606, Polysciences), cover slipped, imaged using an Andor Revolution XDi WD spinning disk confocal microscope and analysed using Imaris software (Oxford Instruments).

Formalin-inflated lung lobes from *Krt8-creER*;*Rosa*^tdT/+ mice were cryoprotected in 20% sucrose in PBS containing 10% OCT compound (4583, Tissue-Tek) overnight on a rocker at 4 °C and embedded in OCT. The next day, 10 µm cryosections were blocked in PBS with 0.3% Triton X-100 and 5% normal donkey serum (017-000-121, Jackson ImmunoResearch) and incubated overnight in a humidified chamber at 4 °C with primary antibodies diluted in PBS with 0.3% Triton X-100 and raised against NKX2-1 (sc-13040, Santa Cruz, 1:1000), LAMP3 (same as above) and KRT8 (same as above). The next morning, sections were washed followed by incubation with secondary antibodies (Jackson ImmunoResearch) and DAPI. Slides were then washed, cover slipped as described

above and imaged using a Nikon A1plus confocal microscope. Cell counter ImageJ plugin was used to count tdT[+] cells within lesions and cells in normal-appearing areas, namely: AT2 cells (LAMP3[+]), tdT[+] AT2 cells (tdT[+]LAMP3[+]), AT1 cells (LAMP3[−]NKX2-1[+], avoiding noticeable airways) and tdT[+] AT1 cells (tdT[+]NKX2-1[+]LAMP3[−]). Percentages of tdT[+]LAMP3[+] and tdT[+]NKX2-1[+]LAMP3[−] cells out of total tdT[+] cells were computed. Counts were averages of triplicate images taken at ×20 magnification for each time point. The percent regional surface area covered by tdT[+] cells in normal-appearing regions was estimated by examining the tdT expression across entire lobe sections for each replicate.

### 3D culture and analysis of AT2-derived organoids

$Gprc5a^{−/−}$;$Sftpc^{creER/+}$;$Rosa^{Sun1GFP/+}$ were treated with NNK or saline and tamoxifen as described above, and they were euthanized at EOE (4 saline-treated and 5 NNK-treated mice) or at 3 months after exposure (10 saline-treated and 13 NNK-treated mice). Lungs were collected, dissociated into single cells (see mouse single-cell derivation in the Methods section 'Single-cell isolation from tissue samples'), and live (Sytox Blue-negative) GFP[+] single cells were collected by flow cytometry using a FACS Aria I instrument as previously described[66]. GFP[+] AT2 cells from NNK-treated or saline-treated groups were immediately washed and resuspended at a concentration of 5,000 cells per 50 µl of 3D medium (F12 medium supplemented with insulin, transferrin and selenium, 10% FBS, penicillin–streptomycin and L-glutamine). GFP[+] cells were mixed at a 1:1 ratio (by volume) with 50,000 mouse endothelial cells (collected from mouse lungs by CD31 selection and expanded in vitro as previously described[67]) and resuspended in 50 µl of Geltrex reduced growth factor basement membrane matrix (A1413301, Gibco). Next, 100 µl of 1:1 GFP[+]:endothelial cell mixture was plated on Transwell inserts with 0.4 µm pores and allowed to solidify for 30 min in a humidified $CO_2$ incubator (EOE: $n$ = 3 wells per condition; 3 months after exposure: $n$ = 4 wells for saline-derived organoids and $n$ = 12 wells for NNK-derived organoids). Each well was then supplemented with 3D medium containing ROCK inhibitor (Y-27632, Millipore) and recombinant mouse FGF-10 (6224-FG, R&D Systems), and plates were incubated at 37 °C in a humidified $CO_2$ incubator. Wells were replenished with 3D medium every other day. For GFP[+] organoids derived from mice exposed to NNK, 200 nM KRAS(G12D)-specific inhibitor MRTX1133 or DMSO vehicle was added to the medium and replenished 3 times a week ($n$ = 6 wells per condition). Organoids were monitored and analysed twice a week using an EVOS M7000 imaging system (Thermo Fisher Scientific), whereby the numbers and sizes of organoids greater than 100 µm in diameter were recorded. At end point, 3D organoids were collected from the basement membrane matrix using Gentle Cell Dissociation reagent (100-0485, StemCell Technologies), fixed with 4% paraformaldehyde, permeabilized, blocked and stained overnight at 4 °C with a mixture of IF primary antibodies raised against LAMP3, GFP, KRT8 and cavin 3. The next day, organoids were washed and stained with fluorophore-conjugated secondary antibodies overnight at 4 °C while being protected from light. Organoids were washed and stained with DAPI nuclear stain for 30 min, after which they were collected in Aqua-Poly/Mount (18606-20, Polysciences) and transferred to slides. Images of organoids were captured using an Andor Revolution XDi WD spinning disk confocal microscope and analysed using Imaris software (Oxford Instruments).

### 2D viability assays

Mouse mycoplasma-free LUAD cell lines LKR13 (mutant $Kras^{G12D}$-driven[31]) and MDA-F471 ($Gprc5a^{−/−}$ and $Kras^{G12D}$ mutant[27]) were plated on 96-well plates (10[3] cells per well) and grown in DMEM (Gibco) supplemented with 10% FBS, 1% antibiotic antimycotic solution (A5955, Sigma-Aldrich) and 1% L-glutamine (G7513, Sigma-Aldrich). The next day, cells were cultured for up to 4 days with medium containing 0.5% FBS, 0.5% FBS with 50 ng ml[−1] epidermal growth factor (EGF) (E5160, Sigma-Aldrich), or 0.5% FBS with EGF and varying concentrations of MRTX1133 (Mirati Therapeutics). alamarBlue Cell Viability reagent (25 µl; DAL1025,

ThermoFisher) was added to each well. At 4 days after treatment, viability was assessed by fluorescence spectrophotometry at 570 nm (and 600 nm as a reference). For the wells showing net positive absorbances relative to blank wells (at least 3 wells per cell line and condition), the percent differences in reduction between treated and control wells were calculated.

### Western blot analysis

LKR13 and MDA-F471 cells were plated in 6-well plates (10[6] cells per well) and grown under different conditions as described above. Protein lysates were extracted at 3 h after treatment and analysed by western blotting following overnight incubation with antibodies to the following primary proteins: vinculin (E1E9V, rabbit, Cell Signaling Technology, 13901; 1:1,000); phosphorylated p44/42 MAPK (ERK1/2, rabbit, Cell Signaling Technology, 9101; 1:2,000); phosphorylated S6 ribosomal protein (Ser 235/236, rabbit, Cell Signaling Technology, 4858; 1:2,000); p44/42 MAPK (ERK1/2, rabbit, Cell Signaling Technology, 9102; 1:2,000); or S6 (E.573.4, rabbit, Invitrogen, MA5-15164; 1:1,000). This was followed by 1 h of incubation with diluted secondary antibody (1706515 goat anti-rabbit IgG-HRP conjugate, Bio-Rad). Protein lysates from each cell line were analysed on multiple gels (four per cell line) with Precision Plus Protein Dual Color Standard (1610394, Bio-Rad) as the ladder and blotted to membranes to separately probe for phosphorylated and total forms of the same proteins, which have highly similar molecular weights (using phospho-specific antibodies or antibodies targeting total version of same protein). Vinculin protein levels were evaluated as loading control on each of the blots. Four blots (phospho-ERK, total ERK, phospho-S6 and total S6) for each of LKR13 and MDA-F471 are shown in Supplementary Fig. 9, each with its own analysis of equal protein loading (vinculin blot) and whereby only the ones indicated with green rectangles are presented in Extended Data Fig. 12c. Membranes were cut horizontally using molecular weight marker as a guide, and cut membranes were incubated with the specified antibodies (see Supplementary Fig. 9 for site of cutting and for overlay of colorimetric and chemiluminescent images of the same blot to display ladder and the analysed protein, respectively). Blots were imaged using the ChemiDoc Touch Imaging System (Bio-Rad) with Chemiluminescence and Colorimetric (for protein ladder) applications and auto expose or manual settings.

### Chemicals and reagents

Tobacco-specific carcinogen (NNK) with a purity of 99.96% by HPLC was purchased from TargetMol. Tamoxifen and H&E staining reagents were purchased from Sigma Aldrich. The KRAS(G12D) inhibitor MRTX1133 was provided by J. Christensen (Mirati Therapeutics).

### Statistical analyses

In addition to the algorithms and statistical analyses described above, all other basic statistical analyses were performed in the R statistical environment (v.4.0.0). The Kruskal–Wallis $H$-test was used to compare variables of interests across three or more groups. Wilcoxon rank-sum test was used for paired comparisons among matched samples from the same patients. Wilcoxon rank-sum test was used to compare other continuous variables such as gene expression levels and signature scores between groups. Spearman's correlation coefficient was calculated to assess associations between two continuous variables (for example, cellular proportions and gene signature scores). Fisher's exact test was used to identify differences in frequencies of groups based on two categorical variables. Ordinal logistic regression was performed using the polr function in the built-in R package MASS (v.7.3). Benjamin–Hochberg method was used to control for multiple hypothesis testing. All statistical tests performed in this study were two-sided. Results were considered significant at $P$ values or FDR $q$ values < 0.05. When a $P$ value reported by R was smaller than 2.2e-16, it was reported as $P < 2.2 \times 10^{-16}$.

## Ethics declarations

All human LUAD and normal lung tissues were obtained from patients who provided informed consent and under institutional review board-approved protocols at The University of Texas MD Anderson Cancer Center. All human data in this manuscript are deidentified to ensure patient privacy. All animal studies were conducted under IACUC-approved protocols at the University of Texas MD Anderson Cancer Center.

## Reporting summary

Further information on research design is available in the Nature Portfolio Reporting Summary linked to this article.

## Data availability

Sequencing data for P1–P5 were previously generated[15] and deposited in the European Genome–phenome Archive (EGA) under the accession number EGAS00001005021. Human scRNA-seq (P6–P16) and ST data generated in this study have been deposited into the EGA under the same accession number (EGAS00001005021). Mouse scRNA-seq and ST data generated in this study have been deposited into the NCBI's GEO with accession number GSE222901. Source data are provided with this paper.

## Code availability

Codes for analysis of scRNA-seq, WES and ST data are available at Zenodo (https://doi.org/10.5281/zenodo.8280138) and GitHub (https://github.com/guangchunhan/LUAD_Code).

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

**Acknowledgements** We appreciate and thank all the patients who provided samples for this research under approved consent; and H. Chapman for providing *Sftpc^creER/+;Rosa^Sun1GFP/+* mice. This work was supported in part by funding from Johnson & Johnson (to H.K.), the University of Texas MD Anderson Cancer Center Office of Strategic Research Programs (to H.K.), National Cancer Institute grants R01CA205608 (to H.K.), R01CA272863 (to H.K.), 1U2CCA233238 (to H.K., S.M.D. and A.E.S.), U01CA264583 (to H.K. and L.W.) and T32CA217789 MD Anderson Cancer Center postdoctoral fellowship (to A.S.), University of Texas SPORE in Lung Cancer P50CA070907, as well as the Cancer Prevention and Research Institute of Texas grants RP150079 (to H.K.) and RP220101 (to H.K.). L.W. and H.K. acknowledge support from the start-up research fund from the University of Texas MD Anderson Cancer Center. T.C., L.W. and H.K. are Andrew Sabin Family Foundation Fellows at the University of Texas MD Anderson Cancer Center. The Flow Cytometry & Cellular Imaging Core Facility is supported by the National Cancer Institute through MD Anderson's Cancer Center Support Grant P30CA016672. The Sequencing Core Facility at the MD Anderson Cancer Center is supported by the National Institutes of Health grant 1S10OD024977.

**Author contributions** G.H. and A.S. are co-first authors, and Z.R. and A.M.L. contributed equally. G.H., A.S., L.W. and H.K. designed the study, interpreted the data and wrote the original draft of the manuscript. All authors reviewed the final version of the manuscript. G.H. led computational analyses relating to scRNA-seq, WES and ST. G.H., A.S., L.W. and H.K. performed data quality control and curation. G.H., X.C. and F.P. processed and aligned scRNA-seq and ST data. G.H. and E.D. performed meta-program analysis. G.H. and Yunhe Liu performed *KRAS* mutation screening using the scRNA-seq data. G.H., Yunhe Liu and G.P. curated bioinformatics pipelines for ST analysis. G.H., A.S., L.W. and H.K. annotated cells, governed overall analysis and interpretation of scRNA-seq, ST and WES data as well as performed data visualization. P.S. developed workflows for analysis of mutations in normal tissues and assisted in data interpretation. A.S. led the generation of scRNA-seq, WES, ST and IF data. A.S., K.K. and L.M.S. processed tissues for ST analysis. J. Fujimoto performed histopathological analysis of tissues analysed by scRNA-seq. A.G.S., J. Fujimoto and L.M.S. performed spot-level histopathological evaluation of mouse and human tissues analysed by ST. W.L., S.D.H. and L.M.S. performed the DSP of human tissues. L.I.G.-B., E.R.P. and L.M.S. developed workflows and provided resources for DSP and IF analysis. A.S., Z.R., J. Feng and W.T. performed tobacco carcinogenesis experiments. A.S. and Z.R. performed tobacco carcinogenesis experiments in AT2 and KRT8 reporter mice. A.S., Z.R., A.M.L., K.L., J.C. and H.K. analysed AT2 and KRT8 tracing in reporter-containing tobacco carcinogenesis experiments. A.M.L., S.J.M. and J.C. developed workflows for KRT8 reporter tracing and IF analysis of mouse lung tissues. J.H. and M.L. developed workflows and tools for ST analysis used in this study. M.L. assisted in human ST data interpretation and analysis. A.S., Yuejiang Liu, C.A. and M.C. performed experiments with cell lines and organoids. T.C., B.S., M.V.N., J.V.H. and J. Fujimoto provided human tissue resources and clinical annotations. M.V.N. and J.V.H. provided advice for KRAS targeted inhibition studies. S.M.D., C.S.S. and A.E.S. provided administrative support, data interpretation and resources pertaining to lung cancer cohorts. J.C., L.W. and H.K. supervised the overall study. H.K. provided strategic oversight and conceived the study.

**Competing interests** C.S.S. and A.E.S. are employees of Johnson & Johnson. H.K. reports research funding from Johnson & Johnson. M.V.N. receives research funding to institution from Mirati, Novartis, Checkmate, Alaunos/Ziopharm, AstraZeneca, Pfizer and Genentech, and consultant/advisory board fees from Mirati, Merck/MSD and Genentech. T.C. reports speaker fees/honoraria from The Society for Immunotherapy of Cancer, Bristol Myers Squibb, Roche, Medscape and PeerView; travel, food and beverage expenses from Dava Oncology and Bristol Myers Squibb; advisory role/consulting fees from MedImmune/AstraZeneca, Bristol Myers Squibb, EMD Serono, Merck & Co., Genentech, Arrowhead Pharmaceuticals and Regeneron; and institutional research funding from MedImmune/AstraZeneca, Bristol Myers Squibb, Boehringer Ingelheim and EMD Serono. S.J.M. reports funding from Arrowhead Pharma and Boehringer Ingelheim outside the scopes of submitted work. B.S. reports consulting and speaker fees from PeerView, AstraZeneca and Medscape, and institutional research funding from Bristol Myers Squibb. J.V.H. reports fees for advisory committees/consulting from AstraZeneca, EMD Serono, Boehringer-Ingelheim, Catalyst, Genentech, GlaxoSmithKline, Hengrui Therapeutics, Eli Lilly, Spectrum, Sanofi, Takeda, Mirati Therapeutics, BMS, BrightPath Biotherapeutics, Janssen Global Services, Nexus Health Systems, Pneuma Respiratory, Kairos Venture Investments, Roche, Leads Biolabs, RefleXion, Chugai Pharmaceuticals; research support from AstraZeneca, Bristol-Myers Squibb, Spectrum and Takeda, and royalties and licensing fees from Spectrum. I.I.W. reports grants and personal fees from Genentech/Roche, grants and personal fees from Bayer, grants and personal fees from Bristol-Myers Squibb, grants and personal fees from AstraZeneca, grants and personal fees from Pfizer, grants and personal fees from HTG Molecular, personal fees

from Asuragen, grants and personal fees from Merck, grants and personal fees from GlaxoSmithKline, grants and personal fees from Guardant Health, personal fees from Flame, grants and personal fees from Novartis, grants and personal fees from Sanofi, personal fees from Daiichi Sankyo, grants and personal fees from Amgen, personal fees from Oncocyte, personal fees from MSD, personal fees from Platform Health, grants from Adaptive, grants from Adaptimmune, grants from EMD Serono, grants from Takeda, grants from Karus, grants from Johnson & Johnson, grants from 4D, from Iovance and from Akoya, outside the submitted work. All other authors declare no competing interests.

**Additional information**

**Correspondence and requests for materials** should be addressed to Jichao Chen, Linghua Wang or Humam Kadara.

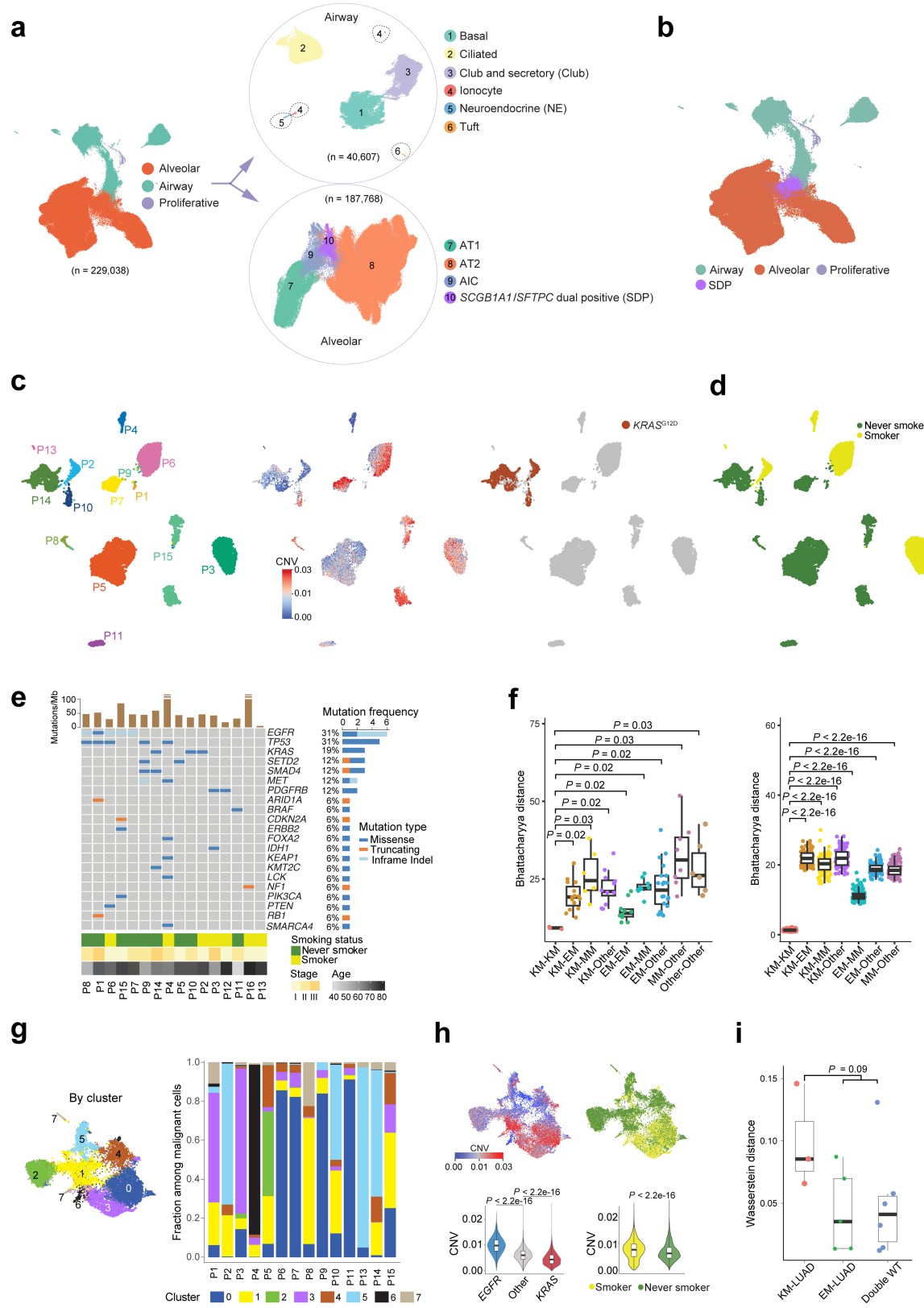

**Extended Data Fig. 1** | See next page for caption.

**Extended Data Fig. 1 | Analysis of normal lung epithelial and malignant subsets in early-stage LUADs. a**,**b**, UMAP plots of 229,038 normal epithelial cells from 63 samples. Each dot represents a single cell coloured by major cell lineage (**a**, left), airway sub-lineage (**a**, top right) and alveolar sub-lineages (**a**, bottom right). SCGB1A1/SFTPC dual positive cells (SDP) cells were separately coloured to show their position on the UMAP (**b**). **c**,**d** UMAP plots of 17,064 malignant cells coloured by patient ID (**c**, left), CNV score (**c**, middle), presence of $KRAS^{G12D}$ mutation (**c**, right) and smoking status (**d**). **e**, Analysis of recurrent driver mutations identified by WES. **f**, Transcriptomic variances quantified by Bhattacharyya distances at the sample (left) and cell (right) levels among LUADs with driver mutations in $KRAS$ (KM), $EGFR$ (EM), and $MET$ (MM), or LUADs that are wild type (WT) for these genes. Box, median ± interquartile range; whiskers, 1.5× interquartile range; centre line: median. $n$ cells in each box-and-whisker in the left panel: KM-KM = 3; KM-EM = 15; KM-MM = 6; KM-Other = 12; EM-EM = 10; EM-MM = 10; EM-Other = 20; MM-Other = 8; Other-Other = 6. $n$ cells in each box-and-whisker in the right panel: 100. $P$ values were calculated by two-sided Wilcoxon Rank-Sum test with a Benjamini–Hochberg correction. **g**, Harmony-corrected UMAP plot of malignant cells coloured by cluster ID (left) and cluster distribution by sample (right). **h**, UMAP plots of malignant cells coloured by CNV scores (top left), smoking status (top right). Comparison of CNV scores between malignant cells from samples carrying different driver mutations (bottom left) or between smokers and never smokers (bottom right). Box-and-whisker definitions are similar to panel **f**. $n$ cells in each box-and-whisker: $EGFR$ = 5,457; Other = 9,135; $KRAS$ = 2,472; Smoker = 5,999; Never smoker = 11,065. $P$ values were calculated by two-sided Wilcoxon Rank-Sum test with a Benjamini–Hochberg correction. **i**, Analysis of Wasserstein distances among KM-LUADs, EM-LUADs, and LUADs with WT $KRAS$ and $EGFR$ (Double WT). Box-and-whisker definitions are similar to panel **f**. $n$ samples in each box-and-whisker: 3; 5; 6. $P$ value was calculated by a two-sided Wilcoxon Rank-Sum test.

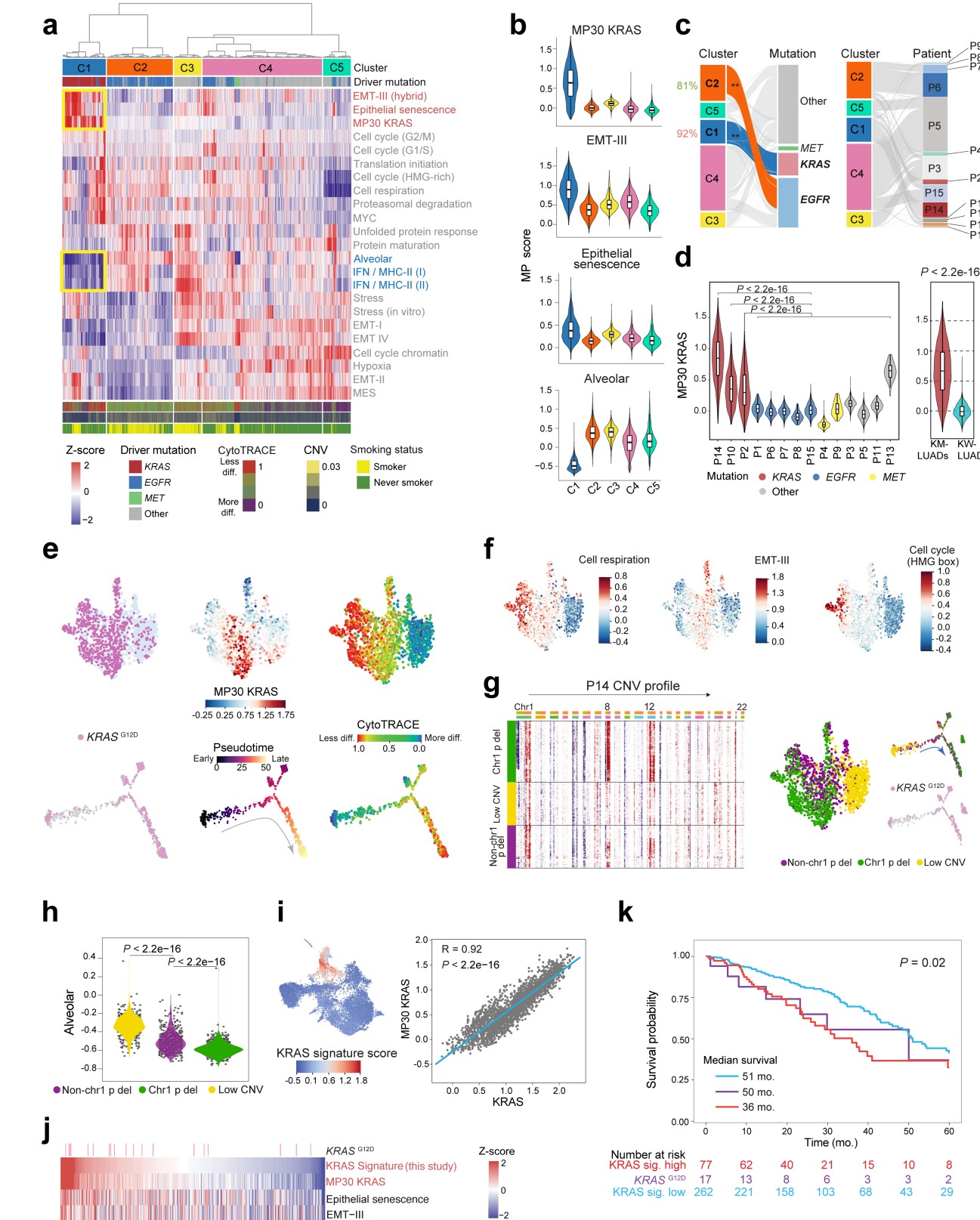

**Extended Data Fig. 2** | See next page for caption.

**Extended Data Fig. 2 | Characterization of inter- and intra-tumour heterogeneity of LUAD malignant cells. a**, Unsupervised clustering of malignant cells based on expression of 23 previously defined consensus cancer cell meta-programs (MPs). **b**, Distribution of signature scores of 4 representative MPs across clusters from **a**. Box-and-whisker definitions similar to Extended Data Fig. 1f. $n$ cells in each box-and-whisker: C1 = 2,600; C2 = 3,968; C3 = 1,647; C4 = 7,182; C5 = 1,667. **c**, Enrichment of clusters (C1-C5) in cells colour coded by recurrent driver mutation status (left) and patients (right). **: $P < 2.2 \times 10^{-16}$. $P$ value was calculated using two-sided Fisher's exact test with a Benjamini–Hochberg correction. **d**, MP30 was computed in malignant cells in each patient (left) and in KM-LUADs versus $KRAS$ WT LUADs (KW-LUADs, right). $n$ cells in each box-and-whisker: P14 = 1,614; P10 = 326; P2 = 532; P1 = 64; P6 = 2,604; P7 = 823; P8 = 147; P15 = 1,819; P4 = 404; P9 = 25; P3 = 2,419; P5 = 5,872; P11 = 375; P13 = 40; KM-LUADs = 2,472; KW-LUADs = 14,592. Box-and-whisker definitions are similar to Extended Data Fig. 1f. $P$ values were calculated using two-sided Wilcoxon Rank-Sum test with a Benjamini–Hochberg correction. **e**, Profiling of ITH in malignant cells from P14 LUAD. UMAP plots show malignant cells coloured by (top left to top right) $KRAS^{G12D}$ mutation status, KRAS signature expression, and cell differentiation status (CytoTRACE). Trajectories of P14 malignant cells coloured by (bottom left to bottom right) the presence of $KRAS^{G12D}$ mutation, inferred pseudotime, and differentiation status. **f**, UMAP plots showing P14 malignant cells coloured by expression of the 3 indicated MPs. **g**, Unsupervised clustering analysis of P14 malignant cells based on inferred CNV profiles (left). UMAP of P14 malignant cells (middle) and inferred trajectory (top right) coloured by CNV clusters, as well as $KRAS^{G12D}$ mutation expression status along pseudotime trajectory (bottom right). **h**, Alveolar MP expression across the CNV clusters shown in panel **g**. $n$ cells in each group: 477; 464; 673. $P$ values were calculated using two-sided Wilcoxon Rank-Sum test with a Benjamini–Hochberg correction. **i**, Harmony-corrected UMAP plot of malignant cells coloured by KRAS signature score (left). Correlation between MP30 expression and KRAS signature score in malignant cells of KM-LUADs (right). $P$ value was calculated with Spearman correlation test. R denotes the Spearman correlation coefficient. **j**, Heatmap showing score distribution of the indicated MPs and signatures in TCGA LUAD samples. **k**, Kaplan-Meier plot showing differences in the survival probability between samples with high and low levels of KRAS signature (KRAS sig.), and those with $KRAS^{G12D}$ mutation. OS: overall survival. $KRAS$ sig. high: samples within top quartile of KRAS signature score. $KRAS$ sig. low: samples below the third quartile of KRAS signature score. mo.: months. $P$ value was calculated with logrank test.

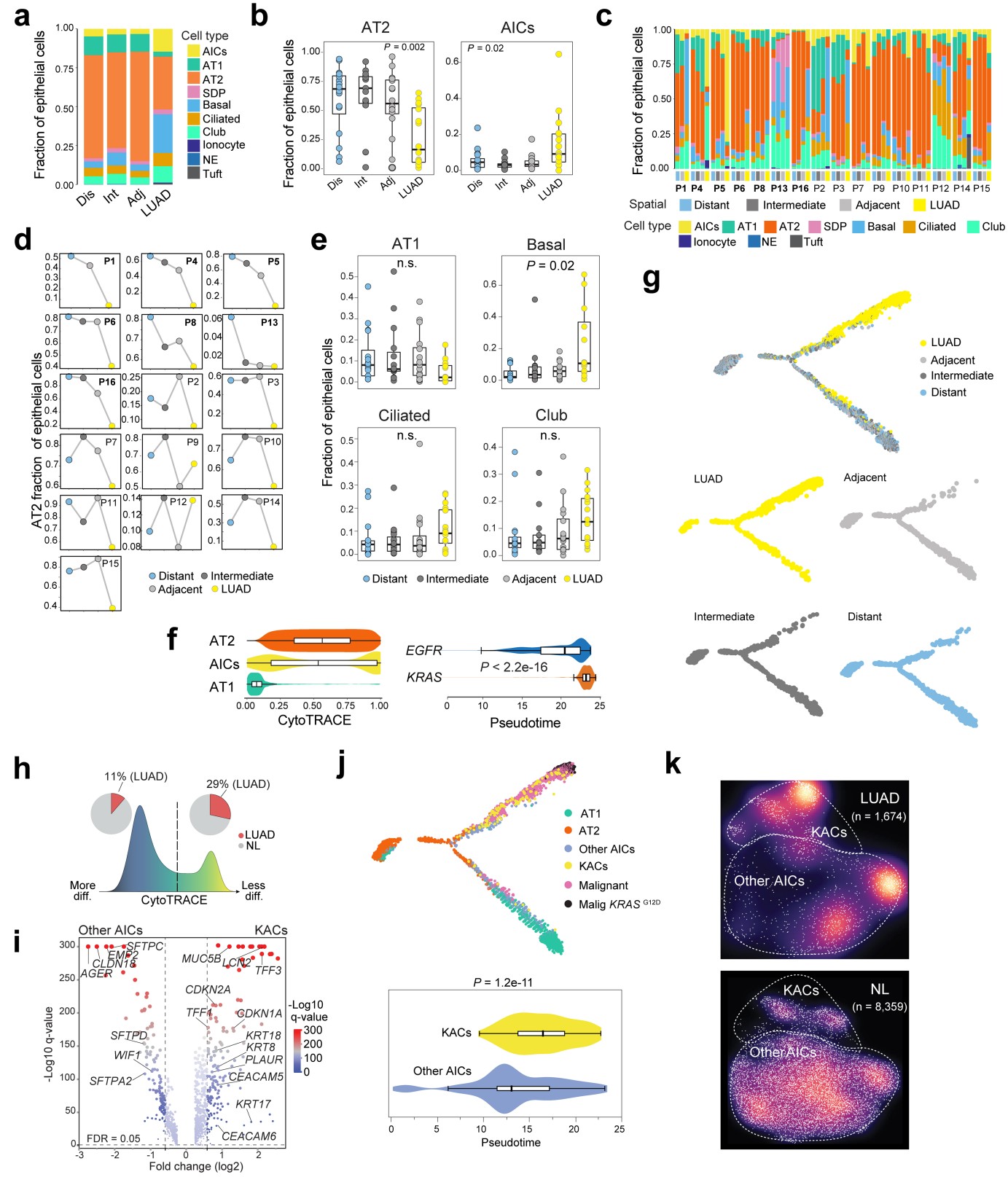

**Extended Data Fig. 3** | See next page for caption.

**Extended Data Fig. 3 | Phenotypic diversity and states of human normal lung epithelial cells. a**, Composition of normal epithelial lineages across spatial regions as defined in Fig. 1a. Dis: distant normal. Int: intermediate normal. Adj: adjacent normal. NE: neuroendocrine. **b**, Changes in cellular fractions of AT2 cells (left) and AICs (right) across the spatial samples. Box-and-whisker definitions are similar to Extended Data Fig. 1f. *n* samples in each box-and-whisker (left to right): 16; 15; 16; 16. *P* values were calculated with Kruskal-Wallis test. **c**, Composition of normal epithelial lineages across the spatial regions at the sample level. **d**, Fractional changes of AT2 cells among all epithelial cells across the spatial regions at the patient level. **c** and **d**: Cases showing gradually reduced AT2 fractions with increasing tumour proximity (7 of the 16 patients; *P* = 0.004 by ordinal regression analysis in **d**). **e**, Fractions of AT1, basal, ciliated, and club and secretory cells along the continuum of the spatial samples. Box-and-whisker definitions are similar to Extended Data Fig. 1f. *n* samples in each box-and-whisker (left to right): 16; 16; 15; 16. *P* values

were calculated with Kruskal-Wallis test. **f**, Distribution of CytoTRACE scores in AICs, AT1 and AT2 cells (left). Distribution of pseudotime scores in malignant cells from *EGFR*- or *KRAS*-mutant tumours (right). *P* value was calculated with two-sided Wilcoxon Rank-Sum test. Box-and-whisker definitions are similar to Extended Data Fig. 1f with *n* cells: AT2 = 14,649; AICs = 974; AT1 = 2,529; *EGFR* = 1,711; *KRAS* = 1,326. **g**, Pseudotime trajectory analysis of alveolar and malignant subsets coloured by tissue location. **h**, Distribution and composition of AICs with low (left) or high (right) CytoTRACE score. **i**, DEGs between KACs and other AICs. **j**, Pseudotime trajectory analysis of malignant and alveolar subsets colour-coded by cell lineage and presence of *KRAS*$^{G12D}$ mutation (top). Pseudotime score in KACs versus other AICs (bottom). Box-and-whisker definitions are similar to Extended Data Fig. 1f. *n* cells in each box-and-whisker: KACs = 157; Other AICs = 817. *P* value was calculated by two-sided Wilcoxon Rank-Sum test. **k**, Differences in cell densities between LUAD (top) and NL tissues (bottom).

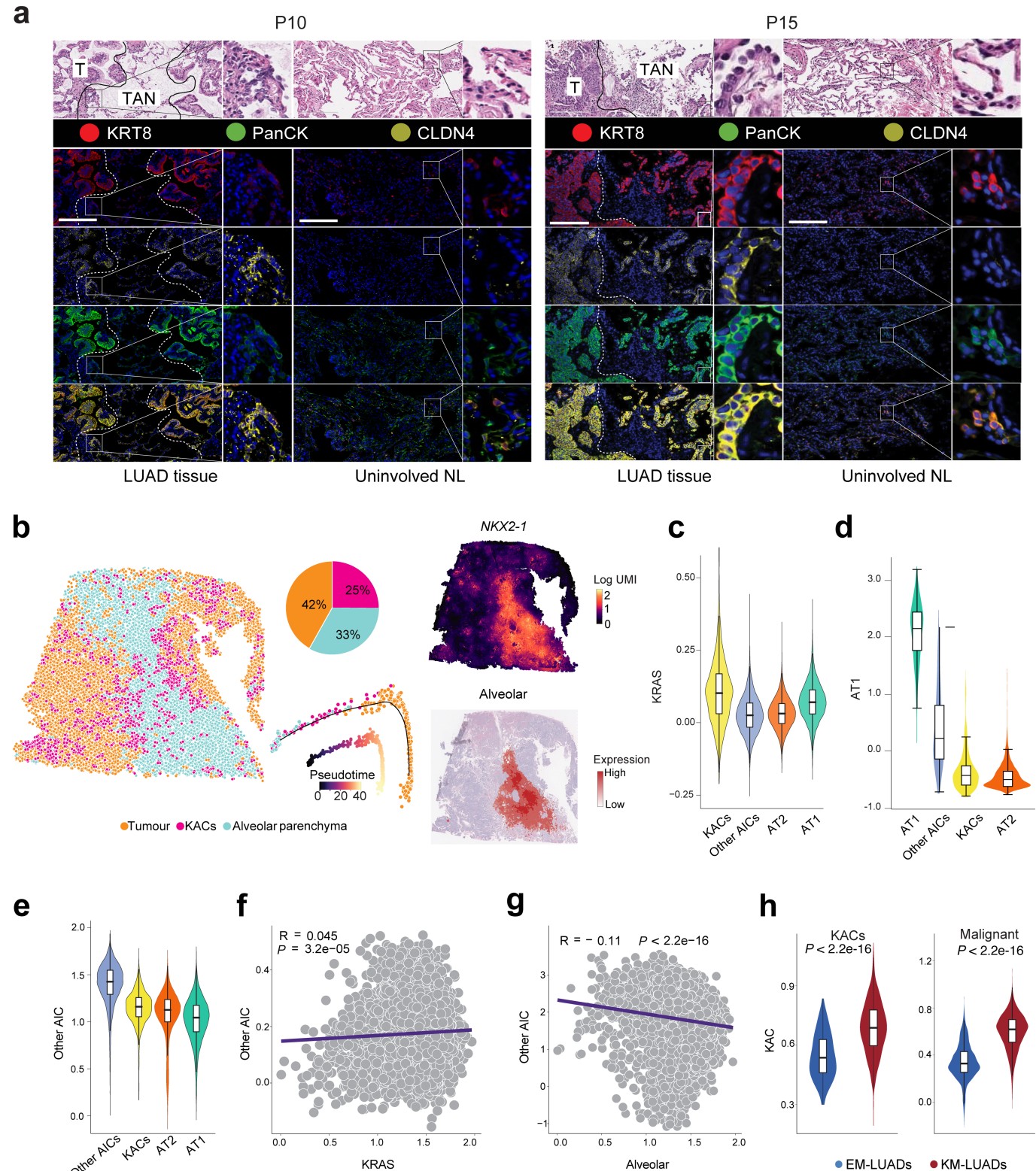

**Extended Data Fig. 4 |** See next page for caption.

**Extended Data Fig. 4 | Spatial and molecular attributes of human KACs.**
**a**, Microphotographs of P10 (left) and P15 (right) LUAD and paired uninvolved NL tissues. Top panels: H&E staining showing LUAD T and TAN (left columns) regions, and uninvolved NL (right columns). DSP analysis of KRT8 (red), CLDN4 (yellow), and pan-cytokeratin (PanCK; green) in LUAD, TAN, and NL regions. Blue nuclear staining was done using Syto13. Magnification, ×20. Scale bar = 200 μm. Staining was repeated four times with similar results. **b**, CytoSPACE deconvolution and trajectory analysis of P14 LUAD ST data. The left spatial map is coloured by deconvoluted cell types. Top middle panel shows the neighbouring cell composition of KACs, and the bottom middle panel depicts inferred trajectory and pseudotime prediction using Monocle 2. Scaled expression of *NKX2-1* and alveolar signature are shown in the rightmost top and bottom panels, respectively. **c**–**e**, Expression of KRAS (**c**), AT1 (**d**), and other AIC (**e**) signatures across AT1, AT2, KACs and other AICs. Box-and-whisker definitions are similar to Extended Data Fig. 1f. *n* cells in each group: KACs = 1,440; Other AICs = 8,593; AT2 = 146,776; AT1 = 25,561. **f**, **g**, Correlation analysis between Other AIC and KRAS (**f**) or alveolar (**g**) signature scores. *P* values were calculated with Spearman correlation test. R denotes the Spearman correlation coefficients. **h**, Enrichment of KAC signature among KACs (left) and malignant cells (right) from KM- or EM-LUAD samples. Box-and-whisker definitions are similar to Extended Data Fig. 1f. *n* cells in each box-and-whisker (left to right): KACs, EM-LUADs = 135; KACs, KM-LUADs = 719; Malignant, EM-LUADs = 5,457; Malignant, KM-LUADs = 2,472. *P* values were calculated by two-sided Wilcoxon Rank-Sum test.

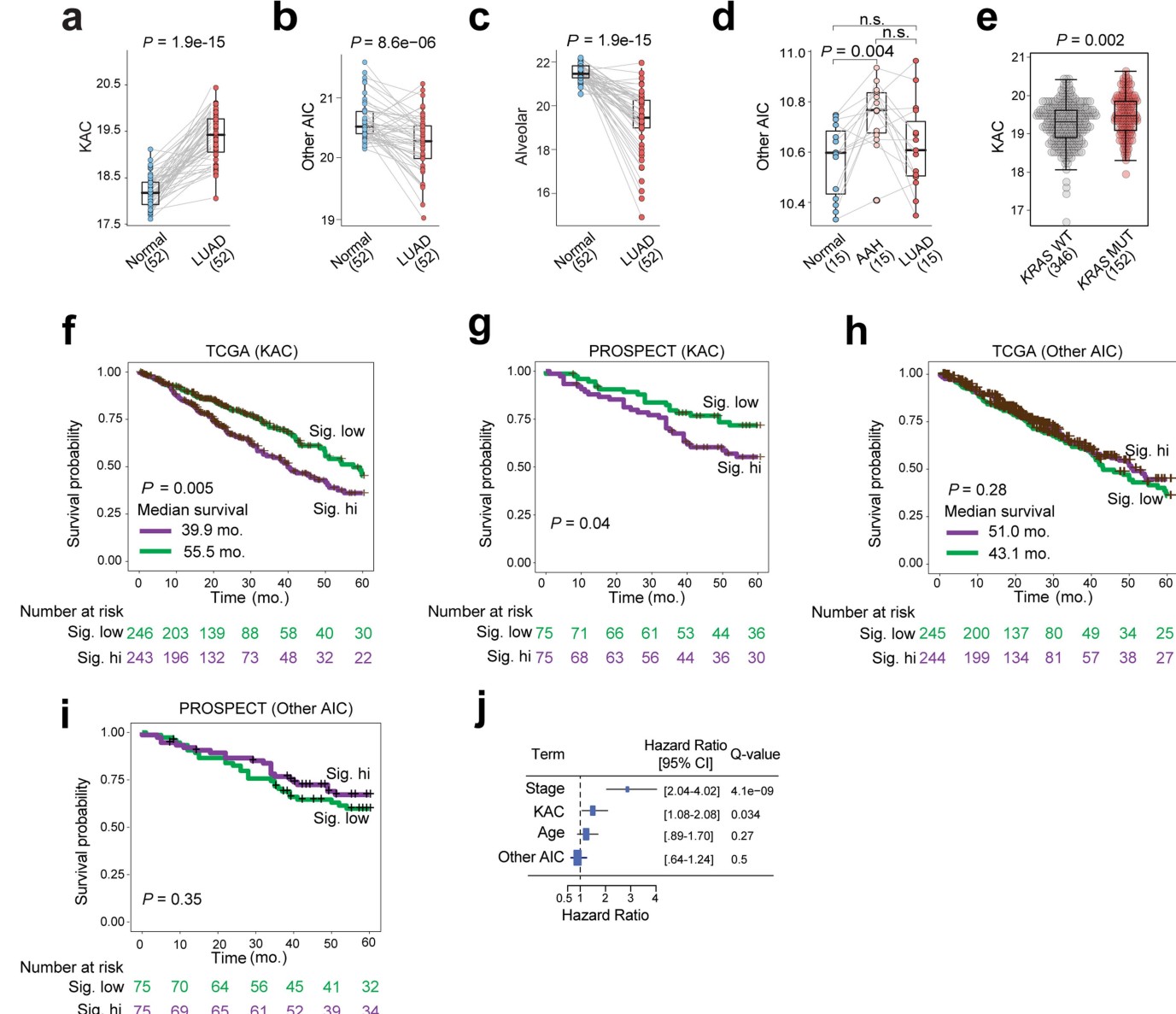

**Extended Data Fig. 5 | Enrichment and clinical relevance of KAC, Other AIC, and alveolar signatures in LUAD. a–e**, Expression of KAC (**a**), other AIC (**b**) and alveolar (**c**) signatures in TCGA LUAD samples and matched NL tissues, of other AIC signature in a lung preneoplasia cohort (**d**), as well as of KAC signature in TCGA LUAD samples grouped by *KRAS* mutation status (**e**). Box-and-whisker definitions are similar to Extended Data Fig. 1f. *n* samples in each group: TCGA Normal = 52; TCGA LUAD = 52; preneoplasia Normal, AAH, and LUAD: 15 each; TCGA LUAD *KRAS* WT = 346; TCGA LUAD *KRAS* MUT = 152. *P* values were calculated by two-sided Wilcoxon Rank-Sum test. Benjamini–Hochberg method was used for multiple testing correction. n.s.: non-significant

(*P* > 0.05). **f–i**, Kaplan-Meier plots showing differences in overall survival probability across TCGA (**f**) and PROSPECT (**g**) samples with high versus low KAC signature scores, or with high versus low scores for other AIC signature (**h**: TCGA; **i**: PROSPECT). Sig. low: LUAD samples with signature scores lower than the group median value. Sig. hi: LUAD samples with signature scores higher than the group median value. *P* values were calculated with the logrank test. **j**, Multivariate Cox proportional hazard regression analysis including pathologic stage, age, as well as KAC and other AIC signatures. Center: estimated Hazard Ratio; error bars: 95% CI. *q* values were calculated by Cox proportional hazards regression model and adjusted with Benjamini–Hochberg method.

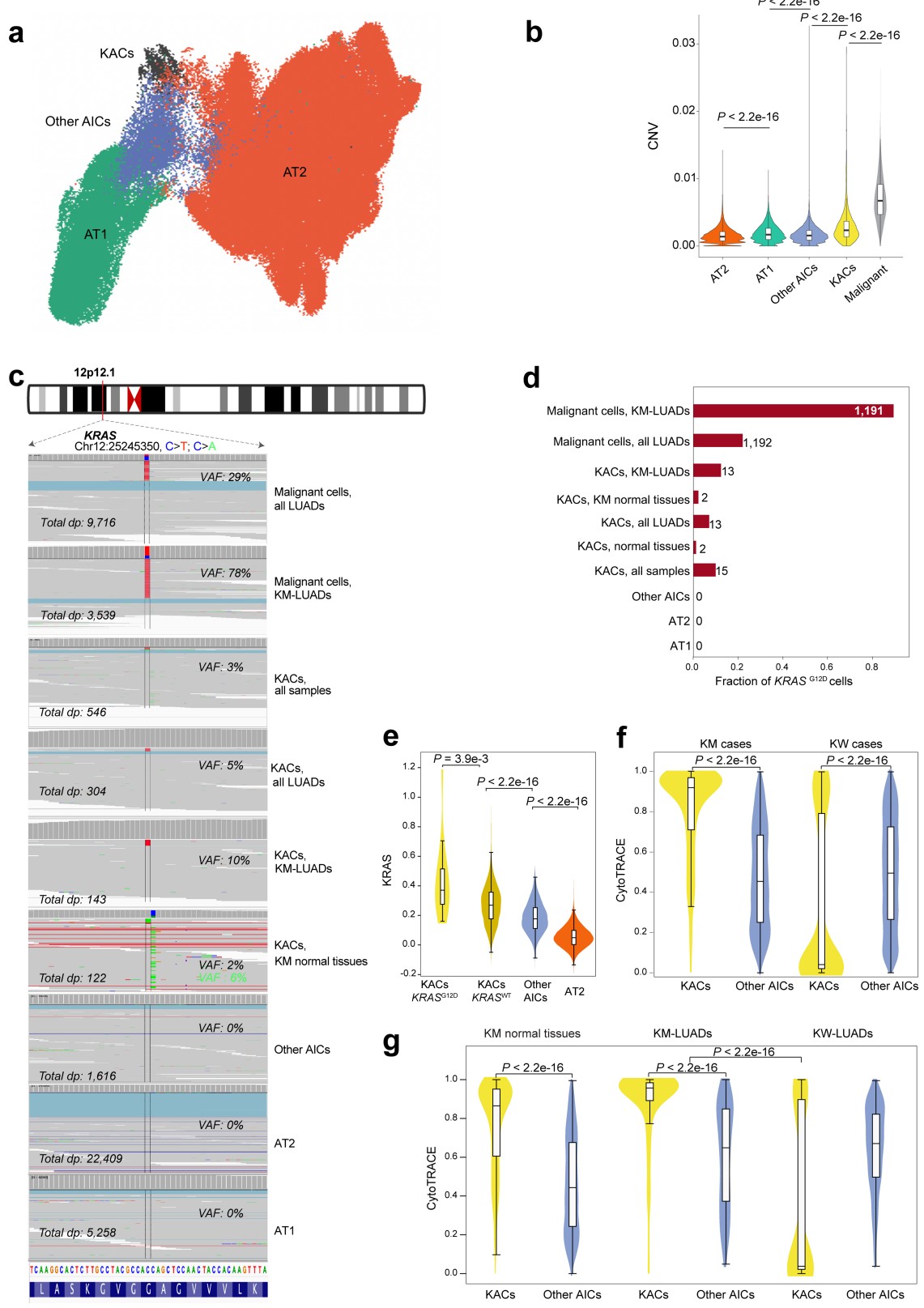

**Extended Data Fig. 6** | See next page for caption.

**Extended Data Fig. 6 | Prevalence of *KRAS*<sup>G12D</sup> mutant KACs in LUAD.**
**a**, UMAP clustering of alveolar subsets. **b**, Quantification of CNV scores across AT1, AT2, KACs and other AICs. Box-and-whisker definitions are similar to Extended Data Fig. 1f. *n* cells in each group: AT2 = 146,776; AT1 = 25,561, Other AICs = 8,593; KACs = 1,440; Malignant = 17,064. *P* values were calculated using two-sided Wilcoxon Rank-Sum test with a Benjamini–Hochberg correction. *KRAS*<sup>G12D</sup> variant allele frequencies (**c**) and fractions of *KRAS*<sup>G12D</sup> mutant cells (**d**) in alveolar and malignant cells from LUAD and normal samples and analysed by scRNA-seq. VAF for *KRAS*<sup>G12C</sup> variant in KACs from KM normal tissues is shown in green (**c**). *n* on top of each bar in **d**: number of *KRAS*<sup>G12D</sup> mutant cells. **e**, KRAS activation signature was statistically compared across *KRAS*<sup>G12D</sup> mutant KACs, *KRAS*<sup>wt</sup> KACs, AICs, and AT2 cells. Box-and-whisker definitions are similar to Extended Data Fig. 1f. *n* cells in each box-and-whisker: KACs *KRAS*<sup>G12D</sup> = 15;

KACs *KRAS*<sup>wt</sup> = 1,425; Other AICs = 8,593; AT2 = 146,776. *P* values were calculated using the two-sided Wilcoxon Rank-Sum test with a Benjamini–Hochberg correction. **f, g**, CytoTRACE scores in KACs versus other AICs from all cells of KM (**f**, left) and KW cases (**f**, right), in cells from normal lung tissues of patients with KM-LUAD (**g**, left), and cells from KM-LUAD (**g**, middle) and KW-LUAD (**g**, right) tissues. Box-and-whisker definitions are similar to Extended Data Fig. 1f. *n* cells in each box-and-whisker: KM cases, KACs = 719; KM cases, Other AICs = 2,414; KW cases, KACs = 721; KM cases, Other AICs = 6,179; KM normal tissues, KACs = 408; KM normal tissues, Other AICs = 2,286; KM-LUADs, KACs = 311; KM-LUADs, Other AICs = 128; KW-LUADs, KACs = 295; KW-LUADs, Other AICs = 940. *P* values were calculated using two-sided Wilcoxon Rank-Sum tests with Benjamini–Hochberg adjustment for multiple testing correction.

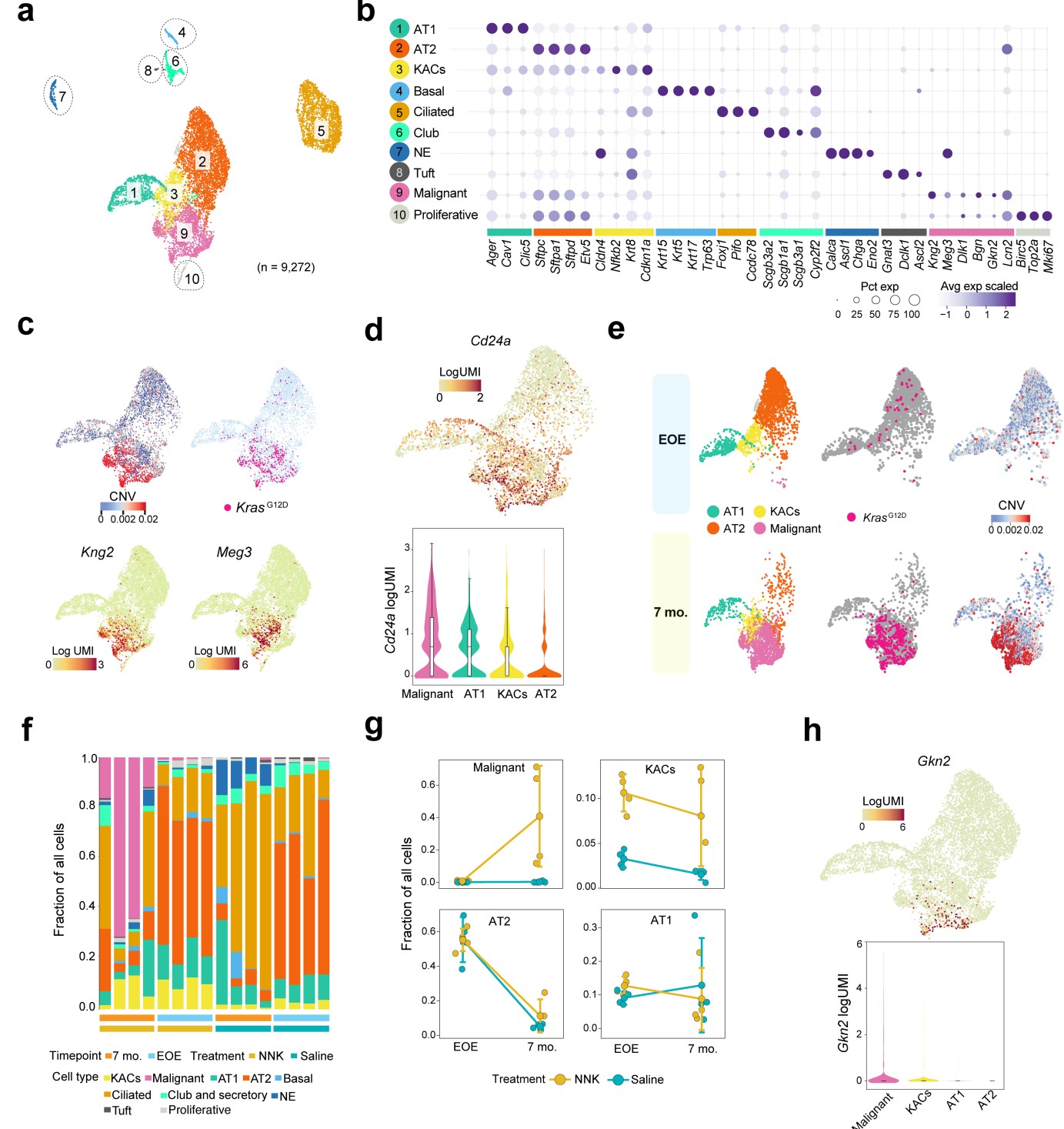

**Extended Data Fig. 7 | scRNA-seq analysis of epithelial subsets in a tobacco carcinogenesis mouse model of KM-LUAD. a**, UMAP distribution of mouse epithelial cell subsets. **b**, Proportions and average expression levels of select marker genes for mouse normal epithelial cell lineages and malignant cell clusters as defined in panel **a**. **c**, UMAP plots of alveolar and malignant cells coloured by CNV score, presence of *Kras*^G12D mutation, or expression levels of *Kng2* and *Meg3*. **d**, UMAP (top) and violin (bottom) plots showing expression level of *Cd24a* in malignant and alveolar subsets. Box-and-whisker definitions are similar to Extended Data Fig. 1f. *n* cells in each group: Malignant = 1,693;

AT1 = 580; KACs = 636; AT2 = 1,791. **e**, UMAP distribution of alveolar and malignant cells coloured by cell lineage, *Kras*^G12D mutation status, and CNV score at EOE or 7 months following NNK. **f**, Proportions of normal epithelial cell lineages and malignant cells in each sample. **g**, Fractional changes of malignant cells, KACs, AT2 and AT1 cells between EOE and 7 months post treatment with NNK or saline; n = 4 biologically independent samples in each group. Whiskers, 1.5× interquartile range; Center dot: median. **h**, UMAP (top) and violin (bottom) plots showing expression levels of *Gkn2* in malignant and alveolar cell subsets. *n* cells in each group: Malignant = 1,693; AT1 = 580; KACs = 636; AT2 = 1,791.

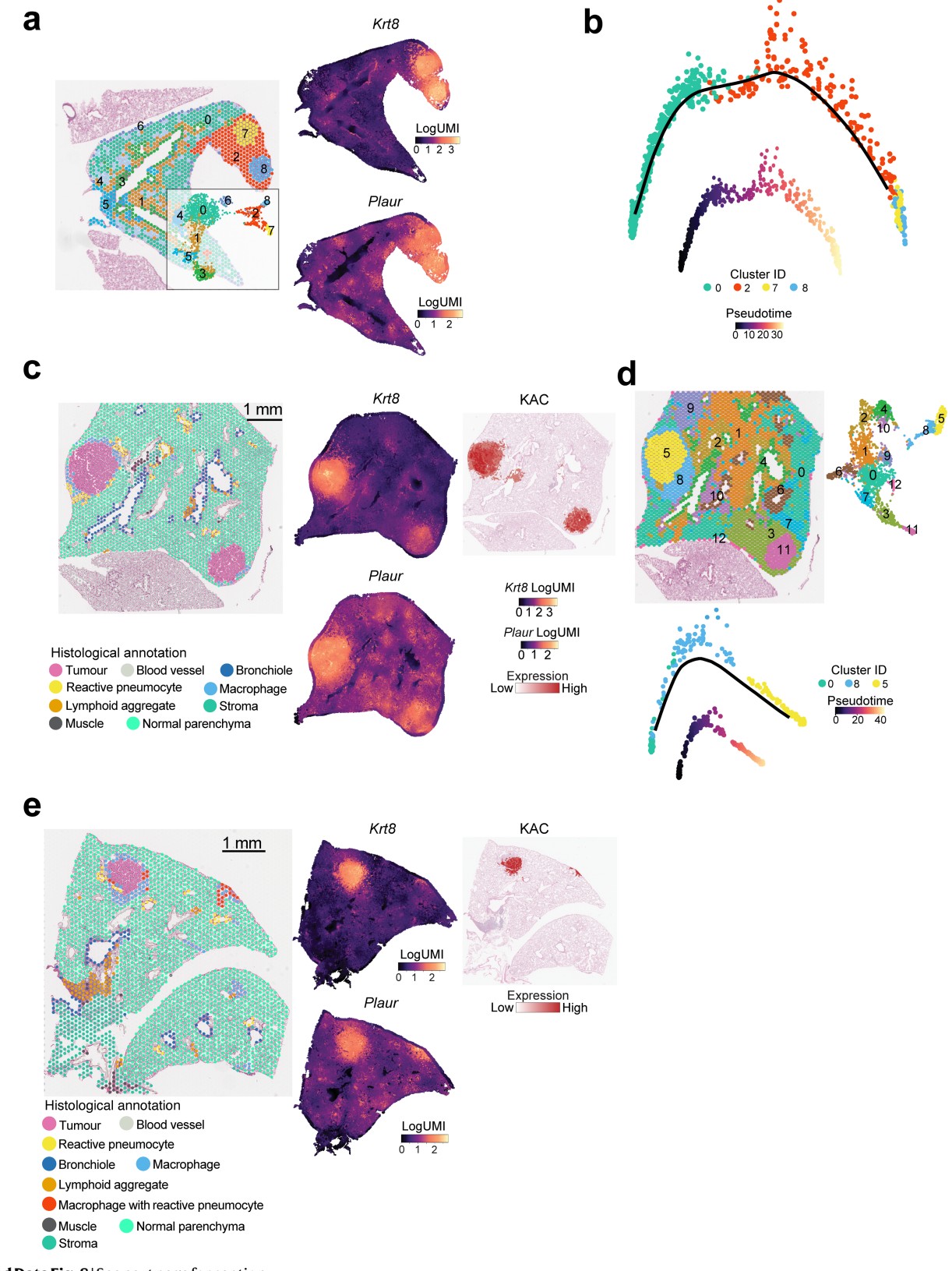

**Extended Data Fig. 8** | See next page for caption.

**Extended Data Fig. 8 | ST analysis of KACs in tobacco-associated development of KM-LUAD. a**, ST analysis of the same tumour-bearing mouse lung in Fig. 3e with cell clusters identified by Seurat (inlet) and mapped spatially (left). Spatial maps with scaled expression of *Krt8* and *Plaur* are shown on the right. **b**, Pseudotime trajectory analysis of C0 (alveolar parenchyma), C2 (reactive area with KACs nearby tumours), and clusters C7 and C8 (representing two tumours) from the same tumour-bearing mouse lung in **a. c**, ST analysis of another tumour-bearing lung region from the same NNK-exposed mouse as in panel a, and showing histological spot-level annotation of H&E-stained images (left) followed by spatial maps with scaled expression of *Krt8*, *Plaur*, and KAC signature (right). **d**, Cell clusters identified by Seurat (top left) and mapped spatially (top right) from the same mouse tumour-bearing lung in **c**. bottom of panel k: Pseudotime trajectory analysis of C0 (alveolar parenchyma), C8 (reactive area with KACs nearby the tumour), and C5 (representing one tumour) from the mouse tumour-bearing lung in **c. e**, ST analysis of a tumour-bearing lung from an additional mouse at 7 months following NNK showing histological spot-level annotation of H&E-stained images (left) followed by spatial maps with scaled expression of *Krt8* (middle, top), *Plaur* (middle, bottom), and KAC signature (right).

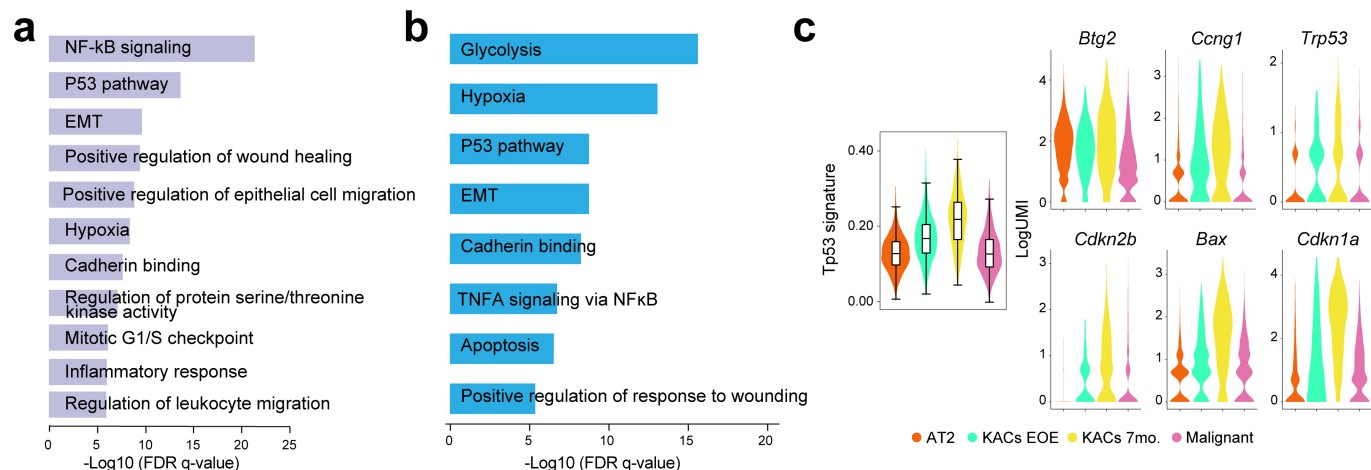

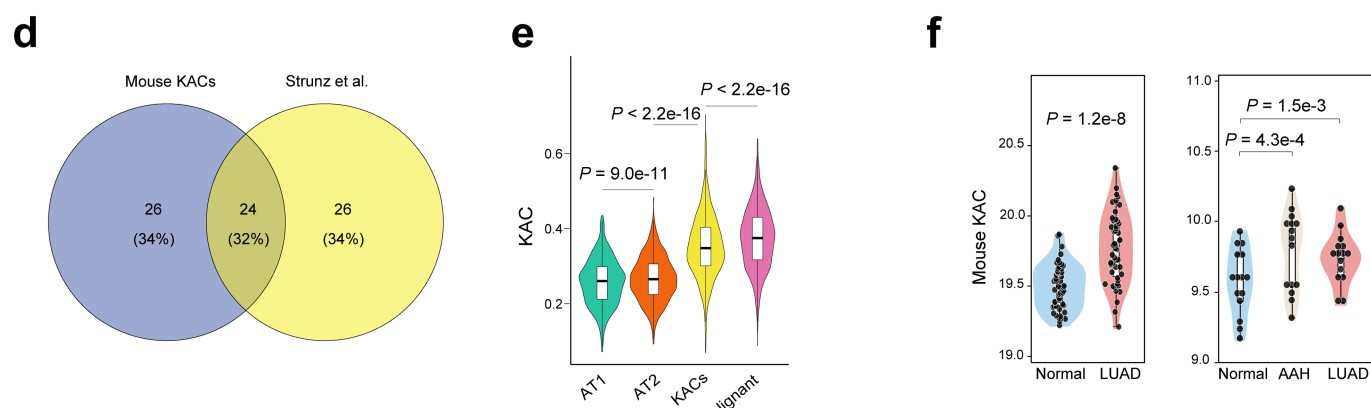

**Extended Data Fig. 9 | Mouse KAC signatures and pathways are relevant to both injury models and human KM-LUAD. a,b,** Pathway enrichment analysis of KACs relative to other alveolar cell subsets and malignant cells in tumour-bearing mice at 7 months following NNK (**a**) and in the human LUAD scRNA-seq dataset from this study (**b**). **c,** Enrichment of *Tp53* signature derived from mouse KACs, and expression of *Btg2*, *Ccng1*, *Cdkn2b*, *Bax*, *Cdkn1a*, as well as *Trp53* itself, across AT2 cells, malignant cells, and KACs at EOE or at 7 months following NNK or saline. *n* cells in each group: AT2 = 1,791; KACs EOE = 301; KACs 7mo. = 335; Malignant = 1,693. **d,** Pie chart showing percentages of unique and overlapping DEG sets between mouse KACs from this study and *Krt8*⁺

transitional cells identified by Strunz and colleagues. **e,f,** Expression of the mouse KAC signature across alveolar and malignant cell subsets from this study (**e**), in normal lung (Normal) and LUAD tissues from the TCGA cohort (**f**, left), as well as in normal lung (Normal), AAH, and LUAD tissues of our premalignancy cohort (**f**, right). *n* cells in each group of panel **e**: AT2 = 1,791; KACs EOE = 301; KACs 7mo. = 335; Malignant = 1,693. *n* samples in each group of panel **f** left: Normal = 52; LUAD = 52. *n* samples in each group of panel **f** right: Normal = 15; AAH = 15; LUAD = 15. Box-and-whisker definitions are similar to Extended Data Fig. 1f. *P* values were calculated using two-sided Wilcoxon Rank-Sum test with a Benjamini–Hochberg correction.

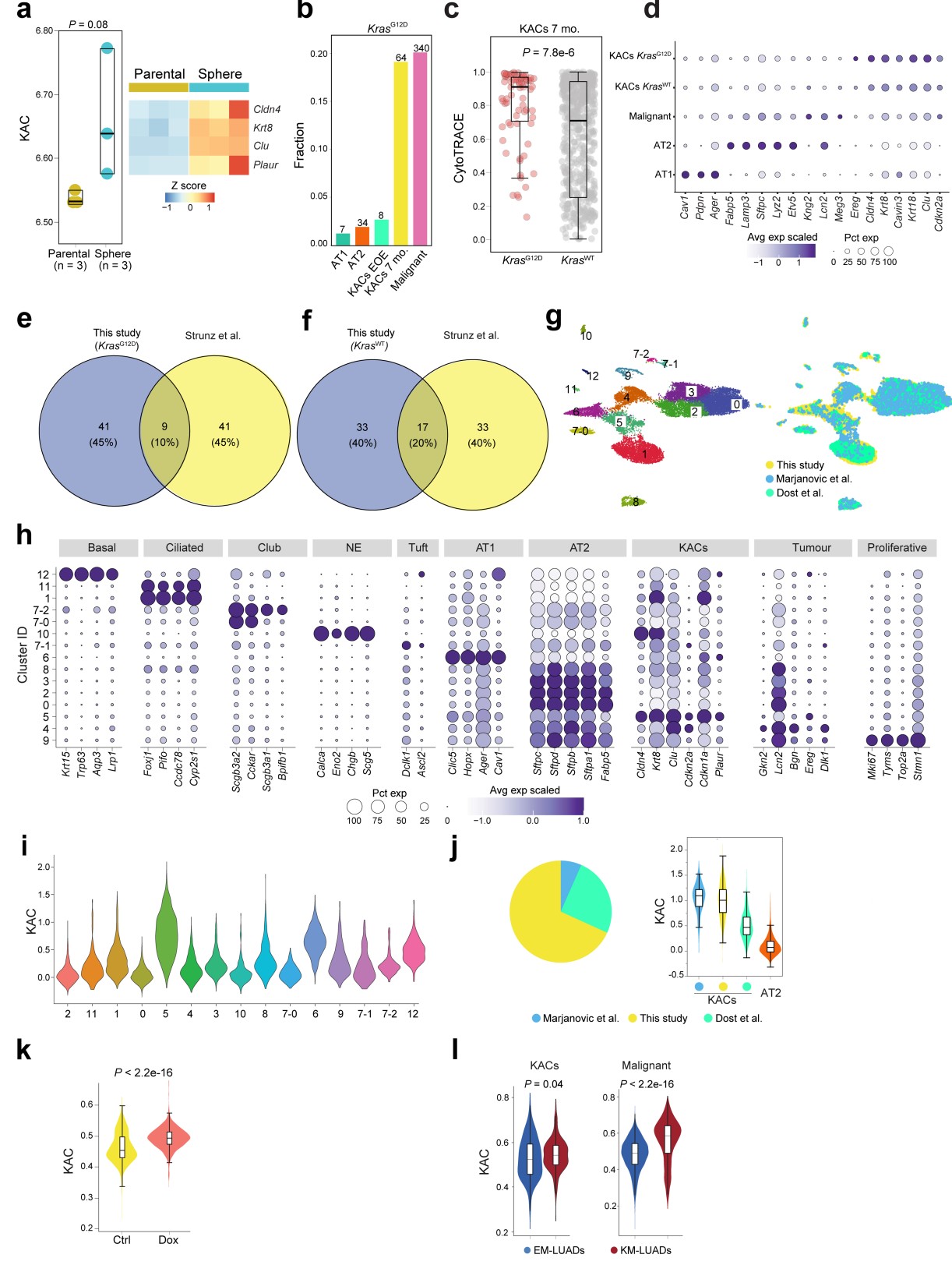

**Extended Data Fig. 10** | See next page for caption.

**Extended Data Fig. 10 | Mouse KACs exist in a continuum, bear strong resemblance to human KACs, and are present in independent *KRAS*^G12D-driven mouse models of LUAD. a**, Mouse KAC signature score (left) and heatmap showing expression of select KAC marker genes (right) in bulk transcriptomes of MDA-F471-derived 3D spheres versus parental MDA-F471 cells grown in 2D. *P* value was calculated using two-sided Wilcoxon Rank-Sum test. Box-and-whisker definitions are similar to Extended Data Fig. 1f. **b**, Fraction of *Kras*^G12D mutant cells in different mouse alveolar cell subsets including when separating KACs into early KACs at EOE and late KACs at 7 months following NNK. Numbers of *Kras*^G12D mutant cells are indicated on top of each bar. **c**, CytoTRACE scores in late KACs with *Kras*^G12D mutation and in those with wild type *KRAS* (*Kras*^wt). *P* value was calculated using two-sided Wilcoxon Rank-Sum test. Box-and-whisker definitions are similar to Extended Data Fig. 1f. *n* cells in each box-and-whisker: *Kras*^G12D = 72; *Kras*^wt = 564. **d**, Proportions and average expression levels of select marker genes for the different subsets indicated. Pie charts showing percentages of unique and overlapping DEG sets between *Krt8*^+ transitional cells identified by Strunz and colleagues and either *Kras*^G12D (**e**) or *Kras*^wt (**f**) KACs from this study. **g**, UMAP clustering of cells integrated from our mouse cohort with cells in the scRNA-seq datasets from studies by Marjanovic et al. and Dost et al. **h**, Proportions and average expression levels of select marker genes for diverse alveolar and tumour cell subsets and across clusters defined in panel **g** with cluster 5 (C5) shown to be enriched with KAC markers. **i**, KAC signature expression across clusters defined in panel **g**. *n* cells in each cluster: 2 = 2,463; 11 = 154; 1 = 3,480; 0 = 4,396; 5 = 1,362; 4 = 1,513; 3 = 2,392; 10 = 219; 8 = 577; 7-0 = 382; 6 = 1,042; 9 = 285; 7-1 = 141; 7-2 = 115; 12 = 119. **j**, Distribution of cells from C5 across the three indicated cohorts (left). KAC signature enrichment across KACs from the three cohorts and relative to pooled AT2 cells (right). Box-and-whisker definitions are similar to Extended Data Fig. 1f. *n* cells in each box-and-whisker: KACs, Marjanovic et al = 90; This study = 485; Dost et al = 343; AT2 = 3,762. **k**, KAC signature score in human AT2 cells with induced expression of *KRAS*^G12D (Dox) relative to *KRAS*^wt cells (Ctrl) from the Dost et al. study. Dox: Doxycycline. Box-and-whisker definitions are similar to Extended Data Fig. 1f. *n* cells in each box-and-whisker: Ctrl = 802; Dox = 1,341. *P* value was calculated using two-sided Wilcoxon Rank-Sum test. **l**, Mouse KAC signature expression in KACs (left) and malignant cells (Malignant, right) from KM-LUADs relative to EM-LUADs in our human scRNA-seq dataset. Box-and-whisker definitions are similar to Extended Data Fig. 1f. *n* cells in each box-and-whisker: KACs, EM-LUADs = 135; KACs, KM-LUADs = 719; Malignant, EM-LUADs = 5,457; Malignant, KM-LUADs = 2,472. *P* values were calculated using two-sided Wilcoxon Rank-Sum test.

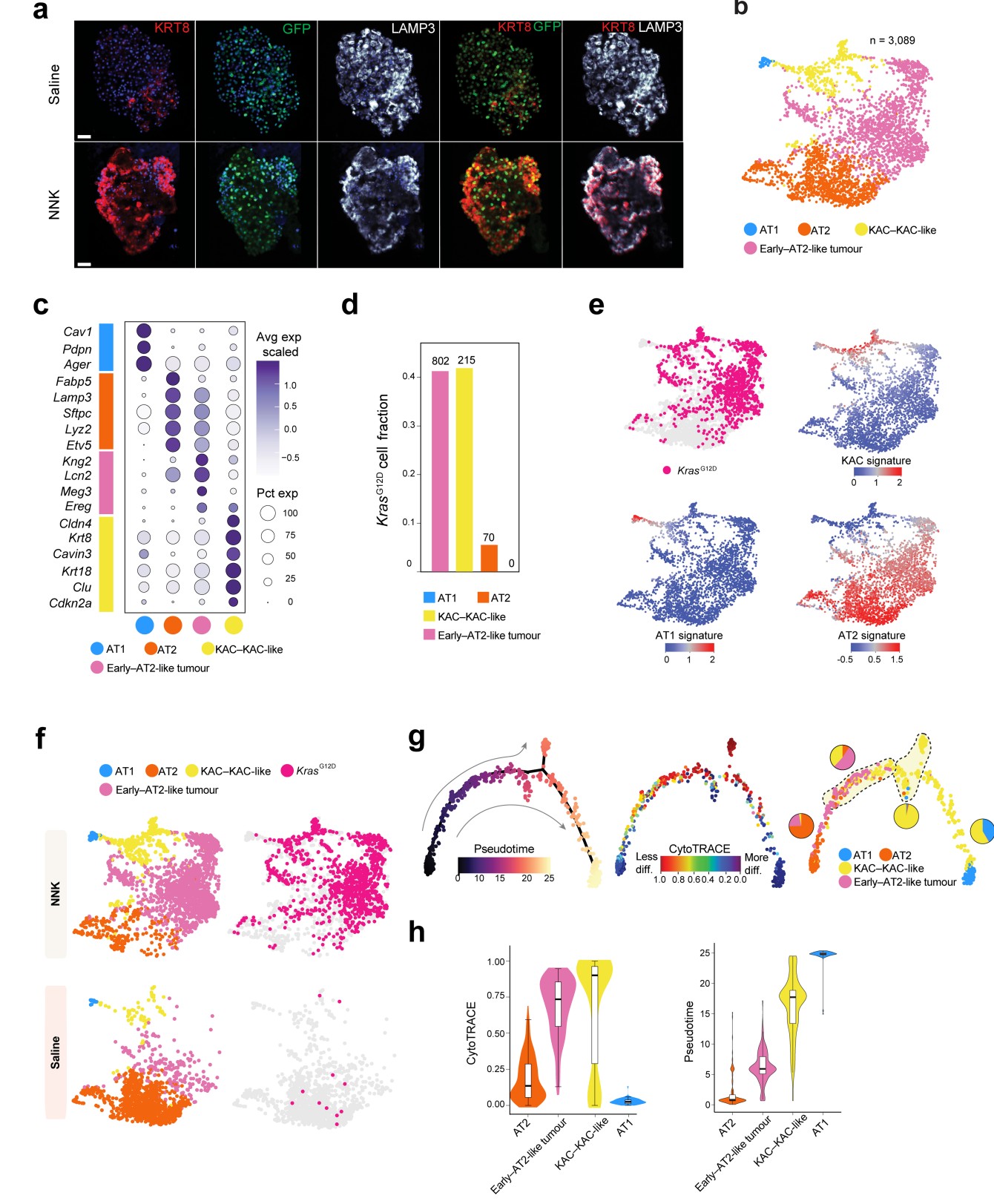

**Extended Data Fig. 11** | See next page for caption.

**Extended Data Fig. 11 | KACs are enriched in lungs and they precede the formation of _Kras_<sup>G12D</sup> tumours in an AT2 lineage reporter tobacco carcinogenesis mouse model. a**, Representative IF analysis of KRT8, GFP, and LAMP3 in GFP-labelled AT2-derived mouse lung organoids (n = 3 wells per condition) derived from tamoxifen-exposed AT2 reporter mice at EOE to saline (n = 4 mice) or NNK (n = 5 mice). Scale bar: 10 μm. **b**, UMAP distribution of GFP⁺ cells at 3 months following NNK exposure or saline and coloured by alveolar or tumour subsets. **c**, Proportions and average expression levels of select marker genes for mouse normal alveolar cell lineages and tumour cells defined in **b**. **d**, Fraction of _Kras_<sup>G12D</sup> cells across alveolar and early tumour subsets. Absolute numbers of _Kras_<sup>G12D</sup> cells are indicated on top of each bar. **e**, UMAPs of GFP⁺ cells from tumour-bearing AT2 reporter mice at 3 months following NNK or saline and coloured by presence of _Kras_<sup>G12D</sup> mutation or expression of KAC, AT1, and AT2 signatures. **f**, UMAPs showing distribution of alveolar and tumour cell subsets (left) as well as cells with _Kras_<sup>G12D</sup> mutation (right) by treatment (saline or NNK). **g**, Trajectories of GFP⁺ cells from tumour-bearing reporter mice at 3 months following NNK or saline coloured by inferred pseudotime (left), differentiation (middle), and cell lineage and showing subset composition (right). **h**, CytoTRACE (left) and pseudotime (right) scores across GFP⁺ subsets. Box-and-whisker definitions are similar to Extended Data Fig. 1f. *n* cells in each box-and-whisker: AT2 = 144; Early–AT2-like tumour = 144; KAC–KAC-like = 288; AT1 = 72.

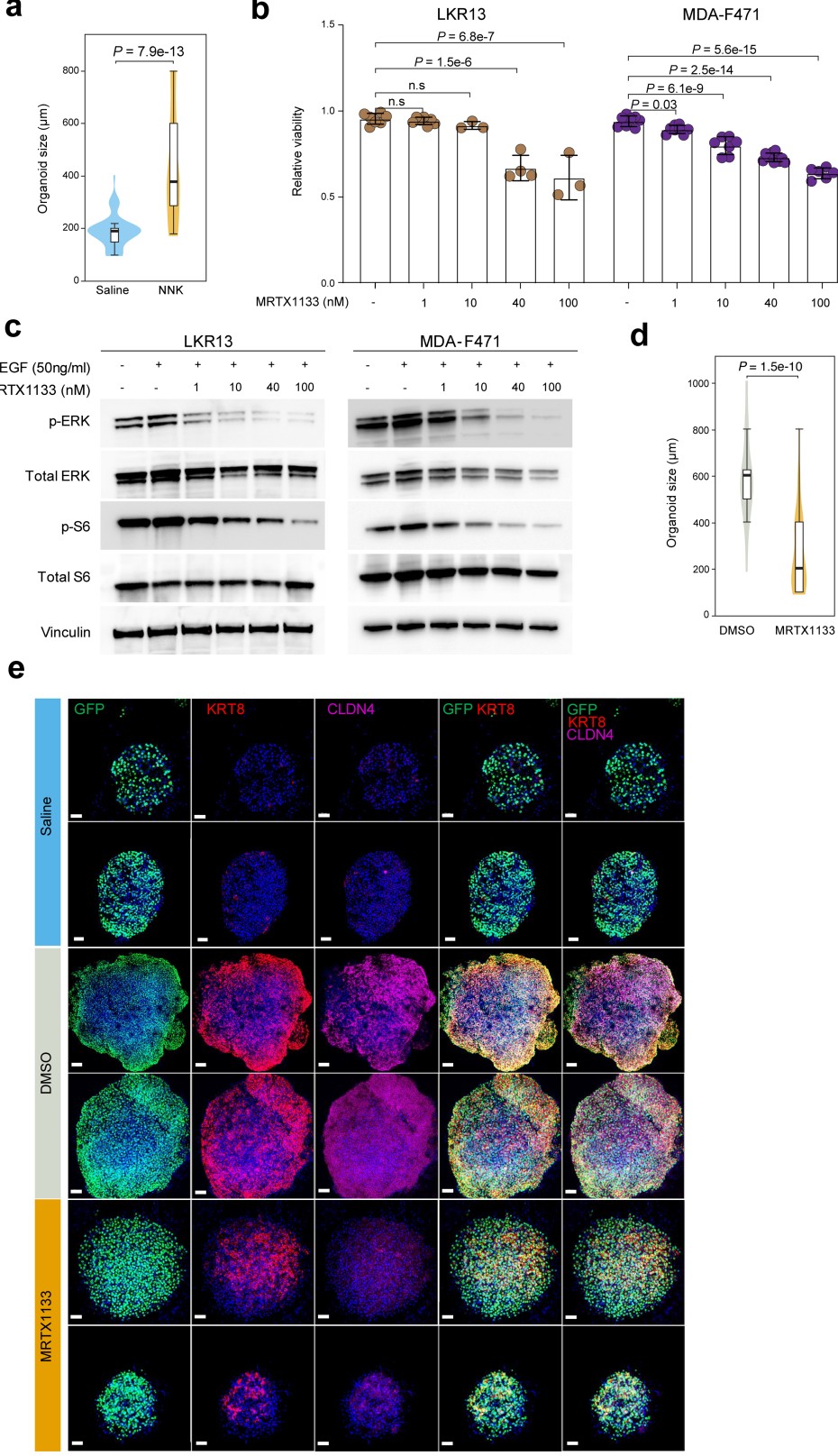

**Extended Data Fig. 12** | See next page for caption.

**Extended Data Fig. 12 | KAC-rich organoids are sensitive to targeted inhibition of KRAS. a**, Size quantification of organoids derived from GFP⁺ lungs cells of mice treated with saline (derived from 10 mice and plated into 4 wells) or NNK (derived from 13 mice and plated into 12 wells) at 3 months post-exposure. Box-and-whisker definitions are similar to Extended Data Fig. 1f. *n* organoids in each group: Saline = 63; NNK = 66. *P* value was calculated using two-sided Wilcoxon Rank-Sum test. **b**, Analysis of relative viability 4 days post treatment of LKR13 and MDA-F471 cells following treatment with increasing concentrations of MRTX1133. *n* samples in each group of LKR13 cells: - = 7; 1 = 7; 10 = 3; 40 = 4; 100 = 3. *n* samples in each group of MDA-F471 cells: - = 8; 1 = 8; 10 = 7; 40 = 11; 100 = 6. n.s: non-significant (*P* > 0.05). Error-bars: standard deviations of means. *P* values were calculated using an ordinary one-way ANOVA with Dunnett's post-test. Results are representative of two independent experiments. **c**, Western blot analysis for the indicated proteins and phosphorylated proteins at 3 h post-treatment to EGF without or with increasing concentrations of the KRASG12D inhibitor MRTX1133 (from Mirati Therapeutics, Inc.). Proteins were run on additional gels (4 per cell line) to separately blot with antibodies against phosphorylated and total forms of each

of the indicated proteins (Supplementary Fig. 9). Vinculin protein levels were analysed as loading control for each gel whereby four LKR13 and four MDA-F471 blots are shown in Supplementary Fig. 9. For lysates from each of the two cell lines, vinculin blots from Gel 1 (Supplementary Fig. 9) are selected and shown in this figure panel. Uncropped images of western blots with molecular weight ladder are also shown in Supplementary Fig. 9. Results are representative of three independent experiments. EGF: epidermal growth factor. **d**, Size quantification of organoids derived from GFP⁺ lungs cells of NNK-treated AT2 reporter mice and treated with 200 nM MRTX1133 or control DMSO in vitro (n = 6 wells per condition). Box-and-whisker definitions are similar to Extended Data Fig. 1f. *n* samples (organoids) in each group: DMSO = 38; MRTX1133 = 53. *P* value was calculated using two-sided Wilcoxon Rank-Sum test. **e**, IF analysis showing representative organoids derived from sorted GFP⁺ cells from AT2 reporter mice that were exposed to saline (top two rows; n = 4 wells) or exposed to NNK and then treated ex vivo with DMSO (middle two rows; n = 6 wells) or 200 nM MRTX1133 (bottom two rows; n = 6 wells). Scale bars = 50 μm except for the first DMSO-treated organoid (third row) whereby scale bar = 100 μm. Staining was repeated three times with similar results.

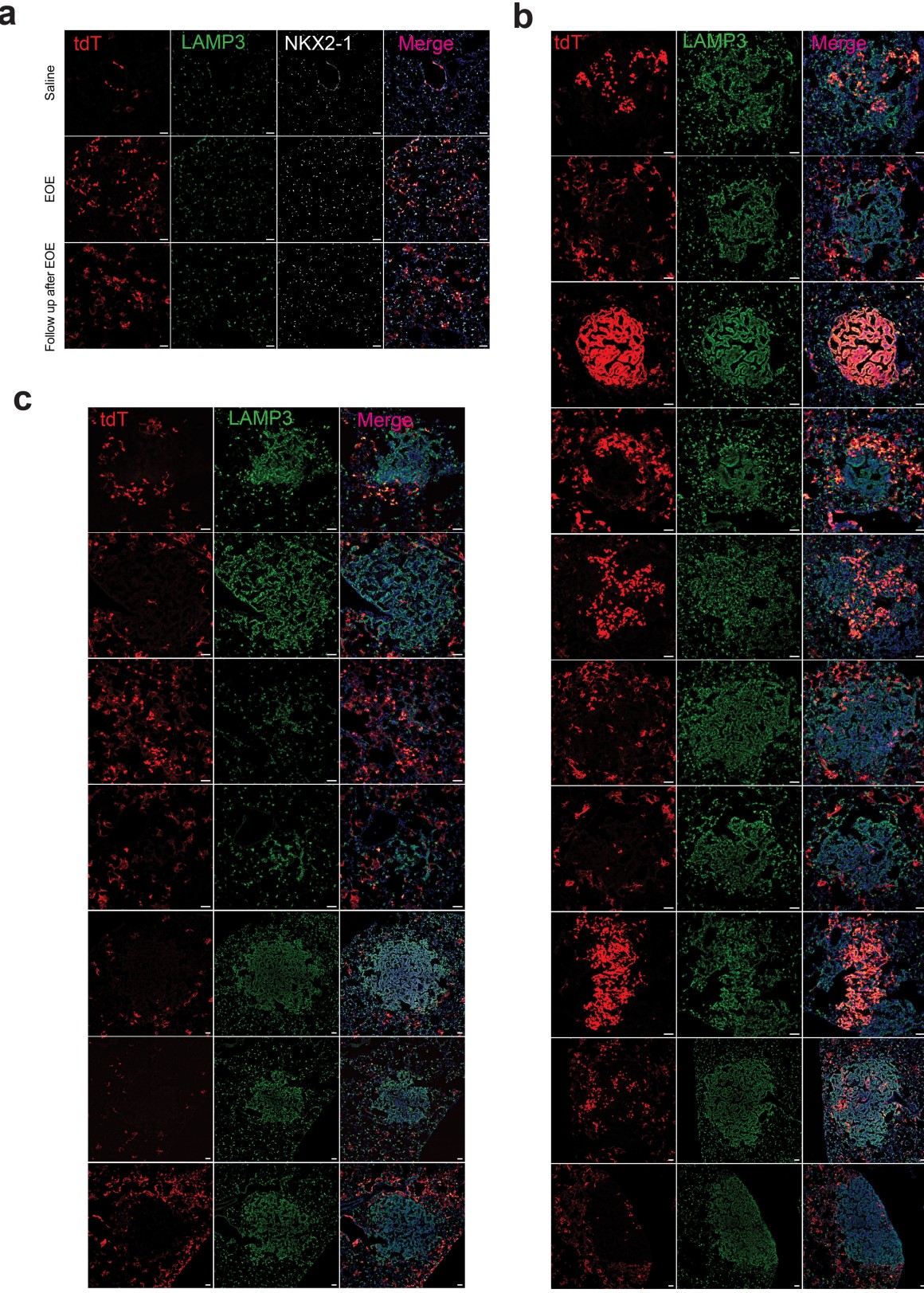

**Extended Data Fig. 13 | Analysis of labelled Krt8⁺ cells following tobacco carcinogen exposure. a**, Representative images of IF analysis of tdT, LAMP3, and NKX2-1 in lung tissues of control saline-treated mice (upper row; n = 2), in non-tumour (normal) lung regions of mice at end of an 8-week NNK exposure (middle row; n = 3), as well as in non-tumour (normal) lung regions of mice at 8–12 weeks following EOE to NNK (lower row; n = 3), and in *Gprc5a⁻/⁻;Krt8-creER; Rosa^{tdTi/+}* mice. IF analysis of tdT and LAMP3 in tumours detected in *Gprc5a⁻/⁻; Krt8-creER;Rosa^{tdTi/+}* mice and showing strong (**b**, n = 10) and negative/low (**c**, n = 7) tdT labelling in tumour cells. Scale bars = 10 μm.

# Reporting Summary

## Statistics

For all statistical analyses, confirm that the following items are present in the figure legend, table legend, main text, or Methods section.

| n/a | Confirmed | |
|---|---|---|
| ☐ | ☒ | The exact sample size (*n*) for each experimental group/condition, given as a discrete number and unit of measurement |
| ☐ | ☒ | A statement on whether measurements were taken from distinct samples or whether the same sample was measured repeatedly |
| ☐ | ☒ | The statistical test(s) used AND whether they are one- or two-sided<br>*Only common tests should be described solely by name; describe more complex techniques in the Methods section.* |
| ☐ | ☒ | A description of all covariates tested |
| ☐ | ☒ | A description of any assumptions or corrections, such as tests of normality and adjustment for multiple comparisons |
| ☐ | ☒ | A full description of the statistical parameters including central tendency (e.g. means) or other basic estimates (e.g. regression coefficient) AND variation (e.g. standard deviation) or associated estimates of uncertainty (e.g. confidence intervals) |
| ☐ | ☒ | For null hypothesis testing, the test statistic (e.g. *F*, *t*, *r*) with confidence intervals, effect sizes, degrees of freedom and *P* value noted<br>*Give P values as exact values whenever suitable.* |
| ☒ | ☐ | For Bayesian analysis, information on the choice of priors and Markov chain Monte Carlo settings |
| ☐ | ☒ | For hierarchical and complex designs, identification of the appropriate level for tests and full reporting of outcomes |
| ☐ | ☒ | Estimates of effect sizes (e.g. Cohen's *d*, Pearson's *r*), indicating how they were calculated |

*Our web collection on statistics for biologists contains articles on many of the points above.*

## Software and code

Policy information about availability of computer code

| | |
|---|---|
| Data collection | A detailed description of patient cohorts, clinical characteristics, data collection and filtering is included in the Methods.<br>All data from single cell RNA sequencing and spatial transcriptomics analysis of human and mouse lung tissues were generated in-house in laboratory of Dr. Humam Kadara at MD Anderson Cancer Center and as described in the Methods section.<br>Cell Ranger Single Cell Software Suite (version 3.0.1) was used to process raw scRNA-seq data.<br>Spaceranger was used for analysis of ST data (version 1.3.0 for human ST data and version 2.0.0 for mouse ST data).<br>ImageScope software (Leica Microsystems, Inc.) was used to visualize H&E stained slides.<br>HALO software (Indica Labs, Albuquerque, NM) was used for pathological annotation of tissue slides.<br>Imaris software was used to analyze IF images obtained with confocal microscopy.<br>Data was recorded using Microsoft Excel v.2016.<br>Codes for analysis of scRNA-seq, WES, and ST data are available at Zenodo (https://doi.org/10.5281/zenodo.8280138) and GitHub (https://github.com/guangchunhan/LUAD_Code). |
| Data analysis | Single-cell RNA-seq data processing and quality control<br>Raw scRNA-seq data were pre-processed (demultiplex cellular barcodes, read alignment and generation of gene count matrix) using CellRanger Single Cell Software Suite (v.3.0.1) provided by 10X Genomics. For read alignment of human and mouse scRNA-seq data, human reference GRCh38 (hg38) and mouse reference GRCm38 (mm10) genomes were used, respectively. Detailed quality control metrics were generated and evaluated, and cells were carefully and rigorously filtered to obtain high-quality data for downstream analyses. In brief, for basic quality filtering, cells with low-complexity libraries (in which detected transcripts were aligned to <200 genes such as cell debris, empty drops and low-quality cells) were filtered out and excluded from subsequent analyses. Probable dying or apoptotic cells in which > 15% of transcripts derived from the mitochondrial genome were also excluded. For scRNA-seq analysis of Gprc5a-/-;SftpcCreER/+;RosaSun1GFP/+ mice, cells mice, cells with more than 500 detected genes or with a mitochondrial gene fraction that is greater or equal to 15% were filtered out using Seurat.<br><br>Doublet detection and removal, and batch effect evaluation and correction |

Probable doublets or multiplets were identified and carefully removed through a multi-step approach as described in previous studies. In brief, doublets or multiplets were identified based on library complexity, whereby cells with high-complexity libraries in which detected transcripts are aligned to >6,500 genes were removed and, based on cluster distribution and marker gene expression,whereby doublets or multiplets forming distinct clusters with hybrid expression features and/or exhibiting an aberrantly high gene count were also removed. Expression levels and proportions of canonical lineage-related marker genes in each identified cluster were carefully reviewed. Clusters co-expressing discrepant lineage markers were identified and removed. Doublets or multiplets were also identified using the doublet detection algorithm DoubletFinder. The proportion of expected doublets was estimated based on cell counts obtained before scRNA-seq library construction. Data normalization was then performed using Seurat. on the filtered gene–cellmatrix. Statistical assessment of possible batch effects was performed on non-malignant epithelial cells using the R package ROGUE, an entropy-based statistic, as described in previous studies, and Harmony was run with default parameters to remove batcheffects present in the PCA space.

Unsupervised clustering and subclustering analysis
The function FindVariableFeatures of Seurat was applied to identify highly variable genes for unsupervised cell clustering. PCA was performed on the top 2,000 highly variable genes. The elbow plot was generated with the ElbowPlot function of Seurat and, based on which, the number of significant principal components (PCs) was determined. The FindNeighbors function of Seurat was used to construct the shared nearest neighbour (SNN) graph based on unsupervised clustering performed using the Seurat function FindClusters. Multiple rounds of clustering and subclustering analyses were performed to identify major epithelial cell types and distinct cell transcriptional states. Dimensionality reduction and 2D visualization of cell clusters was performed using UMAP and the Seurat function RunUMAP. The number of PCs used to calculate the embedding was the same as that used for clustering. For analysis of human epithelial cells, ROGUE was used to quantify cellular transcriptional heterogeneity of each cluster. Subclustering analysis was then performed for low-purity clusters identified by ROGUE. Hierarchical clustering of major epithelial subsets was performed on the Harmony batch-corrected PCA dimension reduction space. For malignant cells, except for global UMAP visualization, downstream analyses, including identification of large-scale CNVs, inference of cancer cell differentiation states, quantification of meta-program expression, trajectory analysis and mutation analysis, were performed without Harmony batch correction. The hierarchical tree of human epithelial cell lineages was computed based on Euclidean distance using the Ward linkage method, and the dendrogram was generated using the R function plot.hc. For scRNA-seq analysis of Gprc5a-/- mice, the top-ranked ten PCs were selected using the elbowplot function. SNN graph construction was performed with resolution parameter = 0.4, and UMAP visualization was performed with default parameters. For scRNA-seq analysis of Gprc5a-/-;SftpcCreER/+;RosaSun1GFP/+ mice, the top-ranked 20 Harmony-corrected PCs were used for SNN graph construction, and unsupervised clustering was performed with resolution parameter = 0.4. UMAP visualization was performed with the RunUMAP function with min.dist = 0.1. Differentially expressed genes (DEGs) of clusters were identified using the FindAllMarkers function with FDR-adjusted P value < 0.05 and log (fold change) > 1.2.

Identification of malignant cells and mapping KRAS codon 12 mutations
Malignant cells were distinguished from non-malignant subsets based on information integrated from multiple sources as described in previous studies. The following strategies were used to identify malignant cells. (1) Cluster distribution: owing to the high degree of inter-patient tumour heterogeneity, malignant cells often exhibit distinct cluster distribution compared with normal epithelial cells. Although non-malignant cells derived from different patients are often clustered together by cell type, malignant cells from different patients probably form separate clusters. (2) CNVs: we applied inferCNV (v.1.3.2) to infer large-scale CNVs in each individual cell with T cells as the reference control. To quantify CNVs at the cell level, CNV scores were aggregated using a previously described strategy. In brief, arm-level CNV scores were computed based on the mean of the squares of CNV values across each chromosomal arm. Arm-level CNV scores were further aggregated across all chromosomal arms by calculating the arithmetic mean value of the arm-level scores using the R function mean. (3) Marker gene expression: expression of lung epithelial lineage-specific genes and LUAD-related oncogenes were determined in epithelial cell clusters. (4) Cell-level expression of KRASG12D mutations: as we previously described, BAM files were queried for KRASG12D mutant alleles, which were then mapped to specific cells. KRASG12D mutations,along with cluster distribution, marker gene expression and inferred CNVs as described above, were used to distinguish malignant cells from non-malignant cells. Following clustering of malignant cells from all patients, an absence of malignant cells that were identified from P12 or P16 was noted. This can be possibly attributed to the low number of epithelial cells captured in tumour samples from these patients (Supplementary Table 2).

Mapping KRAS codon 12 mutations
To map somatic KRASG12D mutations at single-cell resolution, alignment records were extracted from the corresponding BAM files using mutation location information. Unique mapping alignments (MAPQ = 255) labelled as either PCR duplication or secondary mapping were filtered out. The resulting somatic variant carrying reads were evaluated using Integrative Genomics Viewer (IGV) and the CBtags were used to identify cell identities of mutation-carrying reads. To estimate the VAF of KRASG12D mutation and cell fraction of KRASG12D-carrying cells within malignant and non-malignant epithelial cell subpopulations (for example, malignant cells from all LUADs, malignant cells from KM-LUADs, KACs from KM-LUADs), reads were first extracted based on their unique cell barcodes and BAM files were generated for each subpopulation using samtools (v.1.15). Mutations were then visualized using IGV, and VAFs were calculated by dividing the number of KRASG12D -carrying reads by the total number of uniquely aligned reads for each subpopulation. A similar approach was used to visualize KRASG12C-carrying reads and to calculate the VAF of KRASG12C in KACs of normal tissues from KM-LUAD cases. To calculate the mutation-carrying cell fraction, extracted reads were mapped to the KRASG12D locus from BAM files using AlignmentFile and fetch functions in pysam package. Extracted reads were further filtered using the'Duplicate' and 'Quality' tags to remove PCR duplicates and low-quality mappings. The number of reads with or without KRASG12D mutation in each cell was summarized using the CB tag in read barcodes. Mutation-carrying cell fractions were then calculated as the ratio of the number of cells with at least one KRASG12D read over the number of cells with at least one high-quality read mapped to the locus.

PCA analysis of malignant cells and quantification of transcriptome similarity
Raw unique molecular identifier counts of identified malignant cells were log-normalized and used for PCA analysis using Seurat (RunPCA function). PCA dimension reduction data were extracted using the Embeddings function. The top three most highly ranked PCs were exported for visualization using JMP (v.15). 3D scatterplots of PCA data were generated using the scatterplot 3D tool in JMP (v.15). Bhattacharyya distances were calculated using the bhattacharyya.dist function from the R package fpc (v.2.2-9). The top 25 highly ranked PCs were used for both patient-level and cell-level distance calculations. For Bhattacharyya distance quantification at the cell level,100 cells were randomly sampled for each patient group defined by driver mutations (for example, KM-LUADs). The random sampling process was repeated 100 times, and pairwise Bhattacharyya distances were then calculated between patient groups. Differences in Bhattacharyya distances between patient groups were tested using Wilcoxon rank-sum tests, and boxplots were generated using the geom_boxplot function from the R package ggplot2 (v.3.2.0).

Determination of non-malignant cell types and states
Non-malignant cell types and states were determined based on unsupervised clustering analysis following batch effect correction using

Harmony. Two rounds of clustering analysis were performed on non-malignant cells to identify major cell types and cell transcriptional states within major cell types. Clustering and UMAP visualization of human normal epithelial cells (Extended Data Fig. 1a) were performed using Seurat with default parameters. Specifically, the parameters k.param = 20 and resolution = 0.4 were used forSNN graph construction and cluster identification, respectively. UMAP visualization was performed with default parameters (min.dist = 0.3). For clustering analysis of airway and alveolar epithelial cells, the RunPCA function was used to determine the most contributing top PCs for each subpopulation and similar clustering parameters (k.param = 20 and resolution = 0.4) were used for SNN graph construction and cluster identification. UMAP plots were generated with min.dist = 0.3 using the RunUMAP function in Seurat. Density plots of alveolar intermediate cells were generated using the stat_densit_2d function in the R package ggplot2 (v.3.3.5) with the first two UMAP dimension reduction data as the input. DEGs for each cluster were identified using the FindMarkers function in Seurat with a FDR-adjusted P < 0.05 and a fold change cut-off > 1.2. Canonical epithelial marker genes from previously published studies by our group and others were used to annotate normal epithelial cell types and states. Bubble plots were generated for select DEGs and canonical markers to define AT1 cells (AGER1+ ETV5+ PDPN+), AT2 cells (SFTPB, SFTPC, ETV5+), SCGB1A1+SFTPC+ dual-positive cells, AICs (AGER1+ETV5+PDPN+ and SFTPB+SFTPC+), club and secretory cells (SCGB1A1 +SCGB3A1+CYP2F1+), basal cells (KRT5+TP63+), ciliated cells (CAPS+PIFO+FOXJ1+), ionocytes (ASCL3+FOXI+), neuroendocrine cells (CALCA +ASCL1+) and tuft cells (ASCL2+MGST2+PTGS1+). KACs were identified by unsupervised clustering of AICs and defined based on previously reported marker genes, including significant upregulation of the following genes relative to other alveolar cells: KRT8, CLDN4, PLAUR, CDKN1A and CDKN2A.

Scoring of curated gene signatures
Genes in previously defined ITH MPs were downloaded from the original study. Among a total of 41 consensus ITH MPs identified, MPs with unassigned functional annotations (unassigned MPs 38–41; n = 4), neural and haematopoietic lineage-specific MPs (MPs 25–29, MPs 33–37; n = 10) and cell-type-specific MPs irrelevant to LUAD (MPs 22–24 secreted/cilia, MP 32 skin-pigmentation; n = 4) were filtered out, resulting in 23 MPs that closely correlated with hallmarks of cancer and which were used for further analysis. Signature scores were computed using the AddModuleScore function in Seurat as previously described. The KRAS signature used in this study was derived by calculating DEGs between the KRAS mutant malignant-cell-enriched cluster and other malignant cells (FDR-adjusted P value < 0.05, log(fold change) > 1.2; Extended Data Fig. 2i). Human and mouse KAC signatures and the human 'other AIC'signature were derived by calculating DEGs using FindAllMarkers among alveolar cells (FDR-adjusted P value < 0.05, log(fold change) > 1.2). Mouse genes in the p53 pathway were downloaded from the Molecular Signature Database (MSigDB; https://www.gsea-msigdb.org/gsea/msigdb/mouse/geneset/HALLMARK_P53_PATHWAY; MM3896). Signature scores for KACs, other AICs, KRAS andp53 were calculated using the AddModuleScore function in Seurat.

Analysis of alveolar cell differentiation states and trajectories
Analysis of differentiation trajectories of lung alveolar and malignant cells was performed using Monocle 2 by inferring the pseudotemporal ordering of cells according to their transcriptome similarity. Monocle 2 analysis of malignant cells from P14 was performed using default parameters with the detectGenes function. Detected genes were further required to be expressed by at least 50 cells. For construction of the differentiation trajectory of lineage-labelled epithelial cells (GFP), the top 150 DEGs (FDR-adjusted P value < 0.05, log(fold change) > 1.5, expressed in 50 cells or more) ranked by fold-change of each cell population from NNK-treated samples were used for ordering cells with the setOrderingFilter function. Trajectories were generated using the reduceDimension function with the method set to 'DDRTree'. Trajectory roots were selected based on the following criteria: (1) inferred pseudotemporal gradient; (2) CytoTRACE score prediction; and (3) careful manual review of the DEGs along the trajectory. To depict expression dynamics of ITH MPs, ITH MP scores were plotted along the pseudotime axis and smoothed lines were generated using the smoother tool in JMP Pro(v.15). Using the raw counts without normalization as input, CytoTRACE was applied with default parameters to infer cellular differentiation states to complement trajectory analysis and further understand cellular differentiation hierarchies. The normalmixEM function from the R package mixtools was used to determine the CytoTRACE score threshold in AICs with k = 2. A final threshold of 0.58was used to dichotomize AICs into high-differentiation and low-differentiation groups. The Wasserstein distance metric was applied usingR package transport (v.0.13) to quantify the variability of distribution of CytoTRACE scores. The function wasserstein1d was used tocalculate the distance between the distribution of actual CytoTRACE scores of one patient and the distribution of simulated data with identical mean and standard deviation. The robustness of Monocle 2-based pseudotemporal ordering prediction was validated by independent pseudotime prediction tools including Palantir, Slingshot and Cellrank. Slingshot (v.2.6.0) pseudotime prediction was performed using slingshot function with reduceDim parameter set to 'PCA' and other parameters set to defaults. Cellrank prediction was performed using the CytoTRACEKernel function with default parameters from Cellrank python package (v.1.5.1). Palantir prediction was performed using Palantir python package (v.1.0.1). A diffusion map was generated using run_diffusion_maps function withn_components = 5. Palantir prediction was generated using run_palantir function with num_waypoints = 500 and other parameters set to defaults. Inferred pseudotime by the three independent methods was then integrated with that generated by Monocle 2 for each single cell,followed by pairwise mapping and correlation analysis. Cell density plots were generated using Contour tool in JMP (v.15) with n = 10 gradient levels and contour type parameter set to 'Nonpar Density'. To assess the pseudotime prediction consistency between Monocle 2and the three independent methods, Spearman's correlation coefficients were calculated and statistically tested using cor.test function in R.

ST data generation and analysis
ST profiling of formalin fixation and paraffin-embedding (FFPE) tissues of P14 LUAD sample and of three lung tissues from two Gprc5a-/- mice was performed using the Visium platform from 10X Genomics according to the manufacturer's recommendations and as previously reported. P14 FFPE tissues were collected from areas adjacent to the tissues analysed by scRNA-seq. Regions of interest per tissue or sample,each comprising a 6.5 × 6.5 mm capture area, were selected based on careful annotation of H&E-stained slides that were digitally acquired using an Aperio ScanScope Turbo slide scanner (Leica Microsystems). HALO software (Indica Labs) was used for pathological annotation (tumour areas, blood vessels, bronchioles, lymphoid cell aggregates, macrophages, muscle tissue, normal parenchyma and reactive pneumocytes) of H&E histology images. Spot-level histopathological annotation and visualization was generated using loupebrowser (v.6.3.0). In brief, cloupe files generated from Space Ranger were loaded into the loupe browser. Visualization of annotation was then generated in svg formats using the export plot tool. ST RNA-seq libraries were generated according to the manufacturer'sinstructions, each with up to about 3,600 uniquely barcoded spots. Libraries were sequenced on an Illumina NovaSeq 6000 platform to achieve a depth of at least 50,000 mean read pairs per spot and at least 2,000 median genes per spot.
Demultiplexed raw sequencing data were aligned, and gene level expression quantification was generated with analysis pipelines as previously described. In brief, demultiplexed clean reads were aligned against the UCSC human GRCh38 (hg38) or the GRCm38(mm10) mouse reference genomes by Spaceranger (v.1.3.0 for human ST data and v.2.0.0 for mouse ST data) and using default settings. Generated ST gene expression count matrices were then analysed using Seurat (v.4.1.0) to perform unsupervised clustering analysis. Using default parameters, the top-ranked 30 PCA components were used for SNN graph construction and clustering and for UMAP low-dimension space embedding with default parameters. UMAP analysis was performed using the RunUMAP function. The SpatialDimPlotfunction was used to visualize unsupervised clustering. The R package inferCNV was used for copy number analysis. Reference spots used in CNV analysis were selected on the basis of careful review of cluster marker genes using the DotPlot function from Seurat and inspection of pathological annotation. CNV scores were

calculated by computing the standard deviations of CNVs inferred across 22autosomes. Loupe browser (v.6.3.0) was used for visualization of pathological annotation results. Expression levels of genes of interest (for example, KRT8) as well as signatures of interest (for example, KAC and KRAS) were measured and directly annotated on histology images with pixel-level resolution using the TESLA (v.1.2.2) machine learning framework (https://github.com/jianhuupenn/TESLA). TESLA can compute superpixel-level gene expression and detect unique structures within and surrounding tumours by integrating information from high-resolution histology images. The annotation and visualize_annotation functions were used to annotate regions with high signature signals. KRT8, PLAUR, CLDN4, CDKN1A and CDKN2A were used for 'KAC markers' signature annotation in the human ST analysis. For mouse ST data, Krt8, Plaur, Cldn4, Cdkn1a and Cdkn2a were used for 'KAC signature' annotation. Gene level expression visualization of Krt8 and Plaur was generated using the scatter function from scanpy (v.1.9.1). Deconvolution analysis was conducted using CytoSPACE (https://github.com/digitalcytometry/cytospace). Annotated scRNA-seq data were first transformed intoa compatible format using function generate_cytospace_from_scRNA_seurat_object. Visium spatial data were prepared using the functiongenerate_cytospace_from_ST_seurat_object. Deconvolution was performed using CytoSpace function (v.1.0.4) with default parameters. To determine neighbouring cell composition for a specific cell population in Visium data, CytoSPACE was first applied to annotate every spot with the most probable cell type. Neighbouring spots were defined as the six spots surrounding each spot and, accordingly, the neighbouring cell composition for specific cell types were computed. Trajectory construction of ST data was performed using Monocle 2 with the DDRTree method using DEGs with FDR-adjusted P value < 0.05.

Bulk DNA extraction and WES
Total DNA was isolated from homogenized cryosections of human lung tissues and, when available, from frozen peripheral blood mononuclear cells (PBMCs) using a Qiagen AllPrep mini kit (80204) or a DNeasy Blood and Tissue kit (69504), respectively (both from Qiagen) according to the manufacturer's recommendations. Qubit 4 Fluorometer (Thermo Fisher Scientific) was used for measurement of DNA yield. TWIST-WES was performed on a NovaSeq 6000 platform at a depth of 200x for tumour samples and 100x for NL and PBMCs to analyse recurrent driver mutations and using either PBMCs or distant NL tissues when blood draw was not consented, as germline control. WES data were processed and mapped to the human reference genome, and somatic mutations were identified and annotated as previously described with further filtration steps. In brief, only MuTect calls marked as 'KEEP' were selected and taken into the next step. Mutations with a low VAF (<0.02) or low alt allele read coverage (<4) were removed. Then, common variants reported by ExAc (the Exome Aggregation Consortium, http://exac.broadinstitute.org), Phase-3 1000 Genome Project (http://phase3browser.1000genomes.org/Homo_sapiens/Info/Index) or the NHLBI GO Exome Sequencing Project (ESP6500) (http://evs.gs.washington.edu/EVS/) with minor allele frequencies greater than 0.5% were further removed. Intronic mutations, mutations at 3' or 5' UTR or UTR-flanking regions, and silent mutations were also removed. The mutation load in each tumour was calculated as the number of nonsynonymous somatic mutations (nonsense, missense, splicing, stop gain, stop loss substitutions as well as frameshift insertions and deletions).

Survival analysis
Analysis of OS in the TCGA LUAD and PROSPECT cohorts was performed as previously described. KRAS mutation status in TCGA LUAD samples was downloaded from cBioPortal (https://www.cbioportal.org, study ID: luad_tcga_pan_can_atlas_2018). For TCGA dataset, clinical data were downloaded from the PanCanAtlas study. The logrank test and Kaplan–Meier methods were used to calculate P values between groups and to generate survival curves, respectively. Statistical significance testing for all survival analyses was two-sided. To control for multiple hypothesis testing, Benjamini–Hochberg method was applied to correct P values, and FDR q values were calculated where applicable. Results were considered significant at P value or FDR q value of <0.05. Multivariate survival analysis was performed using a Cox proportional hazards regression model that calculated the hazard ratio, the 95% confidence interval and P values when using pathologic stage, age, KAC and 'other AIC' signatures as covariables.

Analysis of public datasets
Publicly available datasets were obtained from the Gene Expression Omnibus (GEO) database (https://www.ncbi.nlm.nih.gov/geo/) under accession numbers GSE149813, GSE154989, GSE150263, GSE102511 and GSE219124. Details of the studies analysed are as follows: GSE149813 investigated single lung cells from KrasLSL-G12D;LSL-YFP mice with Ad5CMV-Cre infection; GSE154989 studied AT2 lineage-labelled cells from lungs of KrasLSL-G12D/+;Rosa26LSL-tdTomato/+ mice. Gene expression count matrices of dataset interrogating KrasG12D-driven mouse model from GSE149813 were pre-processed using Seurat following the same filtering steps in that original report. For the GSE154989 dataset, cells used for analysis were the ones labelled as "PASSED_QC" in supplementary table S7 in that study. For the GSE149813 dataset, cells with >500 median number of genes detected and <10% fraction of mitochondrial genome derived reads, and according to the pre-processing methods described in their original report, were retained for analysis. Cells with >7,500 number of genes detected were further filtered to remove potential doublets or multiplets, resulting in 8,304 cells in total for downstream analysis. Both datasets were integrated with mouse cell data generated in this study using Harmony with default parameters settings. The top ranked 20 Harmony-corrected PCs were used for clustering with the FindClusters function using resolution = 0.4. UMAP dimension reduction embedding was performed using the RunUMAP function with the same set of Harmony-corrected PCs. Gene expression levels and frequencies of representative cluster marker genes were visualized using DotPlot function from Seurat. The KAC signature score was calculated using the AddModuleScore function from Seurat. The mouse KAC signature was also studied in human AT2 cells with and without inducible KRASG12D (dataset GSE150263). Cell filtration criteria described in the original report were followed to filter out potential dead cells and doublets (number of detected genes > 800 and the percent of mitochondrial gene reads fraction < 25%). The 20top-ranked PCs were used for clustering using the FindClusters function with resolution = 0.1. UMAP dimension reduction embeddings were computed using the same SNN graph. The KAC signature score was calculated using AddModuleScore function from Seurat package.
The bulk RNA-seq dataset GSE102511 was a previously published dataset by our group and comprised normal lung tissues, precursor AAHs and matched LUADs (n = 15, each). The previously published bulk RNA-seq data GSE219124 were generated on cancer stem cell and stem cell-like progenitor cells, in the form of spheres, and their parental MDA-F471 counterparts (a cell line we had developed and cultured from a KM-LUAD of an NNK-exposed Gprc5a-/- mouse). To interrogate the association of KACs with tumour formation, gene expression matrices of bulk RNA-seq data GSE102511 (TPM count matrix) and GSE219124 (FPKM count matrix) were extracted and used for quantification of KAC signature expression using MCPcounter (v.1.2.0) R package. Heatmaps were generated using pheatmap (v.1.0.12) R package.
Mouse KACs from this study were compared to mouse Krt8 transitional cells involved in alveolar regeneration post-acute lung injury from a previous study. Overlapping marker genes between mouse KACs and the previously reported Krt8 transitional cells were statistically evaluated using the ggvenn (v.0.1.9) R package using the top-ranked 50 marker genes based on fold change from each study.

Histopathological and IF analysis of mouse lung tissues
Lungs of Gprc5a-/- mice (n= 2 per treatment and time point) were inflated with formalin by gravity drip inflation, excised, examined for lung surface lesions by macroscopic observation and processed for FFPE, sectioning and H&E staining. Stained slides were digitally scanned using an Aperio ScanScope Turbo slide scanner (Leica Microsystems) at ×200 magnification, and visualized using ImageScope software (Leica Microsystems). Unstained lung tissue sections were obtained for IF analysis of LAMP3 (clone 391005, Synaptic Systems), KRT8 (TROMA-I clone from the University of Iowa DSHB) and PDPN (clone 8.1.1, from the University of Iowa DSHB). Lung FFPE tissue samples were obtained in the

same manner from Gprc5a-/-;SftpcCreER/+;RosaSun1GFP/+ mice at 3 months after exposure to saline or NNK (n = 2 mice per condition) and following injection with tamoxifen.

Tissue sections were obtained for H&E staining and assessment of tumour development, and unstained sections were used for IF analysis using antibodies against GFP (AB13970, Abcam, 1:5000), LAMP3 (391005, Synaptic Systems, 1:10,000), KRT8 (TROMA-I,University of Iowa Developmental Studies Hybridoma Bank, 1:100), PDPN (clone 8.1.1, University of Iowa Developmental Studies Hybridoma Bank, 1:100), claudin 4 (ZMD.306, Invitrogen, 1:250), and PRKCDBP (cavin 3, Proteintech, 1:250). Slides werethen stained with fluorophore-conjugated secondary antibodies and 4',6'-diamidino-2-phenylindole (DAPI). Sections were mounted with Aquapolymount (18606, Polysciences), cover slipped, imaged using an Andor Revolution XDi WD spinning disk confocal microscope and analysed using Imaris software (Oxford Instruments).

Formalin-inflated lung lobes from Krt8-creER;RosatdT/+ mice were cryoprotected in 20% sucrose in PBS containing 10% OCT compound (4583, Tissue-Tek) overnight on a rocker at 4 °C and embedded in OCT. The next day, 10 micrometer cryosections were blocked in PBS with 0.3% Triton X-100 and 5% normal donkey serum (017-000-121, Jackson ImmunoResearch) and incubated overnight in a humidified chamber at4 °C with primary antibodies diluted in PBS with 0.3% Triton X-100 and raised against NKX2-1 (sc-13040, Santa Cruz, 1:1000),LAMP3 (same as above) and KRT8 (same as above). The next morning, sections were washed followed by incubation with secondary antibodies (Jackson ImmunoResearch) and DAPI. Slides were then washed, cover slipped as described above and imaged using a NikonA1plus confocal microscope. Cell counter ImageJ plugin was used to count tdT+ cells within lesions and cells in normal-appearing areas,namely: AT2 cells (LAMP3+), tdT+ AT2 cells (tdT+LAMP3+), AT1 cells (LAMP3+NKX2-1+, avoiding noticeable airways) and tdT+ AT1 cells (tdT+NKX2-1+LAMP3+). Percentages of tdT+LAMP3+ and tdT+NKX2-1+LAMP3+ cells out of total tdT+ cells were computed. Counts were averages of triplicate images taken at ×20 magnification for each time point. The percent regional surface area covered by tdT cells in normal-appearing regions was estimated by examining the tdT expression across entire lobe sections for each replicate.

## 3D culture and analysis of AT2-derived organoids

Gprc5a-/-;SftpcCreER/+;RosaSun1GFP/+ were treated with NNK or saline and tamoxifen as described above, and they were euthanized at EOE (4 saline-treated and 5 NNK-treated mice) or at 3 months after exposure (10 saline-treated and 13 NNK-treated mice). Lungs were collected, dissociated into single cells (see mouse single-cell derivation in the Methods section 'Single-cell isolation from tissuesamples'), and live (Sytox Blue-negative) GFP+ single cells were collected by flow cytometry using a FACS Aria I instrument as previously described. GFP+ AT2 cells from NNK-treated or saline-treated groups were immediately washed and resuspended at a concentration of 5,000 cells per 50 microliters of 3D medium (F12 medium supplemented with insulin, transferrin and selenium, 10% FBS,penicillin–streptomycin and l-glutamine). GFP+ cells were mixed at a 1:1 ratio (by volume) with 50,000 mouse endothelial cells (collected from mouse lungs by CD31 selection and expanded in vitro as previously described and resuspended in 50 microliters of Geltrex reduced growth factor basement membrane matrix (A1413301, Gibco). Next, 100 microliters of 1:1 GFP+:endothelial cell mixture was plated on Transwell inserts with 0.4 micrometer pores and allowed to solidify for 30 min in a humidified CO2 incubator (EOE: n = 3 wells per condition; 3 months after exposure: n = 4 wells for saline-derived organoids and n = 12 wells for NNK-derived organoids). Each well was then supplemented with 3D medium containing ROCK inhibitor (Y-27632, Millipore) and recombinant mouse FGF-10 (6224-FG, R&D Systems), and plates were incubated at 37 °C in a humidified CO2 incubator. Wells were replenished with 3D medium every other day. For GFP+ organoids derived from mice exposed to NNK, 200 nM KRAS(G12D)-specific inhibitor MRTX1133 or DMSO vehicle was added to the medium and replenished 3 times a week (n = 6 wells per condition). Organoids were monitored and analysed twice a week using an EVOS M7000 imaging system (Thermo Fisher Scientific), whereby the numbers and sizes of organoids greater than 100 micrometers in diameter were recorded. At end point, 3D organoids were collected from the basement membrane matrix using Gentle Cell Dissociation reagent(100-0485, StemCell Technologies), fixed with 4% paraformaldehyde, permeabilized, blocked and stained overnight at 4 °C with a mixture of IF primary antibodies raised against LAMP3, GFP, KRT8 and cavin 3. The next day, organoids were washed and stained with fluorophore-conjugated secondary antibodies overnight at 4 °C while being protected from light. Organoids were washed and stained with DAPI nuclear stain for 30 min, after which they were collected in Aqua-Poly/Mount (18606-20, Polysciences) and transferred to slides. Images of organoids were captured using an Andor Revolution XDi WD spinning disk confocal microscope and analysed using Imaris software (Oxford Instruments).

## 2D viability assays

Mouse mycoplasma-free LUAD cell lines LKR13 (mutant KrasG12D-driven) and MDA-F471 (Gprc5a-/- and KrasG12D mutant) were plated on 96-well plates (10 cells per well) and grown in DMEM (Gibco) supplemented with 10% FBS, 1% antibiotic antimycotic solution (A5955, Sigma-Aldrich) and 1% l-glutamine (G7513, Sigma-Aldrich). The next day, cells were cultured for up to 4 days with medium containing 0.5% FBS, 0.5% FBS with 50 ng/ml epidermal growth factor (EGF) (E5160, Sigma-Aldrich), or 0.5% FBS with EGF and varying concentrations of MRTX1133 (Mirati Therapeutics). alamarBlue Cell Viability reagent (25 microliters; DAL1025, ThermoFisher)was added to each well. At 4 days after treatment, viability was assessed by fluorescence spectrophotometry at 570 nm (and 600 nm as a reference). For the wells showing net positive absorbances relative to blank wells (at least 3 wells per cell line and condition), the percent differences in reduction between treated and control wells were calculated.

## Western blot analysis

LKR13 and MDA-F471 cells were plated in 6-well plates (10 cells per well) and grown under different conditions as described above. Protein lysates were extracted at 3 h after treatment and analysed by western blotting following overnight incubation with antibodies tot he following primary proteins: vinculin (E1E9V, rabbit, Cell Signaling Technology, 13901; 1:1,000); phosphorylated p44/42 MAPK(ERK1/2, rabbit, Cell Signaling Technology, 9101; 1:2,000); phosphorylated S6 ribosomal protein (Ser 235/236, rabbit, Cell Signaling Technology, 4858; 1:2,000); p44/42 MAPK (ERK1/2, rabbit, Cell Signaling Technology, 9102; 1:2,000); or S6 (E.573.4, rabbit, Invitrogen, MA5-15164; 1:1,000). This was followed by 1 h of incubation with diluted secondary antibody (1706515 goat anti-rabbit IgG-HRP conjugate, Bio-Rad). Protein lysates from each cell line were analysed on multiple gels (four per cell line) with Precision Plus Protein Dual Color Standard (1610394, Bio-Rad) as the ladder and blotted to membranes to separately probe for phosphorylated and total forms of the same proteins, which have highly similar molecular weights (using phospho-specific antibodies or antibodies targeting total version of same protein). Vinculin protein levels were evaluated as loading control on each of the blots. Four blots (phospho-ERK, total ERK, phospho-S6 and total S6) for each of LKR13 and MDA-F471 are shown in Supplementary Fig. 9 each with its own analysis of equal protein loading (vinculin blot) and whereby only the ones indicated with green rectangles are presented in Extended Data Fig. 12c. Membranes were cut horizontally using molecular weight marker as a guide, and cut membranes were incubated with the specified antibodies (see Supplementary Fig. 9 for site of cutting and for overlay of colorimetric and chemiluminescent images of the same blot todisplay ladder and the analysed protein, respectively). Blots were imaged using the ChemiDoc Touch Imaging System (Bio-Rad) with Chemiluminescence and Colorimetric (for protein ladder) applications and auto expose or manual settings.

## Statistical analyses

In addition to the algorithms and statistical analyses described above, all other basic statistical analyses were performed in the R statistical environment (v.4.0.0). The Kruskal–Wallis H -test was used to compare variables of interests across three or more groups. Wilcoxon rank-sum

test was used for paired comparisons among matched samples from the same patients. Wilcoxon rank-sum test was used to compare other continuous variables such as gene expression levels and signature scores between groups. Spearman's correlation coefficient was calculated to assess associations between two continuous variables (for example, cellular proportions and gene signature scores). Fisher'sexact test was used to identify differences in frequencies of groups based on two categorical variables. Ordinal logistic regression was performed using the polr function in the built-in R package MASS (v.7.3). Benjamin–Hochberg method was used to control for multiple hypothesis testing. All statistical tests performed in this study were two-sided. Results were considered significant at P values or FDR q values < 0.05. When a P value reported by R was smaller than 2.2e-16, it was reported as $P < 2.2 \times 10\text{-}16$.

For manuscripts utilizing custom algorithms or software that are central to the research but not yet described in published literature, software must be made available to editors and reviewers. We strongly encourage code deposition in a community repository (e.g. GitHub). See the Nature Portfolio guidelines for submitting code & software for further information.

## Data

Policy information about availability of data

All manuscripts must include a data availability statement. This statement should provide the following information, where applicable:
- Accession codes, unique identifiers, or web links for publicly available datasets
- A description of any restrictions on data availability
- For clinical datasets or third party data, please ensure that the statement adheres to our policy

Sequencing data for P1 - P5 were previously generated and deposited in the European Genome–phenome Archive (EGA) under the accession number EGAS0000100502115. Human scRNA-seq (P6 – P16) and ST data generated in this study have been deposited in EGA under the same accession number (EGAS00001005021). Mouse scRNA-seq and ST data generated in this study have been deposited in NCBI under GEO accession number GSE222901. Source data are provided with this paper.

# Field-specific reporting

Please select the one below that is the best fit for your research. If you are not sure, read the appropriate sections before making your selection.

☒ Life sciences  ☐ Behavioural & social sciences  ☐ Ecological, evolutionary & environmental sciences

For a reference copy of the document with all sections, see nature.com/documents/nr-reporting-summary-flat.pdf

# Life sciences study design

All studies must disclose on these points even when the disclosure is negative.

| | |
|---|---|
| Sample size | No statistical methods were used to predetermine sample size. We performed single-cell RNA sequencing (scRNA-seq) on n = 63 tissue samples from n = 16 patients with early-stage lung adenocarcinomas, and performed spatial transcriptomics analysis on tumour sample from one patient in this cohort. We also performed whole exome sequencing on tumour and normal lung tissues and, when available, peripheral blood mononuclear cells from the same cohort (n = 64). We also analyzed by scRNA-seq lungs of n = 16 Gprc5a-/- mice at end of exposure or at 7 months post-exposure to tobacco carcinogen or control saline (4 mice each). Additionally, we analyzed three lung tissue of 2 mice from 7 months post-exposure to tobacco carcinogen by spatial transcriptomics analysis. Furthermore, we performed scRNA-seq on GFP+ cells isolated (by sorting) from n = 4 AT2 reporter mice at 3 months post-exposure to tobacco carcinogen or saline (2 mice per group). We also stained by IF, lungs of Gprc5a-/-; Krt8-creER;RosatdT/+ reporter mice exposed to NNK or control saline and tamoxifen for activation of Cre recombinase in Krt8-expressing cells, and which we analyzed immediately following labelling with tamoxifen (EOE, NNK n = 3, saline n = 2) or at 8-12 weeks post NNK (follow-up after EOE, n = 3). |
| Data exclusions | No data were excluded from this study |
| Replication | The findings of the KAC signature were observed in human scRNAseq cohort and then validated in 1) TCGA LUAD cohort with matched normal controls (n = 52), 2) the PROSPECT cohort (n = 45), 3) two published scRNA-seq datasets from studies interrogating KrasG12D-driven mouse models by scRNA-seq, 4) a dataset human AT2 cells with and without inducible KRASG12D expression, and 5) our previously reported bulk RNA-sequencing expression dataset comprised of cancer stem cell/stem cell-like progenitor cells, in the form of 3D spheres, and their parental MDA-F471 counterparts (grown in 2D).<br><br>For mouse scRNA-seq analysis experiments, we analyzed n = 4 Gprc5a-/- mice per treatment group and timepoint (16 mice in total), and n = 2 AT2 reporter mice per group (4 mice in total). The presence of KACs and the expression of KAC markers and/or KAC signature were validated in both human and mouse tissues by IF (n = 3 human LUAD cases, n = 4 for mouse) and by spatial transcriptomics analysis (n = 1 human LUAD tissue, n = 3 lungs from 2 mice). We also confirmed the presence of KACs in organoids derived from AT2 reporter mice at two timepoints, at the end of exposure (n = 4 or 5 mice at EOE to saline or NNK) or at 3 months post-exposure (n = 10 or 13 mice at 3 months post-saline or NNK, respectively),  and in organoids with or without targeted KRAS inhibition in vitro. We also analyzed by IF Krt8 expression in tumour-bearing lungs of Krt8 reporter mice exposed to tobacco carcinogen (n = 6 mice, 17 total tumours), and compared to saline exposed animals (n = 2). All IF staining (tissue and organoids) were repeated at least 3 times with similar results. In vitro viability and western blot analyses were repeated two and three times, respectively, with similar results. |
| Randomization | This study does not involve samples from human subjects in clinical trials. This study utilized de-identified genomic and clinical data derived from patients undergoing surgical lung resection, and thus randomization does not apply to this study. In all animal studies, sex- and age-matched animals were randomized to treatment groups and analysis of mouse data was blinded. |
| Blinding | This study does not involve human subjects in clinical trials, blinding does not apply to this study. |

# Reporting for specific materials, systems and methods

We require information from authors about some types of materials, experimental systems and methods used in many studies. Here, indicate whether each material, system or method listed is relevant to your study. If you are not sure if a list item applies to your research, read the appropriate section before selecting a response.

## Materials & experimental systems

| n/a | Involved in the study |
|-----|----------------------|
| ☐ | ☒ Antibodies |
| ☐ | ☒ Eukaryotic cell lines |
| ☒ | ☐ Palaeontology and archaeology |
| ☐ | ☒ Animals and other organisms |
| ☒ | ☐ Human research participants |
| ☒ | ☐ Clinical data |
| ☒ | ☐ Dual use research of concern |

## Methods

| n/a | Involved in the study |
|-----|----------------------|
| ☒ | ☐ ChIP-seq |
| ☒ | ☐ Flow cytometry |
| ☒ | ☐ MRI-based neuroimaging |

## Antibodies

| | |
|---|---|
| Antibodies used | For sorting of epithelial cells: Human: EPCAM-PE (347198, BD Biosciences); Mouse: CD45-PE/Cy7 (BioLegend 103114), ICAM2-A647 (Life Technologies A15452), EPCAM-BV421 (Biolegend 118225), and ECAD-A488 (eBioscience 53-3249-80). <br><br> For immunofluorescence:  LAMP3 (clone 391005, Synaptic Systems), KRT8 (TROMA-I clone from the University of Iowa DSHB), CLDN4 (ZMD.306 from Invitrogen), PRKCDBP (cavin3, Proteintech), GFP (Abcam, AB13970), PDPN (clone 8.1.1, University of Iowa DSHB), and NKX2-1 (sc-13040, Santa Cruz). <br><br> For digital spatial imaging: : Claudin 4 (clone 3E2C1, AF594, LSBio, catalog number LS-C354893 concentration 0.5 µg/ml), Keratin 8 (clone EP1628Y, AF647, Abcam, catalog number ab192468, concentration 0.25 µg/ml), PanCk (clone AE1/AE3, AF532, concentration 0.25 µg/ml, from GeoMx Solid Tumour Morp Kit HsP, 121300301, Novus Biologicals, Littleton, CO). |
| Validation | All antibodies were acquired from commercial vendors and used according the manufacturer's instructions. Antibody optimizations for digital spatial imaging of human tissues were performed with different dilutions using colorectal carcinoma and lung adenocarcinoma tissue. |

## Eukaryotic cell lines

Policy information about cell lines

| | |
|---|---|
| Cell line source(s) | mouse |
| Authentication | MDA-F471 were established in house. We had developed and cultured MDA-F471 cells from a KM-LUAD of an NNK-exposed Gprc5a-/- mouse. LKR13 is a cell line that was previously derived from a tumour in the KrasLA1 model of KrasG12D-mutant LUAD (PMID: 15833854) and was a kind donation from Dr. Jonathan Kurie. |
| Mycoplasma contamination | These cells are routinely tested and validated to be free of mycoplasma contamination. Latest test was in July 2023 and is available upon request. |
| Commonly misidentified lines (See ICLAC register) | Cell lines used are not among the commonly misidentified cell lines. |

## Animals and other organisms

Policy information about studies involving animals; ARRIVE guidelines recommended for reporting animal research

| | |
|---|---|
| Laboratory animals | All animals were Mus Musclus and with B6/SV129 mixed background. Krt8-creER (stock number 017947) and RosatdT/+  (Ai14; stock number 007914) mice were obtained from the Jackson Laboratory. SftpcCreER/+; RosaSun1GFP/+ mice were obtained from Dr. Harold Chapman (University of California, San Francisco). We studied three strains: Gprc5a-/- mice, Gprc5a-/-; SftpcCreER/+; RosaSun1GFP/+, and Gprc5a-/-;Krt8-creER;RosatdT/+ mice. Mice used in experiments were 8 weeks old and of both sexes. |
| Wild animals | This study did not involve wild animals. |
| Field-collected samples | This study did not involve animals collected from the field. |
| Ethics oversight | Mouse handling and care followed the NIH Guide for Care and Use of Laboratory Animals. All animal procedures followed the guidelines of and were approved by the MDACC Institutional Animal Care and Use Committee (IACUC protocol 00000800, PI: Kadara). |

Note that full information on the approval of the study protocol must also be provided in the manuscript.

