## [Peer Review File · Nature]

Manuscript Title: An atlas of epithelial cell states and plasticity in lung adenocarcinoma

Reviewer Comments & Author Rebuttals

Reviewer Reports on the Initial Version:

Referee expertise:

Referee #1: single cell analyses, computational expertise

Referee #2: lung cancer, mouse models, plasticity

Referee #3: lung development and injury, single cell analyses

Referees' comments:

Referee #1 (Remarks to the Author):

Han et al. profiled 246, 102 single cells from 16 early stage lung adenocarcinomas and 47 matched normal lung samples, and established a cell atlas for LUAD. The authors discussed the cell states and cell plasticity in the early development of LUAD. They found the diversity of cancer cells were related to patient specific oncogenic drivers. In particular, they found KRAS-mutant (KM) cancer cells had unique transcriptomic features, and the local niche was enriched with intermediary cells in lung alveolar differentiation transitioning to KRAS mutant cancer cells. A subset of these intermediary cells with KRT8+ expression "KACs" had increased plasticity. To confirm their finding, they used mouse models of LUAD exposed to tobacco carcinogen.

The cohort and experiments consisting of a geospatial mapping to achieve a cross-regional continuum of cell state dynamics were well designed. The focus on epithelial cells, and the integration of WES and single-cell data, and single-cell inference of KRAS somatic mutations are innovative. The manuscript was very well-organized and I greatly enjoyed reading it.

My major concerns are with regard to the quantification of heterogeneity and plasticity described below. It would have been great if the code for the analyses was accessible for reviewers.

- The conclusion that KM-LUADs are closer compared to other LUADs seems to be based on the UMAP (Fig 1e) while we know that UMAPs can lead to distortion in global distances (see Chari, Tara, Joejya Banerjee, and Lior Pachter. bioRxiv (2021)). The authors need to provide better quantification metrics for distances between patients, e.g. using the Bhattacharyya distance.
- The statements regarding a high degree of inter-patient heterogeneity in differentiation states (Fig. 2) could be biased by the integration of data using Harmony and/or Cytotrace. Comparing Fig. 1d-e to post-Harmony Fig. 2a-c clearly shows that the majority of the variance in the data is ignored when projecting cells to a shared latent space. The authors need to quantify the % of variance

explained by the shared latent (or low-dimensional) space in Harmony and confirm that the distributions of differentiation (e.g. bimodal pattern in P14, P2) hold in data without integration. Can the authors clarify if they applied CytoTRACE on Harmony-integrated data? Additionally, they need to compare their results to a variety of different integration methods such as MNN, Seurat, scVI, STACAS, LIGER, quantify the heterogeneity with each (e.g. using LISI score) and interpret the results in the context of the assumptions of each method (Argelaguet, NatBiotech 2021). The higher variability in differentiation in KM should also be quantified e.g. using Wasserstein distance and the statistical significance should be confirmed. Similarly, Fig. 2d (and Fig. 1g if that is also based on Harmony?) should be repeated on data without integration.

- Is the increased MP30 activation observed in the other two KMs (P2, P1)? Why is only one patient shown?
- Can the authors clarify if they also used Harmony or integration for annotating cell types in Fig. 3a-d? Are the continuums and order of cell states apparent without integration?
- Please provide patient breakdowns for Fig. 3a. Is the reduction in AT2 cells in proximity of LUADs seen in all/majority of patients and if so, can the authors test the significance at the patient level using matched samples? Please also include Fig. 3c colored by location (origin).
- In Figure 3e, both KRT8 and CLDN4 were enriched in KACs. The authors didn't motivate their focus on KRT8 rather than CLDN4.
- To confirm the robustness of trajectories in Fig. 5a, can the authors compare the results to other trajectory inference methods such as Slingshot, CellRank, or Palantir?

Minor comments:

- Can the authors confirm (and clarify in caption/text) that the hierarchical cluster in Fig 1c was done at the transcriptomic level and not just based on the shown markers?
- Please provide the entire map in Fig. 1e (right) to validate the mutation calling pipeline and confirm no/fewer KRAS mutation were inferred in other clusters.
- Line 116: reference of Extended Data Fig. 1e seems to be irrelevant?
- Clarify what the ribbons in Extended Data Fig. 2b are representing.
- Use different colors/shades for the yellow ribbons in Fig. 2e to avoid confusion.
- Fig. 2g top right is missing a title (Differentiation)
- Please show all cells in subplots of Fig. 2c (e.g. in light grey color) to provide context about the entire trajectory.
- I would personally prefer to see numbers on colorbars instead of low/high or less/more to provide more information on the range or values. Not sure why this information is removed.

Referee #2 (Remarks to the Author):

This manuscript by Han, Sinjab, and co-workers investigates cell states in lung adenocarcinoma in patients. The authors first present scRNA-seq data from 16 patients. Notably, 3 tumors have a KRAS mutation, and these tumors cluster close to each other in these analyses (possibly because they are all KRASG12D). In addition, KRAS-mutant tumors seem to be less differentiated compared to EGFR-

mutant tumors. The most “differentiated” tumors are found in patients who are never smokers and neither KRAS nor EGFR mutant. A signature derived from KRAS-mutant tumors correlates with lower overall survival. Next the authors further examined different cell states in tumors and outside tumors. This led them to focus on AICs (alveolar intermediate cells). Notably, they identified a subset of AICs expressing KRT8 (“KACs”), which are enriched in tumors compared to normal lungs. A KAC signature correlates with KRAS activation as well as with lower overall signature. KACs were also present in a mouse model of LUAD induced by loss of Gprc5a and NNK exposure.

There is a clear consensus in the lung cancer field that a better understanding of early stages of the disease may help diagnose and treat lung cancer better. This study, which describes cell states in early-stage lung adenocarcinoma, is thus of general interest. The first two figures of the paper describe the human datasets using a variety of computational tools. It is interesting to see that tumors with different oncogenic drivers may have different levels of heterogeneity and differentiation. The identification of a subset of alveolar intermediate cells builds upon the previous description of these intermediate cells but these cells, based on the current analyses, may be closer to a more relevant intermediate state in the development of LUAD. The presence of KAC-like cells in a mouse model also supports the idea that this intermediate state is generally relevant in the development of LUAD. Thus, overall, there should be broad interest in this study, especially in the context of similar studies in humans and mice, so we can reach a consensus in the stages critical for SCLC development.

Some of the limitations of this study includes its largely descriptive nature, as well as some need for clarification between the human and mouse studies.

Specific comments:

1) Maybe the statement associated with Fig. 1c is too strong (“loss of lineage identity”). Clearly, for example, TTF1/NKX2.1 is used to identify LUAD in clinical samples, and the low expression by scRNA-seq may not reflect the protein expression of this marker.

2) In Fig. 3, the authors could include a control using a signature for the “other AICs” instead of the “low KAC signature”, this may help emphasize that KACs are truly different from other AICs and more relevant to tumor progression.

3) The authors never explicitly state whether human KACs are wild-type for all oncogenic drivers and do not have any CNVs; if true, this should be clarified in the text to help readers understand that these cells are just “normal” epithelial cells in a transition state. This is relevant because, in the mouse models, KACs seem to have mutant KRAS based on the data presented by the authors. This seems an important point: are KACs a “normal” intermediate that provides an epigenetic state favorable for transformation by oncogenic drivers, or are these cells already expressing these oncogenic drivers (in that latter case, “other AICs” may represent the previous stage).

4) Related to #3: Because of the high prevalence of KRAS mutations in the mouse model and the lower prevalence in the human samples studied, is it possible that the KACs in both species are different cell populations? (The 21% overlap in the signature analysis seems low). Are there data

from for example EGFR-mutant mice with LUAD that could help resolve this question?

5) Related to #3: In Figure 4, why are non-KAC AICs absent from the analysis? Or do mice in this model not have them?

6) The statement “tumor-associated KACs, a subset of AICs that arises from AT2 cells” is not directly supported by experimental evidence: the GFP reporter was activated after 6 weeks of treatment and may not label specifically AT2 cells then. Could the authors label AT2 cells first and then induce tumors to verify that these tumors are indeed from AT2 cells and that KACs are GFP+ in this context?

7) Gprc5a knockout mice develop tumors without NKK (a bit slower), do these tumors have KACs or KACs specific to NNK-induced tumors? (Again, are the KACs in the mouse model simply early KRAS-mutant cells?)

8) Are KACs present in genetically engineered mouse models in which tumors are initiated by mutant KRASG12D? (a number of datasets are available to address this question). One could imagine that in these models, there is no “need” for a KAC intermediate as tumors can readily reach the KRAS mutant state.

9) Can mouse KACs be isolated using cell surface markers, do they form organoids or tumors when isolated?

Referee #3 (Remarks to the Author):

In Han et al., the team sequenced 16 early stage LUAD samples and 47 normal lung controls. They found that tumor cells primarily clustered based on their donor origin. KRAS mutant cell clusters are more similar to each other, and are less differentiated compared to tumors cells carrying other classes of mutations. A KRT8+ alveolar cell type was identified near KRAS mutant cells, and are predicted to be in a cellular state between AT2 and AT1. These cells are also found in a mouse model of LUAD, and are deemed “likely transitioned to malignant cells.”

This is a comprehensive study of early stage LUAD single cell transcriptomic signature, combined with genotyping data. The findings reveal single cell resolution transcriptomic differences between tumors with different mutations, and also heterogeneity within tumors with the same mutation. More in-depth analysis and validation is needed to yield novel insights of broad interest.

1. The presence of KRT8 cells in the alveolar region of carcinogen injured lungs is not surprising, as these cells have been widely found in injured lungs (Kobayashi et al., NCB 2020; Strunz et al., Nat Comm 2020, Choi et al., Cell Stem Cell 2020). If the stipulation is that these cells are precursor for LUAD, additional validation is needed. TP53 signature was found in these cells, similar to previously described AT2-to-AT1 transitional cell. However, it is the increase of TP53 signature that is associated with these cells as a feature of their senescence property. How a cell with an increased

tumor suppressor signature could be preferentially transformed needs to be addressed.

2. It seems that the so called KAC (KRT8+ alveolar cells) may be distinct from injury-associated KRT8+ transitional cells, in that KACs already have KRAS mutation and are on their way to be transformed. In this scenario, it will be important to address whether KAC is really transitional between AT2 to AT1, or rather, they are just de-differentiated, as would be expected from the transforming action of activated KRAS. In addition, KRT8+ is widely expressed in normal airway cells. A recent study demonstrated that AT2 to airway transition is also an alveolar cell behavior observed following injury (Kathiriya et al., Cell Stem Cell 2021).

3. In the abstract, "local niche" was mentioned. However, only EpCam+ epithelial cells were sorted and analyzed. For a true niche, cells in other lineages need to be analyzed.

4. Figure 1b, the bronchioalveolar cell designation needs to be revisited, and will be better defined in clustering using both alveolar and airway cells.

5. Unclear why the specific Gprc5a mutant was selected as a mouse model to validate the KAC involvement in tumor formation.

6. Fig 5a, it will be important to genotype the KACs used in the organoid assay. Have they already acquired mutation that transform them?

Author Rebuttals to Initial Comments:

Dear Editor and Reviewers,

We would like to express our sincerest gratitude to the insightful reviews and constructive comments on our manuscript #2022-04-06272A. The insights and queries provided by the reviewers guided us to return what we believe is a strengthened manuscript. All the comments we received on this study have been considered in improving the quality and impact of this article. We performed several additional experiments and analyses that have been added to the manuscript in the form of new figures, extended data figures, as well as supplementary material, and we have amended the text in accordance with the reviewers' recommendations. The major modifications to the manuscript are outlined below and fall under the following vignettes:

- **We further examined the biology of human KACs and now incorporate the following:**

- 1) We found by spatial transcriptomics (ST) analysis that KACs are in the local vicinity of the tumour and highly express *KRAS* and KAC mutation signatures *in situ*.
- 2) Human KACs exhibit driver *KRAS* mutations that are absent in other alveolar subsets such as other alveolar intermediate cells (AICs) – *KRAS*-mutant KACs were only found in samples from *KRAS*-mutant patients.
- 3) Human *KRAS*-mutant KACs are less differentiated compared with other AICs or *KRAS* wild type (wt) KACs.
- 4) We performed additional analyses that demonstrated unique features of KACs relative to other alveolar intermediates. While the human KAC signature is elevated in LUADs (including *KRAS*-mutant LUADs) and preneoplasias, is markedly positively associated with *KRAS* signature and inversely associated with an alveolar signature, and associated with poor survival, we found that a signature derived from other AICs showed none of these features. A signature from other AICs was in fact lower in LUADs, was not associated with *KRAS* nor alveolar signatures, and was not associated with survival.

- **We also performed additional and more in-depth analysis of mouse KACs to show their high pertinence to the biology of *KRAS*-mutant LUAD:**

- 1) Like their human counterparts, ST analysis showed that mouse KACs were surrounding tumours, and both KACs and tumours displayed high KAC and *Kras* signatures *in situ*.
- 2) Mouse KACs existed in a continuum where KACs at later timepoints post-tobacco carcinogen were much more enriched with driver *Kras* mutations.
- 3) Like their human counterparts, *Kras*-mutant KACs were less differentiated than *Kras* wt KACs. Interestingly, *Kras*-mutant KACs compared with their *Kras*-

wt counterparts showed less transcriptional overlap with recently reported injury-associated Krt8+ intermediate cells.

4) We performed integrative analysis of publicly available datasets of *Kras*^{G12D}-driven mouse models with our study and found evidence for KACs in *Kras*^{G12D}-driven models albeit at a reduced frequency.

5) We found that mouse KAC signature was markedly increased in human KACs and malignant cells from patients in our cohort with *KRAS*-mutant LUADs relative to those from patients with *EGFR*-mutant LUADs.

6) We performed analyses of AT2 lineage labeled mice followed up until three months post-NNK and found that organoids-derived NNK-exposed mice not only grew much larger and had markedly increased expression of KAC markers than those from saline-treated animals, but that they were also highly sensitive to targeted *KRAS* inhibition.

- **We also performed additional validation by scRNA-seq and immunofluorescence analysis in lineage-labeled mice:**

1) We similarly labeled (treated with tamoxifen at end of NNK/saline exposure) *Gprc5a*^{-/-}; *Sftpc*^{CreER/+}; *Rosa*^{Sun1GFP/+} mice at EOE to NNK or saline and which were then followed up to 3 months post-NNK or -saline. We performed scRNA-seq of GFP⁺ and GFP⁻ cell fractions from NNK- and saline-treated mice. We also performed immunofluorescence (IF) analysis using additional mice. We summarize findings from these experiments and analyses that allow us to surmise that we were able to label AT2 cells and that KACs likely reside in the trajectory from AT2 to tumour cells.

- **We validated using several other computational methods various trajectory analyses as well as conclusions with respect to the uniqueness of *KRAS*-mutant malignant cells relative to cells from LUADs with other driver mutations.**

- **We have deposited the code used for all analyses in GitHub.**

We have addressed each comment and **point-by-point** responses are provided below:

Reviewer #1: _____ page
3

Reviewer #2: _____ page
17

Reviewer #3: _____ page
43

In our letter, we have inserted some of the revised and newly added figures and corresponding legends, and we have in this letter quoted excerpts (highlighted in gray) from the revised manuscript for ease of accessibility.

Referees' comments:

Referee #1 (Remarks to the Author):

General comment: *Han et al. profiled 246, 102 single cells from 16 early stage lung adenocarcinomas and 47 matched normal lung samples, and established a cell atlas for LUAD. The authors discussed the cell states and cell plasticity in the early development of LUAD. They found the diversity of cancer cells were related to patient specific oncogenic drivers. In particular, they found KRAS-mutant (KM) cancer cells had unique transcriptomic features, and the local niche was enriched with intermediary cells in lung alveolar differentiation transitioning to KRAS mutant cancer cells. A subset of these intermediary cells with KRT8+ expression "KACs" had increased plasticity. To confirm their finding, they used mouse models of LUAD exposed to tobacco carcinogen.*

The cohort and experiments consisting of a geospatial mapping to achieve a cross-regional continuum of cell state dynamics were well designed. The focus on epithelial cells, and the integration of WES and single-cell data, and single-cell inference of KRAS somatic mutations are innovative. The manuscript was very well-organized and I greatly enjoyed reading it.

My major concerns are with regard to the quantification of heterogeneity and plasticity described below. It would have been great if the code for the analyses was accessible for reviewers.

General response: We are delighted that the reviewer greatly enjoyed reading the manuscript. We provide below point-by-point responses to address reviewer 1's concerns on quantification of heterogeneity and plasticity.

The code is now available at GitHub (https://github.com/quangchunhan/LUAD_Code).

Comment 1: *The conclusion that KM-LUADs are closer compared to other LUADs seems to be based on the UMAP (Fig 1e) while we know that UMAPs can lead to distortion in global distances (see Chari, Tara, Joeyta Banerjee, and Lior Pachter. bioRxiv (2021)). The authors need to provide better quantification metrics for distances between patients, e.g. using the Bhattacharyya distance.*

Response 1: We thank the reviewer for this insightful comment and great suggestion. To comply with the reviewer's suggestion, we applied the Bhattacharyya distance metric to quantify transcriptional similarity at both the sample and cell levels. As expected, we observed that Bhattacharyya distances within the KM-LUAD group (KM-KM) were significantly smaller when compared to distances between KM-LUADs and other *KRAS* wildtype (WT) LUADs, such as *EGFR*-mutant LUADs (KM-EM), *MET*-mutant LUADs (KM-MET), and other remaining LUADs (KM-WT) (**Extended Data Fig. 1e**, left), suggesting relatively higher transcriptional similarity within KM-LUADs. To adjust for sample-specific variation, we randomly sampled 100 cells from each patient group with the same driver mutation (e.g., KM-LUADs) for 100 times and calculated the Bhattacharyya distance between different patient groups. In accordance with sample-level Bhattacharyya analysis as described above, we also noted significantly smaller distances between cells within KM-LUADs (KM-KM) relative to the distances between cells of KM-LUADs and *KRAS* WT

LUADs (i.e., KM-EM, KM-MET, KM-WT) (**Extended Data Fig. 1e**, right). These data support our clustering analyses and show higher transcriptome similarity among KM-LUADs compared to other LUADs.

Extended Data Fig. 1e. Analysis of Bhattacharyya distances calculated at the sample (left) and cell (right) levels and comparing variances between LUADs with driver mutations in *KRAS* (KM), *EGFR* (EM), or *MET* (MM) relative to those WT for these genes. *P*-values were calculated by Wilcoxon Rank-Sum test.

These new data are now described in the revised manuscript as follows:

Results section (lines 128 – 132): “We also applied the Bhattacharyya distance metric to quantify transcriptomic similarity and noted significantly smaller distance (i.e., higher similarity) at both sample (**Extended Data Fig. 1e, left**) and cell (**Extended Data Fig. 1e, right**) levels among KM-LUADs compared to EM- or MM-LUADs or other tumours with wild type *KRAS* (*KRAS*-WT) ($P < 2.2 \times 10^{-16}$).”

Methods section (lines 989 – 996): “The Bhattacharyya distance was calculated using the *bhattacharyya.dist* function from R package *fpc* (v2.2-9). Top 25 most highly ranked PCs were used for both patient level and cell level distance calculations. For Bhattacharyya distance quantification at the cell level, 100 cells were randomly for each patient group that was defined by driver mutations (e.g., KM-LUADs) by repeating 100 times and the Bhattacharyya distances were then calculated between patient groups in a pairwise fashion. Differences in Bhattacharyya distance between patient groups were tested using Wilcoxon Rank-Sum tests and the boxplot visualization was generated using the *geom_boxplot* function from R package *ggplot2* (v3.2.0).”

Comment 2: The statements regarding a high degree of inter-patient heterogeneity in differentiation states (Fig. 2) could be biased by the integration of data using Harmony and/or Cytotrace. Comparing Fig. 1d-e to post-Harmony Fig. 2a-c clearly shows that the majority of the variance in the data is ignored when projecting cells to a shared latent space. The authors need to quantify the % of variance explained by the shared latent (or low-dimensional) space in Harmony and confirm that the distributions of differentiation (e.g. bimodal pattern in P14, P2) hold in data without integration. Can the authors clarify if they

applied CytoTRACE on Harmony-integrated data? Additionally, they need to compare their results to a variety of different integration methods such as MNN, Seurat, scVI, STACAS, LIGER, quantify the heterogeneity with each (e.g. using LISI score) and interpret the results in the context of the assumptions of each method (Argelaguet, NatBiotech 2021). The higher variability in differentiation in KM should also be quantified e.g. using Wasserstein distance and the statistical significance should be confirmed. Similarly, Fig. 2d (and Fig. 1g if that is also based on Harmony?) should be repeated on data without integration.

Response 2: We thank the reviewer for their astute comment, and we apologize for failing to have clearly mentioned that our CytoTRACE inference of cell differentiation (ridge plots in previously Fig. 2c now in **Fig. 1g**, far right), meta-program analyses of malignant cells (previously Fig. 2d now in **Fig. 1h**) as well as analysis of P14 malignant cells (previously Fig. 2g now **Fig. 1k**) were performed on normalized count matrix without Harmony-based correction considering that application of data integration (e.g. Harmony) may reduce or wipe out the intertumoral heterogeneity of malignant cells. Thus, the bimodal pattern observed in P14 and P2 was not influenced by data integration. We appreciate this opportunity to clarify that Harmony-corrected data were only used for UMAP-based visualization of malignant cells as shown previously in Fig. 2b,c (now UMAPs in **Fig. 1f,g**) and Extended Data Fig. 2a (now **Extended Data Fig. 1f**). We apologize for not clearly explaining the above in the original manuscript. We now accentuate in the revised manuscript that Harmony-corrected data were only used for visualization and that inference of differentiation and analysis of metaprograms were performed on normalized count matrices prior to Harmony-based correction as such:

Methods section (lines 928 - 932): “For malignant cells, except for the global UMAP visualization, downstream analyses including identification of large-scale copy number variations (CNVs), inference of cancer cell differentiation states, quantification of meta-program expressions, trajectory analysis, and mutation analysis were performed without Harmony batch correction.”

We appreciate the reviewer’s constructive comment on the need to quantify the variability in differentiation of KM-LUADs. Accordingly, we quantified patient level variability of differentiation using the Wasserstein distance, a metric commonly used to quantify distance between probability distributions ¹, between the distribution of actual CytoTRACE scores of malignant cells from a patient and the distribution of simulated data with identical mean and standard deviation. As shown below, malignant cells of KM-LUADs showed a trend of higher Wasserstein distance relative to those of *EGFR*-mutant LUADs (EM-LUAD) or other *KRAS* and *EGFR* WT LUADs (Double WT), albeit without reaching statistical significance, possibly due to the limited number of patients analyzed per group (**Extended Data Fig. 1g**). This analysis suggests increased variability in distribution of malignant cell differentiation states among KM-LUADs relative to other LUADs examined.

Extended Data Fig. 1g. Analysis of Wasserstein distances among KM-LUADs, EM-LUADs, and LUADs with WT *KRAS* and *EGFR* (Double WT). *P*-value was calculated by the Wilcoxon Rank-Sum test.

These new findings are now described in the revised manuscript as such:

Results section (lines 150 - 152): "...with KM-LUADs showing a trend for higher variability in differentiation (indicated by greater Wasserstein distances) relative to EM- or other LUADs"

Methods section (lines 1063 - 1067): "Wasserstein distance metric was applied using R package *transport* v0.13 to quantify the variability of distribution of CytoTRACE scores. Function *wasserstein1d* was used to calculate the distance between the distribution of actual CytoTRACE scores of one patient and the distribution of simulated data with identical mean and standard deviation."

Comment 3: Is the increased MP30 activation observed in the other two KMs (P2, P1)? Why is only one patient shown?

Response 3: We thank the reviewer for this comment. MP30 was shown in P14 to accentuate intratumoral heterogeneity in transcriptional programs in KM-LUAD cells. P14 was an exemplar because it comprised the largest number of malignant cells among KM-LUADs and which allowed comprehensive assessment of intra-tumoral heterogeneity. To address the reviewer's concerns, we now present the expression of MP30 signature scores in all patients including in the three KM-LUADs (P2, P10 and P14). KM-LUADs overall showed significantly higher expression of the MP30 metaprogram relative to *KRAS* WT LUADs (**Extended Data Fig. 2b** and **Supplementary Fig. 4**).

Extended Data Fig. 2b. MP30 signature score was computed in each patient and color-coded by driver mutation. Box, median \pm interquartile range; whiskers, 1.5 \times interquartile range; center line: median.

Supplementary Fig. 4. MP30 KRAS signature score in malignant cells from KM-LUADs and *KRAS* wild type LUAD (KW-LUADs). Box, median \pm interquartile range; whiskers, 1.5 \times interquartile range; center line: median. *P*-value was calculated by Wilcoxon rank-sum test.

Accordingly, we revised the manuscript as such:

Results section (lines 167 - 169): “Notably, malignant cells from KM-LUAD patients P2, P10, and P14 showed increased expression of MP30 relative to malignant cells from all but one of the LUADs.”

Comment 4: Can the authors clarify if they also used Harmony or integration for annotating cell types in Fig. 3a-d? Are the continuums and order of cell states apparent without integration?

Response 4: We applied Harmony during the process of identifying non-malignant cell types. Generation of continuums was based on data without Harmony-based correction (Monocle 2). We apologize for not making this clear in the original manuscript and we now better describe our computational workflow in the revised manuscript.

Comment 5: Please provide patient breakdowns for Fig. 3a. Is the reduction in AT2 cells in proximity of LUADs seen in all/majority of patients and if so, can the authors test the significance at the patient level using matched samples? Please also include Fig. 3c colored by location (origin).

Response 5: We thank the reviewer for this constructive comment. To comply with the reviewer's comment, we now provide a complementary version to the original figure (now in **Fig. 2a**) but broken down by patient (**Extended Data Fig. 3a**). We also plotted fractions of AT2 cells for samples in each patient using line plots (**Extended Data Fig. 3b**). We found that fractions of AT2 cells were overall reduced with increasing proximity to LUADs in 7 out of the 16 patients (P1, P4, P5, P6, P8, P13 and P16; **Extended Data Fig. 3a,b**). For 6 patients (P2, P7, P10, P11, P14 and P15), AT2 fractions were lower in LUADs versus normal lungs. Only two patients (P9 and P12) showed fluctuating AT2 cells fractions across the geospatial samples – although for P9, fractions of AT2 cells were higher in distant and intermediate (from tumor) normal lung tissues when compared to the tumor and adjacent-to-tumor normal sample (**Extended Data Fig. 3a,b**). Also, using ordinal logistic regression we statistically evaluated changes in AT2 fractions across the geospatial samples. We found that, for at least for the 7 patients above, AT2 levels were significantly reduced from distant normal sites to the tumours ($P = 0.004$). These findings suggest that fractions of AT2 cells are lower across normal tissues in at least a subset of early-stage LUAD patients. We thus now revised our conclusion in the revised manuscript and instead state that AT2 cell fractions are gradually reduced in samples with increasing proximity to tumors in at least a subset of the patients studied. These results are now described in the revised manuscript as such:

Extended Data Fig. 3a. Composition of normal epithelial lineages across the spatial regions at the sample level. SDP: *SCGB1A1/SFTPC* dual positive cells. NE: neuroendocrine. Patients showing increased AT2 fractions in NL samples with closer proximity to tumours are bolded on the x-axis.

Extended Data Fig. 3b. Fractional changes of AT2 cells among all epithelial cells across the spatial regions at the patient level.

Accordingly we revised the manuscript as such:

Results section (lines 227 - 230): “Among the 16 patients, 7 showed gradually reduced AT2 fractions with increasing tumour proximity ($P = 0.004$ by ordinal regression analysis), 6 had reduced AT2 fractions in LUADs relative to NL, and only 2 showed fluctuated AT2 fractions in geospatial samples.”

We have also modified original Fig. 3c, now **Supplementary Fig. 5**, by color coding cells according to their corresponding spatial locations, or the spatial sample they originated from.

Supplementary Fig. S5. Pseudotime trajectory analysis of alveolar and malignant subsets colored by tissue location.

Comment 6: In Figure 3e, both KRT8 and CLDN4 were enriched in KACs. The authors didn't motivate their focus on KRT8 rather than CLDN4.

Response 6: We thank the reviewer for their comment. In our data, we have identified an intermediate lung alveolar cell state that was distinct from AT1 and AT2 cells, and that was associated with tumour formation. To better characterize those intermediate cells, we interrogated differentially expressed lineage genes that could mark those cells and distinguish them from other alveolar cells. Therefore, we made use of this property to coin this unique population of cells as *KRT8*-expressing alveolar cells, or KACs for short. In agreement with our findings, *Krt8* expression has also been reported in alveolar states that have been linked to lung injury and inflammation – these cells were referred to as Krt8+ intermediate cells (e.g., the report by Strunz and colleagues²). Indeed, we found significant overlap between the KAC cells from our study and the Krt8+ intermediate cells from previous reports on lung injury (report by Strunz et al (**Extended Fig. 7a**)). While the rest of the manuscript focuses on understanding KAC biology, the interrogation of *KRT8* itself serves as a marker of those cells. We now performed spatial transcriptomics (ST) analysis of P14 (a KM-LUAD) tumour tissue and *Gprc5a*^{-/-} mouse lung tissues using the 10X Visium platform (~3600 uniquely barcoded spots, > 50k reads per spot on average with > 2000 genes detected per spot). We examined the distribution of KAC and KRAS signatures as well as KAC marker gene expression *in situ*. We quantified expression of these genes and gene signatures and projected their expression levels directly on high-resolution histology images using the TESLA algorithm (<https://github.com/jianhuupenn/TESLA.git>). We show that, for both human and mouse tissues, the KAC signature that includes *CLDN4* is increased in tumours as well as in surrounding/adjacent KACs (**Fig. 2j**; **Fig. 3f**; **Extended data Fig. 6h**).

Fig. 2j. ST analysis of P14 LUAD showing histological annotation of H&E-stained Visium slide (10X Genomics) (left) and spatial heatmaps showing CNV score, as well as scaled expression of *KRT8*, KAC markers, and KRAS signature.

Fig. 3f. ST analysis of lung tissue at 7 mo post-exposure to NNK and showing histological annotation of H&E-stained Visium slide (10X Genomics) (left) and spatial heatmaps showing scaled expression of *KRT8* as well as KAC and KRAS signatures.

Extended Data Fig. 6h. ST analysis of the same tumour-bearing mouse lung in **Fig. 3f** with UMAP of cell clusters identified by Seurat (left) and mapped spatially (middle). Spatial maps with scaled expression of *Krt8* and *Plaur* are shown on the right.

Accordingly, we revised the manuscript as such:

Results section (lines 267 - 277): “Spatial transcriptomics (ST) analysis of P14 tumour tissue using the Visium platform (10X Genomics) demonstrated increased expression of *KRT8* in tumour regions with high CNV scores, as well as in tumour-adjacent normal-appearing regions that were histopathologically evaluated to comprise highly reactive pneumocytes and that exhibited low/moderate CNV scores (**Fig. 2j**)..... Similarly, we found that a signature based on the 5 KAC markers (see **Fig. 2e**) was high in tumour and in adjacent-to-tumour regions with reactive pneumocytes, with a similar and spatially overlapping pattern for *KRAS* signature.”

Results section (lines 379 - 386): “We then studied by ST lung tissues from mice at 7 months post-exposure to NNK. We found that areas histologically-annotated as tumour regions had significantly increased expression of *Krt8*, *Plaur*, as well as KAC and *KRAS* signatures which were also overlapping, further confirming KACs enrichment in mouse tumours induced by tobacco carcinogen exposure (**Fig. 3f and Extended Data Fig. 6h**). In line with our data from human LUAD patients (**Fig. 2j**), *Krt8* high KACs with elevated expression of KAC and *KRAS* signatures were enriched in “reactive”, non-neoplastic regions surrounding tumours of NNK-exposed mice”.

We also provided a new detailed subsection in the **Methods section** that describes the ST workflow (**lines 1083 – 1128**).

Comment 7: *To confirm the robustness of trajectories in Fig. 5a, can the authors compare the results to other trajectory inference methods such as Slingshot, CellRank, or Palantir?*

Response 7: To address this constructive comment, we ran 3 trajectory inference methods including Slingshot, CellRank, and Palantir. We then quantified correlation between our original Monocle 2-derived pseudotime included in the original manuscript (now **Fig. 4a**) and pseudotime inferred by Slingshot, CellRank or Palantir. We not only found that Slingshot, CellRank and Palantir resulted in similar trajectories (see the next page, upper panels) to the Monocle 2-derived plot in **Fig. 4a**, but that there was strong correlation between each of those three trajectory inference methods and Monocle 2 (lower).

Supplementary Fig. 6. Robustness of the pseudotime trajectory analysis of mouse alveolar cell subsets and tumour cells. Trajectory analysis of alveolar and malignant subsets using three trajectory inference methods (top left to right: Palantir, Slingshot, and CellRank), as well as quantified correlation of each of these methods with results obtained from Monocle 2-derived pseudotime (bottom). *P*-values were calculated with Spearman correlation test. *R* denotes Spearman correlation coefficient. The cell density estimation (see Methods) denotes the cell distribution density on the scatterplot.

Accordingly, we revised the manuscript as such:

Methods section (1067 - 1081): “The robustness of Monocle 2-based pseudotemporal ordering prediction was validated by independent pseudotime prediction tools including Palantir³, Slingshot⁴ and Cellrank⁵. Slingshot (v2.6.0) pseudotime prediction was performed using *slingshot* function with *reduceDim* parameter set to ‘PCA’ and other parameters set to defaults. Cellrank prediction was performed using the *CytoTRACEKernel* function with default parameters from Cellrank python package (v1.5.1). Palantir prediction was performed using palantir python package (v1.0.1). A diffusion map was generated using *run_diffusion_maps* function with *n_components* = 5. Palantir prediction was generated using *run_palantir* function with *num_waypoints* = 500 and other parameters set to defaults.

Inferred pseudotimes by the three independent methods were then integrated with that generated by Monocle2 for each single cell, followed by pairwise mapping and correlation analysis. The cell density plot was generated using *Contour* tool in JMP v15 with $n = 10$ gradient levels and contour type parameter set to 'Nonpar Density'. To assess the pseudotime prediction consistency between Monocle 2 and the three independent methods, Spearman's correlation coefficients were calculated and statistically tested using *cor.test* function in R."

Minor comments:

Comment 8: 1 Can the authors confirm (and clarify in caption/text) that the hierarchical cluster in Fig 1c was done at the transcriptomic level and not just based on the shown markers?

Response 8: We apologize for not clearly describing the basis for the hierarchical clustering in Fig. 1c in our original manuscript (now **Fig. 1b**). Indeed, the hierarchical clustering shown in **Fig. 1b** was performed at the transcriptomic level by using principal components derived from batch-corrected PCA dimension reduction data. For each cell lineage defined by unsupervised clustering analysis, we selected a few representative markers and generated the bubble plot. We now explain that clustering was performed at the transcriptomic level in the revised manuscript.

Comment 9: Please provide the entire map in Fig. 1e (right) to validate the mutation calling pipeline and confirm no/fewer KRAS mutation were inferred in other clusters.

Response 9: We thank the reviewer for this important comment. We originally found and showed $KRAS^{G12D}$ in malignant cells from P2, P10, and P14 (shown as an inlet from the map in original Fig. 1e, now **Fig 1c**). We analyzed $KRAS^{G12D}$ mutations (see **Methods**, section "**Identification of malignant cells and mapping KRAS G12 mutations**") calling across all malignant samples from all patients in the map. We found no additional $KRAS^{G12D}$ mutations in malignant cells from other samples/patients (now **Extended Data Fig. 1b, third UMAP on the right**) thereby supporting the specificity of our KRAS mutation analysis and its corroboration with whole-exome sequencing of tissues from the same patients (now **Extended Data Fig. 1b**).

Extended Data Fig. 1b. UMAP plot of 17,064 malignant cells colored by $KRAS^{G12D}$ mutation status.

Accordingly, we revised the manuscript as such:

Results section (lines 117 - 121): “Annotation based on genomic profiling (by WES) showed that malignant cells from 3 $KRAS$ -mutant (KM-) LUAD patients in our cohort (P2, P10, P14) clustered closely together and in comparison to malignant cells from 5 $EGFR$ -mutant (EM-), 2 MET -mutant (MM-), or other LUADs, all of which showed a more dispersed clustering pattern.”

Comment 10: Line 116: reference of Extended Data Fig. 1e seems to be irrelevant?

Response 10: We thank the reviewer for pointing this out and apologize for the inappropriate reference arrangement. Extended Data Fig. 1e (now **Extended Data Fig. 1c**) was included to show the clustering pattern with respect to smoking. We now correct the reference of the extended data figure.

Comment 11: 4 Clarify what the ribbons in Extended Data Fig. 2b are representing.

Response 11: We thank the reviewer for pointing this out. We have revised the corresponding legend to Extended Data Fig. 2b (now **Extended Data Fig. 2a**) to clarify what the ribbons represent.

Comment 12: 5 Use different colors/shades for the yellow ribbons in Fig. 2e to avoid confusion.

Response 12: We thank the reviewer for pointing this out. We have changed the color of the two originally colored yellow ribbons now making them distinct and accordingly updated the legend of Fig. 2e (now **Extended Data Fig. 2a**, left).

Extended Data Fig. 2a. Enrichment of clusters (C1-C5) from **Fig. 1h** in samples color-coded and labeled by different driver mutations. **: $P < 0.01$. P -value was calculated using Fisher's exact test.

Comment 13: *Fig. 2g top right is missing a title (Differentiation)*

Response 13: We thank the reviewer for pointing this out. We have amended this by adding a title.

Comment 14: *Please show all cells in subplots of Fig. 2c (e.g. in light grey color) to provide context about the entire trajectory.*

Response 14: The reviewer's comment is most likely related to Fig. 3c showing the cell subpopulations along the inferred trajectory. We updated the right part of original Fig. 3c (now **Fig. 2c**) as suggested to show the trajectory positioning pattern of each population. Each population is highlighted with unique colors and shown in the context of all cells along the entire trajectory which were dimmed in the background. The resultant plot better positions the populations of interest with respect to other populations. These changes are now reflected in **Fig. 2c** of the revised manuscript.

Fig. 2c subplot. Pseudotime trajectory analysis of alveolar and malignant cells colored by the cell subpopulations in the context of all cells along the entire trajectory which were dimmed in the background.

Comment 15: I would personally prefer to see numbers on color bars instead of low/high or less/more to provide more information on the range or values. Not sure why this information is removed.

Response 15: We thank the reviewer for this comment. We have now updated the scale bars where appropriate (**Fig. 1f-h,k, Fig. 2c, Fig. 3d, Fig. 4a; Extended Data Fig. 1b, 2d, 2g, 2h, 6c, 6d, 6g**) to include numerical quantification of “low/high” as well as “less/more” ranges.

Referee #2 (Remarks to the Author):

General comment: This manuscript by Han, Sinjab, and co-workers investigates cell states in lung adenocarcinoma in patients. The authors first present scRNA-seq data from 16 patients. Notably, 3 tumors have a KRAS mutation, and these tumors cluster close to each other in these analyses (possibly because they are all KRASG12D). In addition, KRAS-mutant tumors seem to be less differentiated compared to EGFR-mutant tumors. The most “differentiated” tumors are found in patients who are never smokers and neither KRAS nor EGFR mutant. A signature derived from KRAS-mutant tumors correlates with lower overall survival. Next the authors further examined different cell states in tumors and outside tumors. This led them to focus on AICs (alveolar intermediate cells). Notably, they identified a subset of AICs expressing KRT8 (“KACs”), which are enriched in tumors compared to normal lungs. A KAC signature correlates with KRAS activation as well as with lower overall signature. KACs were also present in a mouse model of LUAD induced by loss of Gprc5a and NNK exposure.

There is a clear consensus in the lung cancer field that a better understanding of early stages of the disease may help diagnose and treat lung cancer better. This study, which describes cell states in early-stage lung adenocarcinoma, is thus of general interest. The first two figures of the paper describe the human datasets using a variety of computational tools. It is interesting to see that tumors with different oncogenic drivers may have different levels of heterogeneity and differentiation. The identification of a subset of alveolar intermediate cells builds upon the previous description of these intermediate cells but these cells, based on the current analyses, may be closer to a more relevant intermediate state in the development of LUAD. The presence of KAC-like cells in a mouse model also supports the idea that this intermediate state is generally relevant in the development of LUAD. Thus, overall, there should be broad interest in this study, especially in the context of similar studies in humans and mice, so we can reach a consensus in the stages critical for SCLC development.

Some of the limitations of this study includes its largely descriptive nature, as well as some need for clarification between the human and mouse studies.

General response: We are very happy that reviewer #2 finds that there should be broad interest in our study. We provide below a point-by-point response to reviewer #2's concerns including with respect to clarification between human and mouse studies.

Comment 1: Maybe the statement associated with Fig. 1c is too strong (“loss of lineage identity”). Clearly, for example, TTF1/NKX2.1 is used to identify LUAD in clinical samples, and the low expression by scRNA-seq may not reflect the protein expression of this marker.

Response 1: We thank the reviewer for this comment. While NKX2-1 is commonly considered as a hallmark of LUADs, not all LUADs have high expression of NKX2-1 as previously reported for LUADs with gastric differentiation or of the mucinous variant⁶. In our dataset, we have noted significantly reduced NKX2-1 expression in malignant cells. In order to further investigate this notion (and to address other concerns), we performed spatial transcriptomics (ST) analysis using the 10X Genomics Visium platform and we investigated the spatial distribution of NKX2-1 expression. ST analysis showed that NKX2-1 expression is

very specific to AT2 cells and almost absent in the tumoural regions (now **Extended Data Fig. 4a**).

Fig. 2j. ST analysis of P14 LUAD showing histological annotation of H&E-stained Visium slide (10X Genomics) (left) and spatial heatmaps showing CNV score (right).

Extended Data Fig. 4a. ST analysis of P14 LUAD using the Visium platform from 10X Genomics showing scaled expression of *NKX2-1* (top) and alveolar signature (bottom).

Accordingly we revised the manuscript as such:

Results section (lines 274 - 277): “Notably, ST analysis showed that tumoural regions from this KM-LUAD had markedly reduced expression of *NKX2-1* and alveolar signature (**Extended Data Fig. 4a**) in line with our findings above on reduced alveolar differentiation in KM-LUAD (**Fig. 1j**).”

We also provided a new detailed subsection in the **Methods section** that describes the ST workflow (**lines 1083 – 1128**).

We do acknowledge that gene expression by scRNA-seq may not reflect protein expression. Thereby, we rephrased the statement in the revised manuscript in which we now describe this observation as “diminished lineage identity” (**line 112 of the Results section**).

Comment 2: In Fig. 3, the authors could include a control using a signature for the “other AICs” instead of the “low KAC signature”, this may help emphasize that KACs are truly different from other AICs and more relevant to tumor progression.

Response 2: We thank the reviewer for this constructive comment. We derived a signature from “other AICs” as suggested. We first computed expression levels of the other AICs signature in alveolar subsets and in KACs to confirm pertinence of this signature to other AICs. We found that “other AICs” signature scores were markedly higher in AICs that are non-KACs (i.e., in other AICs), compared to KACs, AT1, and AT2 cells (now **Extended Data Fig. 4d**).

Extended Data Fig. 4d. Expression of other AICs signature across AT1, AT2, KACs and other AICs. Box, median \pm interquartile range; whiskers, 1.5 \times interquartile range; center line: median.

Next, we further queried the AICs signature. In sharp contrast to what we originally observed with the KAC signature (now **Fig. 2k**), the other AICs score showed no or minimal correlation with *KRAS* or alveolar signature (now **Extended Data Fig. 4e, f**).

Fig. 2k. Correlation analysis between KAC signature scores and KRAS signature scores or MP31 alveolar lineage signature scores. P -values were calculated with Spearman correlation test. R denotes the Spearman correlation coefficients.

Extended Data Fig. 4e,f. Correlation analysis between other AICs signature and KRAS signature scores (e) or alveolar signature (f). P -values were calculated with Spearman correlation test. R denotes the Spearman correlation coefficients.

Also, while we had found in our original manuscript that the KAC signature was increased in LUADs or in lung preneoplasias when compared to normal lung tissues (now **Fig. 2m**), the opposite was true with the other AICs signature; it was significantly reduced in LUADs relative to normal lung tissues and showed no pattern in lung preneoplasias (now **Extended Data Fig. 4g, i**).

Fig. 2m. Enrichment of KAC signature across samples of TCGA LUAD (left) and premalignancy (right) cohorts. AAH: atypical adenomatous hyperplasia. *P*-values were calculated using Wilcoxon signed-rank test.

Extended Data Fig. 4g,i. Expression of other AICs signature in TCGA LUAD samples and matched NL tissues (**g**), as well as in a lung preneoplasia cohort (**i**). *P*-values were calculated using Mann-Whitney U test. Box, median \pm interquartile range; whiskers, 1.5 \times interquartile range; center line: median. AAH: atypical adenomatous hyperplasia. n.s.: non-significant ($P > 0.05$).

Another important difference between the KAC and other AICs signature was their association with prognosis. While patients with higher KAC signature displayed significantly poorer survival (now **Fig. 2o** and **Extended Data Fig. 4j,k**), there were no differences in survival among patients based on the other AICs signature (now **Extended Data Fig. 4l,m**). These observations were consistent among independent datasets: TCGA (**Extended Data Fig. 4j,l**) and MD Anderson's PROSPECT cohort (**Extended Data Fig. 4k,m**). Importantly, we performed multivariate Cox proportional hazard regression analysis by including covariates stage, age, KAC signature as well as the other AICs signature in the model. While the KAC signature was associated with poorer survival even after accounting for stage, the other AICs signature was not significantly associated with survival outcome (**Fig. 2o**).

Extended Data Fig. 4j-m. Kaplan-Meier plots showing differences in overall survival probability in TCGA (**j**) and PROSPECT (**k**) samples with high versus low KAC signature scores, or with high versus low scores for other AICs signature (**l**: TCGA; **m**: PROSPECT). OS: overall survival. Sig. low: LUAD samples with signature scores lower than the group median value. Sig. hi: LUAD samples with signature scores higher than the group median value. Mo: months. P -values were calculated with the log-rank test.

Fig. 2o. Multivariate Cox proportional hazard regression analysis including stage, age, KAC signature and “other AICs” signature. Q-values were calculated using a Cox proportional hazards (PH) regression model and adjusted with Benjamini–Hochberg method.

Our findings further support the stark distinction between KAC and other AICs, as well as the association between KACs and pathogenesis of LUAD. Accordingly, we revised the manuscript as such:

Results section (lines 280 - 308): “To further emphasize the distinction between KACs and other AICs, we found that a signature pertinent to “other AICs” was evidently lower in KACs relative to AICs that are non-KACs (**Extended Data Fig. 4d**). The expression levels of KAC signature significantly and positively correlated with that of KRAS signature ($r = 0.45$; $P < 2.2 \times 10^{-16}$) and inversely correlated with alveolar signature ($r = -0.77$; $P < 2.2 \times 10^{-16}$; **Fig. 2k**), unlike “other AICs” signature which showed no correlation with KRAS ($r = 0.045$; $P = 3.2 \times 10^{-5}$) or alveolar ($r = -0.11$; $P < 2.2 \times 10^{-16}$) signatures (**Extended Data Fig. 4e, f**). Of note, KAC signature was significantly higher in KACs and in malignant cells from KM-LUAD tumour tissues relative to those from EM-LUADs ($P < 2.2 \times 10^{-16}$; **Fig. 2l**). We then found that the KAC signature was significantly enriched in bulk transcriptomes of human TCGA LUADs compared to uninvolved NL tissues ($P = 1.9 \times 10^{-15}$; **Fig. 2m, left**), and in contrast to “other AICs” and alveolar signatures which were both significantly reduced in LUADs ($P = 8.6 \times 10^{-6}$ and $P = 1.9 \times 10^{-15}$, respectively; **Extended Data Fig. 4g, h**). By analysis of our independent cohort of lung preneoplasias, we found that KAC signature was significantly and progressively increased along the pathologic continuum of NL, atypical adenomatous hyperplasia (AAH; the earliest precursor of LUAD), and matching invasive LUAD (**Fig. 2m, right**), whereas there was no such pattern for “other AICs” signature (**Extended Data Fig. 4i**). Furthermore, using TCGA data, we also found that KAC signature was significantly higher in KM-LUADs relative to KRAS-WT LUADs ($P = 0.002$; **Fig. 2n**). Also, patients with KAC signature-high LUADs ($n = 243$) showed significantly reduced survival ($P = 0.005$) relative to those with low expression of the signature (TCGA; $n = 246$; **Extended Data Fig. 4j**). In a separate cohort, KAC signature score was also associated with poor OS (PROSPECT; $n = 150$; $P = 0.04$; **Extended Data Fig. 4k**). In either of these cohorts, there were no differences in survival among patients based on “other AICs” signature (TCGA, $P = 0.28$; PROSPECT, $P = 0.35$; **Extended Data Fig. 4l, m**). Importantly, we performed multivariate Cox proportional hazard regression analysis by including covariables such as stage, age, KAC and “other AICs” signatures in the model. We found that KAC signature was associated with shortened OS even after accounting for stage (FDR adjusted q -value = 0.034), whereas “other AICs” signature showed no significant association with survival outcome (FDR adjusted q -value = 0.5; **Fig. 2o**).”

***Comment 3:** The authors never explicitly state whether human KACs are wild-type for all oncogenic drivers and do not have any CNVs; if true, this should be clarified in the text to help readers understand that these cells are just “normal” epithelial cells in a transition state. This is relevant because, in the mouse models, KACs seem to have mutant KRAS based on the data presented by the authors. This seems an important point: are KACs a “normal”*

intermediate that provides an epigenetic state favorable for transformation by oncogenic drivers, or are these cells already expressing these oncogenic drivers (in that latter case, “other AICs” may represent the previous stage).

Response 3: We thank the reviewer for this valuable comment. To address this important comment, we first examined larger-scale chromosomal CNVs in all alveolar cell subsets and in malignant cells. We found that KACs exhibited moderately elevated CNV scores relative to AT2, AT1, and other AICs, although their levels of aneuploidy were considerably lower compared to that of malignant cells (**Extended Data Fig. 5a,b**). It is worthwhile to note that this finding is congruent with our original data in mice (now **Fig. 3e**) showing that mouse KACs exhibited modestly increased global CNV burden compared to AT2 and AT1 cells.

Extended Data Fig. 5 a,b. UMAP clustering of alveolar subsets by CNV score (**a**) and quantification of CNV scores across AT1, AT2, KACs and other AICs (**b**). Box, median \pm interquartile range; whiskers, 1.5 \times interquartile range; center line: median. P -values were calculated using Wilcoxon Rank-Sum test.

As mentioned above, we found that the KAC, but not other AICs, signature was strongly associated with a signature of *KRAS* activation (**Fig. 2k and Extended Data Fig. 4e, f**). We also had found in our original manuscript that KACs in mice were *Kras*^{G12D}-mutant (now **Fig. 3d, e**). By virtue of the 5' scRNA-seq assay used in our study, we were able to call genomic alterations codon 12 of *KRAS*. Thus, we extracted and pooled corresponding reads for human KACs, alveolar subsets, and malignant cells by matching their unique cell barcodes and screened *KRAS*^{G12D} mutation in these cell subsets. First, we confirmed that *KRAS*^{G12D} was indeed present in malignant cells from KM-LUADs and with a variant allele frequency (VAF) of 78% (**Fig. 2p and Extended Data Fig. 5c, d**). Importantly, KACs harbored *KRAS*^{G12D} mutations which were absent in AT2, AT1, and other AICs (**Fig. 2p and Extended Data Fig. 5c,d**). Additionally, *KRAS* mutations in KACs were only found in KM-LUAD patients (**Fig. 2p and Extended Data Fig. 5c, d**). The VAF of *KRAS*^{G12D} mutation in KACs from KM-LUADs (10%) was higher when compared to that from all LUADs (5%) or all samples (3%) (**Fig. 2p and Extended Data Fig. 5c, d**). Interestingly, we noted *KRAS*

mutations including additional variants (e.g., $KRAS^{G12C}$) in uninvolved normal lung tissues from KM-LUAD patients. Altogether, these findings support the notion that not all normal alveolar intermediates (AICs) are the same, and that KACs may constitute a later intermediate state that is perhaps more committed to transformation towards a malignant phenotype. Our new findings also further show increased congruence between human and murine KACs with both displaying driver $KRAS$ mutations.

Fig. 2p. Fraction of $KRAS^{G12D}$ cells across malignant cells or KACs and in KM-LUADs, all LUADs, or NL from KM-LUADs.

Extended Data Fig. 5c. *KRAS* variant allele frequencies across diverse alveolar and malignant subsets from our scRNA-seq LUAD and/or normal samples and from patients with KM- and/or *KRAS*-WT LUADs.

Extended Data Fig. 5d. Fractions of $KRAS^{G12D}$ mutant cells across alveolar and malignant cell subsets from the human LUAD and/or normal samples and from patients with KM- and/or $KRAS$ -WT LUADs.

Accordingly, we revised the manuscript as such:

Results section (lines 310 - 324): “Our findings on high expression of $KRAS$ signature in KACs prompted us to evaluate potential copy number changes and inferred $KRAS^{G12D}$ driver mutations in KACs and in comparison to other cell subsets. We found that KACs exhibited moderately elevated CNV burdens relative to AT2, AT1, and other AICs, albeit their CNV scores were considerably lower compared to malignant cells (**Extended Data Fig. 5a, b**). $KRAS^{G12D}$ was indeed present in malignant cells with a variant allele frequency (VAF) of 78% in KM-LUADs (**Fig. 2p** and **Extended Data Fig. 5c** and **Supplementary Table 9**). Importantly, KACs, but not AT2, AT1, or other AICs, harbored $KRAS^{G12D}$ mutations (**Fig. 2p** and **Extended Data Fig. 5c, d**). $KRAS$ -mutant KACs were exclusively found in tissues (primarily tumours) from KM-LUADs and, thus, $KRAS^{G12D}$ VAF (10%) was higher in KACs from KM-LUADs compared to when examined using all LUADs (5%) or all samples (3%) (**Fig. 2p** and **Extended Data Fig. 5c, d**). Intriguingly, $KRAS^{G12D}$ mutations were detected in KACs of NL tissues from KM-LUAD patients (VAF 2%), and other $KRAS$ variants ($KRAS^{G12C}$) were detected in NL of one KM-LUAD, signifying a potential field cancerization effect that is associated with mutant $KRAS$ (**Fig. 2p** and **Extended Data Fig. 5c, d**).”

Comment 4: Related to #3: Because of the high prevalence of $KRAS$ mutations in the mouse model and the lower prevalence in the human samples studied, is it possible that the KACs in both species are different cell populations? (The 21% overlap in the signature analysis seems low). Are there data from for example EGFR-mutant mice with LUAD that could help resolve this question?

Response 4: We thank the reviewer for pointing this out and for suggesting alternative ways to validate our hypothesis. It is important to note that in our original manuscript we showed that mouse KACs displayed similar enriched pathways/transcriptional programs (now **Extended Data Fig. 7c**) when compared to human malignant cells from KM-LUADs (Meta-C1, now **Fig. 1h**), but not to malignant cells from *KRAS* wildtype LUADs, such as EMT, senescence, and poorer alveolar differentiation (now **Fig. 1h, i**). Yet, we do acknowledge that the reader might be perplexed by the modest statistical overlap between KAC human and mouse signatures, and we agree that interrogating other mouse models, say *EGFR*-mutant, may help resolve this issue. We are limited by our capacity to study *EGFR*-mutant mouse models by scRNA-seq, since public scRNA-seq datasets of mouse models of *EGFR*-mutant LUAD were unavailable at the time of this revision. As an alternative, we employed different strategies to address this astute comment.

First, we found that the human KAC signature, which we had derived from our human data (**Fig. 2k**), was significantly enriched in KAC cells from *KRAS*-mutant relative to *EGFR*-mutant patients ($P < 2.2e-16$; **Fig. 2l**, left). This KAC signature was also markedly elevated in human malignant cells from patients with KM-LUAD versus those from *EGFR*-mutant LUAD patients ($P < 2.2e-16$; **Fig. 2l**, right).

Fig. 2l. Enrichment of KAC signature among KACs (left) and malignant cells (right) from KM- or EM-LUAD samples.

Second, we derived a murine KAC signature which we confirmed to be increased in mouse KACs and malignant cells relative to normal alveolar subsets (**Extended Data Fig. 7e**). We then found that this mouse KAC signature was significantly increased in human KACs ($P = 0.04$) and malignant cells ($P < 2.2 \times 10^{-16}$) from *KRAS*-mutant versus *EGFR*-mutant patients of our human scRNA-seq cohort (**Extended Data Fig. 8l**).

Extended Data Fig. 8l. Mouse KAC signature in human KACs (left) and malignant cells (right) from KM-LUADs relative to EM-LUADs in our human scRNA-seq dataset. Box, median \pm interquartile

range; whiskers, 1.5x interquartile range; center line: median. P-values were calculated using Wilcoxon Rank-Sum test.

Third, we performed spatial transcriptomics analysis using the Visium platform from 10X Genomics of human KM-LUAD (P14; **Fig. 2j**) and mouse tumour-bearing lungs from multiple mice (**Fig. 3f** and **Extended Data Fig. 6h-j**) and not only confirmed the *in situ* presence of KACs (histologically annotated as reactive pneumocytes) in normal-appearing regions surrounding tumours, but also found that both human and mouse KACs, like their nearby tumours, displayed enrichment for KRAS and KAC signatures (**Fig. 2j** and **Fig. 3f** and **Extended Data Fig. 6h-j**).

Fig. 2j. ST analysis of P14 LUAD showing histological annotation of H&E-stained Visium slide (10XGenomics) (left) and spatial heatmaps showing CNV score, as well as scaled expression of *KRT8*, KAC markers, and KRAS signature.

Fig. 3f. ST analysis of lung tissue at 7 mo post-exposure to NNK and showing histological annotation of H&E-stained Visium slide (10X Genomics) (left) and spatial heatmaps showing scaled expression of *KRT8* as well as KAC and KRAS signatures.

Extended Data Fig. 6h. ST analysis of the same tumour-bearing mouse lung in **Fig. 3f** with UMAP of cell clusters identified by Seurat (left) and mapped spatially (middle). Spatial maps with scaled expression of *Krt8* and *Plaur* are shown on the right.

Fourth, we performed gene set/pathway enrichment analysis of human KACs relative to other human alveolar cell subsets and malignant cells. We found that, like their mouse counterparts (**Extended Data Fig. 7c**), human KACs displayed increased activation of pathways involving hypoxia, EMT, p53, NFκB, cadherin binding, and positive regulation of wound healing (**Extended Data Fig. 7d**).

Extended Data Fig. 7c,d. Pathway enrichment analysis of KACs relative to other alveolar cell subsets and malignant cells in tumour-bearing mice at 7 months post-NNK (**c**) and in the human LUAD scRNA-seq dataset (**d**).

Fifth, in response to an additional comment (comment #8) by the same reviewer (see below), we interrogated KACs and KAC expression features in publicly available cohorts of mutated *Kras*^{G12D}-driven mouse models. We performed integrated analysis of cells from our

model (shown in **Fig. 3 a,b,d,e**) with cells from *Kras*^{G12D}-driven mice from the study by Marjanovic et al ⁷ (KT mice for short) and from the study by Dost et al ⁸ (KY mice for short). Following clustering analysis, we identified a cluster (C5) of cells from the three studies (our study, and the papers by Marjanovic and Dost) with distinctly high expression of KAC markers (**Extended Data Fig. 8g, h, i**). Although, we found that the majority of KACs in this C5 cluster originated from our model (**Extended Data Fig. 8j**, left), we did find evidence for cells in C5 from the KT and KY mice and with markedly higher KAC marker expression relative to pooled AT2 cells (**Extended Data Fig. 8j**, right) thus providing association between KACs and *Kras* mutation (Sixth, we interrogated the dataset on human AT2 cells with and without inducible expression of mutant *KRAS*^{G12D} from the study by Dost et al ⁸ and found that our mouse KAC signature was markedly and significantly increased in AT2 cells with induced expression of *KRAS*^{G12D} relative to AT2 cells with wild type *KRAS* ($P < 2.2e-16$; **Extended Data Fig. 8k**). **Extended Data Fig. 8g-k are shown below in response to comment #8 by the same reviewer.**

Collectively, these data provide additional support that the KACs we identified in the mouse studies are pertinent to KM-LUAD biology. These new analyses are described in the revised manuscript as follows:

Results section (lines 286 - 288): “Of note, KAC signature was significantly higher in KACs and in malignant cells from KM-LUAD tumour tissues relative to those from EM-LUADs ($P < 2.2 \times 10^{-16}$; **Fig. 2l**).”

Results section (lines 267 - 277): “Spatial transcriptomics (ST) analysis of P14 tumour tissue using the Visium platform (10X Genomics) demonstrated increased expression of *KRT8* in tumour regions with high CNV scores, as well as in tumour-adjacent normal-appearing regions that were histopathologically evaluated to comprise highly reactive pneumocytes and that exhibited low/moderate CNV scores (**Fig. 2j**)..... Similarly, we found that a signature based on the 5 KAC markers (see **Fig. 2e**) was high in tumour and in adjacent-to-tumour regions with reactive pneumocytes, with a similar and spatially overlapping pattern for *KRAS* signature.”

Results section (lines 379 - 386): “We then studied by ST lung tissues from mice at 7 months post-exposure to NNK. We found that areas histologically-annotated as tumour regions had significantly increased expression of *Krt8*, *Plaur*, as well as KAC and *KRAS* signatures which were also overlapping, further confirming KACs enrichment in mouse tumours induced by tobacco carcinogen exposure (**Fig. 3f and Extended Data Fig. 6h**). In line with our data from human LUAD patients (**Fig. 2j**), *Krt8* high KACs with elevated expression of KAC and *KRAS* signatures were enriched in “reactive”, non-neoplastic regions surrounding tumours of NNK-exposed mice”. We also provided a new detailed subsection in the **Methods section** that describes the ST workflow (**lines 1083 – 1128**).

Results section (398 - 403): “We first performed pathway enrichment analysis of KACs relative to other alveolar cell subsets and malignant cells in tumour-bearing mice at 7 months post-NNK and in the human LUAD scRNA-seq dataset, and found that both mouse (**Extended Data Fig. 7c**) and human (**Extended Data Fig. 7d**) KACs displayed increased

activation of pathways involving hypoxia, EMT, p53, NFκB, cadherin binding, and positive regulation of wound healing.”

Results section (lines 437 - 446): “Given the presence of *Kras*^{G12D} mutations in KACs, we next sought to probe for evidence of *Kras*-mutant KACs or cells with KAC expression features in *Kras*^{G12D}-driven mouse models (e.g., *Kras*^{LSL-G12D}). We performed integrated scRNA-seq analysis of cells from our mouse cohort with cells in mice driven by *Kras*^{G12D} from the two separate studies by Marjanovic et al ⁷ and Dost et al ⁸. Following clustering, we identified a shared cluster (C5) of cells from all three studies with distinctly high expression of KAC markers and the KAC signature itself (**Extended Data Fig. 8g-i**). Notably, the overwhelming fraction of C5 cells were from our study (**Extended Data Fig. 8j, left**). Despite representing a smaller proportion of C5, cells from *Kras*^{G12D}-driven mice still expressed higher KAC signature compared with normal AT2 cells from all studies (**Extended Data Fig. 8j, right**).”

Results section (446 - 450): “To further probe the association of KAC gene features with mutant *KRAS*, we analyzed our KAC signature in the dataset of human AT2 cells with and without inducible *KRAS* from the Dost et al study. We found that the KAC signature was markedly and significantly increased in AT2 cells with induced expression of *KRAS*^{G12D} relative to those with wild type *KRAS* ⁸ ($P < 2.2 \times 10^{-16}$, **Extended Data Fig. 8k**).”

Results section (lines 450 - 453): “In agreement with these findings, we also found that our murine KAC signature was significantly enriched in KACs and malignant cells from KM-LUADs relative to EM-LUADs in our human scRNA-seq dataset ($P = 0.04$ and $P < 2.2 \times 10^{-16}$, respectively; **Extended Data Fig. 8l**).”

Comment 5: Related to #3: In Figure 4, why are non-KAC AICs absent from the analysis? Or do mice in this model not have them?

Response 5: We thank the reviewer for this insightful comment. Unlike our human dataset, our LUAD mouse model enables longitudinal assessment of KACs along the course of LUAD progression. Our carcinogenesis mouse model confers a follow-up post-exposure to tobacco carcinogen (NNK) or control saline, and whereby we analyzed murine lungs by scRNA-seq at both end-of-exposure (EOE) as well as at 7 months post-exposure to saline or NNK (now **Fig. 3a**). Early on at EOE, carcinogen-exposed mouse lungs are normal-appearing and devoid of any lesions, in contrast to lungs analyzed at 7 months post-exposure to NNK which harbor LUADs (**Fig. 3b** and **Extended Data Fig. 6f** and in line with our previous studies ⁹. In our original manuscript, we studied KAC markers and other features (e.g., *Kras* mutation) in all KACs lumped together. Now, to interrogate the continuum of KAC progression with time and following exposure to tobacco carcinogen, we compared and contrasted different features of KACs across both time points in the NNK treated mice.

We found that the fraction of KACs with *Kras*^{G12D} mutation at EOE is low (~0.02), whereas this fraction is significantly higher at 7 months post-NNK (~0.19) (**Extended Data Fig. 8b**). We then compared and contrasted *Kras*^{G12D}-mutant and *Kras* wild type (*Kras*^{wt}) KACs at 7 months post-NNK. *Kras*^{G12D}-mutant KACs exhibited significantly higher cytoTRACE score

(i.e., were less differentiated) compared with *Kras^{wt}* KACs (**Extended Data Fig. 8c**). *Kras^{G12D}*-mutant KACs also showed higher levels of KAC compared with *Kras^{wt}* KACs (**Extended Data Fig. 8d**).

Extended Data Fig. 8b-d. **b**, Fraction of *Kras^{G12D}* mutations in different mouse alveolar cell subsets including when separating KACs at EOE (early) and 7 months post-NNK (late KACs). **c**, CytoTRACE score in late KACs with (mut) or without (wt) *Kras^{G12D}* mutation. P-value was calculated using Wilcoxon Rank-Sum test. Box, median \pm interquartile range; whiskers, 1.5 \times interquartile range; center line: median. **d**, Expression (proportions and average expression levels) of selected marker genes for *Kras^{G12D}* mutant KACs (KAC *Kras^{G12D}*), *Kras^{wt}* KACs (KAC *Kras^{wt}*) as well as malignant, AT1 and AT2 subsets.

It is important to note that we profiled AT2 lineage-labeled cells by scRNA-seq, in response to another comment (comment #6) by the same reviewer, in mice at 3 months post-NNK (**Fig. 4b-e,g** and **Extended Data Fig. 10a, b**) and found that KACs at this intermediate time point (3 months post-NNK) had already acquired high frequency of *Kras* mutations (**figures are shown below for comment #6 by the same reviewer**).

These results collectively suggest that the identified KACs in our tobacco carcinogenesis mouse model most likely exist in a continuum. Mouse KACs at EOE are perhaps closer to an AIC phenotype and more reflective of a response to injury (i.e., injury to tobacco carcinogen) and similar in concept to what was reported in lung injury models such in the study by Strunz and colleagues using bleomycin as a stimulus for lung injury². On the other hand, “later” KACs at 7 months post-NNK especially those that are *Kras^{G12D}*-mutant are perhaps more advanced in the transformation trajectory towards KM-LUAD and thus, are more representative of KACs found at the time of frank lung malignancy as observed in our human analysis. Indeed, when we compared our differentially expressed genes (DEGs) in our mouse KACs with Krt8⁺ intermediate cells that were described by Strunz et al to be associated with acute lung injury, we found that the overlap between *Kras^{wt}* KACs and these previously reported cells was much higher (20.5%) when compared with *Kras^{G12D}*-mutant KACs (9.9%) (**Extended Data Fig. 8e, f**).

Extended Data Fig. 8e,f. Pie charts showing percent of unique and overlapping DEG sets between *Krt8*+ transitional cells identified by Strunz and colleagues and either *Kras*^{G12D} (e) or *Kras*^{wt} (f) KACs from this study.

Accordingly, we revised the manuscript as such:

Results section (lines 422 - 436): “With the capacity to look at temporal dynamics of alveolar transitions and the emergence of KACs in our carcinogenesis mouse model, we took a closer look at KACs by timepoint and *Kras*^{G12D} mutation status and found that those at EOE were somewhat less differentiated relative to the more differentiated KACs at 7 months post-exposure (Fig. 4a, bottom right). Interestingly, the fraction of KACs with *Kras*^{G12D} mutation at EOE was low (~0.02), whereas this fraction was significantly increased at 7 months post-NNK (~0.19) (Extended Data Fig. 8b). By further investigating KACs from the late timepoint, we found that those with *Kras*^{G12D} were significantly less differentiated ($P = 9.2 \times 10^{-6}$; Extended Data Fig. 8c) and showed higher expression of KAC signature genes such as *Cldn4*, *Krt8*, *Cavin3*, and *Cdkn2a* relative to KACs without the mutation (Extended Data Fig. 8d). Intrigued by these findings, we separately compared our *Kras*^{wt} and *Kras*-mutant KACs to previously reported *Krt8*+ intermediate cells (Strunz et. al²). Despite the overall similarities we had previously noted between KACs at large and these cells (31.6%; Extended Data Fig. 7a), *Kras*^{wt} KACs were more similar to *Krt8*+ intermediate cells than *Kras*-mutant KACs (20.5 % overlap versus 9.9%) (Extended Data Fig. 8e, f).”

Results section (lines 456 - 473): “We further investigated the biology of KACs using *Gprc5a*^{-/-} mice with reporter labeled-AT2 cells (*Gprc5a*^{-/-}; *Sftpc*^{CreER/+}; *Rosa*^{Sun1GFP/+}) at EOE to NNK or saline (Fig. 4b).....We also monitored reporter mice up to 3 months post-NNK or -saline and analyzed GFP⁺ and GFP⁻ cell fractions by scRNA-seq (2 mice/group). We found that GFP⁺ cells (n = 3,089) almost exclusively comprised AT2, early tumour/AT2, KAC/KAC-like cells, and few AT1 cells, all of which were nearly absent in the GFP⁻ fraction (Fig. 4c, d; Supplementary Fig. 7). There were markedly increased fractions of GFP⁺ AT1, KACs, and, expectedly, tumour cells from NNK- versus saline-treated mice (Fig. 4e).....Notably, GFP⁺ KACs from this time point, that coincides with formation of preneoplasias⁹, harbored driver *Kras*^{G12D} mutations at similar fractions when compared with early tumour/AT2 cells (Fig. 4g and Extended Data Fig. 10a, b).”

Comment 6: The statement “tumor-associated KACs, a subset of AICs that arises from AT2 cells” is not directly supported by experimental evidence: the GFP reporter was activated

after 6 weeks of treatment and may not label specifically AT2 cells then. Could the authors label AT2 cells first and then induce tumors to verify that these tumors are indeed from AT2 cells and that KACs are GFP+ in this context?

Response 6: We thank the reviewer for this constructive comment. We apologize for inadvertently failing to accentuate that there are no tumors or tumor cells in the tobacco lung carcinogenesis mouse model we employed at the time of cessation of tobacco carcinogen (End of NNK timepoint; EOE), which is when we labeled cells. Thus, we had labeled cells prior to emergence of tumors which typically happens several months after cessation of tobacco carcinogen (NNK). During the time the manuscript was being reviewed, we had an ongoing experiment where we similarly labeled *Gprc5a*^{-/-}; *Sftpc*^{CreER/+}; *Rosa*^{Sun1GFP/+} mice at EOE to NNK or saline and which were then followed up to 3 months post-NNK or -saline (now **Fig. 4b**). In an attempt to address the reviewer's important concern, we collected GFP⁺ and GFP⁻ cellular compartments from saline and NNK-exposed *Gprc5a*^{-/-}; *Sftpc*^{CreER/+}; *Rosa*^{Sun1GFP/+} mice at 3 months post-exposure (n = 2 per group) (**Fig. 4b**). We used some GFP⁺ cells from NNK- and saline-treated mice to derive and culture organoids in air liquid interface (ALI). We performed scRNA-seq of GFP⁺ and GFP⁻ cell fractions from NNK- and saline-treated mice. Using additional mice (n = 2 per group), we also performed immunofluorescence (IF) analysis of GFP, Krt8, Cldn4 or Cavin3 (KAC markers), Pdpn (an AT1 marker), as well as, in some cases, Lamp3 (an AT2 marker). We herein summarize findings from these experiments and analyses that allow us to respectfully suppose that we were able to label AT2 cells and that KACs likely reside in the trajectory from AT2 to tumor cells.

In our scRNA-seq analysis, not only did we find AT2 cells in the GFP⁺ compartment, but that AT2, other alveolar cell subsets (including KACs), and tumor cells were almost exclusively found in the GFP⁺ compartment (from both saline and NNK-treated mice), i.e., there were no non-labeled AT2 or alveolar cells in the GFP⁻ fraction (**Fig. 4c,d** and **Supplementary Fig. 7**).

Fig. 4c,d. **c**, UMAP distribution of GFP⁺ cells at 3 months following NNK exposure or saline and colored by alveolar or tumour subsets. **d**, Expression (proportions and average expression levels) of selected marker genes for mouse normal alveolar cell lineages and tumour cells defined in panel c.

Supplementary Fig. 7. Proportions and average expression levels of selected marker genes for alveolar, tumour, and basal cell clusters in GFP⁺ and GFP⁻ samples from saline- and NNK-treated AT2 lineage-labeled mice.

Our scRNA-seq analysis of these sorted cells also showed that there were markedly increased fractions of AT1, KACs, and, expectedly, tumor cells in GFP⁺ cells from NNK- versus saline-treated mice (**Fig. 4e**).

Fig. 4e. Fractions of AT1, AT2, KAC/KAC-like, and early tumour/AT2 cells across GFP⁺ cells from lungs of 2 NNK and 2 saline-exposed mice analyzed at 3 months-post exposure.

Notably, GFP⁺ KACs from this time point (3 months post-NNK, coincides with formation of preneoplasias, see ⁹) harbored driver *Kras*^{G12D} mutations (**Fig. 4g** and **Extended Data Fig. 10a, b**). Our pseudotime analysis also inferred that KACs were along the trajectory from early AT2 cells, with tumor cells, and prior to generation of AT1 cells (**Extended Data Fig. 10c**), in ways very similar to our findings with *Gprc5a*^{-/-} mice (**Fig. 4a**).

Fig. 4g. Fraction of *KRAS*^{G12D} cells across alveolar and early tumour subsets. Absolute numbers of *Kras*^{G12D} cells are indicated on top of each bar.

Extended Data Fig. 10a-c. **a**, UMAPs of GFP⁺ cells from tumour-bearing reporter mice at 7 months post-NNK or saline colored by presence of *Kras*^{G12D} mutation or expression of KAC, AT1, and AT2 signatures. **b**, UMAPs (top) showing distribution of alveolar and tumour cell subsets as well as cells with *Kras*^{G12D} mutation by treatment (saline or NNK). **c**, Pseudotime (left) and differentiation trajectories (middle) of GFP⁺ cells from tumour-bearing reporter mice at 7 months post-NNK or saline. Right panel shows pseudotemporal trajectory color-coded by subset composition (right).

Our IF analysis of tumor tissues also showed the presence of GFP⁺/Lamp3⁺/Krt8^{low} AT2 cells in normal (non-tumoral) lung regions from NNK-exposed *Gprc5a*^{-/-}; *Sftpc*^{CreER/+}; *Rosa*^{Sun1GFP/+} mice. Normal lung regions also included KACs that were GFP⁺/Lamp3⁺/Krt8⁺ or GFP⁺/Krt8⁺ that were also KAC marker positive⁺ (Cldn4/Cavin3) (**Fig. 4f** and **Extended Fig. 9c, d**). Importantly, tumor cells from the same mice were almost entirely GFP⁺ as well as Krt8⁺ and KAC marker⁺ (Cldn4/Cavin3) (**Fig. 4f** and **Extended Fig. 9c, d**).

Analysis of organoids from GFP⁺ sorted fractions not only showed significantly and markedly enhanced growth of cells from NNK- versus saline-treated mice, but that cells from NNK-exposed mice at three months post-exposure, much like what we showed previously at EOE (now **Extended Data Fig. 9a**) were almost exclusively GFP⁺, Krt8⁺, and KAC marker⁺ (**Extended Data Fig. 11a, e**).

Extended Data Fig. 11a. Size quantification of organoids derived from GFP⁺ lungs cells of saline- or NNK-treated mice at 3 months post-exposure. Box, median \pm interquartile range; whiskers, 1.5 \times interquartile range; center line: median. *P*-value was calculated using Wilcoxon Rank-Sum test.

While we agree with the reviewer that ideally labelling prior to commencing NNK exposure will result in labeling AT2 cells, we surmise that the nature of tumour development in our tobacco model (absence of tumor cells at EOE to NNK), and our new findings from analysis of *Gprc5a*^{-/-}; *Sftpc*^{CreER/+}; *Rosa*^{Sun1GFP/+} mice suggest adequate labeling of AT2 cells and that KACs reside along the trajectory from AT2 to tumor cells (or are at least associated with tumour development).

These new findings are described in the following sections of the revised manuscript as such:

Results section (lines 456 - 477): We further investigated the biology of KACs using *Gprc5a*^{-/-} mice with reporter labeled-AT2 cells (*Gprc5a*^{-/-}; *Sftpc*^{CreER/+}; *Rosa*^{Sun1GFP/+}) at EOE to NNK or saline (**Fig. 4b**).....We also monitored reporter mice up to 3 months post-NNK or -saline and analyzed GFP⁺ and GFP⁻ cell fractions by scRNA-seq (2 mice/group). We found that GFP⁺ cells (n = 3,089) almost exclusively comprised AT2, early tumour/AT2, KAC/KAC-like cells, and few AT1 cells, all of which were nearly absent in the GFP⁻ fraction (**Fig. 4c, d; Supplementary Fig. 7**). There were markedly increased fractions of GFP⁺ AT1, KACs, and, expectedly, tumour cells from NNK- versus saline-treated mice (**Fig. 4e**). IF analysis showed that GFP expression was almost exclusive to alveolar regions (**Extended Data Fig. 9b**) and tumours (**Extended Data Fig. 9c**) and that tumours were almost entirely GFP⁺ as well as Krt8⁺ and KAC marker⁺ (Cldn4, Cavin3) (**Fig. 4f and Extended Data Fig. 9c**). Normal lung regions included KACs that were GFP⁺/Krt8⁺ and also KAC marker positive⁺ (Cldn4, Cavin3) (**Fig. 4f and Extended Data Fig. 9c**). GFP⁺/LAMP3⁺/KRT8^{-/low} AT2 cells were evident

including in normal (non-tumoural) lung regions from NNK-exposed reporter mice (**Extended Data Fig. 9d**). Notably, GFP⁺ KACs from this time point, that coincides with formation of preneoplasias⁹, harbored driver *Kras*^{G12D} mutations at similar fractions when compared with early tumour/AT2 cells (**Fig. 4g and Extended Data Fig. 10a, b**). Furthermore, KACs were not only positioned along the trajectory from early AT2 cells, with tumour cells, and prior to generation of AT1 cells, but they were also less differentiated relative to AT2 or early tumour cells (**Extended Data Fig. 10c, d**), in agreement with our previous findings in *Gprc5a*^{-/-} mice (**Fig. 4a**)."

Results section (lines 479 - 482): "We then analyzed organoids which we derived from the same sorted GFP⁺ cells. GFP⁺ organoids from NNK- versus saline-treated mice not only showed significantly and markedly enhanced growth but that they were almost exclusively comprised of GFP⁺, Krt8⁺, and Cldn4⁺ cells (**Extended Data Fig. 11a,e**)."

Comment 7: Gprc5a knockout mice develop tumors without NKK (a bit slower), do these tumors have KACs or KACs specific to NNK-induced tumors? (Again, are the KACs in the mouse model simply early KRAS-mutant cells?)

Response 7: We thank the reviewer for this important comment. We apologize for inadvertently failing to explain tumor status in *Gprc5a*^{-/-} saline-treated animals. In accordance with our previous reports⁹, *Gprc5a*^{-/-} saline-treated animals that we studied in our original submission, did not show any lesions by histopathological analysis nor tumor cells by scRNA-seq analysis (**Fig. 3b and Extended Data Fig. 6f**). Additionally, we found that KACs were far more prevalent in NNK-exposed animals. Autochthonous tumor formation in this model occurs in mice with an age of least 14-16 months¹⁰. We thus agree with the reviewer that KACs in our model are pertinent to exposure to tobacco carcinogen (NNK) and hypothesize that they are early *Kras*-mutant cells (given our above-described findings), while also acknowledging unique attributes to our tobacco carcinogenesis model such as KACs themselves being at a continuum (our response to comment #5 by the same reviewer). We now address the relevant pertinence of KACs in our mouse model to tobacco carcinogen exposure in our revised manuscript as such:

Discussion section (lines 561 – 576): "Temporal analysis of lungs post-tobacco along the spectrum of normal-appearing (tumour-free) to tumour-bearing lungs enabled us to identify evolving features of KACs, most notably, somatic acquisition of *Kras* mutations and reduced differentiation, very much reminiscent of their human counterparts..... Notably, that discriminating expression features and markers of the mouse KACs were not only enriched in profiles of alveolar intermediate cells from mouse models of acute lung injury² but they were also found in cell subsets from animals whose tumours are driven by lung-specific expression of *Kras*^{G12D}^{7,8}, albeit at lesser frequency compared to our tobacco-injury carcinogenesis model. Thus, it is plausible that KACs can arise due to an injury stimulus (here tobacco exposure) or following expression of mutant *Kras* – or to both, since in our tobacco model KACs exist in a continuum, i.e., low versus high frequency of *Kras* mutations. Intriguingly, our high-resolution imaging and ST analyses showed that KACs were evident in normal-appearing areas in the vicinity of lesions in both murine and patient samples, suggesting that the early appearance of these cells (e.g., following tobacco exposure) may represent development of a *field of injury*¹¹."

Comment 8: Are KACS present in genetically engineered mouse models in which tumors are initiated by mutant KRASG12D? (a number of datasets are available to address this question). One could imagine that in these models, there is no “need” for a KAC intermediate as tumors can readily reach the KRAS mutant state.

Response 8: We thank the reviewer for this constructive comment. To address the reviewer’s comment, we opted to analyze two available datasets.

We queried the publicly available dataset from Marjanovic et al ⁷ and analyzed AT2-lineage labeled cells from that study’s cohort of mice that are driven by $Kras^{G12D}$, $Kras^{LSL-G12D/+}$; $Rosa26^{LSL-tdTomato/+}$ (KT mice for short) and infected with AT2 cell-specific *Sftpc* promoter (AdSftpc-Cre). We also analyzed lung cells from $Kras^{LSL-G12D;LSL-YFP}$ (KY mice for short) with Ad5CMV-Cre infection from the study by Dost et al ⁸. We performed integrated analysis of cells from our model with cells from KT and KY mice from the Marjanovic and Dost studies, respectively (**Extended Data Fig. 8g**). Following clustering analysis, we identified a cluster (C5) of cells from the three studies (our study, and the papers by Marjanovic and Dost) with distinctly high expression of KAC markers (**Extended Data Fig. 8g-i**). We found that the overwhelming fraction of KACs or cells with high expression of various KAC markers in C5 were from our study (**Extended Data Fig. 8j**, left). Although, much fewer in number, we did find cells in C5 from the Marjanovic and Dost studies that showed much higher KAC signature expression when compared to pooled AT2 cells suggesting the presence of a small fraction of likely KAC cells from tumor-bearing lungs of $Kras^{G12D}$ -driven mice (**Extended Data Fig. 8j**, right).

Extended Data Fig. 8g. UMAP clustering of cells integrated from our mouse cohort with cells from scRNA-seq studies by Marjanovic et al and Dost et al.

Extended Data Fig. 8h. Expression (proportions and average expression levels) of selected marker genes for diverse alveolar and tumour subsets and across clusters defined in **Extended Data Fig. 8g**.

Extended Data Fig. 8i,j. i, KAC signature score across clusters defined in **Extended Data Fig. 8g**. j, Distribution of cells from C5 across the three cohorts (left). KAC signature enrichment across KACs from the three cohorts and relative to pooled AT2 cells (right). Box, median ± interquartile range; whiskers, 1.5x interquartile range; center line: median.

These new analyses and findings allow us to surmise that KACs are much more frequently encountered in our tobacco lung carcinogenesis model which includes a stimulus (NNK) for an injury response that may parallel or even precede somatic acquisition of *Kras*. Orthogonally, our finding of few cells with KAC expression features (i.e., likely KAC cells) in KT and KY suggest the presence of the intermediate cell state following lung epithelium-specific expression of mutant *Kras* allele. Indeed, when we interrogated the dataset on human AT2 cells with and without inducible expression of mutant *KRAS*^{G12D} from the same study by Dost et al⁸, we found that our mouse KAC signature was markedly and significantly increased in AT2 cells with induced expression of *KRAS*^{G12D} relative to AT2 cells with wildtype *KRAS* (**Extended Data Fig. 8k**).

Extended Data Fig. 8k. KAC signature score in human AT2 cells with induced expression of KRAS^{G12D} (Dox) relative to KRAS^{wt} cells (Ctrl) from the Dost et al study. Dox: Doxycycline. Box, median ± interquartile range; whiskers, 1.5x interquartile range; center line: median.

It is plausible that KACs can arise due to an injury stimulus (e.g., tobacco exposure) or following expression of mutant *Kras* – or to both, since in our tobacco model KACs exist in a continuum, i.e., low versus high frequency of *Kras* mutations. Interestingly, our new findings further lend support to the association of KACs with *KM-LUAD* tumorigenesis (our response to comment #4 by the same reviewer).

These new data are now described in the revised manuscript as such:

Results section (lines 437 - 446): “Given the presence of *Kras*^{G12D} mutations in KACs, we next sought to probe for evidence of *Kras*-mutant KACs or cells with KAC expression features in *Kras*^{G12D}-driven mouse models (e.g., *Kras*^{LSL-G12D}). We performed integrated scRNA-seq analysis of cells from our mouse cohort with cells in mice driven by *Kras*^{G12D} from the two separate studies by Marjanovic et al ⁷ and Dost et al ⁸. Following clustering, we identified a shared cluster (C5) of cells from all three studies with distinctly high expression of KAC markers and the KAC signature itself (**Extended Data Fig. 8g-i**). Notably, the overwhelming fraction of C5 cells were from our study (**Extended Data Fig. 8j, left**). Despite representing a smaller proportion of C5, cells from *Kras*^{G12D}-driven mice still expressed higher KAC signature compared with normal AT2 cells from all studies (**Extended Data Fig. 8j, right**).”

Results section (446 - 450): “To further probe the association of KAC gene features with mutant *KRAS*, we analyzed our KAC signature in the dataset of human AT2 cells with and without inducible *KRAS* from the Dost et al study. We found that the KAC signature was markedly and significantly increased in AT2 cells with induced expression of *KRAS*^{G12D} relative to those with wild type *KRAS* ⁸ ($P < 2.2 \times 10^{-16}$, **Extended Data Fig. 8k**).”

Comment 9: Can mouse KACs be isolated using cell surface markers, do they form organoids or tumors when isolated?

Response 9: We thank the reviewer for this comment. In order to address whether it might be possible to isolate of KACs using cell surface markers, we treated *Gprc5a*^{-/-}; *Sftpc*^{CreER/+}; *Rosa*^{Sun1GFP/+} lineage tracer mice with NNK and collected total lung single cells following tamoxifen treatment and AT2 labeling. We stained cells with fluorescent-tagged antibodies

targeting surface proteins highly expressed in our derived mouse KAC signature, namely, Cavin3 and Cldn4, and analyzed the cells by flow cytometry. Our aim was to gate, out of the GFP⁺ cells, KAC-marker expressing cells. However, the proportion of these cells from the total population of mouse lung cells analyzed was very low, which is not surprising, to assess in animals. We previously derived cancer stem cell (CSC)/stem cell-like progenitor cells in the form of spheres from parental MDA-F471 cells (a cell line we had developed and cultured from a KM-LUAD from an NNK-exposed *Gprc5a*^{-/-} mouse). We had also interrogated gene expression profiles that were different between the spheres and their parental cell counterparts. To address, albeit indirectly, the insightful comment by the reviewer, we queried this previous dataset of ours and compared the mouse KAC signature between the spheres enriched with CSCs and their parental cell counterparts. Interestingly, we found that the mouse KAC signature and levels of individual KAC markers were markedly higher in the CSC-enriched spheres relative to parental mouse KM-LUAD cells grown in 2D (**Extended Data Fig. 8a**). These data suggest that markers of KACs, and thus perhaps KACs themselves, are associated with formation of tumors. We discuss these findings in the revised manuscript as such:

Extended Data Fig. 8a. Murine KAC signature score (left) and heatmap showing expression of select KAC marker genes (right) in bulk transcriptomes of MDA-F471-derived 3D spheres versus parental MDA-F471 cells grown in 2D. *P*-value was calculated using Wilcoxon Rank-Sum test. Box, median \pm interquartile range; whiskers, 1.5 \times interquartile range; center line: median.

We opted to include these findings in the revised manuscript as such:

Results section (lines 419 - 422): "Of note, we also found that the KAC signature was elevated in spheres enriched for cancer stem cell (CSC)-progenitor like cells, and that we had derived from *Gprc5a*^{-/-} LUAD cells (MDA-F471 cell line; previously derived from a LUAD in an NNK-exposed *Gprc5a*^{-/-} mouse; ¹²), relative to parental 2D cells (**Extended Data Fig. 8a**)."

Referee #3 (Remarks to the Author):

General comment: In Han et al., the team sequenced 16 early stage LUAD samples and 47 normal lung controls. They found that tumor cells primarily clustered based on their donor origin. KRAS mutant cell clusters are more similar to each other, and are less differentiated compared to tumors cells carrying other classes of mutations. A KRT8+ alveolar cell type was identified near KRAS mutant cells, and are predicted to be in a cellular state between AT2 and AT1. These cells are also found in a mouse model of LUAD, and are deemed “likely transitioned to malignant cells.”

This is a comprehensive study of early stage LUAD single cell transcriptomic signature, combined with genotyping data. The findings reveal single cell resolution transcriptomic differences between tumors with different mutations, and also heterogeneity within tumors with the same mutation. More in-depth analysis and validation is needed to yield novel insights of broad interest.

General response: We are pleased to know that reviewer #3 finds our study comprehensive in terms of analyzing LUAD single-cell transcriptomics as well as genomics data. We provide below a point-by-point response to reviewer #3’s comments and suggestions by conducting additional validation studies, which altogether broaden the scope of our findings.

Comment 1: *The presence of KRT8 cells in the alveolar region of carcinogen injured lungs is not surprising, as these cells have been widely found in injured lungs (Kobayashi et al., NCB 2020; Strunz et al., Nat Comm 2020, Choi et al., Cell Stem Cell 2020). If the stipulation is that these cells are precursor for LUAD, additional validation is needed. TP53 signature was found in these cells, similar to previously described AT2-to-AT1 transitional cell. However, it is the increase of TP53 signature that is associated with these cells as a feature of their senescence property. How a cell with an increased tumor suppressor signature could be preferentially transformed needs to be addressed.*

Response 1: We thank the reviewer for pointing this important observation. We are excited to note that the Krt8+ cells have been observed as transitional cell states in response to fibrosis-related injury (bleomycin) in other animal studies, as in the papers cited by the reviewer. Indeed, we did cite these highly pertinent reports in our original submission (references # 30, 31, and 34 in our original report, now references # 28, 29, 35). It is reassuring to identify parallels in lung alveolar biology between our findings and those reported in mice exposed to other forms of lung injury. We performed additional analyses to gain better insight into the biology of KACs.

First, we compared KACs by time in our model (7 months post-NNK versus end-of-exposure/EOE to NNK). We found that the fraction of KACs with *Kras*^{G12D} mutation at EOE is low (~0.02), whereas this fraction is significantly higher at 7 months post-NNK (~0.19) (**Extended Data Fig. 8b**). We then compared and contrasted *Kras*^{G12D}-mutant and *Kras* wild type (*Kras*^{wt}) KACs at 7 months post-NNK. *Kras*^{G12D}-mutant KACs exhibited significantly higher cytoTRACE score (i.e., were less differentiated) compared with *Kras*^{wt} KACs (**Extended Data Fig. 8c**). *Kras*^{G12D}-mutant KACs also showed higher levels of KAC compared with *Kras*^{wt} KACs (**Extended Data Fig. 8d**).

Extended Data Fig. 8b-d. **b**, Fraction of *Kras*^{G12D} mutations in different mouse alveolar cell subsets including when separating KACs at EOE (early) and 7 months post-NNK (late KACs). **c**, CytoTRACE score in late KACs with (mut) or without (wt) *Kras*^{G12D} mutation. *P*-value was calculated using Wilcoxon Rank-Sum test. Box, median \pm interquartile range; whiskers, 1.5 \times interquartile range; center line: median. **d**, Expression (proportions and average expression levels) of selected marker genes for Kas-mutant KACs (KAC *Kras*^{G12D}), *Kras* wild type (KAC *Kras*^{wt}) KACs as well as malignant, AT1 and AT2 subsets.

These findings allow us to posit that KACs are in a continuum where at first (EOE to NNK) they are more reminiscent of Krt8⁺ intermediate cells that emerge in response to injury (with injury here caused by tobacco carcinogen exposure) such as those described in the study by Strunz and colleagues using bleomycin as a stimulus for lung injury². On the other hand, “later” KACs at 7 months post-NNK especially those that are *Kras*^{G12D}-mutant are perhaps more advanced in the transformation trajectory towards KM-LUAD and thus, are more representative of KACs found at the time of frank lung malignancy as observed in our human analysis. Indeed, when we compared our differentially expressed genes (DEGs) in our mouse KACs with Krt8⁺ intermediate cells that were described by Strunz et al to be associated with acute lung injury, we found that the overlap between *Kras*^{wt} KACs and these previously reported cells was much higher (20.5%) when compared with *Kras*^{G12D}-mutant KACs (9.9%) (**Extended Data Fig. 8e, f**).

Extended Data Fig. 8e,f. Pie charts showing percent of unique and overlapping DEG sets between Krt8+ transitional cells identified by Strunz and colleagues and either *Kras*^{G12D} (e) or *Kras*^{wt} (f) KACs from this study.

Second, we validated our finding on presence of *Kras* mutations in KACs by analysis of two publicly available scRNA-seq datasets of lung cells from *Kras*-driven mice. We queried the publicly available dataset from Marjanovic et al⁷ and analyzed AT2-lineage labeled cells from that study's cohort of mice that are driven by *Kras*^{G12D}, *Kras*^{LSL-G12D/+}; *Rosa26*^{LSL-tdTomato/+} (KT mice for short) and infected with AT2 cell-specific *Sftpc* promoter (AdSftpc-Cre). We also analyzed lung cells from *Kras*^{LSL-G12D;LSL-YFP} (KY mice for short) with Ad5CMV-Cre infection from the study by Dost et al⁸. We performed integrated analysis of cells from our model with cells from KT and KY mice from the Marjanovic and Dost studies, respectively (**Extended Data Fig. 8g**). Following clustering analysis, we identified a cluster (C5) of cells from the three studies (our study, and the papers by Marjanovic and Dost) with distinctly high expression of KAC markers (**Extended Data Fig. 8g-i**). We found that the overwhelming fraction of KACs or cells with high expression of various KAC markers in C5 were from our study (**Extended Data Fig. 8j**, left). Although, much fewer in number, we did find cells in C5 from the Marjanovic and Dost studies that showed much higher KAC signature expression when compared to pooled AT2 cells suggesting the presence of a small fraction of likely KAC cells from tumor-bearing lungs of *Kras*^{G12D}-driven mice (**Extended Data Fig. 8j**, right).

Extended Data Fig. 8g. UMAP clustering of cells integrated from our mouse cohort with cells from scRNA-seq studies by Marjanovic et al and Dost et al.

Extended Data Fig. 8h. Expression (proportions and average expression levels) of selected marker genes for diverse alveolar and tumour subsets and across clusters defined in **Extended Data Fig. 8g**.

Extended Data Fig. 8i,j. i, KAC signature score across clusters defined in **Extended Data Fig. 8g**. j, Distribution of cells from C5 across the three cohorts (left). KAC signature enrichment across KACs from the three cohorts and relative to pooled AT2 cells (right). Box, median ± interquartile range; whiskers, 1.5× interquartile range; center line: median.

Third, we now further support our observations with scRNA-seq analysis of GFP⁺ fractions and IF of tissues from AT2-lineage labeled mice at 3 months post-exposure to NNK or saline. We similarly labeled *Gprc5a*^{-/-}; *Sftpc*^{CreER/+}; *Rosa^{Sun1GFP/+}* mice at EOE to NNK or saline and which were then followed up to 3 months post-NNK or -saline (**Fig. 4b**). We collected GFP⁺ and GFP⁻ cellular compartments from saline and NNK-exposed *Gprc5a*^{-/-}; *Sftpc*^{CreER/+}; *Rosa^{Sun1GFP/+}* mice at 3 months post-exposure (n = 2 per group) (**Fig. 4b**). We used some GFP⁺ cells from NNK- and saline-treated mice to derive and culture organoids in air liquid interface (ALI). We performed scRNA-seq of GFP⁺ and GFP⁻ cell fractions from NNK- and saline-treated mice. Using additional mice (n = 2 per group), we also performed immunofluorescence (IF) analysis of GFP, Krt8, Cldn4 or Cavin3 (KAC markers), Pdpn (an AT1 marker), as well as, in some cases, Lamp3 (an AT2 marker). We herein summarize findings from these experiments and analyses that allow us to respectfully suppose that we were able to label AT2 cells and that KACs likely reside in the trajectory from AT2 to tumor cells.

In our scRNA-seq analysis, not only did we find AT2 cells in the GFP⁺ compartment, but that AT2, other alveolar cell subsets (including KACs), and tumor cells were almost exclusively found in the GFP⁺ compartment (from both saline and NNK-treated mice), i.e., there were no non-labeled AT2 or alveolar cells in the GFP⁻ fraction (**Fig. 4c,d** and **Supplementary Fig. 7**).

Fig. 4c,d. **c**, UMAP distribution of GFP⁺ cells at 3 months following NNK exposure or saline and colored by alveolar or tumour subsets. **d**, Expression (proportions and average expression levels) of selected marker genes for mouse normal alveolar cell lineages and tumour cells defined in panel **c**.

Supplementary Fig. 7. Proportions and average expression levels of selected marker genes for alveolar, tumour, and basal cell clusters in GFP⁺ and GFP⁻ samples from saline- and NNK-treated AT2-lineage labeled mice.

Our scRNA-seq analysis of these sorted cells also showed that there were markedly increased fractions of AT1, KACs, and, expectedly, tumor cells in GFP⁺ cells from NNK- versus saline-treated mice (**Fig. 4e**).

Fig. 4e. Fractions of AT1, AT2, KAC/KAC-like, and early tumour/AT2 cells across GFP⁺ cells from lungs of 2 NNK and 2 saline-exposed mice analyzed at 3 months-post exposure.

Notably, GFP⁺ KACs from this time point (3 months post-NNK, coincides with formation of preneoplasias, see ⁹) harbored driver *Kras*^{G12D} mutations (**Fig. 4g** and **Extended Data Fig. 10a, b**). Our pseudotime analysis also inferred that KACs were along the trajectory from early AT2 cells, with tumor cells, and prior to generation of AT1 cells (**Extended Data Fig. 10c**), in ways very similar to our findings with *Gprc5a*^{-/-} mice (**Fig. 4a**).

Fig. 4g. Fraction of *KRAS*^{G12D} cells across alveolar and early tumour subsets. Absolute numbers of *KRAS*^{G12D} cells are indicated on top of each bar.

Extended Data Fig. 10a-c. **a**, UMAPs of GFP⁺ cells from tumour-bearing reporter mice at 7 months post-NNK or saline colored by presence of *Kras*^{G12D} mutation or expression of KAC, AT1, and AT2 signatures. **b**, UMAPs (top) showing distribution of alveolar and tumour cell subsets as well as cells with *Kras*^{G12D} mutation by treatment (saline or NNK). **c**, Pseudotime (left) and differentiation trajectories (middle) of GFP⁺ cells from tumour-bearing reporter mice at 7 months post-NNK or saline. Right panel shows pseudotemporal trajectory color-coded by subset composition (right).

Our IF analysis of tumor tissues also showed the presence of GFP⁺/Lamp3⁺/Krt8^{low} AT2 cells in normal (non-tumoral) lung regions from NNK-exposed *Gprc5a*^{-/-}; *Sftpc*^{CreER/+}; *Rosa*^{Sun1GFP/+} mice. Normal lung regions also included KACs that were GFP⁺/Lamp3^{+/}/Krt8⁺ or GFP⁺/Krt8⁺ that were also KAC marker positive⁺ (Cldn4/Cavin3) (**Fig. 4f** and **Extended Fig. 9c, d**). Importantly, tumor cells from the same mice were almost entirely GFP⁺ as well as Krt8⁺ and KAC marker⁺ (Cldn4/Cavin3) (**Fig. 4f** and **Extended Fig. 9c, d**).

Fourth, we performed spatial transcriptomics (ST) analysis using the Visium platform from 10X Genomics of a human KM-LUAD (P14; **Fig. 2j**) and mouse tumour-bearing lungs from multiple mice (**Fig. 3f** and **Extended Data Fig. 6h-j**) and not only confirmed the *in situ* presence of KACs (histologically annotated as reactive pneumocytes) in normal-appearing regions surrounding tumours, but also found that both human and mouse KACs, like their nearby tumours, displayed enrichment for KRAS and KAC signatures (**Fig. 2j** and **Fig. 3f** and **Extended Data Fig. 6h-j**).

Fig. 2j. ST analysis of P14 LUAD showing histological annotation of H&E-stained Visium slide (10XGenomics) (left) and spatial heatmaps showing CNV score, as well as scaled expression of *KRT8*, KAC markers, and KRAS signature.

Fig. 3f. ST analysis of lung tissue at 7 mo post-exposure to NNK and showing histological annotation of H&E-stained Visium slide (10X Genomics) (left) and spatial heatmaps showing scaled expression of *KRT8* as well as KAC and KRAS signatures.

Extended Data Fig. 6h. ST analysis of the same tumour-bearing mouse lung in **Fig. 3f** with UMAP of cell clusters identified by Seurat (left) and mapped spatially (middle). Spatial maps with scaled expression of *Krt8* and *Plaur* are shown on the right.

Fifth, we derived a murine KAC signature which we confirmed to be increased in mouse KACs and malignant cells relative to normal alveolar subsets (**Extended Data Fig. 7e**). We then found that this mouse KAC signature was significantly increased in human KACs ($P = 0.04$) and malignant cells ($P < 2.2e-16$) from *KRAS*-mutant versus *EGFR*-mutant patients of our human scRNA-seq cohort (**Extended Data Fig. 8i**).

Extended Data Fig. 8i. Mouse KAC signature in human KACs (left) and malignant cells (right) from KM-LUADs relative to EM-LUADs in our human scRNA-seq dataset. Box, median \pm interquartile range; whiskers, 1.5x interquartile range; center line: median. P -values were calculated using Wilcoxon Rank-Sum test.

Sixth, by virtue of the 5' scRNA-seq assay used in our study, we were able to call genomic alterations codon 12 of *KRAS*. Thus, we extracted and pooled corresponding reads for human KACs, alveolar subsets, and malignant cells by matching their unique cell barcodes and screened *KRAS*^{G12D} mutation in these cell subsets. First we confirmed that *KRAS*^{G12D} was indeed present in malignant cells from KM-LUADs and with a variant allele frequency (VAF) of 78% (**Fig. 2p** and **Extended Data Fig. 5c,d**). Importantly, KACs harbored *KRAS*^{G12D} mutations which were absent in AT2, AT1, and other AICs (**Fig. 2p** and **Extended Data Fig. 5c,d**). Additionally, *KRAS* mutations in KACs were only found in KM-LUAD patients (**Fig. 2p** and **Extended Data Fig. 5c,d**). The VAF of *KRAS*^{G12D} mutation in KACs

from KM-LUADs (10%) was higher when compared to that from all LUADs (5%) or all samples (3%) (Fig. 2p and Extended Data Fig. 5c, d). Altogether, these findings support the notion that not all normal alveolar intermediates (AICs) are the same, and that KACs may constitute a later intermediate state that is perhaps more committed to transformation towards a malignant phenotype. Our new findings also further show increased congruence between human and murine KACs with both displaying driver *KRAS* mutations.

Fig. 2p. Fraction of *KRAS*^{G12D} cells across malignant cells or KACs and in KM-LUADs, all LUADs, or NL from KM-LUADs.

Extended Data Fig. 5c. *KRAS* variant allele frequencies across diverse alveolar and malignant subsets from our scRNA-seq LUAD and/or normal samples and from patients with KM- and/or *KRAS*-WT LUADs.

Extended Data Fig. 5d. Fractions of *KRAS*^{G12D} mutant cells across alveolar and malignant cell subsets from the human LUAD and/or normal samples and from patients with KM- and/or *KRAS*-WT LUADs.

Seventh, when we queried gene expression changes between cancer stem cell (CSC)/stem cell-like progenitor spheres derived from MDA-F471 cells (a cell line we had developed and cultured from a KM-LUAD from an NNK-exposed *Gprc5a*^{-/-} mouse) and parental MDA-F471 cells¹³, we found that the spheres were markedly enriched with the mouse KAC signature (**Extended Data Fig. 8a**). These data suggest that markers of KACs, and thus perhaps KACs themselves, are associated with formation of tumors.

Extended Data Fig. 8a. Murine KAC signature score (left) and heatmap showing expression of select KAC marker genes (right) in bulk transcriptomes of MDA-F471-derived 3D spheres versus parental MDA-F471 cells grown in 2D. *P*-value was calculated using Wilcoxon Rank-Sum test. Box, median ± interquartile range; whiskers, 1.5× interquartile range; center line: median.

Altogether, these analyses show that while KRT8 expression in alveolar intermediate cells, i.e., KACs, may be related to various forms of injury in the lungs, the emergence of KACs in

response to tobacco carcinogen exposure and their subsequent preferential progression to go on and acquire driver *Kras* mutations supports their implication in or at least association with lung carcinogenesis.

Accordingly, we incorporated these findings in the revised manuscript as such:

Results section (lines 267 - 277): “Spatial transcriptomics (ST) analysis of P14 tumour tissue using the Visium platform (10X Genomics) demonstrated increased expression of *KRT8* in tumour regions with high CNV scores, as well as in tumour-adjacent normal-appearing regions that were histopathologically evaluated to comprise highly reactive pneumocytes and that exhibited low/moderate CNV scores (Fig. 2j)..... Similarly, we found that a signature based on the 5 KAC markers (see Fig. 2e) was high in tumour and in adjacent-to-tumour regions with reactive pneumocytes, with a similar and spatially overlapping pattern for *KRAS* signature.”

Results section (lines 379 - 386): “We then studied by ST lung tissues from mice at 7 months post-exposure to NNK. We found that areas histologically-annotated as tumour regions had significantly increased expression of *Krt8*, *Plaur*, as well as KAC and *KRAS* signatures which were also overlapping, further confirming KACs enrichment in mouse tumours induced by tobacco carcinogen exposure (Fig. 3f and Extended Data Fig. 6h). In line with our data from human LUAD patients (Fig. 2j), *Krt8* high KACs with elevated expression of KAC and *KRAS* signatures were enriched in “reactive”, non-neoplastic regions surrounding tumours of NNK-exposed mice”. We also provided a new detailed subsection in the **Methods section** that describes the ST workflow (lines 1081 – 1128).

Results section (lines 310 - 324): “Our findings on high expression of *KRAS* signature in KACs prompted us to evaluate potential copy number changes and inferred *KRAS*^{G12D} driver mutations in KACs and in comparison to other cell subsets. We found that KACs exhibited moderately elevated CNV burdens relative to AT2, AT1, and other AICs, albeit their CNV scores were considerably lower compared to malignant cells (Extended Data Fig. 5a, b). *KRAS*^{G12D} was indeed present in malignant cells with a variant allele frequency (VAF) of 78% in KM-LUADs (Fig. 2p and Extended Data Fig. 5c and Supplementary Table 9). Importantly, KACs, but not AT2, AT1, or other AICs, harbored *KRAS*^{G12D} mutations (Fig. 2p and Extended Data Fig. 5c, d). *KRAS*-mutant KACs were exclusively found in tissues (primarily tumours) from KM-LUADs and, thus, *KRAS*^{G12D} VAF (10%) was higher in KACs from KM-LUADs compared to when examined using all LUADs (5%) or all samples (3%) (Fig. 2p and Extended Data Fig. 5c, d). Intriguingly, *KRAS*^{G12D} mutations were detected in KACs of NL tissues from KM-LUAD patients (VAF 2%), and other *KRAS* variants (*KRAS*^{G12C}) were detected in NL of one KM-LUAD, signifying a potential field cancerization effect that is associated with mutant *KRAS* (Fig. 2p and Extended Data Fig. 5c, d).”

Results section (lines 419 - 422): “Of note, we also found that the KAC signature was elevated in spheres enriched for cancer stem cell (CSC)-progenitor like cells, and that we had derived from *Gprc5a*^{-/-} LUAD cells (MDA-F471 cell line; previously derived from a LUAD in an NNK-exposed *Gprc5a*^{-/-} mouse;¹²), relative to parental 2D cells (Extended Data Fig. 8a).”

Results section (lines 422 - 436): “With the capacity to look at temporal dynamics of alveolar transitions and the emergence of KACs in our carcinogenesis mouse model, we took a closer look at KACs by timepoint and *Kras*^{G12D} mutation status and found that those at EOE were somewhat less differentiated relative to the more differentiated KACs at 7 months post-exposure (**Fig. 4a, bottom right**). Interestingly, the fraction of KACs with *Kras*^{G12D} mutation at EOE was low (~0.02), whereas this fraction was significantly increased at 7 months post-NNK (~0.19) (**Extended Data Fig. 8b**). By further investigating KACs from the late timepoint, we found that those with *Kras*^{G12D} were significantly less differentiated ($P = 9.2 \times 10^{-6}$; **Extended Data Fig. 8c**) and showed higher expression of KAC signature genes such as *Cldn4*, *Krt8*, *Cavin3*, and *Cdkn2a* relative to KACs without the mutation (**Extended Data Fig. 8d**). Intrigued by these findings, we separately compared our *Kras*^{wt} and *Kras*-mutant KACs to previously reported *Krt8*+ intermediate cells (Strunz et. al²). Despite the overall similarities we had previously noted between KACs at large and these cells (31.6%; **Extended Data Fig. 7a**), *Kras*^{wt} KACs were more similar to *Krt8*+ intermediate cells than *Kras*-mutant KACs (20.5 % overlap versus 9.9%) (**Extended Data Fig. 8e, f**).”

Results section (lines 437 - 446): “Given the presence of *Kras*^{G12D} mutations in KACs, we next sought to probe for evidence of *Kras*-mutant KACs or cells with KAC expression features in *Kras*^{G12D}-driven mouse models (e.g., *Kras*^{LSL-G12D}). We performed integrated scRNA-seq analysis of cells from our mouse cohort with cells in mice driven by *Kras*^{G12D} from the two separate studies by Marjanovic et al⁷ and Dost et al⁸. Following clustering, we identified a shared cluster (C5) of cells from all three studies with distinctly high expression of KAC markers and the KAC signature itself (**Extended Data Fig. 8g-i**). Notably, the overwhelming fraction of C5 cells were from our study (**Extended Data Fig. 8j, left**). Despite representing a smaller proportion of C5, cells from *Kras*^{G12D}-driven mice still expressed higher KAC signature compared with normal AT2 cells from all studies (**Extended Data Fig. 8j, right**).”

Results section (446 - 450): “To further probe the association of KAC gene features with mutant *KRAS*, we analyzed our KAC signature in the dataset of human AT2 cells with and without inducible *KRAS* from the Dost et al study. We found that the KAC signature was markedly and significantly increased in AT2 cells with induced expression of *KRAS*^{G12D} relative to those with wild type *KRAS*⁸ ($P < 2.2 \times 10^{-16}$, **Extended Data Fig. 8k**).”

Results section (lines 456 - 477): We further investigated the biology of KACs using *Gprc5a*^{-/-} mice with reporter labeled-AT2 cells (*Gprc5a*^{-/-}; *Sftpc*^{CreER/+}; *Rosa*^{Sun1GFP/+}) at EOE to NNK or saline (**Fig. 4b**)..... We also monitored reporter mice up to 3 months post-NNK or -saline and analyzed GFP⁺ and GFP⁻ cell fractions by scRNA-seq (2 mice/group). We found that GFP⁺ cells (n = 3,089) almost exclusively comprised AT2, early tumour/AT2, KAC/KAC-like cells, and few AT1 cells, all of which were nearly absent in the GFP⁻ fraction (**Fig. 4c, d**; **Supplementary Fig. 7**). There were markedly increased fractions of GFP⁺ AT1, KACs, and, expectedly, tumour cells from NNK- versus saline-treated mice (**Fig. 4e**). IF analysis showed that GFP expression was almost exclusive to alveolar regions (**Extended Data Fig. 9b**) and tumours (**Extended Data Fig. 9c**) and that tumours were almost entirely GFP⁺ as well as

Krt8⁺ and KAC marker⁺ (Cldn4, Cavin3) (Fig. 4f and Extended Data Fig. 9c). Normal lung regions included KACs that were GFP⁺/Krt8⁺ and also KAC marker positive⁺ (Cldn4, Cavin3) (Fig. 4f and Extended Data Fig. 9c). GFP⁺/LAMP3⁺/KRT8^{-low} AT2 cells were evident including in normal (non-tumoural) lung regions from NNK-exposed reporter mice (Extended Data Fig. 9d). Notably, GFP⁺ KACs from this time point, that coincides with formation of preneoplasias⁹, harbored driver *Kras*^{G12D} mutations at similar fractions when compared with early tumour/AT2 cells (Fig. 4g and Extended Data Fig. 10a, b). Furthermore, KACs were not only positioned along the trajectory from early AT2 cells, with tumour cells, and prior to generation of AT1 cells, but they were also less differentiated relative to AT2 or early tumour cells (Extended Data Fig. 10c, d), in agreement with our previous findings in *Gprc5a*^{-/-} mice (Fig. 4a).”

The reviewer also astutely comments on the unclear role of increased senescence and *Tp53* signaling observed in KACs. To address this, we derived a *Tp53* signature and compared its score in separate epithelial subsets from our mouse dataset, namely, AT2 cells, tumor cells, as well as KACs from NNK-exposed mice. *Tp53* score was increased in KACs at EOE to NNK, and even more so at 7 months post-exposure to NNK. Intriguingly, tumor cells showed much reduced enrichment of *Tp53* score relative to KACs, with levels resembling normal AT2 cells (Extended Data Fig. 7b). We noted a similar pattern in expression of *Tp53* pathway-related genes and senescence markers including *Btg2*, *Ccng1*, *Cdkn2b*, *Bax*, *Cdkn1a*, as well as *Tp53* itself. These results point towards a context-dependent activation of senescence markers whereby it is high in transitional and injury-related KACs but is later diminished or eliminated in transformed tumor cells. This observation and our supposition are reminiscent of previous reports showing increase senescence in preneoplasias compared to uninvolved normal tissues but then diminished senescence in malignancy¹⁴.

Extended Data Fig. 7b. Enrichment of Tp53 signature derived from murine KACs, and expression of *Btg2*, *Ccng1*, *Cdkn2b*, *Bax*, *Cdkn1a*, as well as *Tp53* itself, across AT2 cells, malignant cells, and KACs at EOE or at 7 mo post-NNK or saline.

These new findings are described in the revised manuscript as such:

Results section (lines 392 - 396): “We derived a tp53 signature and found that its expression was significantly elevated in KACs at EOE to NNK, and more so at 7 months post-exposure to NNK, compared to both AT2 as well as tumour cells (Extended Data Fig.

7b, left). We noted a similar pattern in expression of tp53 pathway-related genes and senescence markers including *Btg2*, *Ccng1*, *Cdkn2b*, *Bax*, *Cdkn1a*, as well as *Tp53* itself (**Extended Data Fig. 7b, right**).”

Comment 2: It seems that the so called KAC (KRT8+ alveolar cells) may be distinct from injury-associated KRT8+ transitional cells, in that KACs already have KRAS mutation and are on their way to be transformed. In this scenario, it will be important to address whether KAC is really transitional between AT2 to AT1, or rather, they are just de-differentiated, as would be expected from the transforming action of activated KRAS. In addition, KRT8+ is widely expressed in normal airway cells. A recent study demonstrated that AT2 to airway transition is also an alveolar cell behavior observed following injury (Kathiriya et al., Cell Stem Cell 2021).

Response 2: We thank the reviewer for bringing forth this important argument.

To ascertain whether KACs in our model are distinct from or related to injury-associated *Krt8+* transitional cells, we interrogated a dataset from Strunz and colleagues² and whereby the authors identified a *Krt8+* alveolar differentiation intermediate cell state during alveolar regeneration following bleomycin injury and that persist in human lung fibrosis. We compared and contrasted differentially expressed gene sets in our cohort and in the cohort from Strunz et al. We found a significant fraction of genes that overlapped between the two datasets (14.9%) (**Extended Data Fig. 7a**). Interestingly, KACs from our study comprised gene sets that were previously shown to be enriched in *Krt8+* transitional cells in the report by Strunz and colleagues. These include NF κ B-regulated gene signaling, hypoxia, activated p53 including transcriptional features of cellular senescence (**Extended Data Fig. 7c**).

Extended Data Fig. 7a. Pie chart showing percent of unique and overlapping DEG sets between murine KACs from this study and *Krt8+* transitional cells identified by Strunz and colleagues.

While this shows that the mouse KACs in our study are, at least in part, reminiscent of the *Krt8+* alveolar transitional state in recent reports interrogating mechanisms in response to lung injury (such as that by Strunz et al), we surmise that these KACs are also distinct. In response to several comments by this reviewer and others, we had further studied heterogeneity among the mouse KACs. We noted that KACs have much higher frequencies of *Kras* mutations at the time of tumor development versus earlier (at EOE to NNK) (**Extended Data Fig. 8b**). These *Kras*-mutant KACs had lower differentiation (i.e., higher cytoTRACE score; **Extended Data Fig. 8c**), and increased levels of KAC markers (**Extended Data Fig. 8d**) compared with *Kras* wild type KACs. Notably, expression features of these *Kras*-mutant KACs displayed considerably lower overlap (9.9%) with gene profiles

of the reported Krt8+ alveolar transitional cells compared with Kras wild type KACs (20.5%) (**Extended Data Fig. 8e,f**) – suggesting perhaps that indeed *Kras*-mutant KACs are more closely associated with KM-LUAD development (**Extended Data Fig. 8b-f were shown above in response to comment #1 by the same reviewer**).

In our human dataset, we had found that KACs comprise a fraction of AICs. It is also noteworthy that a human KAC signature we had originally derived showed strong positive correlation with a signature of activated/mutated *KRAS* and pronounced inverse correlation with an alveolar signature in KACs (**Fig. 2k**) as well as was enriched in human *KRAS*-mutant LUAD tissues relative to tumors with wild type *KRAS* (**Fig. 2n**). To extend these analyses, we now derived a signature of other AICs and found that this signature was markedly higher in AICs that are non-KACs (i.e., in other AICs), compared to KACs, AT1, and AT2 cells (now **Extended Data Fig. 4d**).

Next, we further queried the AICs signature. In sharp contrast to what we originally observed with the KAC signature (now **Fig. 2k**), the other AICs score showed no or minimal correlation with *KRAS* or alveolar signature (now **Extended Data Fig. 4e,f**).

Fig. 2k. Correlation analysis between KAC signature scores and *KRAS* signature scores or MP31 alveolar lineage signature scores. *P*-values were calculated with Spearman correlation test. *R* denotes the Spearman correlation coefficients.

Extended Data Fig. 4e,f. Correlation analysis between other AICs signature and KRAS signature scores (**e**) or alveolar signature (**f**). P -values were calculated with Spearman correlation test. R denotes the Spearman correlation coefficients.

Also, while we had found in our original manuscript that the KAC signature was increased in LUADs or in lung preneoplasias when compared to normal lung tissues (now **Fig. 2m**), the opposite was true with the other AICs signature; it was significantly reduced in LUADs relative to normal lung tissues and showed no pattern in lung preneoplasias (now **Extended Data Fig. 4g, i**).

Fig. 2m. Enrichment of KAC signature across samples of TCGA LUAD (left) and premalignancy (right) cohorts. AAH: atypical adenomatous hyperplasia. P -values were calculated using Wilcoxon signed-rank test.

Extended Data Fig. 4g,i. Expression of other AICs signature in TCGA LUAD samples and matched NL tissues (**g**), as well as in a lung preneoplasia cohort (**i**). P -values were calculated using Mann-Whitney U test. Box, median \pm interquartile range; whiskers, 1.5 \times interquartile range; center line: median. AAH: atypical adenomatous hyperplasia. n.s.: non-significant ($P > 0.05$).

Another important difference between the KAC and other AICs signature was their association with prognosis. While patients with higher KAC signature displayed significantly poorer survival (now **Fig. 2o** and **Extended Data Fig. 4j,k**), there were no differences in survival among patients based on the other AICs signature (now **Extended Data Fig. 4l,m**). These observations were consistent among independent datasets: TCGA (**Extended Data Fig. 4j,l**) and MD Anderson's PROSPECT cohort (**Extended Data Fig. 4k, m**). Importantly, we performed multivariate Cox proportional hazard regression analysis by including covariates stage, age, KAC signature as well as the other AICs signature in the model. While the KAC signature was associated with poorer survival even after accounting for stage, the other AICs signature was not significantly associated with survival outcome (**Fig. 2o**).

Extended Data Fig. 4j-m. Kaplan-Meier plots showing differences in overall survival probability in TCGA (**j**) and PROSPECT (**k**) samples with high versus low KAC signature scores, or with high versus low scores for other AICs signature (**l**: TCGA; **m**: PROSPECT). OS: overall survival. Sig. low: LUAD samples with signature scores lower than the group median value. Sig. hi: LUAD samples with signature scores higher than the group median value. Mo: months. P -values were calculated with the log-rank test.

Fig. 2o. Multivariate Cox proportional hazard regression analysis including stage, age, KAC signature and “other AICs” signature. Q-values were calculated using a Cox proportional hazards (PH) regression model and adjusted with Benjamini–Hochberg method.

In response to reviewer #2, we had further analyzed mutational features of KACs in the human tissues and found that – like the mouse KACs – they exhibited increased frequencies of *KRAS* mutations which were absent in other alveolar sets including other AICs as well as in KACs from *KRAS*-wildtype LUADs (**Fig. 2p** and **Extended Data Fig. 5c, d**) (**these were shown above**).

Importantly, human KACs exhibited significantly higher cytoTRACE scores and, thus, less differentiated states, when compared to other AICs (**Fig. 2f**). This was more striking when analyzing KACs and AICs from patients with KM-LUADs (**Extended Data Fig. 5e,f**). Interestingly, we found that KACs were less differentiated when compared to other AICs when analyzing cells from *KRAS*-mutant tumors as well as cells from normal tissues of the same patients (**Extended Data Fig. 5f**). Intriguingly, KACs from KM-LUAD tissues were more de-differentiated relative to KACs from patients with *KRAS*-WT LUADs (**Extended Data Fig. 5f**). These findings suggest that human KACs are transcriptionally closer to KM-LUAD malignant cells than other AICs. Further, KACs relative to other AICs were more closely associated with malignant cells in pseudotime (**Extended Data Fig. 3f**). Similarly, and as mentioned above, we found in our animal studies that mouse *Kras*-mutant KACs had lower differentiation (i.e., higher cytoTRACE score; **Extended Data Fig. 8c**), and increased levels of KAC markers (**Extended Data Fig. 8d**) compared with *Kras* wild type KACs.

Extended Data Fig. 5e,f. CytoTRACE scores in KACs versus other AICs from all cells of KM-LUADs (**e**, left) or *KRAS*-WT LUAD patients (**e**, right), cells from normal lung tissues of KM-LUAD patients (**f**, left), and cells from tumour tissues of KM-LUAD (**f**, middle) and *KRAS*-WT LUAD (**f**, right) patients. *P*-values were calculated using Wilcoxon Rank-Sum tests.

Extended Data Fig. 3f. Pseudotime trajectory of human malignant and alveolar subsets color-coded by cell lineage and presence of *KRAS*^{G12D} mutation (top). Pseudotime score in KACs versus other AICs (bottom). *P*-value was calculated by Wilcoxon Rank-Sum test.

It is important to mention that our findings do not neglect the possibility that KACs following NNK exposure, like in other injury settings, are involved in AT2 to AT1 transition. When we analyzed GFP⁺ fractions from our AT2 lineage-labeled (GFP) mice at 3 months post-NNK versus saline, we found markedly increased fractions of GFP⁺ AT1 cells, along with KACs and tumor cells, in NNK-exposed lungs (**Fig. 4b-e**). Also, our IF analysis showed some GFP⁺/Pdpn⁺/Lamp3⁻ cells (white arrow heads) which were co-existent with yet distinct from GFP⁺/Krt8⁺ KACs in NNK-exposed lungs (**Extended Data Fig. 9d**). Our findings suggest that KACs in our tobacco carcinogenesis model perhaps are implicated in both AT2-AT1 transition and tumorigenesis. It is thus plausible to surmise that KACs originally emerge in response to injury cues and with the intention to repair the damaged lung, but the uniqueness of the tumor-promoting insult here (tobacco carcinogen) compared to other forms of lung damage suggests a link between KACs and tumor development.

Fig. 4b-e. **b**, Schematic overview showing treatment of *Gprc5a*^{-/-} mice with reporter labeled-AT2 cells (*Gprc5a*^{-/-}; *Sftpc*^{CreER/+}; *Rosa*^{Sun1GFP/+}) and analysis of labeled cells at EOE and at 3 months post-NNK or saline. **c**, UMAP distribution of GFP⁺ cells at 3 months following NNK exposure or saline and colored by alveolar or tumour subsets. **d**, Expression (proportions and average expression levels) of selected marker genes for mouse normal alveolar cell lineages and tumour cells defined in panel **c**. **e**, Fractions of GFP⁺ AT1, AT2, KAC/KAC-like, and early tumour/AT2 from lungs of 2 NNK and 2 saline-exposed mice analyzed at 3 months-post exposure.

Extended Data Fig. 9d. IF analysis of GFP, Krt8, Lamp3, and Pdpn in lung parenchyma and tumours of AT2 reporter mouse lungs at 3 months post-exposure to saline or NNK. White arrows; GFP⁺ Krt8^{low} AT2 cells. Yellow arrows; GFP⁺ Krt8⁺ KACs. White arrow heads; GFP⁺ Pdpn⁺ AT1 cells. Scale bars: 30 µm for the first two rows and 10 µm for the higher magnified tumour.

We also thank the reviewer for bringing forth the recent findings by Kathiriya et al. Despite the overall parallels with our study, with respect to reporting modes of lineage plasticity in response to lung injury, we note some major differences between our study and that by Kathiriya et al. We respectfully surmise that the reviewer is referring to the Kathiriya Nature Cell Biology (2021) manuscript¹⁵, since the study led by the same author in Cell Stem Cell¹⁶ focused on a distinct subpopulation of club-like cells with the capacity to differentiate into alveolar subpopulations in response to lung injury. While the Kathiriya 2021 Nature Cell Biology paper did note both AT2-lineage labeled Krt5⁺ basal cells and Krt8⁺ intermediate cells in fibrotic lung injury models that emulate fibrosis, the Krt5⁺ basal cells and the Krt8⁺ intermediates were overall mutually exclusive (Figure 2 in that report). Naturally, these are distinct from our findings reporting on alveolar plasticity in response to specific lung carcinogenic insults and in the context of KM-LUAD formation. Nonetheless, to query the reviewer's comment, we analyzed basal cell markers in GFP⁺ and GFP⁻ fractions in AT2-lineage labeled *Gprc5a*^{-/-}; *Sftpc*^{CreER/+}; *Rosa*^{Sun1GFP/+} mice at 3 months post-NNK or saline by scRNA-seq and tissues by IF. In support of our previous findings, we found that GFP⁺ fractions analyzed by scRNA-seq from our AT2 lineage labeled mice, and which we later found to be enriched with KACs, were devoid of basal/airway cell markers (**Supplementary Figure 7**). Also, our IF analysis from lungs of *Gprc5a*^{-/-}; *Sftpc*^{CreER/+}; *Rosa*^{Sun1GFP/+} mice concordantly showed absence of GFP (i.e., AT2 lineage cells) in conducting airways and bronchioles (**Extended Data Fig. 9b**).

Supplementary Fig. 7. Proportions and average expression levels of selected marker genes for alveolar, tumour, and basal cell clusters in GFP⁺ and GFP⁻ samples from saline- and NNK-treated AT2 lineage-labeled mice.

Extended Data Fig. 9b. IF analysis of lung parenchyma and airway spaces in lung tissues of AT2 reporter mouse lungs at 3 months post-exposure to saline or NNK. White arrows; GFP⁺ AT2 cells. Yellow arrows; GFP⁺ Krt8⁺ KACs. Scale bars: 30 μ m.

Thus, while *Krt8* expression in alveolar subsets seems to be strongly associated with lineage transitions and stem cell features and across multiple models of lung injury, our new analyses and findings support the context-specific biology of KACs identified in our lung carcinogenesis models and their relevance to development of *KRAS*-mutant LUAD. Also, our results lend support to the distinct features of KACs identified in our human cohort as well as mouse experiments analyzed by scRNA-seq and validated by spatial transcriptomics as well as IF staining. We describe and discuss these findings in the revised manuscript as such:

Results section (lines 251 – 253): “KACs and more developmentally late ($P = 1.2 \times 10^{-11}$; Extended Data Fig. 3f).”

Results section (lines 280 - 308): “To further emphasize the distinction between KACs and other AICs, we found that a signature pertinent to “other AICs” was evidently lower in KACs

relative to AICs that are non-KACs (**Extended Data Fig. 4d**). The expression levels of KAC signature significantly and positively correlated with that of KRAS signature ($r = 0.45$; $P < 2.2 \times 10^{-16}$) and inversely correlated with alveolar signature ($r = -0.77$; $P < 2.2 \times 10^{-16}$; **Fig. 2k**), unlike “other AICs” signature which showed no correlation with KRAS ($r = 0.045$; $P = 3.2 \times 10^{-5}$) or alveolar ($r = -0.11$; $P < 2.2 \times 10^{-16}$) signatures (**Extended Data Fig. 4e, f**). Of note, KAC signature was significantly higher in KACs and in malignant cells from KM-LUAD tumour tissues relative to those from EM-LUADs ($P < 2.2 \times 10^{-16}$; **Fig. 2l**). We then found that the KAC signature was significantly enriched in bulk transcriptomes of human TCGA LUADs compared to uninvolved NL tissues ($P = 1.9 \times 10^{-15}$; **Fig. 2m, left**), and in contrast to “other AICs” and alveolar signatures which were both significantly reduced in LUADs ($P = 8.6 \times 10^{-6}$ and $P = 1.9 \times 10^{-15}$, respectively; **Extended Data Fig. 4g, h**). By analysis of our independent cohort of lung preneoplasias, we found that KAC signature was significantly and progressively increased along the pathologic continuum of NL, atypical adenomatous hyperplasia (AAH; the earliest precursor of LUAD), and matching invasive LUAD (**Fig. 2m, right**), whereas there was no such pattern for “other AICs” signature (**Extended Data Fig. 4i**). Furthermore, using TCGA data, we also found that KAC signature was significantly higher in KM-LUADs relative to *KRAS*-WT LUADs ($P = 0.002$; **Fig. 2n**). Also, patients with KAC signature-high LUADs ($n = 243$) showed significantly reduced survival ($P = 0.005$) relative to those with low expression of the signature (TCGA; $n = 246$; **Extended Data Fig. 4j**). In a separate cohort, KAC signature score was also associated with poor OS (PROSPECT; $n = 150$; $P = 0.04$; **Extended Data Fig. 4k**). In either of these cohorts, there were no differences in survival among patients based on “other AICs” signature (TCGA, $P = 0.28$; PROSPECT, $P = 0.35$; **Extended Data Fig. 4l, m**). Importantly, we performed multivariate Cox proportional hazard regression analysis by including covariables such as stage, age, KAC and “other AICs” signatures in the model. We found that KAC signature was associated with shortened OS even after accounting for stage (FDR adjusted q -value = 0.034), whereas “other AICs” signature showed no significant association with survival outcome (FDR adjusted q -value = 0.5; **Fig. 2o**.)”

Results section (lines 310 - 324): “Our findings on high expression of KRAS signature in KACs prompted us to evaluate potential copy number changes and inferred *KRAS*^{G12D} driver mutations in KACs and in comparison to other cell subsets. We found that KACs exhibited moderately elevated CNV burdens relative to AT2, AT1, and other AICs, albeit their CNV scores were considerably lower compared to malignant cells (**Extended Data Fig. 5a, b**). *KRAS*^{G12D} was indeed present in malignant cells with a variant allele frequency (VAF) of 78% in KM-LUADs (**Fig. 2p** and **Extended Data Fig. 5c** and **Supplementary Table 9**). Importantly, KACs, but not AT2, AT1, or other AICs, harbored *KRAS*^{G12D} mutations (**Fig. 2p** and **Extended Data Fig. 5c, d**). *KRAS*-mutant KACs were exclusively found in tissues (primarily tumours) from KM-LUADs and, thus, *KRAS*^{G12D} VAF (10%) was higher in KACs from KM-LUADs compared to when examined using all LUADs (5%) or all samples (3%) (**Fig. 2p** and **Extended Data Fig. 5c, d**). Intriguingly, *KRAS*^{G12D} mutations were detected in KACs of NL tissues from KM-LUAD patients (VAF 2%), and other *KRAS* variants (*KRAS*^{G12C}) were detected in NL of one KM-LUAD, signifying a potential field cancerization effect that is associated with mutant *KRAS* (**Fig. 2p** and **Extended Data Fig. 5c, d**).”

Results section (lines 326 – 332): “We were thus prompted to further explore differentiation states by *KRAS* status. We found that KACs were significantly less differentiated relative to other AICs when analyzing all cells of KM-LUAD patients ($P < 2.2 \times 10^{-16}$), as well as in cells from NL only of the same patients ($P < 2.2 \times 10^{-16}$), and in stark contrast to cells of *KRAS*-WT patients ($P < 2.2 \times 10^{-16}$; **Extended Data Fig. 5e, f**). Intriguingly, KACs from KM-LUAD tissues were less differentiated relative to KACs from *KRAS*-WT LUAD tissues ($P < 2.2 \times 10^{-16}$; **Extended Data Fig. 5f**), all together suggesting that human KACs are transcriptionally closer to KM-LUAD malignant cells than other AICs.”

Results section (lines 422 - 436): “By further investigating KACs from the late timepoint, we found that those with *Kras*^{G12D} were significantly less differentiated ($P = 9.2 \times 10^{-6}$; **Extended Data Fig. 8c**) and showed higher expression of KAC signature genes such as *Cldn4*, *Krt8*, *Cavin3*, and *Cdkn2a* relative to KACs without the mutation (**Extended Data Fig. 8d**).

Results section (lines 456 - 477): We further investigated the biology of KACs using *Gprc5a*^{-/-} mice with reporter labeled-AT2 cells (*Gprc5a*^{-/-}; *Sftpc*^{CreER/+}; *Rosa*^{Sun1GFP/+}) at EOE to NNK or saline (**Fig. 4b**).....We also monitored reporter mice up to 3 months post-NNK or -saline and analyzed GFP⁺ and GFP⁻ cell fractions by scRNA-seq (2 mice/group). We found that GFP⁺ cells (n = 3,089) almost exclusively comprised AT2, early tumour/AT2, KAC/KAC-like cells, and few AT1 cells, all of which were nearly absent in the GFP⁻ fraction (**Fig. 4c, d**; **Supplementary Fig. 7**). There were markedly increased fractions of GFP⁺ AT1, KACs, and, expectedly, tumour cells from NNK- versus saline-treated mice (**Fig. 4e**). IF analysis showed that GFP expression was almost exclusive to alveolar regions (**Extended Data Fig. 9b**) and tumours (**Extended Data Fig. 9c**) and that tumours were almost entirely GFP⁺ as well as Krt8⁺ and KAC marker⁺ (*Cldn4*, *Cavin3*) (**Fig. 4f and Extended Data Fig. 9c**). Normal lung regions included KACs that were GFP⁺/Krt8⁺ and also KAC marker positive⁺ (*Cldn4*, *Cavin3*) (**Fig. 4f and Extended Data Fig. 9c**). GFP⁺/LAMP3⁺/KRT8^{-/low} AT2 cells were evident including in normal (non-tumoural) lung regions from NNK-exposed reporter mice (**Extended Data Fig. 9d**).

Comment 3: In the abstract, “local niche” was mentioned. However, only EpCam+ epithelial cells were sorted and analyzed. For a true niche, cells in other lineages need to be analyzed.

Response 3: We apologize for the inapt use of the term “local niche” when in fact only Epcam+ fractions were analyzed in this context. We thank the reviewer for bringing this up and accordingly we have revised the phrase in question by omitting it from the abstract and replacing it with “Local epithelia surrounding LUADs were enriched with...” (**line 32**).

Comment 4: Figure 1b, the bronchioalveolar cell designation needs to be revisited, and will be better defined in clustering using both alveolar and airway cells.

Response 4: We thank the reviewer for noting that. We have now revised the designation by renaming bronchioalveolar cells as “SCGB1A1/SFTPC dual positive cells-SDP”, as

shown in **Fig. 1b** and **Extended Data Fig. 1a** of the revised manuscript. This cluster is now better defined in the UMAP showing all epithelial cells in the revised manuscript (**Supplementary Fig. 2**), as well as in the UMAP shown in revised **Extended Data Fig. 1a**.

Supplementary Fig. 2. UMAP plots of 229,038 human normal epithelial cells colored by the major cell lineages. SDP cells were separately colored to show their position on the UMAP.

Comment 5: Unclear why the specific Gprc5a mutant was selected as a mouse model to validate the KAC involvement in tumor formation.

Response 5: We thank the reviewer for pointing out that the choice of animal model was not clearly described or justified, and we agree that a better clarification is needed. Based on our findings from the human scRNA-seq and additional KM-LUAD cohorts, we identified a strong link between KACs, and LUADs harboring *Kras*^{G12D} mutations. Given previous association of KACs with injury (as we referenced in our original manuscript, see response to comment #1 by the same reviewer), we wanted to employ an animal model that develops LUADs following exposure to a form of injury that is closely pertinent to lung tumourigenesis and mutant *Kras*, i.e., tobacco carcinogen. In our previous work we have shown that mice with knockout of the lung lineage gene *Gprc5a* not only exhibit enhanced and accelerated development of LUADs post-tobacco carcinogen but that these tumors harbored somatic *Kras* mutations (which were absent in baseline normal lung and germline)⁹. Additionally, we had found that lung expression signatures in NNK-exposed *Gprc5a*^{-/-} mice are markedly enriched in those from human smokers especially those with LUAD¹⁷. Therefore, we deemed the *Gprc5a*^{-/-} mouse as a suitable human-relevant model to temporally interrogate KAC biology early on before tumor development, and up to acquisition of somatic driver mutations and development of KM-LUADs. Nonetheless, in the revised manuscript, we have now analyzed datasets from published studies interrogating other LUAD mouse models harboring genetically altered *Kras*^{G12D} mutations. Our analyses interrogating scRNAseq data from these studies revealed that while markers of KACs were evident in cells from *Kras*-driven models, they were far fewer (**Extended Data Fig. 8g-k, shown above**). This could be attributed to an early enrichment of KACs following lung injury by tobacco carcinogen, a window which may have not been possible to interrogate in these other mouse models, or perhaps the roles of specific forms of carcinogenic lung insults in KAC biology. We now briefly discuss this in the Results and Discussion sections of the revised manuscript as such:

"

Results section (lines 344 – 349): “.....namely mice with knockout of the airway lineage-specific G-protein coupled receptor (*Gprc5a*^{-/-})^{9,18} in the presence or absence of tobacco-carcinogen exposure. LUADs form in *Gprc5a*^{-/-} mice but not from wild type littermates, are accelerated by tobacco-carcinogen exposure, and acquire somatic *Kras*^{G12D} mutations – we surmised that these features are highly pertinent to KM-LUAD development^{9,17,19} and thus to exploring KACs in this setting.”

Discussion section (lines 561 – 576): “Temporal analysis of lungs post-tobacco along the spectrum of normal-appearing (tumour-free) to tumour-bearing lungs enabled us to identify evolving features of KACs, most notably, somatic acquisition of *Kras* mutations and reduced differentiation, very much reminiscent of their human counterparts.....Notably, that discriminating expression features and markers of the mouse KACs were not only enriched in profiles of alveolar intermediate cells from mouse models of acute lung injury² but they were also found in cell subsets from animals whose tumours are driven by lung-specific expression of *Kras*^{G12D}^{7,8}, albeit at lesser frequency compared to our tobacco-injury carcinogenesis model. Thus, it is plausible that KACs can arise due to an injury stimulus (here tobacco exposure) or following expression of mutant *Kras* – or to both, since in our tobacco model KACs exist in a continuum, i.e., low versus high frequency of *Kras* mutations. Intriguingly, our high-resolution imaging and ST analyses showed that KACs were evident in normal-appearing areas in the vicinity of lesions in both murine and patient samples, suggesting that the early appearance of these cells (e.g., following tobacco exposure) may represent development of a *field of injury*¹¹.”

Comment 6: *Fig 5a, it will be important to genotype the KACs used in the organoid assay. Have they already acquired mutation that transform them?*

Response 6: We thank the reviewer for bringing up this important point. To better understand the biology of KACs in the organoid system, we now include results from three separate analyses in the revised manuscript. These analyses are focused on high-level interrogation in *Gprc5a*^{-/-}; *Sftpc*^{CreER+}; *Rosa*^{Sun1GFP/+} mice which were labelled similarly to mice analyzed at end-of-exposure, except that those mice were now followed up to 3 months post-NNK or -saline. At 3 months following exposure to NNK or either saline (control), we separately analyzed, by scRNA-seq, GFP⁺ as well as GFP⁻ cell compartments (**Fig. 4b-e** and **Extended Data Fig. 10**). We noted markedly increased fractions of AT1, KACs, and, expectedly, tumor cells in GFP⁺ cells from NNK- versus saline-treated mice (**Fig. 4e**). Like the findings reported in our original manuscript using *Gprc5a*^{-/-} mice, we noted that GFP⁺ KACs from this time point (3 months post-NNK, coincides with formation of preneoplasias, see ⁹) harbored key driver *Kras*^{G12D} mutations (**Fig. 4g** and **Extended Data Fig. 10a, b**).

KRAS^{G12D} cells across alveolar and early tumour subsets. Absolute numbers of *Kras*^{G12D} cells are indicated next to each bar.

Extended Data Fig. 10 a,b. UMAPs of GFP⁺ cells from tumour-bearing reporter mice at 7 months post-NNK or saline colored by presence of *Kras*^{G12D} mutation or expression of KAC, AT1, and AT2 signatures (**a**) and UMAPs (**b**) showing distribution of alveolar and tumour cell subsets as well as cells with *Kras*^{G12D} mutation by treatment (saline or NNK).

Given these findings, we were intrigued to investigate whether KACs are sensitive to KRAS^{G12D} inhibition. To interrogate that, we derived 3D organoids in air-liquid interface (ALI) from the same pool of GFP⁺ cells of the same *Gprc5a*^{-/-}; *Sftpc*^{CreER/+}; *Rosa*^{Sun1GFP/+} at 3 months following exposure with saline or NNK, in the same manner the EOE organoids that were studied in the original manuscript. We found that GFP⁺ organoids derived from NNK-treated mice were significantly larger than those derived from saline-treated mice (**Extended Data Fig. 11a**) and they highly expressed *Krt8* and *Cldn4* (**Extended Data Fig. 11e**).

Extended Data Fig. 11 a,b. Size quantification of organoids derived from GFP⁺ lungs cells of saline- or NNK-treated mice at 3 months post-exposure (**a**). Box, median \pm interquartile range; whiskers, 1.5 \times interquartile range; center line: median. P -value was calculated using Wilcoxon Rank-Sum test. Relative viability (**b**) analysis of LKR13 and MDA-F471 cells at 6 hours following treatment with increasing concentrations of MRTX1133. n.s: not significant. Error-bars: standard deviations of means. P -values were calculated using paired t-tests.

To further characterize NNK-derived organoids, we sought to examine effects of targeted inhibition of KRAS on these organoids using the KRAS^{G12D} specific inhibitor MRTX1133²⁰. We first examined effects of MRTX1133 on growth of mouse *Kras*-mutant LUAD cancer cells *in vitro*. We found that MRTX1133 resulted in dose-dependent inhibition of MDA-F471 (derived from a LUAD in an NNK-exposed *Gprc5a*^{-/-} mouse) and LKR13 (derived from *Kras*^{LSL-G12D} mice;²¹) cells (**Extended Data Fig. 11b**). We also found that MRTX1133 doses that inhibited cell growth also suppressed phosphorylated levels of ERK1/2 and S6kinase in both cell lines (**Extended Data Fig. 11c**). We then studied the effects of 200 nM MRTX1133 treatment when compared to control DMSO on KAC marker positive organoids from NNK-exposed *Gprc5a*^{-/-}; *Sftpc*^{CreER/+}; *Rosa*^{Sun1GFP/+} mice. We found significantly reduced organoid size upon treatment with the inhibitor and relative to DMSO-treated counterparts, as well as reduced intensity of Krt8 and Cldn4 expression by IF (**Extended Data Fig. 11d,e**). These results allow us to confidently albeit indirectly deduce that *Kras*^{G12D} mutations are enriched in KAC⁺ organoids from NNK-treated *Gprc5a*^{-/-}; *Sftpc*^{CreER/+}; *Rosa*^{Sun1GFP/+} mice and suggest that *Kras* mutation may be transforming in these cells.

Extended Data Fig. 11 b-d. Relative viability (**b**) and western blot (**c**) analysis of LKR13 and MDA-F471 cells at 6 hours following treatment with increasing concentrations of MRTX1133. n.s.: not significant. Error-bars: standard deviations of means. *P*-values were calculated using paired t-tests. **d**, Size quantification of organoids derived from GFP⁺ lungs cells of NNK-mice and treated with 200 nM MRTX1133 or control DMSO in vitro.

e

Extended Data Fig. 11e. IF analysis showing two representative organoids derived from sorted GFP⁺ cells from saline-exposed mice (top two rows), from NNK-exposed mice and then treated with DMSO in culture (middle two rows), and from NNK-exposed mice and then treated with 200 nM MRTX1133 (bottom two rows). Box, median \pm interquartile range; whiskers, 1.5 \times interquartile range; center line: median. *P*-value was calculated using Wilcoxon Rank-Sum test. Scale bars = 50 μ m except for the first DMSO-treated organoid (third row, scale bar = 100 μ m).

Accordingly, we revised the manuscript as such:

Results section (lines 456 - 477): We further investigated the biology of KACs using *Gprc5a*^{-/-} mice with reporter labeled-AT2 cells (*Gprc5a*^{-/-}; *Sftpc*^{CreER/+}; *Rosa*^{Sun1GFP/+}) at EOE to NNK or saline (**Fig. 4b**).....We also monitored reporter mice up to 3 months post-NNK or -saline and analyzed GFP⁺ and GFP⁻ cell fractions by scRNA-seq (2 mice/group). We found that GFP⁺ cells (n = 3,089) almost exclusively comprised AT2, early tumour/AT2, KAC/KAC-like cells, and few AT1 cells, all of which were nearly absent in the GFP⁻ fraction (**Fig. 4c, d; Supplementary Fig. 7**). There were markedly increased fractions of GFP⁺ AT1, KACs, and, expectedly, tumour cells from NNK- versus saline-treated mice (**Fig. 4e**). IF analysis showed that GFP expression was almost exclusive to alveolar regions (**Extended Data Fig. 9b**) and tumours (**Extended Data Fig. 9c**) and that tumours were almost entirely GFP⁺ as well as Krt8⁺ and KAC marker⁺ (Cldn4, Cavin3) (**Fig. 4f and Extended Data Fig. 9c**). Normal lung

regions included KACs that were GFP⁺/Krt8⁺ and also KAC marker positive⁺ (Cldn4, Cavin3) (Fig. 4f and Extended Data Fig. 9c). GFP⁺/LAMP3⁺/KRT8^{-low} AT2 cells were evident including in normal (non-tumoural) lung regions from NNK-exposed reporter mice (Extended Data Fig. 9d). Notably, GFP⁺ KACs from this time point, that coincides with formation of preneoplasias⁹, harbored driver *Kras*^{G12D} mutations at similar fractions when compared with early tumour/AT2 cells (Fig. 4g and Extended Data Fig. 10a, b). Furthermore, KACs were not only positioned along the trajectory from early AT2 cells, with tumour cells, and prior to generation of AT1 cells, but they were also less differentiated relative to AT2 or early tumour cells (Extended Data Fig. 10c, d), in agreement with our previous findings in *Gprc5a*^{-/-} mice (Fig. 4a).”

Results section (lines 479 - 493): “We then analyzed organoids which we derived from the same sorted GFP⁺ cells. GFP⁺ organoids from NNK- versus saline-treated mice not only showed significantly and markedly enhanced growth but that they were almost exclusively comprised of GFP⁺, Krt8⁺, and Cldn4⁺ cells (Extended Data Fig. 11a,e). To further characterize NNK-derived organoids, we sought to examine effects of targeted inhibition of KRAS on these organoids using the KRAS^{G12D} specific inhibitor MRTX1133²². We first examined effects of MRTX1133 on growth of mouse *Kras*-mutant LUAD cancer cells *in vitro* and noted dose-dependent inhibition of growth of MDA-F471 as well as LKR13 (derived from *Kras*^{LSL-G12D} mice;²¹) cells (Extended Data Fig. 11b). This was accompanied by suppression of phosphorylated levels of ERK1/2 and S6kinase in both cell lines (Extended Data Fig. 11c). We then studied the effects of 200 nM MRTX1133 treatment on KAC marker positive organoids (from NNK-treated reporter mice) and found significantly reduced organoid size upon treatment with the inhibitor and relative to DMSO-treated counterparts, as well as reduced intensity of Krt8 and Cldn44 expression by IF (Extended Data Fig. 11d, e). Taken together, our *in vivo* analyses identify KACs as an intermediate cell state in early development of KM-LUAD and following tobacco carcinogen exposure.”

References cited in this response letter:

- 1 Heiser, C. N. & Lau, K. S. A Quantitative Framework for Evaluating Single-Cell Data Structure Preservation by Dimensionality Reduction Techniques. *Cell Rep* **31**, 107576, doi:10.1016/j.celrep.2020.107576 (2020).
- 2 Strunz, M. *et al.* Alveolar regeneration through a Krt8+ transitional stem cell state that persists in human lung fibrosis. *Nature Communications* **11**, 3559, doi:10.1038/s41467-020-17358-3 (2020).
- 3 Setty, M. *et al.* Characterization of cell fate probabilities in single-cell data with Palantir. *Nature biotechnology* **37**, 451-460, doi:10.1038/s41587-019-0068-4 (2019).
- 4 Street, K. *et al.* Slingshot: cell lineage and pseudotime inference for single-cell transcriptomics. *BMC genomics* **19**, 477, doi:10.1186/s12864-018-4772-0 (2018).
- 5 Lange, M. *et al.* CellRank for directed single-cell fate mapping. *Nature methods* **19**, 159-170, doi:10.1038/s41592-021-01346-6 (2022).
- 6 Skoulidis, F. *et al.* Co-occurring genomic alterations define major subsets of KRAS-mutant lung adenocarcinoma with distinct biology, immune profiles, and therapeutic vulnerabilities. *Cancer discovery* **5**, 860-877, doi:10.1158/2159-8290.CD-14-1236 (2015).
- 7 Marjanovic, N. D. *et al.* Emergence of a High-Plasticity Cell State during Lung Cancer Evolution. *Cancer cell* **38**, 229-246.e213, doi:10.1016/j.ccell.2020.06.012 (2020).
- 8 Dost, A. F. M. *et al.* Organoids Model Transcriptional Hallmarks of Oncogenic KRAS Activation in Lung Epithelial Progenitor Cells. *Cell Stem Cell* **27**, 663-678.e668, doi:10.1016/j.stem.2020.07.022 (2020).
- 9 Fujimoto, J. *et al.* Development of Kras mutant lung adenocarcinoma in mice with knockout of the airway lineage-specific gene Gprc5a. *International journal of cancer* **141**, 1589-1599, doi:10.1002/ijc.30851 (2017).
- 10 Tao, Q. *et al.* Identification of the retinoic acid-inducible Gprc5a as a new lung tumor suppressor gene. *J Natl Cancer Inst* **99**, 1668-1682, doi:10.1093/jnci/djm208 (2007).
- 11 Steiling, K., Ryan, J., Brody, J. S. & Spira, A. The field of tissue injury in the lung and airway. *Cancer Prev Res (Phila Pa)* **1**, 396-403, doi:10.1158/1538-7446.APR2008-0174 (2008).
- 12 Fujimoto, J. *et al.* Comparative functional genomics analysis of NNK tobacco-carcinogen induced lung adenocarcinoma development in Gprc5a-knockout mice. *PloS one* **5**, e11847, doi:10.1371/journal.pone.0011847 (2010).
- 13 Daouk, R. *et al.* Genome-Wide and Phenotypic Evaluation of Stem Cell Progenitors Derived From Gprc5a-Deficient Murine Lung Adenocarcinoma With Somatic Kras Mutations. *Frontiers in oncology* **9**, 207, doi:10.3389/fonc.2019.00207 (2019).
- 14 Collado, M. *et al.* Tumour biology: senescence in premalignant tumours. *Nature* **436**, 642, doi:10.1038/436642a (2005).
- 15 Kathiriya, J. J. *et al.* Human alveolar type 2 epithelium transdifferentiates into metaplastic KRT5(+) basal cells. *Nat Cell Biol* **24**, 10-23, doi:10.1038/s41556-021-00809-4 (2022).
- 16 Kathiriya, J. J., Brumwell, A. N., Jackson, J. R., Tang, X. & Chapman, H. A. Distinct Airway Epithelial Stem Cells Hide among Club Cells but Mobilize to

- Promote Alveolar Regeneration. *Cell Stem Cell* **26**, 346-358.e344, doi:10.1016/j.stem.2019.12.014 (2020).
- 17 Kantrowitz, J. *et al.* Genome-Wide Gene Expression Changes in the Normal-Appearing Airway during the Evolution of Smoking-Associated Lung Adenocarcinoma. *Cancer Prev Res (Phila)* **11**, 237-248, doi:10.1158/1940-6207.CAPR-17-0295 (2018).
- 18 Westcott, P. M. *et al.* The mutational landscapes of genetic and chemical models of Kras-driven lung cancer. *Nature* **517**, 489-492, doi:10.1038/nature13898 (2015).
- 19 Cancer Genome Atlas Research, N. Comprehensive molecular profiling of lung adenocarcinoma. *Nature* **511**, 543-550, doi:10.1038/nature13385 (2014).
- 20 Wang, X. *et al.* Identification of MRTX1133, a Noncovalent, Potent, and Selective KRAS(G12D) Inhibitor. *J Med Chem* **65**, 3123-3133, doi:10.1021/acs.jmedchem.1c01688 (2022).
- 21 Wislez, M. *et al.* Inhibition of mammalian target of rapamycin reverses alveolar epithelial neoplasia induced by oncogenic K-ras. *Cancer Res* **65**, 3226-3235, doi:10.1158/0008-5472.Can-04-4420 (2005).
- 22 Hallin, J. *et al.* Anti-tumor efficacy of a potent and selective non-covalent KRASG12D inhibitor. *Nat Med* **28**, 2171-2182, doi:10.1038/s41591-022-02007-7 (2022).

Reviewer Reports on the First Revision:

Referees' comments:

Referee #1 (Remarks to the Author):

The authors have modified and reinforced the conclusions by including studies of:

1. Spatial transcriptomics of LUADs with patient 14 and mice at 7 months post-exposure to NNK, thus further confirming the KACs spatial distributions that correlate with tumor.
2. Large-scale chromosomal CNVs in all alveolar cell subsets and in malignant cells to confirm the increased global CNV compared to AT2 and AT1 cells.
3. IF experiments to further confirm KRT8 cells as the precursor of LUAD.

The major concerns regarding the quantification of heterogeneity and plasticity have been mostly addressed with minor concerns remaining:

1. To establish that KM-LUADs are more similar compared to other LUADs, the authors used Bhattacharyya distance as suggested. However, the authors only compared KM to KM and I don't understand why they failed to include EM-EM or MM-MM in Extended Data Fig. 1e, which are crucial for concluding KM-LUADs have higher similarity in transcriptomic than other LUADs.
2. The authors mentioned that "Harmony-corrected data were only used for visualization and inference of differentiation and analysis of metaprograms was performed on normalized counts". In that case, I find it misleading to present a figure with over-integration that doesn't match the analysis that didn't use the Harmony integrated data. Can the authors add the unintegrated visualization at least in Extended Data Figures?
3. CytoTRACE requires gene expression tables that are unfiltered and unnormalized as input, however, Response 2 mentions Cytotrace was applied to normalized data, which is confusing.
4. Statistical testing should be performed on Extended Data Fig. 2b to confirm P2 and P10 have significantly higher MP30 scores than non-KARS.

Minor comment related to new data in the revised version:

1. While the spatial data showing the KAC, KARS signature, and Krt8 expression has strengthened the results, further analysis could be performed to support the conclusion. For example, to confirm the differentiation from AT2 to AT1, in Fig 2j, the location of AT2 cells and AT1 cells may be identified using deconvolution methods for spatial transcriptomics analysis and their spatial distance to the tumor could be analyzed. The KAC signature in Fig 3f shows clear gradients. The transition between the tumor cell and KM-KAC may be directly shown from spatial transcriptomics by identifying the cell types from the ST data.

Referee #2 (Remarks to the Author):

The authors responded thoroughly to the comments from the three reviewers. The focus on epithelial cell populations in early-stage LUAD is important to gain a better understanding of the mechanisms of cancer initiation. The focus on KRAS-mutant tumors makes sense based on the

preliminary observations across multiple tumors. In the first part of the paper, a number of computational studies are performed that show specific transcriptional programs in KRAS-mutant cells, which is interesting but not really novel (KRAS-mutant signatures were generated before). A second conclusion of the first part is that tumors are heterogeneous at the single-cell level, which is also not really novel. Still, this part provides a resource for the field, and it is likely that a number of readers will find useful information in it. In the second part of the study, the authors focus on AICs, “intermediate” cells between AT1 and AT2 cells. These cells are interesting because they have also been identified in the lung epithelium in response to injury. Specifically, the authors identified “KACs” as a subtype of AICs that is associated with tumor development in KRAS-mutant LUAD in humans and in mouse models. Overall, the study represents an impressive amount of work on a cancer type that is highly prevalent and still very lethal. The identification of cell types implicated in lung cancer initiation may open new ways to inhibit the development and the long-term growth of lung cancer.

There is one additional point that may require some clarification: the authors show that KACs sometimes harbor mutations in KRAS, but not always, and their model in Figure 4h, distinguishes KACs and KRAS-mutant KACs. How KACs score so strongly for the mutant KRAS signature if only a few of these cells are mutant for KRAS? Possibly these cells already show increased KRAS signaling (before they get a KRAS mutation), which may also explain the result with the MRTX1133 inhibitor in the last figure. Is there a way to check for activation of KRAS signaling (in KRAS- wild-type KACs (versus AT2 cells and other AICs, using a transcriptional signature or phospho-MEK or phospho-ERK)? One interesting scenario would be if AT2 cells can reach several AIC states upon injury, including a KAC state; the “reason” why KACs are an important step in the tumorigenic process might be because they already rely on KRAS signaling, and may be “selected” to acquire KRAS mutations.

There are other open questions from this work, including whether non-KAC AICs cannot form tumors for example if activation of KRAS is detrimental in these cells. In addition, it remains unclear if KACs that appear after injury are capable of giving rise to AT1 cells, or if they represent a dead end (unless they acquire KRAS mutations) (and can they revert back to AT2). But these experiments may be more challenging to perform and could be part of future studies in the field.

Minor: In the Reporting Summary, the origin of the LKR13 cell line is not reported and it is unclear where this cell line is used in the manuscript.

Minor: Overall, the abstract is a bit confusing, and the general message diluted by some complicated sentences.

Minor: What is “Supplementary Figure 3” (line 122)?

Minor: what do we really learn from Figure 1h-j? A full page of text concludes that cancer cells show “transcriptional heterogeneity” - not sure these panels belong to the main figure based on the conclusion?

Minor: For ED Figure 2i, was it already known that KRAS-mutant LUAD has a worse survival than

other LUAD subtypes, and the signature is then just a validation of this previous observation?

Minor: For Figure 2a and ED Figure 3c, why is the fraction of non-AT2 cells the same between tumors and non-tumors? One would expect that tumors are enriched in cancer cells and therefore de-enriched in other normal epithelial cells. It is a bit confusing whether the authors use the name of epithelial cells for these epithelial cells or for cells with a signature resembling those cells.

Minor: add KAC on top of the panel in Figure 2g to make it easier for the reader.

Minor: line 369, the authors assume that all KRT8+ cells are KACs, which is not correct.

Minor: line 418, maybe the authors should reconsider using the word tumor “progenitors”, as some readers may interpret this word as “cell of origin” while others may think “cancer stem cell”?

Referee #3 (Remarks to the Author):

I appreciate the extensive effort from the authors in revision, which addressed many critiques. However, in regard to the primary conclusion of the paper, that the Krt8+ alveolar cells are precursors for lung adenocarcinoma, the evidence remains correlative, including data from single cell signature, trajectory analysis and mixed cell input organoid culture. SfpccreERT2-based lineage tracing labels the entire AT2 lineage, not specifically the Krt8+ alveolar cell lineage, and therefore does not directly demonstrate the primary conclusion.

I suggest that you consider Nature Communications as a suitable venue for your work. To transfer your manuscript there, please use our <https://mts-nature.nature.com/cgi-bin/main.plex?el=A1K4DAxh1C5GITf5X5A9ftdjLRCbCkMhhsbgACnTkDhqwZ> manuscript transfer portal. You will not have to re-supply manuscript metadata and files, unless you wish to make modifications, but please note that this link can only be used once and remains active until used. For more information, please see our http://www.nature.com/authors/author_resources/transfer_manuscripts.html?WT.mc_id=EMI_NPG_1511_AUTHORTRANSF&WT.ec_id=AUTHOR manuscript transfer FAQ page.

Note that any decision to opt in to In Review at the original journal is not sent to the receiving journal on transfer. You can opt in to <https://www.nature.com/nature-portfolio/for-authors/in-review> In Review at receiving journals that support this service by choosing to modify your manuscript on transfer. In Review is available for primary research manuscript types only.

Author Rebuttals to First Revision:

Dear referees,

We are truly grateful for your thoughtful, constructive, and positive comments. We also would like to thank you for your generous time spent evaluating our manuscript. We below provide point-by-point responses to the constructive comments you raised. We sincerely hope you now find our manuscript strengthened and suitable for publication in *Nature*.

Reviewer #1

General comment:

“The authors have modified and reinforced the conclusions by including studies of:

- 1. Spatial transcriptomics of LUADs with patient 14 and mice at 7 months post-exposure to NNK, thus further confirming the KACs spatial distributions that correlate with tumor.*
- 2. Large-scale chromosomal CNVs in all alveolar cell subsets and in malignant cells to confirm the increased global CNV compared to AT2 and AT1 cells.*
- 3. IF experiments to further confirm KRT8 cells as the precursor of LUAD.*

The major concerns regarding the quantification of heterogeneity and plasticity have been mostly addressed with minor concerns remaining.”

General response: We are grateful that reviewer #1 positively remarked that **our conclusions were reinforced by including new studies and analyses**. We also note the **positive comment by the reviewer that the major concerns regarding the quantification of heterogeneity and plasticity have been mostly addressed with minor concerns remaining**. In this rebuttal, we present point-by-point responses and plans to address reviewer #1's comments.

Minor Comment 1: “To establish that KM-LUADs are more similar compared to other LUADs, the authors used Bhattacharyya distance as suggested. However, the authors only compared KM to KM and I don’t understand why they failed to include EM-EM or MM-MM in Extended Data Fig. 1e, which are crucial for concluding KM-LUADs have higher similarity in transcriptomic than other LUADs.”

Response 1: We thank the reviewer for pointing this out and apologize for inadvertently forgetting to compare EM to EM. We have updated **Extended Data Fig. 1e, left** by including EM-EM (please see below). As we only have two MM samples (P4, P9) included in this study, the Bhattacharyya distance for these two samples is a single value and, thus, cannot be included in the box plot. As expected, we observed that Bhattacharyya distances between KM-LUADs (KM-KM) were significantly smaller compared to distances between EM-LUADs (EM-EM) ($P = 0.02$) and to distances within all other LUADs (Other-Other; $P = 0.03$).

Minor Comment 2: “The authors mentioned that “Harmony-corrected data were only used for visualization and inference of differentiation and analysis of metaprograms was performed on normalized counts”. In that case, I find it misleading to present a figure with over-integration that doesn’t match the analysis that didn’t use the Harmony integrated data. Can the authors add the unintegrated visualization at least in Extended Data Figures?”

Response 2: We apologize for inadvertently misleading this reviewer whom we thank for this astute comment. The unintegrated visualization of malignant cells was originally included in **Extended Data Fig. 1b,c** of our revised manuscript (please see below). We apologize for not pointing this out specifically in the previous response letter. We would also like to clarify that Harmony was used to correct potential batches among samples in **Fig. 1f-g** in order to better assess transcriptome similarity among tumor cells.

Minor Comment 3: *“CytoTRACE requires gene expression tables that are unfiltered and unnormalized as input, however, Response 2 mentions Cytotrace was applied to normalized data, which is confusing.”*

Response 3: We thank the reviewer for the careful review and for pointing this out, we rerun CytoTRACE using raw counts as suggested and observed only subtle changes in the corresponding figures and results, and which do not impact our main conclusions. To support our statement, we present below representative figures comparing various original results (before) and the updated results using raw counts (after). Note: previous **Fig. 1j** is now removed to comply with a comment by another reviewer on redundancy.

Figure 1g. CytoTRACE scores in LUADs between the original result using normalized count (Before) and the updated result using raw counts (After).

Figure 1g. Per sample distribution of cell CytoTRACE scores in the original results (Before) and the updated results using raw counts (After).

Figure 2f. CytoTRACE score in human KACs versus other AICs. Before: the original results. After: CytoTRACE scores calculated using raw counts.

Now Extended Data Fig. 5f. CytoTRACE scores in KACs versus other AICs in *KRAS*-mutant (KM) LUADs and *KRAS* wild-type LUADs. Before: original results. After: CytoTRACE scores calculated using raw counts.

Extended Data Fig. 8c. CytoTRACE scores in late mouse KACs with (mut) or without (wt) *Kras*^{G12D} mutation. Before: the original results. After: CytoTRACE scores calculated using raw counts.

Extended Data Fig. 10d. CytoTRACE scores across mouse GFP+ subsets. Before: the original results. After: CytoTRACE scores calculated using raw counts.

Minor Comment 4: *“Statistical testing should be performed on Extended Data Fig. 2b to confirm P2 and P10 have significantly higher MP30 scores than non-KARS.”*

Response 4: We thank the reviewer for this comment and have applied statistical tests as suggested. We observed significantly increased MP30 scores in P2 and P10 compared to non-KRAS samples ($P < 2.2 \times 10^{-16}$ for both comparisons). We now updated **Extended Data Fig. 2b** accordingly in the revised manuscript.

Minor comment related to new data in the revised version: *“While the spatial data showing the KAC, KARS signature, and Krt8 expression has strengthened the results, further analysis could be performed to support the conclusion. For example, to confirm the differentiation from AT2 to AT1, in Fig 2j, the location of AT2 cells and AT1 cells may be identified using deconvolution methods for spatial transcriptomics analysis and their spatial distance to the tumor could be analyzed. The KAC signature in Fig 3f shows clear gradients. The transition between the tumor cell and KM-KAC may be directly shown from spatial transcriptomics by identifying the cell types from the ST data.”*

Response: We are pleased that the reviewer finds that the **spatial data has strengthened the results**, and we agree with the reviewer’s astute suggestions on additional spatial analysis that could further support the conclusion.

Using CytoSpace, we found that spatial distances between KACs and tumour cells were indeed smaller compared to distances between alveolar cells in the nearby parenchymal region and tumour cells. Following trajectory analysis of a defined region extracted from the

ST data composed of a tumoural region as well as nearby KACs and alveolar cells, we noted that KACs were indeed intermediary in the transition of alveolar parenchyma to tumour cells. These findings are now depicted in **Extended Data Fig. 4a**.

Extended Data Fig. 4a. CytoSpace deconvolution and trajectory analysis of P14 LUAD ST data. The left spatial map is colored by deconvoluted cell types. Top right panel shows the neighboring cell composition of KACs and the bottom right panel depicts inferred trajectory and pseudotime prediction using monocle 2.

We describe these findings in lines 267-271 of the revised manuscript.

We performed similar trajectory analysis of the mouse ST data. For instance, for the mouse in **Extended Data Fig. 6h**, we had identified different clusters (C) that mapped to tumors (C7, C8), an area enriched with reactive pneumocytes/KACs (C2) and normal alveolar parenchyma (C0). Trajectory analysis of these clusters showed that, like the human data, KACs were intermediary in the transition of alveolar cells in the normal parenchyma to tumour cells. These data are now depicted in the **Extended Data Fig. 6i**. We also noted similar findings when analyzing defined tumour cells, KAC-enriched regions and alveolar parenchyma extracted from ST data of other mouse samples (now depicted in **Extended Data Fig. 6k**).

h**i**
Extended Data Fig. 6h, ST analysis of the same tumour-bearing mouse lung in **Fig. 3f** with cell clusters identified by Seurat (inlet) and mapped spatially (left). Spatial maps with scaled expression of Krt8 and Plaur are shown in the middle. **i**, Trajectory analysis of C0 (alveolar parenchyma), C2 (reactive area with KACs nearby tumours), and C7/C8 (representing two tumours) from the same tumour-bearing mouse lung in **h**.

We now describe the additional analysis of the mouse ST data in lines 392-396 of the revised manuscript.

We hope the reviewer finds that these revisions to analysis of the ST data to further strengthen our manuscript.

Reviewer #2

General comment: *“The authors responded thoroughly to the comments from the three reviewers. The focus on epithelial cell populations in early-stage LUAD is important to gain a better understanding of the mechanisms of cancer initiation. The focus on KRAS-mutant tumors makes sense based on the preliminary observations across multiple tumors. In the first part of the paper, a number of computational studies are performed that show specific transcriptional programs in KRAS-mutant cells, which is interesting but not really novel (KRAS-mutant signatures were generated before). A second conclusion of the first part is that tumors are heterogeneous at the single-cell level, which is also not really novel. Still, this part provides a resource for the field, and it is likely that a number of readers will find useful information in it. In the second part of the study, the authors focus on AICs, “intermediate” cells between AT1 and AT2 cells. These cells are interesting because they have also been identified in the lung epithelium in response to injury. Specifically, the authors identified “KACs” as a subtype of AICs that is associated with tumor development in KRAS-mutant LUAD in humans and in mouse models. Overall, the study represents an impressive amount of work on a cancer type that is highly prevalent and still very lethal. The identification of cell types implicated in lung cancer initiation may open new ways to inhibit the development and the long-term growth of lung cancer.”*

Response to general comment: We are pleased that reviewer #2 deemed that **we have responded thoroughly to the comments from the three reviewers**. We very much appreciate the reviewer’s overall positive tone which included comments such as that **the first part provided a resource for the field, and it is likely that a number of readers will find useful information in it; the study represents an impressive amount of work on a cancer type that is highly prevalent and still very lethal**. We are grateful for the reviewer’s understanding of the impact of our study in such that **the identification of cell types implicated in lung cancer initiation may open new ways to inhibit the development and long-term growth of lung cancer**. Indeed we found (Extended Data Fig. 11) that organoids enriched with KACs were highly responsive to targeted KRAS-G12D inhibition, thus raising the possible conjecture that targeting KRAS early on during lung

oncogenesis (e.g., at preneoplastic or 'KAC' stage, may be a viable approach for interception of *KRAS*-mutant LUAD.

The reviewer commented that the first part of the study pointing to specific transcriptional programs in *KRAS*-mutant cells and showing that tumours are heterogeneous at the single-cell level are interesting but not really novel. We agree with the reviewer that at first glance these conclusions and efforts may not come as novel -- we believe this is a fault on our part for inadvertently masking the importance of results in the first part of our study and their novelty. We do not only demonstrate *KRAS* signatures in tumour cells, but show that *KRAS*-mutant LUAD cells compared to malignant cells from other molecular subtypes of LUADs are unique in various different ways. For one, we found that transcriptional similarity between *KRAS*-mutant LUAD cells is considerably higher when compared to similarity indices between tumour cells from other molecular subgroups of LUAD (e.g., *EGFR*-mutant LUAD). We found that *KRAS*-mutant tumour cells almost universally displayed loss of alveolar dedifferentiation which was strikingly very different from *EGFR*-mutant LUADs. Notably, this property (loss of alveolar differentiation) is reciprocated in the KACs which we find in the second part of the study to lose alveolar gene programs and to acquire (at least some of the KACs) the same *KRAS* mutations in the tumour cells.

To comply with a suggestion by the same reviewer below (minor comment #4), we reduced discussion of the transcriptional heterogeneity of malignant cells focusing instead on unique features of *KRAS*-mutant LUAD cells. These features (e.g., reduced alveolar differentiation) of *KRAS*-mutant LUADs tie in well with our novel findings on properties of KACs (second part of the study) which also happen to exhibit elevated *KRAS* activation -- which we now show in response to another astute comment by the same reviewer. We also now better emphasize novel aspects in the first part of the study with respect to *KRAS*-mutant LUAD in the Discussion section (**page 25**) of the revised manuscript.

Additional point that requires some clarification: *"There is one additional point that may require some clarification: the authors show that KACs sometimes harbor mutations in KRAS, but not always, and their model in Figure 4h, distinguishes KACs and KRAS-mutant KACs. How KACS score so strongly for the mutant KRAS signature if only a few of these cells are mutant for KRAS? Possibly these cells already show increased KRAS signaling (before they get a KRAS mutation), which may also explain the result with the MRTX1133 inhibitor in the last figure. Is there a way to check for activation of KRAS signaling (in KRAS-wild-type KACs (versus AT2 cells and other AICs, using a transcriptional signature or phospho-MEK or phospho-ERK)? One interesting scenario would be if AT2 cells can reach several AIC states upon injury, including a KAC state; the "reason" why KACs are an important step in the tumorigenic process might be because they already rely on KRAS signaling, and may be "selected" to acquire KRAS mutations.*

Response to additional point that requires some clarification: We thank the reviewer for this very astute and constructive comment. Indeed, our new data demonstrate that some but not

all human KACs exhibit *KRAS* mutations (at a variant allele frequency of close to 5% when considering all LUAD tissues, and around 10% when considering *KRAS*-mutant LUAD tissues). This finding is reminiscent of earlier reported observations by our group and others of driver mutations in uninvolved tissue surrounding tumours (field cancerization) and which were discovered by bulk-sequencing approaches – the novelty here being the identification of the specific cell type in “normal” regions that solely harbors *KRAS* mutations and perhaps signifies clonal field cancerization.

To comply with the reviewer’s suggestion we compared the same *KRAS* activation signature that we derived and studied in human LUADs among *KRAS*-mutant KACs, *KRAS*-wild type KACs, other AICs, and AT2 cells. We found that the *KRAS* activation gene signature from our study was significantly increased in *KRAS*-mutant KACs relative to *KRAS*-WT counterparts ($P = 3.9 \times 10^{-3}$). Interestingly, the *KRAS* activation signature was also significantly increased in *KRAS*-WT KACs relative to other AICs ($P < 2.2 \times 10^{-16}$) and in other AICs relative to AT2 cells ($P < 2.2 \times 10^{-16}$), further pointing towards increased *KRAS* signaling along the AT2-AIC-KAC spectrum and prior to acquisition of *KRAS* mutations. These data were added as a new **Extended Data Fig. 5e**.

Extended Data Fig. 5e. *KRAS* activation signature was statistically compared across *KRAS*^{G12D} mutant KACs, *KRAS*^{G12D} -wild type KACs, AICs, and AT2 cells. Box, median ± interquartile range; whiskers, 1.5x interquartile range; center line: median. P-values were calculated using the Wilcoxon Rank-Sum test.

These new data are discussed in lines 326-330 and in lines 565-567 of the revised manuscript.

It is worthwhile to mention that this finding is concordant with our observation of a subset of human *KRAS* wild-type LUADs from the TCGA cohort that exhibit relatively high expression of *KRAS* activation signatures/metaprogram (Extended Data Figure 2h) and that display poorer survival compared with *KRAS* wild-type LUADs with relatively lower expression of the *KRAS* activation signature (Extended Data Figure 2i). The reviewer also mentions that “AT2 cells can reach several AIC states upon injury, including a KAC state; the “reason” why KACs is an important step in the tumorigenic process might be because they already rely on *KRAS* signaling, and may be “selected” to acquire *KRAS* mutations.” This perceptive supposition by the reviewer is supported by our new data in the last revised submission showing that KACs *in vivo* with time and during lung oncogenesis display increased frequency of somatic *Kras* mutations (**Extended Data Fig. 8b**).

Comment regarding open questions from this work: *“There are other open questions from this work, including whether non-KAC AICs cannot form tumors for example if activation of *KRAS* is detrimental in these cells. In addition, it remains unclear if KACs that appear after injury are capable of giving rise to AT1 cells, or if they represent a dead end (unless they acquire *KRAS* mutations) (and can they revert back to AT2). But these experiments may be more challenging to perform and could be part of future studies in the field.”*

Response to comment regarding open questions from this work: We thank the reviewer for pointing out these open questions in our study which can certainly guide the field to perform additional studies. In response to the comment by reviewer #3 (see below), and to further confirm that KACs indeed give rise to tumour cells, we studied *Gprc5a*^{-/-}; *Krt8*-CreER; *Rosa*^{tdT/+} mice exposed to NNK followed by tamoxifen for labeling of *Krt8*⁺ cells. Mice harboring *Krt8*-CreER/+; *Rosa*^{tdT/+} were first used to examine extent of labeling of *Krt8*⁺ cells following NNK or saline. We found markedly increased tdT⁺ labeling in the lung parenchyma at the end of an 8-week NNK exposure followed by tamoxifen compared to control saline-treated which is consistent with the increased levels of KACs that we noted by single-cell sequencing at the end of an 8-week NNK exposure. To examine the relevance of *Krt8*⁺ cells to tumour development, we studied *Gprc5a*^{-/-}; *Krt8*-CreER; *Rosa*^{tdT/+} mice at 8-12 weeks following an 8-week NNK exposure and that were injected with tamoxifen right after completing NNK treatment. Lung tissue areas containing tdT⁺ cells were increased at 8-12 weeks post-NNK compared with at the end of the 8-week NNK exposure. Ten out of 17 tumours showed tdT⁺/*Krt8*⁺ cells at varying levels, with some tumours showing strong extent of tdT labeling. The percentage of tdT⁺/*Lamp3*⁺ cells out of total tdT⁺ cells was similar between end of the 8-week NNK exposure and at 8-12 weeks following NNK in tumour-bearing lungs. The overwhelming majority of tdT⁺ tumour cells were *Lamp3*⁺ implying that these tumours developed from AT2 cells, in line with our previous observations in AT2 labeled lungs (now **Fig. 4c**). Of note, normal-appearing regions also showed tdT⁺ AT1 cells (*Nkx2-1*⁺/*Lamp3*⁻) suggesting possible turn-over of AT2 cells and KACs to AT1 cells in non-tumour regions.

These new data, coupled with our findings using AT2-lineage labeled mice (now **Fig. 4c** and **Extended Data Fig. 9,10**) and that we included in the previous round of revisions, suggest that KACs are involved in AT2-to-tumour transformation. It is worthwhile to mention that not all tumours in *Gprc5a*^{-/-}; *Krt8*-CreER; *Rosa*^{tdT/+} mice were tdT+. One possible reason for this is the difficulty in labeling all KACs, compared with parent AT2 cells, since KACs likely arise continuously during/post-NNK. Despite the technical challenges in labeling *Krt8*+ cells post carcinogen-exposure, the presence of early lung tumours that are tdT+/Krt8+ provides supportive evidence for the role of KACs in AT2-to-tumour transformation. These new data are now included as new **Fig. 4d,e** and **Supplementary Fig. 11** and are described in lines 507-526 of the revised manuscript.

Fig. 4d. IF analysis of tdT and Krt8 expression in lung tissues at the end of an 8-week NNK exposure (first row) as well as at 8-12 weeks following an 8-week NNK-exposure in normal-appearing regions (second row) and in tumours of *Gprc5a*^{-/-}; *Krt8*-CreER; *Rosa*^{tdT/+} (last two rows). Tamoxifen was delivered right after NNK exposure for six continuous days at 1 mg/dose. Images were captured at 20X magnification, scale bar: 10 μ m.

Fig. 4e. Percentages of lung tissue areas containing tdT+ cells were computed as described in the Methods and plotted (upper). Quantification of %tdT+/Lamp3+ cells among total tdT+ cells in normal-appearing regions at the end of an 8-week NNK exposure as well as in normal-appearing regions and tumours at 8-12 weeks following an 8-week NNK-exposure (lower).

Supplementary Figure 11. a, Representative images of IF analysis of tdT, Lamp3, and Nkx2-1 in lung tissues of saline-treated mice (upper row), in non-tumour (normal) lung regions of mice at end of an 8-week NNK exposure (middle row), as well as in non-tumour (normal) lung regions of mice at 8-12 weeks following an 8-week NNK exposure in *Gprc5a*^{-/-}; *Krt8-CreER*; *Rosa*^{tdT/+} mice (lower row). **b**, IF analysis of tdT and Lamp3 in 17 tumours detected in *Gprc5a*^{-/-}; *Krt8-CreER*; *Rosa*^{tdT/+} mice and showing negative (left, n = 7) and positive (right, n = 10) tdT labeling. Scale bars = 10µm.

We agree with the reviewer that there are open questions from this work that would be important to pursue. For instance, it is not clear whether KACs are a dominant or obligatory path in the AT2-to-tumour transformation. Also, we do not know the effects of expressing mutant oncogenes, *Kras* or others, in KACs. We also do not know whether introduction of mutant *Kras* in KACs that arise following different lung injury stimuli could enhance the likelihood of their transition to tumour cells rather than be involved in AT1 regeneration. We now discuss these open questions in the Discussion section (lines 591-596) of the revised manuscript.

Minor Comment 1: *"In the Reporting Summary, the origin of the LKR13 cell line is not reported and it is unclear where this cell line is used in the manuscript."*

Response to Minor Comment 1: We apologize for inadvertently not including the LKR13 cell line in the reporting summary. LKR13 is a cell line that was previously derived from a tumour in the *Kras*^{LA1} model of *Kras*^{G12D}-mutant LUAD (PMID: 15833854). We used it, in addition to our *Gprc5a*^{-/-} LUAD cell line MDA-F471, to first examine and confirm the effects of the KRAS-G12D inhibitor MTRX1133 on cancer cell growth of murine LUAD lines and on KRAS signaling (**Extended Data Figure 11b,c**). We now include this cell line in the data reporting summary.

Minor Comment 2: *"Overall, the abstract is a bit confusing, and the general message diluted by some complicated sentences."*

Response to Minor Comment 2: We apologize for rendering the abstract confusing and we now made effort to provide what we believe to be a clearer abstract.

Minor Comment 3: *"What is "Supplementary Figure 3" (line 122)?"*

Response to Minor Comment 3: We apologize for causing this confusion. Supplementary Figure 3 shows analysis of CNV burden and *KRAS*^{G12D} mutations when we clustered both tumour and normal cells together. The supplementary figure shows the presence of CNVs and *KRAS*^{G12D} mutations in only patient-specific clusters of tumour cells and not in normal epithelial cells of different lineages (e.g., ciliated). We inadvertently forgot to cite **Extended Data Figure 1b** (right panel) in this sentence to mention that also no *KRAS*^{G12D} mutations were found in other LUADs which confirmed our whole-exome sequencing (WES) analysis. We now revise this sentence to include citation of the proper Extended Data Figure panel and Supplementary Figure 3 (lines 122-126 of the revised manuscript).

Minor Comment 4: *“what do we really learn from Figure 1h-j? A full page of text concludes that cancer cells show “transcriptional heterogeneity” - not sure these panels belong to the main figure based on the conclusion?”*

Response to Minor Comment 4: We apologize for the lengthy discussion of Figure 1h-j. We apologize for inadvertently failing to accentuate the conclusions from Figure 1h. Hierarchical clustering based on metaprograms in Figure 1h (rows represent metaprograms and each column denotes a tumour cell) demonstrates that *KRAS*-mutant tumour cells cluster tightly together (C1) compared to tumour cells from say *EGFR*-mutant LUADs (dispersed through clusters C2, C3, C4, and C5) – likely because, as the heatmap shows, *KRAS*-mutant tumour cells show marked differences in the levels of several metaprograms such as EMT, senescence and alveolar programs – important to mention here that all these metaprograms are features of KACs. To comply with the reviewer’s suggestion we opted to keep **Fig. 1h**, and **moved Fig. 1i to Extended Data Fig. 2a**. In further reviewing our manuscript to comply with the reviewer’s suggestion we opted to **remove panel j** due to redundancy with previous panels in the figure. Accordingly, **we substantially shortened the text pertaining to panel h and its accompanying extended figure panels.**

Minor Comment 5: *“For ED Figure 2i, was it already known that *KRAS*-mutant LUAD has a worse survival than other LUAD subtypes, and the signature is then just a validation of this previous observation?”*

Response to Minor Comment 5: We apologize for inadvertently not clearly discussing this conclusion. Extended Data Figure 2i does not only include *KRAS*-mutant LUADs from the TCGA cohort but also *KRAS* wild-type LUADs with varying levels of expression of a *KRAS* signature (Extended Data Figure 2h) (as mentioned above in response to an astute comment by the same reviewer, similar in concept to the supposition that *KRAS* wild-type KACs, despite being *KRAS* wild-type, may still exhibit aberrant *KRAS* signaling). Extended Data Fig. 2i shows that *KRAS* wild-type LUADs with increased expression of *KRAS* signature (red color) exhibit significantly poorer survival compared to *KRAS* wild-type LUADs with lower levels of the signature (brown color). We now emphasize these points in the revised version.

Minor Comment 6: *“For Figure 2a and ED Figure 3c, why is the fraction of non-AT2 cells the same between tumors and non-tumors? One would expect that tumors are enriched in cancer cells and therefore de-enriched in other normal epithelial cells. It is a bit confusing whether the authors use the name of epithelial cells for these epithelial cells or for cells with a signature resembling those cells.”*

Response to Minor Comment 6: We thank the reviewer for raising this perceptive observation. Our findings here are concordant with earlier reported findings by our group on single-cell analysis of early-stage LUAD (Sinjab et al, Cancer Discovery, 2021) in which we found trends for enhanced fractions of non-alveolar epithelial cells in the LUAD ecosystem. It is important to mention to the reviewer that LUAD tissues that are digested to derive single-cell suspensions seldom are comprised of purely tumour cells. We found that resected

tissues of LUADs that arise in the lung periphery/parenchyma often are enriched with normal epithelial cells (mostly alveolar). It is plausible to suppose that since alveolar (e.g., AT2 and AT1) fractions are overall lower in the LUAD tissue/ecosystem, fractions (out of total epithelial cells) for other epithelial cell types, such as basal cells, are rendered to be increased in the LUAD tissue/ecosystem. It is important to note that these changes are trends that we found for the most part (except for basal cells) to be statistically insignificant.

Minor Comment 7: *“add KAC on top of the panel in Figure 2g to make it easier for the reader.”*

Response to Minor Comment 7: We now add ‘KAC’ on top of the panel in Figure 2g in the revised manuscript.

Minor Comment 8: *“line 369, the authors assume that all KRT8+ cells are KACs, which is not correct.”*

Response to Minor Comment 8: We corrected this in the revised manuscript.

Minor Comment 9: *“line 418, maybe the authors should reconsider using the word tumor “progenitors”, as some readers may interpret this word as “cell of origin” while others may think “cancer stem cell”?”*

Response to Minor Comment 9: We agree with the reviewer and we now avoid the term progenitors in the revised version and mention that KACs may be implicated in tumour development (line 429).

Reviewer #3

Comment: “I appreciate the extensive effort from the authors in revision, which addressed many critiques. However, in regard to the primary conclusion of the paper, that the Krt8+ alveolar cells are precursors for lung adenocarcinoma, the evidence remains correlative, including data from single cell signature, trajectory analysis and mixed cell input organoid culture. SftpcCreERT2-based lineage tracing labels the entire AT2 lineage, not specifically the Krt8+ alveolar cell lineage, and therefore does not directly demonstrate the primary conclusion.”

Response: We thank the reviewer for acknowledging **our extensive efforts in revision that have addressed many of their critiques**. We appreciate the reviewer's comment that SftpcCreERT2 labels the entire AT2 lineage and not specifically Krt8+ cells, and thus does not directly demonstrate the primary conclusion. We would be grateful if the reviewer kindly considers the ensemble of different types of correlative data that we provided which lend support to the supposition that KACs are implicated in lung tumour development including but not exclusive to: uniqueness of KACs with respect to other AICs; presence of driver KRAS mutations in both human and mouse KACs; the spatial proximity of KACs to human and mouse tumours which is reminiscent of pertinent clonal driver mutations in field cancerization (PMID: 33220180) and as schematized in Figure 4h; and the presence of KACs in other *Kras*-driven models.

Having said that, we agree with the reviewer that our conclusion would be strengthened by specific labeling of Krt8+ alveolar cells. Thus, **to address the reviewer's concern**, and to confirm that KACs indeed give rise to tumour cells, we studied *Gprc5a*^{-/-}; Krt8-CreER; *Rosa*^{tdT/+} mice exposed to NNK followed by tamoxifen for labeling of Krt8+ cells. Mice harboring Krt8-CreER/+; *Rosa*^{tdT/+} were first used to examine extent of labeling of Krt8+ cells following NNK or saline. We found markedly increased tdT+ labeling in the lung parenchyma at the end of an 8-week NNK exposure followed by tamoxifen compared to control saline-treated which is consistent with the increased levels of KACs that we noted by single-cell sequencing at the end of an 8-week NNK exposure. To examine the relevance of Krt8+ cells to tumour development, we studied *Gprc5a*^{-/-}; Krt8-CreER; *Rosa*^{tdT/+} mice at 8-12 weeks following an 8-week NNK exposure and that were injected with tamoxifen right after completing NNK treatment. Lung tissue areas containing tdT+ cells were increased at 8-12 weeks post-NNK compared with at the end of the 8-week NNK exposure. Ten out of 17 tumours showed tdT+/Krt8+ cells at varying levels, with some tumours showing strong extent of tdT labeling. The percentage of tdT+/Lamp3+ cells out of total tdT+ cells was similar between end of the 8-week NNK exposure and at 8-12 weeks following NNK in tumour-bearing lungs. The overwhelming majority of tdT+ tumour cells were Lamp3+ implying that these tumours developed from AT2 cells, in line with our previous observations in AT2 labeled lungs (now **Fig. 4c**). Of note, normal-appearing regions also showed tdT+ AT1 cells (Nkx2-1+/Lamp3-) suggesting possible turn-over of AT2 cells and KACs to AT1 cells in non-tumour regions.

These new data, coupled with our findings using AT2-lineage labeled mice (now **Fig. 4c** and **Extended Data Fig. 9,10**) and that we included in the previous round of revisions, suggest that KACs are involved in AT2-to-tumour transformation. It is worthwhile to mention that not all tumours in *Gprc5a*^{-/-}; *Krt8-CreER*; *Rosa*^{tdT/+} mice were tdT+. One possible reason for this is the difficulty in labeling all KACs, compared with parent AT2 cells, since KACs likely arise continuously during/post-NNK. Despite the technical challenges in labeling *Krt8*+ cells post carcinogen-exposure, the presence of early lung tumours that are tdT+/Krt8+ provides supportive evidence for the role of KACs in AT2-to-tumour transformation. These new data are now included as new **Fig. 4d,e** and **Supplementary Fig. 11** and are described in lines 507-526 of the revised manuscript.

Fig. 4d. IF analysis of tdT and Krt8 expression in lung tissues at the end of an 8-week NNK exposure (first row) as well as at 8-12 weeks following an 8-week NNK-exposure in normal-appearing regions (second row) and in tumours of *Gprc5a*^{-/-}; *Krt8-CreER*; *Rosa*^{tdT/+} (last two rows). Tamoxifen was delivered right after NNK exposure for six continuous days at 1 mg/dose. Images were captured at 20X magnification, scale bar: 10 μ m.

Fig. 4e. Percentages of lung tissue areas containing tdT+ cells were computed as described in the Methods and plotted (upper). Quantification of %tdT+/Lamp3+ cells among total tdT+ cells in normal-appearing regions at the end of an 8-week NNK exposure as well as in normal-appearing regions and tumours at 8-12 weeks following an 8-week NNK-exposure (lower).

Supplementary Figure 11. a, Representative images of IF analysis of tdT, Lamp3, and Nkx2-1 in lung tissues of saline-treated mice (upper row), in non-tumour (normal) lung regions of mice at end of an 8-week NNK exposure (middle row), as well as in non-tumour (normal) lung regions of mice at 8-12 weeks following an 8-week NNK exposure in *Gprc5a*^{-/-}; *Krt8*-CreER; *Rosa*^{tdT/+} mice (lower row). **b**, IF analysis of tdT and Lamp3 in 17 tumours detected in *Gprc5a*^{-/-}; *Krt8*-CreER; *Rosa*^{tdT/+} mice and showing negative (left, n = 7) and positive (right, n = 10) tdT labeling. Scale bars = 10µm.

We sincerely thank you for your time to read this long rebuttal letter and we hope that you and the editors find our manuscript strengthened and suitable for publication.

Sincerely,

Humam Kadara, PhD

Department of Translational Molecular Pathology

The University of Texas MD Anderson Cancer Center

Linghua Wang, MD, PhD

Department of Genomic Medicine

The University of Texas MD Anderson Cancer Center

Reviewer Reports on the Second Revision:

Referees' comments:

Referee #2 (Remarks to the Author):

The authors responded very well to the last round of reviews, including with new mouse data. The data in early human tumors and the validation of key observations in mouse models is a great combination.

Referee #3 (Remarks to the Author):

I appreciate the authors adding in Krt8creERT2 lineage tracing experiments. While it should be noted by the authors that Krt8creERT2 is not specific for KACs (they label airway cells and also some AT2s), the comparison between saline and treated does support the increased labeling of KACs and their contribution to tumor. I applaud the persistence of the authors, and would now support the publication of this study.